# Distributed Stochastic $K$-Level Optimization Over Networks

**Xinwen Zhang** [1]   **Yihan Zhang** [1]   **Hongchang Gao** [1]   **Heng Huang** [2]

## Abstract

In recent years, decentralized optimization has gained significant attention for solving machine learning problems where data are distributed across multiple devices. However, existing decentralized optimization algorithms are primarily designed for single-level and two-level optimization tasks, limiting their application to more complex problems such as decentralized stochastic $K$-level optimization, where $K > 2$. In this work, we propose a novel decentralized stochastic $K$-level variance-reduced gradient descent algorithm to address the significant computation and communication overhead caused by the multi-level structure in decentralized stochastic $K$-level optimization problems. Moreover, we propose a novel theoretical analysis to tackle the recursive dependence issue caused by the multi-level structure when establishing the convergence rate of our algorithm. Finally, the experimental results confirm the effectiveness of our proposed algorithm.

## 1. Introduction

In real-world machine learning tasks, data are often distributed across devices, making central aggregation difficult due to privacy and legal constraints. Decentralized optimization (Lian et al., 2017) addresses this challenge via peer-to-peer communication, offering greater reliability than the parameter-server framework. Meanwhile, a special stochastic 2-level optimization, i.e., stochastic bilevel optimization (SBO), which minimizes an upper-level objective subject to a nested lower-level problem, has seen growing interest in tasks such as hyperparameter tuning (Feurer & Hutter, 2019), meta-learning (Rajeswaran et al., 2019), and reinforcement learning (Giovannelli et al., 2025). Building on this, stochastic multi-level optimization (SMO)

[1]Department of Computer and Information Sciences, Temple University, Philadelphia, USA [2]Department of Computer Science, University of Maryland, College Park, Maryland, USA. Correspondence to: Hongchang Gao <hongchang.gao@temple.edu>.

*Proceedings of the 43rd International Conference on Machine Learning*, Seoul, South Korea. PMLR 306, 2026. Copyright 2026 by the author(s).

where $K > 2$, involving multiple lower-level problems, has emerged in applications such as compositional optimization (Yang et al., 2019) and graph neural networks (Liu et al., 2021). However, most existing work remains limited to bilevel settings, restricting applicability to more complex and general $K$-level problems.

In this paper, we aim to develop an efficient distributed optimization algorithm to solve the general stochastic $K$-level optimization problem, i.e., the distributed stochastic multi-level optimization (SMO) problem:

$$
\min_{x \in \mathbb{R}^{d_0}} \underbrace{\frac{1}{N} \sum_{n=1}^{N} f_{K+1}^{(n)}(x, y_K^*(x))}_{F_{K+1}(x) \triangleq f_{K+1}(x, y_K^*(x))}
$$

$$
s.t.\ y_K^*(x) = \arg \min_{y_K \in \mathbb{R}^{d_K}} \underbrace{\frac{1}{N} \sum_{n=1}^{N} f_K^{(n)}(y_K, y_{K-1}^*(x))}_{F_K(y_K) \triangleq f_K(y_K, y_{K-1}^*(x))}
$$

$$
\cdots \tag{1}
$$

$$
s.t.\ y_1^*(x) = \arg \min_{y_1 \in \mathbb{R}^{d_1}} \underbrace{\frac{1}{N} \sum_{n=1}^{N} f_1^{(n)}(y_1, y_0^*(x))}_{F_1(y_1) \triangleq f_1(y_1, y_0^*(x))},
$$

where $y_0^*(x) = x$. This paper primarily investigates distributed SMO problem in the decentralized setting, where a system with $N$ workers performs peer-to-peer communication. The $n$-th worker has its own loss functions: $f_{K+1}^{(n)}(\cdot, \cdot) = \mathbb{E}[f^{(n)}(\cdot, \cdot; \xi_{K+1}^{(n)})]$ is the upper-level loss function on the $n$-th worker and $f_k^{(n)}(\cdot, \cdot) = \mathbb{E}[f_k^{(n)}(\cdot, \cdot; \xi_k^{(n)})]$ is the $k$-th ($k \in \{1, 2, \cdots, K\}$) lower-level loss function on the $n$-th worker, $\{\xi_k^{(n)}\}_{k=1}^{K+1}$ denotes the random samples on the $n$-th worker.

There has been increasing interest in developing optimization algorithms for the special case of Eq. (1), *i.e.*, decentralized SBO problems (Yang et al., 2022; Chen et al., 2023; Gao et al., 2023a; Zhang et al., 2026b; Zhu et al., 2024; Chen et al., 2024; Kong et al., 2025). For example, (Chen et al., 2024) proposed a decentralized stochastic bilevel gradient method; (Yang et al., 2022) incorporated momentum techniques, and (Gao et al., 2023a) applied variance reduction to accelerate convergence. A key challenge in decentralized SBO lies in estimating the hypergradient $\nabla_x f_{K+1}^{(n)}(x, y_K^*(x))$, which requires computing Jacobian

and Hessian matrices across all workers. This challenge is further amplified in decentralized SMO problems due to the **recursive dependence** across levels — both **within individual workers** and **between workers** — particularly when multiple lower-level subproblems ($K > 1$) are involved.

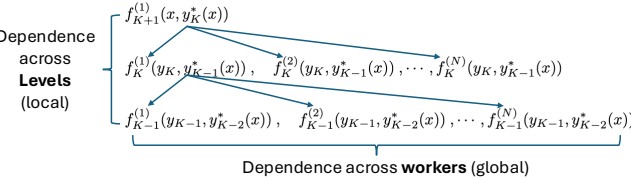

*Figure 1.* An illustration of the recursive dependence across levels and workers in Eq. (1).

Specifically, as shown in Figure 1, the recursive dependence in Eq. (1) manifests in two ways in the decentralized setting: **locally**, across levels within a single worker; and **globally**, through communication across workers. This local and global recursive structure introduces two key challenges in estimating the hypergradient on each worker:

- **Computation Challenge**: Due to *locally recursive dependence*, each worker must compute the Hessian-inverse–vector product *at every level*, incurring a large computation overhead.
- **Communication Challenge**: Due to *globally recursive dependence*, each worker has to perform communication between different workers *at every level* when calculating the Hessian-inverse-vector product, leading to a high communication cost.

Due to these unique challenges from the multi-level structure, existing decentralized SBO algorithms cannot be directly applied to decentralized SMO. To address this, (Yang et al., 2023) proposed a decentralized stochastic multi-level gradient descent algorithm using moving average to estimate gradients, Jacobian matrices, and Hessian matrices. However, this method suffers from two key issues: (1) it needs to communicate the high-dimensional Jacobian and Hessian matrices, which is not practical for real-world machine learning models; (2) its convergence analysis relies on some assumptions that do not hold simultaneously. In particular, (Yang et al., 2023) assumes both uniformly bounded gradients and strong convexity of the lower-level objective. However, strongly convex functions generally do not satisfy globally bounded gradient assumptions, making these conditions potentially contradictory. Beyond (Yang et al., 2023), no existing methods solve Eq. (1). Hence, it remains unclear how to effectively handle the **locally and globally recursive dependence across levels** to efficiently optimize the decentralized SMO problem in Eq. (1) and what theoretical convergence rate can be achieved in this setting.

Therefore, to address the computational and communication challenges in solving Eq. (1), we propose an efficient decentralized multi-level gradient descent algorithm with rigorous theoretical guarantees under mild assumptions. Specifically, to reduce computation overhead, we design a fast-converging algorithm that finds stationary points in fewer iterations. To lower communication cost, we compress communicated variables and gradients without impairing the convergence performance. Notably, no prior work has explored how to reduce both computation overhead and communication cost for decentralized SMO problems, and existing methods cannot address the unique challenges caused by the multi-level structure.

### 1.1. Contribution

We propose a computation- and communication-efficient decentralized stochastic multi-level variance-reduced gradient descent algorithm (CE-DMSGD-VR) to optimize Eq. (1), with the following contributions:

- Algorithm design: We introduce a novel **recursive Hessian-inverse-vector product** to estimate the hypergradient, computed locally and communicated across workers to address both locally and globally recursive dependence. A recursive variance reduction strategy is further used to accelerate convergence. To reduce communication cost, we **compress variables and recursive Hessian-inverse-vector product** using error-feedback compression, updating each with local full-precision gradients and global compensated error.

- Convergence analysis: The **triple-recursive dependence** across *levels*, *workers*, and *iterations* brings significant challenges. We derive *recursive upper bound for optimization errors, gradient estimation errors, consensus errors, and compression errors* to capture those triple-recursive dependencies, and construct *a novel potential function with carefully designed coefficients*. This allows us to establish a convergence rate of $O(1/(N\epsilon^3(1-\lambda)^4\delta^2))$, where $\epsilon > 0$ is the accuracy of the solution, $1 - \lambda$ is the spectral gap, and $\delta \in (0, 1)$ reflects compression quality. To the best of our knowledge, this is the first work to propose decentralized SMO algorithms with rigorous theoretical guarantees.

At last, we apply our algorithm to hyperparameter optimization in a novel way, demonstrating broader applicability. Extensive experimental results confirm the computational and communication efficiency of our method.

## 2. Related Works

As a special case of decentralized multi-level optimization, decentralized bilevel optimization has been extensively explored recently. Numerous decentralized bilevel optimization algorithms (Yang et al., 2022; Gao et al., 2023a; Chen et al., 2023; Liu et al., 2022; Lu et al., 2022; Liu et al., 2023;

Zhang et al., 2026b; Zhu et al., 2024; Chen et al., 2024; Dong et al., 2025; Kong et al., 2025; Zhang et al., 2026a) and several extensions in federated learning (Gao, 2022; Zhang et al., 2025) have been proposed and their theoretical convergence rates have been established. In particular, (Lu et al., 2022; Liu et al., 2022; 2023) focus on the decentralized bilevel optimization problem where the lower-level subproblem is not distributed on different workers, while the other existing methods focus on the case where both the upper-level and lower-level problems are distributed on different workers. For example, (Liu et al., 2022) proposed to directly compute the Hessian inverse to estimate the hypergradient in each worker and then to compute the SPIDER gradient estimator to update local variables. Obviously, the computation overhead is large when directly computing the Hessian inverse matrix. (Yang et al., 2022) proposed to use Neumman expansion series approach to approximately compute the Hessian inverse matrix. However, it requires to communicate the high-dimensional Jacobian and Hessian matrices, which can incur high communication costs. (Chen et al., 2024) proposed to compute and communicate the Jacobian-Hessian-inverse product to estimate the hypergradient on each worker. However, the Jacobian-Hessian-inverse product is still a high-dimensional matrix, so there is no advantage in saving communication costs compared to (Yang et al., 2022). (Gao et al., 2023a) focuses on the homogeneous setting so that the hypergradient can be estimated with the local Jacobian and Hessian matrices. Recently, to address the problem of high communication cost, (Zhang et al., 2026b) developed a communication-efficient single-loop decentralized bilevel optimization algorithm based on the Hessian-inverse-vector product, where this product and two variables are communicated between workers. As a result, (Zhang et al., 2026b) can significantly save communication costs compared to (Yang et al., 2022; Chen et al., 2023; 2024). Due to its efficacy in saving communication costs, the Hessian-inverse-vector product based single-loop algorithm has been studied in different settings recently, such as the deterministic setting (Dong et al., 2025), the gossip communication setting (Kong et al., 2025), and various communication strategies (Zhu et al., 2024). However, all these existing methods focus solely on the bilevel problem.

Only a recent work (Yang et al., 2023) explored the decentralized multi-level problem. However, as an extension of (Yang et al., 2022), it still needs to communicate the high-dimensional Jacobian and Hessian matrices. With increasing levels, the issue of high communication costs becomes more severe. Moreover, several important machine learning models can be formulated as a special case of multi-level optimization, namely compositional optimization problems (Zhang & Xiao, 2021; Jiang et al., 2022; 2025), which have been extensively studied in decentralized (Gao & Huang, 2021; Gao, 2024a) and federated learning

settings (Gao et al., 2022; Gao, 2024b). In addition, deep AUC maximization (Yuan et al., 2022; Zhang & Gao, 2025b) can also be formulated as a multi-level compositional minimax optimization problem (Zhang et al., 2023; 2024; Zhang & Gao, 2025a). Motivated by these applications, this paper considers a more general decentralized multi-level optimization framework. Consequently, developing communication-efficient algorithms together with rigorous theoretical guarantees remains an important open problem.

To save communication costs in distributed optimization, performing compression for variables or gradients has been actively studied recently (Seide et al., 2014; Stich et al., 2018; Alistarh et al., 2017; Wen et al., 2017; Karimireddy et al., 2019; Richtárik et al., 2021; Gao & Huang, 2020; Gao et al., 2021; 2023b). However, compressing variables or gradients could degrade the convergence performance. To address this issue, the error-feedback mechanism (Seide et al., 2014) was proposed, where the compression residual is used to compensate for the compression loss. After that, the error-feedback compression mechanism has been applied to different settings. As for decentralized optimization, (Koloskova et al., 2019) studied how to apply the error-feedback mechanism to the commonly used gossip communication method. Later, (Zhao et al., 2022; Yau & Wai, 2023; Yan et al., 2023) developed decentralized stochastic gradient descent algorithms for minimization problems based on different gradient estimators. In particular, (Zhao et al., 2022) focuses on the standard stochastic gradient, while (Yau & Wai, 2023) uses the variance-reduced gradient estimator (Cutkosky & Orabona, 2019). (Yan et al., 2023) focuses on the stochastic proximal gradient descent algorithm. As far as we know, the error-feedback compression mechanism has never been explored for decentralized bilevel or multi-level optimization problems. We are only aware of a work (He et al., 2024) that used the error-feedback compression mechanism under the centralized setting. Therefore, it is necessary to develop communication efficient algorithms for decentralized multi-level optimization problems because the communication cost becomes higher when the number of levels is large.

## 3. Preliminaries

### 3.1. Problem Setup

We first introduce several standard assumptions commonly adopted in existing works (Gao et al., 2023a; Yang et al., 2023; Zhang et al., 2026b).

**Assumption 3.1.** For $k \in \{1, \cdots, K\}$, the $k$-th lower-level loss function $f_k(y_k, y_{k-1})$ is $\mu$-strongly-convex with respect to $y_k \in \mathbb{R}^{d_k}$ for any given $y_{k-1} \in \mathbb{R}^{d_{k-1}}$, where $\mu > 0$ is a constant value.

**Assumption 3.2.** For $n \in \{1, \cdots, N\}$, the upper-level loss

**Deterministic hypergradient:**

$$\nabla F_{K+1}(x) \triangleq \frac{1}{N}\sum_{n=1}^{N}\nabla F_{K+1}^{(n)}(x) = \frac{1}{N}\sum_{n=1}^{N}\nabla_1 f_{K+1}^{(n)}(x, y_K^*(x)) + \frac{1}{N}\sum_{n=1}^{N}\left(\frac{\partial y_K^*(x)}{\partial x}\right)^T \nabla_2 f_{K+1}^{(n)}(x, y_K^*(x))$$

where

$$\left(\frac{\partial y_K^*(x)}{\partial x}\right)^T = (-1)^K \prod_{k=1}^{K}\left(\frac{1}{N}\sum_{i_k=1}^{N}J_k^{(i_k)}\right)^T\left[\frac{1}{N}\sum_{j_k=1}^{N}H_k^{(j_k)}\right]^{-1}, \tag{3}$$

$$J_k^{(n)} = \nabla_{12}^2 f_k^{(n)}(y_k^*(x), y_{k-1}^*(x)), \quad H_k^{(n)} = \nabla_{11}^2 f_k^{(n)}(y_k^*(x), y_{k-1}^*(x)),$$

$J_k^{(n)}$ and $H_k^{(n)}$ denote the Jacobian and Hessian matrices of the $k$-th level loss function on the $n$-th worker.

**Hessian-inverse-Jacobian product $\hat{z}_k^*(x)$:**

$$\underbrace{[H_1]^{-1}J_2^T}_{\hat{z}_1^*(x)\in\mathbb{R}^{d_1\times d_2}} \cdots \underbrace{[H_{K-1}]^{-1}J_K^T}_{\hat{z}_{K-1}^*(x)\in\mathbb{R}^{d_{K-1}\times d_K}} \underbrace{[H_K]^{-1}\nabla_2 f_{K+1}(x, y_K^*(x))}_{\hat{z}_K^*(x)\in\mathbb{R}^{d_K\times 1}}, \tag{4}$$

where $J_k = \frac{1}{N}\sum_{n=1}^{N}J_k^{(n)}$ and $H_k = \frac{1}{N}\sum_{n=1}^{N}H_k^{(n)}$.

*Figure 2.* Deterministic hypergradient and Hessian-inverse-Jacobian product.

function $f_{K+1}^{(n)}(\cdot, \cdot)$ satisfies the $L_f$-Lipschitz smoothness in expectation, where $L_f > 0$ is a constant value. Additionally, $\mathbb{E}[\|\nabla_2 f_{K+1}^{(n)}(\cdot, \cdot; \xi_{K+1})\|^2] \leq C_f^2$ where $C_f > 0$ is a constant value.

**Assumption 3.3.** For $n \in \{1, \cdots, N\}$, $k \in \{1, \cdots, K\}$, $(y_k, y_{k-1}) \in \mathbb{R}^{d_k} \times \mathbb{R}^{d_{k-1}}$ and $(y_k', y_{k-1}') \in \mathbb{R}^{d_k} \times \mathbb{R}^{d_{k-1}}$, the $k$-th lower-level loss function $f_k^{(n)}(\cdot, \cdot)$ satisfies the $L_g$-Lipschitz smoothness in expectation with respect to the first variable, where $L_g > 0$ is a constant value. Additionally, the Jacobian matrix $\nabla_{12}^2 f_k^{(n)}(\cdot, \cdot)$ is $L_J$-Lipschitz continuous in expectation, where $L_J > 0$ is a constant value. Moreover, the Hessian matrix $\nabla_{11}^2 f_k^{(n)}(\cdot, \cdot)$ is $L_H$-Lipschitz continuous in expectation, where $L_H > 0$ is a constant value. Furthermore, $\mu\mathbf{I} \preceq \nabla_{11}^2 f_k^{(n)}(y_k, y_{k-1}; \xi_k) \preceq L_g\mathbf{I}$, $\mathbb{E}[\|\nabla_{12}^2 f_k^{(n)}(y_k, y_{k-1}; \xi_k)\|] \leq C_J$.

**Assumption 3.4.** For $n \in \{1, \cdots, N\}$ and $k \in \{1, \cdots, K + 1\}$, the stochastic gradient, stochastic Jacobian matrix, and stochastic Hessian matrix are unbiased estimators of their deterministic counterparts respectively. Additionally, their variances are upper bounded by $\sigma > 0$.

When considering the decentralized setting, we have the following assumptions (Koloskova et al., 2019; Zhao et al., 2022; Yan et al., 2023):

**Assumption 3.5.** The adjacency matrix $W = [w_{ij}] \in \mathbb{R}_+^{N \times N}$ satisfies: $W = W^T$ and $W\mathbf{1} = \mathbf{1}$. Here, $w_{ij} > 0$ indicates the $i$-th worker and the $j$-th worker are connected. Otherwise, $w_{ij} = 0$. Moreover, the absolute value of its eigenvalues satisfies: $|\lambda_n| \leq \cdots \leq |\lambda_2| < |\lambda_1| = 1$. The spectral gap of $W$ is denoted by $1 - \lambda$, where $\lambda = |\lambda_2|$.

**Assumption 3.6.** The compression operator $\mathcal{C} : \mathbb{R}^d \to \mathbb{R}^d$ satisfies the following condition:

$$\|x - \mathcal{C}(x)\|^2 \leq (1 - \delta)\|x\|^2, \tag{5}$$

where $\delta \in (0, 1]$.

Of the widely used compression operators, both the Top-K operator (Stich et al., 2018) and the scaled sign operator (Karimireddy et al., 2019) meet this requirement.

### 3.2. Challenges of DSMO

Following (Yang et al., 2023), for Eq. (1), the deterministic hypergradient is given in Figure 2. From Eq. (3), it is clear that **the deterministic hypergradient $\nabla F_{K+1}^{(n)}(x)$ on worker $n$ relies on all $\{(J_k^{(n)}, H_k^{(n)})\}_{k=1,n=1}^{K,N}$ across $K$ levels and $N$ workers**, resulting in high computation and communication costs. For example, (Yang et al., 2023) directly communicate $(J_k^{(n)}, H_k^{(n)})$ at each level $k$ and compute the Hessian inverse via Neumann series expansion. However, this approach incurs substantial overhead due to high dimensionality of matrices and iterative nature of Neumann series expansion.

An alternative approach for (Yang et al., 2023) is to extend the method designed for decentralized bilevel optimization problems to compute the hypergradient in Eq. (2). Specifically, a straightforward extension for (Zhang et al., 2026b; Zhu et al., 2024; Kong et al., 2025) is to introduce the **Hessian-inverse-Jacobian product $\hat{z}_k^*(x)$ for each level $k$ independently, as Eq. (4)**. Then, by viewing each $\hat{z}_k^*(x)$ as the optimal solution of a quadratic strongly convex optimization problem as (Zhang et al., 2026b; Zhu et al., 2024;

Kong et al., 2025), this approach can avoid introducing additional loops to compute the Hessian inverse. However, this straightforward extension still suffers from high communication costs because the dimensionality of $\hat{z}_k^*(x)$ is still large.

## 4. Algorithm

### 4.1. Recursive Hessian-Inverse-Vector Product

To address the aforementioned computation and communication challenges, we introduce the **recursive Hessian-inverse-vector product** $z_k^*(x) \in \mathbb{R}^{d_k \times 1}$ as follows:

$$[H_1]^{-1} J_2^T \cdots [H_{K-1}]^{-1} J_K^T \underbrace{[H_K]^{-1} \nabla_2 f_{K+1}(x, y_K^*(x))}_{\hat{z}_K^*(x) \in \mathbb{R}^{d_K \times 1}} . \quad (6)$$

$$\underbrace{\phantom{[H_1]^{-1} J_2^T \cdots [H_{K-1}]^{-1} J_K^T [H_K]^{-1} \nabla_2 f_{K+1}(x, y_K^*(x))}}_{\hat{z}_{K-1}^*(x) \in \mathbb{R}^{d_{K-1} \times 1}}$$

$$\underbrace{\phantom{[H_1]^{-1} J_2^T \cdots [H_{K-1}]^{-1} J_K^T [H_K]^{-1} \nabla_2 f_{K+1}(x, y_K^*(x))}}_{z_1^*(x) \in \mathbb{R}^{d_1 \times 1}}$$

It can be seen that $z_k^*(x)$ is computed recursively. More specifically, it is computed as follows:

$$z_k^*(x) = \left[ \frac{1}{N} \sum_{i_k=1}^N H_k^{(i_k)} \right]^{-1} \left[ \frac{1}{N} \sum_{j_k=1}^N J_k^{(j_k)} \right] z_{k+1}^*(x) . \quad (7)$$

Obviously, the **recursive Hessian-inverse-vector product** $z_k^*(x) \in \mathbb{R}^{d_k \times 1}$ has a much smaller dimensionality than the **Hessian-inverse-Jacobian product** $\hat{z}_k^*(x) \in \mathbb{R}^{d_{k-1} \times d_k}$. As a result, communicating $z_k^*(x)$ can save communication costs significantly than communicating $\hat{z}_k^*(x)$.

Based on this novel recursive Hessian-inverse-vector product, we can compute the hypergradient as follows.

**Lemma 4.1.** *Given Assumptions 3.1-3.4, the deterministic hypergradient can be computed as follows:*

$$\nabla F_{K+1}(x) \triangleq \frac{1}{N} \sum_{n=1}^N \nabla F_{K+1}^{(n)}(x) = \frac{1}{N} \sum_{n=1}^N \nabla_1 f_{K+1}^{(n)}(x, y_K^*(x))$$

$$+ (-1)^K \left( \frac{1}{N} \sum_{i=1}^N \nabla_{12}^2 f_1^{(i)}(y_1^*(x), x) \right)^T z_1^*(x) . \quad (8)$$

It is worth noting that $z_k^*(x)$ at the $k$-th level **depends on the** $\{(J_k^{(n)}, H_k^{(n)})\}_{n=1}^N$ **at the $k$-th level for all workers and** $z_{k+1}^*(x)$ **at the $(k+1)$-th level**, which indicates both the locally and globally recursive dependence. Such recursive dependence does not occur in traditional SBO problems, highlighting the unique challenges.

**Notations.** Throughout the paper, we use capital notation to represent all variables in the decentralized network, such as $X = [x^{(1)}, x^{(2)}, \cdots, x^{(N)}]$. Additionally, we denote $J = \frac{1}{N} \mathbf{1} \mathbf{1}^T$.

### 4.2. CE-DSMGD-VR

We propose a novel computation- and communication-efficient decentralized stochastic multi-level gradient de-

scent with variance reduction (**CE-DSMGD-VR**) algorithm in Algorithm 1. The detail is shown below.

---

**Algorithm 1** CE-DSMGD-VR

---

**Input:** $x_0$, $\{y_{k,0}\}_{k=1}^K$, $\{z_{k,0}\}_{k=1}^K$, $\eta > 0$, $\alpha_x > 0$, $\alpha_y > 0$, $\alpha_z > 0$, $\gamma_x > 0$, $\gamma_y > 0$, $\gamma_z > 0$, $\beta_x > 0$, $\beta_y > 0$, $\beta_z > 0$.

1: **for** $t = 0, \cdots, T-1$ **do**
2:     Update variables:
$$X_{t+1} = X_t - \gamma_x \eta P_t + \rho_x \eta \hat{X}_t (W - I),$$
$$Y_{k,t+1} = Y_{k,t} - \gamma_y \eta Q_{k,t} + \rho_y \eta \hat{Y}_{k,t}(W - I),$$
$$Z_{k,t+1} = \mathcal{P}(Z_{k,t} - \gamma_z \eta R_{k,t} + \rho_z \eta \hat{Z}_{k,t}(W - I)),$$
3:     Variance-reduced gradient estimator:
    Update $\{M_{t+1}^x\}$ as Eq. (15) ,
    Update $\{M_{k,t+1}^y\}_{k=1}^K$ as Eq. (13) ,
    Update $\{M_{k,t+1}^z\}_{k=1}^K$ as Eq. (10) ,
4:     Gradient tracking:
$$P_{t+1} = P_t + M_{t+1}^x - M_t^x + \beta_x \hat{P}_t(W - I),$$
$$Q_{k,t+1} = Q_{k,t} + M_{k,t+1}^y - M_{k,t}^y + \beta_y \hat{Q}_{k,t}(W - I),$$
$$R_{k,t+1} = R_{k,t} + M_{k,t+1}^z - M_{k,t}^z + \beta_z \hat{R}_{k,t}(W - I),$$
5:     Error Feedback and Communication:
$$\hat{X}_{t+1} = \hat{X}_t + \mathcal{C}(X_{t+1} - \hat{X}_t), \quad \hat{P}_{t+1} = \hat{P}_t + \mathcal{C}(P_{t+1} - \hat{P}_t),$$
$$\hat{Y}_{k,t+1} = \hat{Y}_{k,t} + \mathcal{C}(Y_{k,t+1} - \hat{Y}_{k,t}), \quad \hat{Q}_{k,t+1} = \hat{Q}_{k,t} + \mathcal{C}(Q_{k,t+1} - \hat{Q}_{k,t}),$$
$$\hat{Z}_{k,t+1} = \hat{Z}_{k,t} + \mathcal{C}(Z_{k,t+1} - \hat{Z}_{k,t}), \quad \hat{R}_{k,t+1} = \hat{R}_{k,t} + \mathcal{C}(R_{k,t+1} - \hat{R}_{k,t}),$$
6: **end for**

---

**Update recursive Hessian-inverse-vector product.** To alleviate both the globally and locally recursive dependence issue, we estimate $z_k^*(x)$ locally and then communicate it across workers. Specifically, $z_k^*(x)$ can be interpreted as the optimal solution of the following strongly-convex quadratic optimization problem:

$$\min_{\substack{z_k, \\ k \in \{1, \cdots, K-1\}}} h_k(z_k) \triangleq \frac{1}{N} \sum_{n=1}^N \left( \underbrace{\frac{1}{2} z_k^T H_k^{(n)} z_k - z_k^T J_{k+1}^{(n)} z_{k+1}^*(x)}_{h_k^{(n)}(z_k)} \right),$$

$$h_K(z_K) = \frac{1}{N} \sum_{n=1}^N (\underbrace{\frac{1}{2} z_K^T H_k^{(n)} z_K - z_K^T \nabla_2 f_{K+1}^{(n)}(x, y_K^*(x))}_{h_K^{(n)}(z_K)}) .$$

On the $n$-th worker, we compute a local $z_{k,t}^{(n)}$ to approximate the global $z_k^*(\bar{x}_t)$ at the $t$-th iteration with the following local gradient:

$$v_{k,t}^{(n)} = \nabla_{11}^2 f_k^{(n)}(y_{k,t}^{(n)}, y_{k-1,t}^{(n)}) z_{k,t}^{(n)}$$
$$- \nabla_{12}^2 f_{k+1}^{(n)}(y_{k+1,t}^{(n)}, y_{k,t}^{(n)}) z_{k+1,t}^{(n)} , \quad (9)$$

where $\bar{x}_t = \frac{1}{N} \sum_{n=1}^N x_t^{(n)}$ denotes the global variable, $z_{k,t}^{(n)}$ and $y_{k,t}^{(n)}$ represent the variables on the $n$-th worker at the $t$-th iteration for any $k \in \{1, \cdots, K\}$.

In practice, to update $\{z_{k,t}^{(n)}\}_{k=1}^K$, we compute a stochastic *recursive variance-reduced gradient estimator* $\{M_{k,t+1}^z\}_{k=1}^K$

as follows:

$$M_{k,t+1}^z = (1 - \alpha_z\eta^2)(M_{k,t}^z - \tilde{V}_{k,t,t+1}) + \tilde{V}_{k,t+1,t+1}, \quad (10)$$

where $\alpha_z > 0$ and $\alpha_z\eta^2 < 1$, the matrix $\tilde{V}_{k,t+1,t+1} = [\tilde{v}_{k,t+1,t+1}^{(1)}, \cdots, \tilde{v}_{k,t+1,t+1}^{(N)}]$ includes the local stochastic gradient on all workers (so do $\tilde{V}_{k,t+1,t+1}$, $M_{k,t}^z$, $M_{k,t+1}^z$), and the local stochastic gradient $\tilde{v}_{k,t,t+1}^{(n)}$ and $\tilde{v}_{k,t+1,t+1}^{(n)}$ are defined as:

$$\tilde{v}_{k,t,t+1}^{(n)} = \nabla_{11}^2 f_k^{(n)}(y_{k,t}^{(n)}, y_{k-1,t}^{(n)}; \xi_{k,t+1}^{(n)})z_{k,t}^{(n)}$$
$$- \nabla_{12}^2 f_{k+1}^{(n)}(y_{k+1,t}^{(n)}, y_{k,t}^{(n)}; \xi_{k+1,t+1}^{(n)})z_{k+1,t}^{(n)},$$
$$\tilde{v}_{k,t+1,t+1}^{(n)} = \nabla_{11}^2 f_k^{(n)}(y_{k,t+1}^{(n)}, y_{k-1,t+1}^{(n)}; \xi_{k,t+1}^{(n)})z_{k,t+1}^{(n)}$$
$$- \nabla_{12}^2 f_{k+1}^{(n)}(y_{k+1,t+1}^{(n)}, y_{k,t+1}^{(n)}; \xi_{k+1,t+1}^{(n)})z_{k+1,t+1}^{(n)}, \quad (11)$$

where $\{\xi_{k,t+1}^{(n)}\}_{k=1}^K$ denotes the random sample at the $t+1$-th iteration on the $n$-th worker.

We then apply the gradient tracking approach to communicate the gradients between workers, as shown in step 4 in Algorithm 1. Here, $R_{k,t+1}$ estimates the global variance-reduced gradient, $\hat{R}_{k,t}$ is an auxiliary variable in step 5, which stores the compression residual as follows:

$$\hat{R}_{k,t+1} = \hat{R}_{k,t} + \mathcal{C}(R_{k,t+1} - \hat{R}_{k,t}), \quad (12)$$

where the compression operator $\mathcal{C}(\cdot)$ compresses the difference between the full-precision variable $R_{k,t+1}$ and the auxiliary variable $\hat{R}_{k,t}$. Furthermore, $\hat{R}_{k,t}(W - I)$ denotes the compressed residual communicated between each worker and its neighbors to reduce communication cost. Similarly, variable $z_{k,t+1}^{(n)}$ is also compressed.

After obtaining $\{R_{k,t+1}\}_{k=1}^K$, our algorithm updates the local variable $\{Z_{k,t}\}_{k=1}^K$ using the local gradient estimator and its neighbor's compression residuals as shown in step 2 in Algorithm 1. Here, $\mathcal{P}(\cdot)$ denotes a projection step since $\|z_k^*(x)\|$ is upper bounded (Lemma D.4), and $\rho_x > 0$ is a hyperparameter that controls the update from the compression residual. Each worker $n$ communicates $\mathcal{C}(z_{k,t+1}^{(n)} - \hat{z}_{k,t}^{(n)})$ with its neighbors and stores their compression residuals $\{\hat{z}_{k,t+1}^{(n')} : w_{nn'} > 0\}$. Thus, $\hat{Z}_{k,t}(W - I)$ is computed locally without involving communication.

**Update lower-level variables.** Similarly, to update $\{y_{k,t}^{(n)}\}_{k=1}^K$ we use the current iterate $y_{k,t}$ to approximate the optimal $y_{k,t}^*(x_t)$, and compute *stochastic recursive variance-reduced gradient estimators* $\{M_{k,t+1}^y\}_{k=1}^K$ as:

$$M_{k,t+1}^y = (1 - \alpha_y\eta^2)(M_{k,t}^y - \tilde{U}_{k,t,t+1}) + \tilde{U}_{k,t+1,t+1}, \quad (13)$$

where $\alpha_y > 0$ and $\alpha_y\eta^2 < 1$, the matrix $\tilde{U}_{k,t,t+1} = [\tilde{u}_{k,t,t+1}^{(1)}, \cdots, \tilde{u}_{k,t,t+1}^{(N)}]$ includes the stochastic gradient on all workers (so do $\tilde{U}_{k,t,t+1}$, $M_{k,t+1}^y$, and $M_{k,t}^y$), and the stochastic gradient $\tilde{u}_{k,t,t+1}^{(n)}$ and $\tilde{u}_{k,t+1,t+1}^{(n)}$ are defined as:

$$\tilde{u}_{k,t,t+1}^{(n)} = \nabla_1 f_k^{(n)}(y_{k,t}^{(n)}, y_{k-1,t}^{(n)}; \xi_{k,t+1}^{(n)}),$$
$$\tilde{u}_{k,t+1,t+1}^{(n)} = \nabla_1 f_k^{(n)}(y_{k,t+1}^{(n)}, y_{k-1,t+1}^{(n)}; \xi_{k,t+1}^{(n)}). \quad (14)$$

**Update upper-level variable.** Based on $\{z_{k,t}^{(n)}\}_{k=1}^K$ and $\{y_{k,t}^{(n)}\}_{k=1}^K$ at the $t$-th iteration, we can compute a recursive variance-reduced gradient estimator for hypergradient as:

$$M_{t+1}^x = (1 - \alpha_x\eta^2)(M_t^x - \tilde{G}_{t,t+1}) + \tilde{G}_{t+1,t+1}, \quad (15)$$

where $\alpha_x > 0$ and $\alpha_x\eta^2 < 1$, the matrix $\tilde{G}_{t,t+1} = [\tilde{g}_{t,t+1}^{(1)}, \cdots, \tilde{g}_{t,t+1}^{(N)}]$ includes the stochastic gradient on all workers (so do $\tilde{G}_{t,t+1}$, $M_{t+1}^x$, and $M_t^x$), and the stochastic hypergradient $\tilde{g}_{t,t+1}^{(n)} \triangleq \hat{\nabla}F_{K+1}^{(n)}(x_t^{(n)}; \hat{\xi}_{t+1}^{(n)})$ and $\tilde{g}_{t+1,t+1}^{(n)} \triangleq \hat{\nabla}F_{K+1}^{(n)}(x_{t+1}^{(n)}; \hat{\xi}_{t+1}^{(n)})$ are defined as follows:

$$\hat{\nabla}F_{K+1}^{(n)}(x_t^{(n)}; \hat{\xi}_{t+1}^{(n)}) = \nabla_1 f_{K+1}^{(n)}(x_t^{(n)}, y_{K,t}^{(n)}; \xi_{K+1,t+1}^{(n)})$$
$$+ (-1)^K \nabla_{12}^2 f_1^{(n)}(y_{1,t}^{(n)}, x_t^{(n)}; \xi_{1,t+1}^{(n)})z_{1,t}^{(n)}, \quad (16)$$
$$\hat{\nabla}F_{K+1}^{(n)}(x_{t+1}^{(n)}; \hat{\xi}_{t+1}^{(n)}) = \nabla_1 f_{K+1}^{(n)}(x_{t+1}^{(n)}, y_{K,t+1}^{(n)}; \xi_{K+1,t+1}^{(n)})$$
$$+ (-1)^K \nabla_{12}^2 f_1^{(n)}(y_{1,t+1}^{(n)}, x_{t+1}^{(n)}; \xi_{1,t+1}^{(n)})z_{1,t+1}^{(n)},$$

where $\hat{\xi}_{t+1}^{(n)} = \{\xi_{K+1,t+1}^{(n)}, \xi_{1,t+1}^{(n)}\}$ denotes random samples at the $t+1$-th iteration on the $n$-th worker.

The other variables and gradient estimators are compressed, communicated, and updated in the same manner.

# 5. Theoretical Analysis

## 5.1. Convergence Rate

For simplicity, we use *opt-err$_z$* to represent the optimization estimation error for $\{z_{k,0}\}_{k=1}^K$ at initialization summed across all levels, *i.e.*, $\sum_{k=1}^K \mathbb{E}[\|\bar{z}_{k,0} - z_k^*(\bar{x}_0)\|^2]$. Similarly, we use *opt-err$_y$* to represent $\sum_{k=1}^K \mathbb{E}[\|\bar{y}_{k,0} - y_k^*(\bar{x}_0)\|^2]$.

**Theorem 5.1.** *Given Assumptions 3.1-3.5, by setting $\eta \leq \min\{\frac{1}{2\gamma_x L_F}, \frac{1}{\gamma_z\mu}, \frac{1}{\sqrt{\alpha_x}}, \frac{1}{\sqrt{\alpha_y}}, \frac{1}{\sqrt{\alpha_z}}, \frac{1}{\rho}, 1\}$, $\alpha_y > 0$, and $\alpha_z > 0$, $\beta_x = \beta_y = \beta_z = \beta$, $\rho_x = \rho_y = \rho_z = \rho$, when the conditions in Eq. (58) are satisfied, Algorithm 1 has the following convergence upper bound:*

$$\frac{1}{T}\sum_{t=0}^{T-1}\mathbb{E}[\|\nabla F_{K+1}(\bar{x}_t)\|^2] \leq \frac{2(F_{K+1}(x_0) - F_{K+1}(x_*))}{\eta\gamma_x T}$$

$$+ O\left(\frac{\alpha_z}{\alpha_x\eta\gamma_y T}\right)opt\text{-}err_y + O\left(\frac{1}{\gamma_z\eta T}\right)opt\text{-}err_z$$

$$+ O\left(\frac{\gamma_z}{\alpha_x N\eta T}\right)opt\text{-}err_z + O\left(\frac{\gamma_z}{\beta^2(1-\lambda)^2\eta T}\right)opt\text{-}err_z$$

$$+ O\left(\frac{1}{\beta(1-\lambda)\eta T}\right)\frac{1}{N}\sum_{n=1}^N\mathbb{E}[\|\nabla_1 f_{K+1}^{(n)}(x_0, y_{K,0})\|^2]$$

$$+ O\Big(\frac{1}{\beta(1-\lambda)\eta T}\Big)\frac{1}{N}\sum_{n=1}^{N}\sum_{k=1}^{K}\mathbb{E}[\|\nabla_1 f_k^{(n)}(y_{k,0}, y_{k-1,0})\|^2]$$

$$+ O\Big(\frac{\alpha_z}{\alpha_x\alpha_y\eta^2 T}\frac{1}{NS_0}\Big) + O\Big(\frac{1}{\alpha_x\eta^2 T}\frac{1}{NS_0}\Big) + O\Big(\frac{\alpha_z^2\eta^2}{\alpha_x N}\Big)$$

$$+ O\Big(\frac{\alpha_z\alpha_y\eta^2}{\alpha_x N}\Big) + O\Big(\frac{\alpha_x\eta^2}{N}\Big) + O\Big(\frac{1}{\beta^2(1-\lambda)^2\eta TS_0}\Big)$$

$$+ O\Big(\frac{1}{\beta^2(1-\lambda)^2\eta T}\Big) + O\Big(\frac{\alpha_x^2\eta^3}{\beta^2(1-\lambda)^2}\Big)$$

$$+ O\Big(\frac{\alpha_y^2\eta^3}{\beta^2(1-\lambda)^2}\Big) + O\Big(\frac{\alpha_z^2\eta^3}{\beta^2(1-\lambda)^2}\Big) .$$

*Remark* 5.2. $\alpha_y > 0$ and $\alpha_z > 0$ are free hyperparameters. The upper bound of $\alpha_x$, $\gamma_x$, $\gamma_y$, and $\gamma_y$, $\beta$ in Eq. (58) only relies on the Lipschitz constant, the strong convexity constant and compression hyperparameter $\delta$, which does not affect the order of the convergence rate.

*Remark* 5.3. According to Eq. (58), for $\epsilon \in (0, 1)$, by setting $T = O\left(\frac{\delta^{-2}}{N(1-\lambda)^4\epsilon^3}\right)$, $S = O(\frac{1}{\epsilon})$, $\eta = O(N\epsilon)$, $\gamma_x = \gamma_y = \gamma_z = O(\delta^2(1-\lambda)^4)$, $\beta = O(\delta(1-\lambda))$, $\rho = O(\delta(1-\lambda))$, $\alpha_x = O(1/N)$, $\alpha_y = O(1/N)$, $\alpha_z = O(1/N)$, Algorithm 1 can achieve the $\epsilon$-accuracy solution:

$$\frac{1}{T}\sum_{t=0}^{T-1}\mathbb{E}[\|\nabla F(\bar{x}_t)\|^2] \leq O(\epsilon^2) .$$

To the best of our knowledge, this is the first communication-efficient decentralized multi-level optimization algorithm with theoretical guarantees. Moreover, this is the first time to reveal how the hyperparameter $\delta$ of the compression operator affects the convergence rate.

*Remark* 5.4. The iteration complexity $T = O\left(\frac{1}{N(1-\lambda)^4\epsilon^3}\right)$ surpasses that of the decentralized SMO algorithm proposed in (Yang et al., 2023) and is consistent with the state-of-the-art results for decentralized SBO algorithms as presented in (Zhang et al., 2026b). To our knowledge, this is the first time that the iteration complexity $T = O\left(\frac{1}{N(1-\lambda)^4\epsilon^3}\right)$ has been achieved for decentralized multi-level ($K > 1$) optimization problems. Here, we omit the compression parameter $\delta$ when comparing with DSMO/DSBO baselines, since those methods do not employ compressor operators.

### 5.2. Unique Challenges and Proof Sketch

Unique challenges and proof sketch are provided in Appendix B, and the complete proof is given in Appendix C-G.

## 6. Experiment

In this section, we apply our algorithm to the hyperparameter optimization task in a novel manner to demonstrate its

performance. We believe that this novel application can advance the development of multi-level optimization.

### 6.1. Hyperparameter Optimization via Multi-Level Optimization

In this experiment, we propose to use the multi-level optimization algorithm to solve the hyperparameter optimization task, where the lower-level optimization problem is nonconvex. Specifically, given a binary classification task where the classifier is a deep neural network, the lower-level optimization problem learns the weight of the neural network, which is a nonconvex optimization problem, and the upper-level optimization problem learns the hyperparameter, which is the coefficient of the regularization term. Formally, we have the following optimization problem:

$$\min_x \frac{1}{N}\sum_{n=1}^{N}\mathcal{L}(w^*(x); \mathcal{V}^{(n)})$$

$$s.t.\ w^*(x) = \arg\min_w \frac{1}{N}\sum_{n=1}^{N}\mathcal{L}(w; \mathcal{D}^{(n)}) + \mathcal{R}(x, w) , \quad (17)$$

where $\mathcal{R}(x, w)$ represents the regularization term $\frac{1}{d_1 d_2}\sum_{p=1}^{d_1}\sum_{q=1}^{d_2}\exp(x_{l,q})w_{l,pq}^2$ for the weight of each layer of the neural network and $l$ is the layer index, $\mathcal{D}^{(n)}$ denotes the training set, while $\mathcal{V}^{(n)}$ denotes the validation set on the $n$-th worker, $w$ denotes the weights of all layers.

Since the lower-level optimization problem is nonconvex, most of existing decentralized bilevel optimization algorithms cannot be used to solve it because they are designed for the bilevel optimization problem with a strongly-convex lower problem. To handle this practical model, we propose to convert the bilevel optimization problem in Eq. (17) into a multi-level optimization problem and then we can use the multi-level optimization algorithm to solve it. Specifically, we propose to use the gradient descent algorithm to update $w$ for one step and enforce the new update to satisfy the regularization explicitly by solving the following problem:

$$\min_y \frac{1}{N}\sum_{n=1}^{N}\frac{1}{|\mathcal{D}^{(n)}|}\sum_{i\in\mathcal{D}^{(n)}}\|y - (w - \theta\nabla\mathcal{L}(w; \mathcal{D}_i^{(n)}))\|^2$$

$$+ \mathcal{R}(x, y) . \quad (18)$$

Here, $w - \theta\nabla\mathcal{L}(w; \mathcal{D}_i^{(n)})$ denotes the one-step update via stochastic gradient of the loss term and $\theta > 0$ is the learning rate. Then, *the first term in this loss function is to enforce the new weight $y$ close to the update from gradient descent, while the second term $\mathcal{R}(x, y)$ is to enforce the new weight $y$ to be effectively regularized by the regularization*. Based on this reformulation, we perform multiple updates on the model weights, leading to a multi-level optimization problem, whose full expression is given in Eq. (19). Note that in each iteration we reset the weight $w$ of the neural network to $y_K^*(x)$ in the last iteration. Obviously, each lower-level

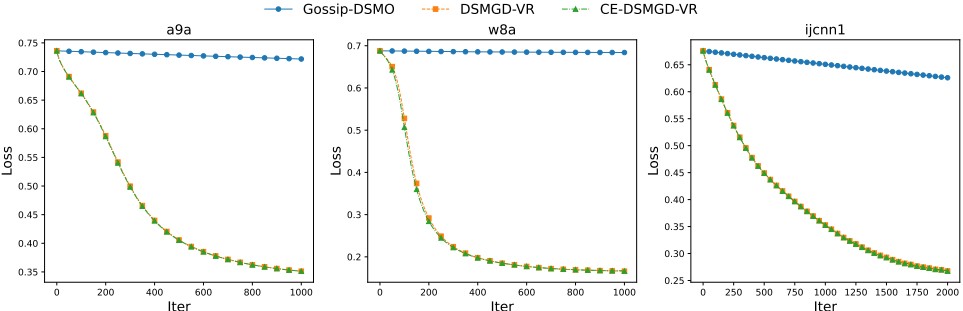

*Figure 3.* The upper-level loss function value with respect to the number of iterations.

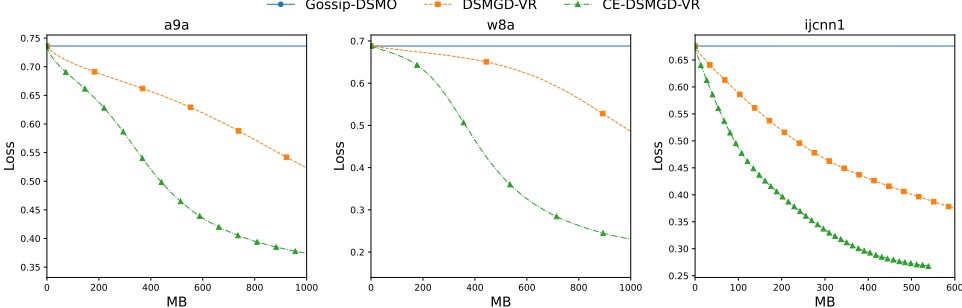

*Figure 4.* The upper-level loss function value with respect to the communicated megabytes (MB).

optimization problem is strongly convex, which satisfies our problem setting. Then, we can use our Algorithm 1 to solve it.

### 6.2. Comparison with Multi-Level Baselines

**Experimental Settings.** The datasets used in this experiment include a9a, w8a, and ijcnn1, which are obtained from LIBSVM[1]. The neural network used in this experiment is a two-layer fully connected neural network. The size of the hidden layer has 10, and the size of the output layer is 1. Additionally, the number of levels $K$ is set to 2. Therefore, it is a three-level optimization problem. As for the baseline method, to the best of our knowledge, there exists only one baseline method, Gossip-DSMO (Yang et al., 2023). In addition, to evaluate the impact of compression on performance and communication cost, we include an ablated variant of our method that replaces the compressed operation with full communication while retaining the recursive Hessian-inverse-vector product, referred to as DSMGD-VR. Therefore, we compare our CE-DSMGD-VR with these two baseline methods for the multi-level optimization problem in Eq. (19). Throughout this experiment, we set the solution accuracy $\epsilon$ to 0.1 for those three datasets. Then, according to the convergence analysis in (Yang et al., 2023), we set its learning rate to $\epsilon^2$ and those of DSMGD-VR and CE-DSMGD-VR to $\epsilon$. For all methods, we set the coefficient

[1] https://www.csie.ntu.edu.tw/~cjlin/libsvmtools/datasets/

of the momentum to 0.1. Moreover, for our Algorithm 1, we set $\gamma_x = \gamma_y = \gamma_z = 0.9$ and select $\alpha_x$, $\alpha_y$, and $\alpha_z$ from $\{0.1, 0.3, 0.5, 0.7, 0.9\}$. Then, we run all these algorithms on eight workers, which are connected via a ring graph, and set their batch size to 100 on each worker. The compression operator is Top-20%.

**Experimental Results.** We plot the upper-level loss function with respect to the number of iterations in Figure 3 and the communicated megabytes (MB) in Figure 4. From Figure 3, we can observe that both DSMGD-VR and CE-DSMGD-VR, converge much faster than Gossip-DSMO, which confirms the superior convergence rate of our algorithm. Moreover, we can observe that DSMGD-VR and CE-DSMGD-VR have almost the same convergence performance in terms of the number of iterations, which confirms that the compression operation in our CE-DSMGD-VR does not impair the convergence performance. Furthermore, from Figure 4, we can observe that DSMGD-VR has a much smaller communication cost than the baseline method. The reason is that our proposed recursive Hessian-inverse-vector product can save communication costs in each iteration significantly. Moreover, our algorithm CE-DSMGD-VR has a much less communication cost than the full-precision DSMGD-VR, which confirms the communication efficiency of CE-DSMGD-VR.

### 6.3. Comparison with Bilevel Baselines

Given the availability of more baselines for decentralized bilevel optimization, we further evaluate Algorithm 1 on this

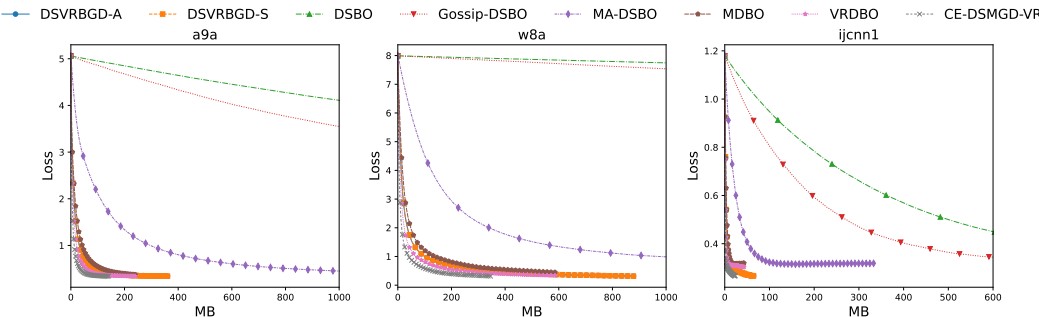

*Figure 5.* The upper-level loss function value with respect to the communicated megabytes (MB).

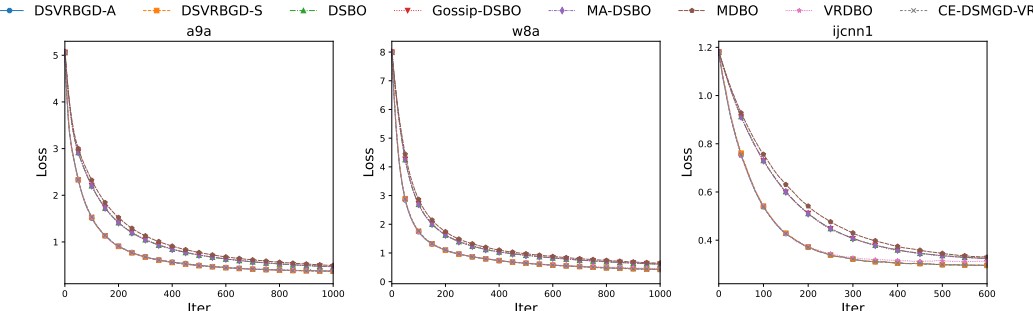

*Figure 6.* The upper-level loss function value with respect to the number of iterations.

problem. We still consider the hyperparameter optimization problem in Eq. (17), but use logistic regression instead of deep neural networks. As a result, the lower-level problem is strongly convex, while the upper-level problem remains nonconvex. This setting enables comparison with existing decentralized bilevel optimization algorithms designed under this assumption, including Algorithm 1.

In particular, we compare our Algorithm 1 against state-of-the-art full-precision decentralized bilevel methods, including DSBO (Chen et al., 2024), gossip-DSBO (Yang et al., 2022), MA-DSBO (Chen et al., 2023), MDBO (Gao et al., 2023a), VRDBO (Gao et al., 2023a), DSVRBGD-S (Zhang et al., 2026b), and DSVRBGD-A (Zhang et al., 2026b). Throughout this experiment, we set the solution accuracy $\epsilon$ to $0.05$ for the those three datasets. Then, according to the convergence analysis in (Yang et al., 2022; Chen et al., 2023; Gao et al., 2023a; Chen et al., 2024), we set the learning rate of DSBO, gossip-DSBO, and MDBO to $\epsilon^2$ and that of VRDBO, DSVRBGD-S, DSVRBGD-A, and our Algorithm 1 to $\epsilon$. For all momentum-based methods, we set the coefficient of the momentum to $0.1$. Moreover, for our Algorithm 1, we use the same hyperparameter settings as in the previous experiments. With these experimental settings, we run all these algorithms on eight workers, which are connected via a ring graph, and set their batch size to $100$ on each worker.

In Figure 5, we plot the upper-level loss function value with respect to the communicated megabytes (MB) when using

the Top-K compression operator with $K$ being $20\%$. It can be observed that our Algorithm 1 has a much smaller communication cost than all baseline methods for convergence, which confirms the communication efficiency of CE-DSMGD-VR. Moreover, we plot the upper-level loss function value with respect to the number of iterations in Figure 6. It can be observed that the compression operation in CE-DSMGD-VR does not degrade the convergence performance compared to all full-precision baseline methods, which further confirms the effectiveness of our algorithm.

Additional experiments on hyperparameter sensitivity are provided in Appendix A.

## 7. Conclusion

In this paper, we developed a novel decentralized stochastic multi-level optimization algorithm, aiming to address the large computation and communication cost caused by the multi-level structure. Specifically, we developed a novel recursive Hessian-inverse-vector product to reduce the computation and communication cost. Moreover, we proposed to compress the communicated variables to further save communication costs. With these novel algorithm designs, our algorithm achieves superior performance in practical applications. Moreover, we established the convergence rate of our algorithm, which is the first time to provide theoretical guarantees for decentralized multi-level optimization under mild assumptions.

## Acknowledgements

We thank anonymous reviewers for constructive comments. X. Zhang, Y. Zhang, and H. Gao was partially supported by U.S. NSF CAREER 2339545, NSF IIS 2416607, NSF CNS 2107014. H. Huang was partially supported by NSF IIS 2347592, 2348169, DBI 2405416, CCF 2348306, CNS 2347617, RISE 2536663.

## Impact Statement

This paper presents work whose goal is to advance the field of Machine Learning. There are many potential societal consequences of our work, none which we feel must be specifically highlighted here.

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

# Contents

## A. More Experiments

In our experiments, we consider the following hyperparameter optimization via multi-level optimization:

$$\min_x \frac{1}{N} \sum_{n=1}^{N} \mathcal{L}(y_K^*(x); \mathcal{V}^{(n)})$$

$$s.t., y_K^*(x) = \arg\min_{y_K} \mathcal{R}(x, y_K) + \frac{1}{N} \sum_{n=1}^{N} \frac{1}{m_n} \sum_{i \in \mathcal{D}^{(n)}} \|y_K - (y_{K-1}^*(x) - \theta \nabla \mathcal{L}(y_{K-1}^*(x); \mathcal{D}_i^{(n)}))\|^2$$

$$\cdots \tag{19}$$

$$y_2^*(x) = \arg\min_{y_2} \mathcal{R}(x, y_2) + \frac{1}{N} \sum_{n=1}^{N} \frac{1}{m_n} \sum_{i \in \mathcal{D}^{(n)}} \|y_2 - (y_1^*(x) - \theta \nabla \mathcal{L}(y_1^*(x); \mathcal{D}_i^{(n)}))\|^2$$

$$y_1^*(x) = \arg\min_{y_1} \mathcal{R}(x, y_1) + \frac{1}{N} \sum_{n=1}^{N} \frac{1}{m_n} \sum_{i \in \mathcal{D}^{(n)}} \|y_1 - (w - \theta \nabla \mathcal{L}(w; \mathcal{D}_i^{(n)}))\|^2 .$$

where $m_n = |\mathcal{D}^{(n)}|$ denotes the number of training samples on the $n$-th worker.

We further conduct experiments to study the impact of different hyperparameters on the convergence of our algorithm, including the learning-rate coefficient $\gamma$, momentum coefficient $\alpha\eta^2$, and compression-related coefficients $\beta$ and $\rho$. We also investigate different compression ratios and communication topologies for Algorithm 1.

In Figure 7, we run our two algorithms on a9a dataset with $\gamma \in \{0.1, 0.3, 0.5, 0.7, 0.9\}$, while keeping all other settings unchanged. From its two subplots, we can conclude that a larger $\gamma$ can accelerate the convergence speed because it is the coefficient of the learning rate. In Figure 8, we run our two algorithms on a9a dataset with $\alpha\eta^2 \in \{0.1, 0.3, 0.5, 0.7, 0.9\}$, while keeping all other settings unchanged. From its two subplots, we can conclude that the coefficient in the variance-reduced gradient estimator does not have a significant influence on the convergence rate.

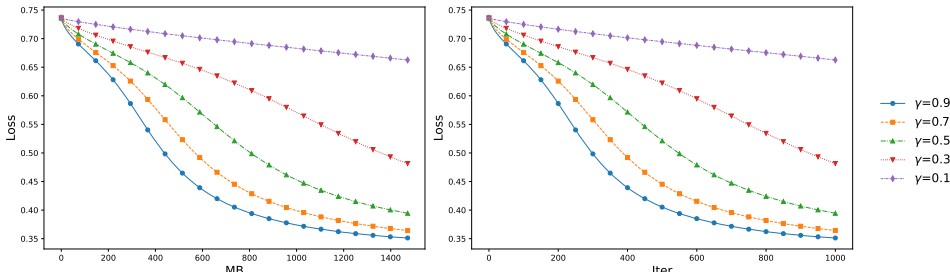

*Figure 7.* The upper-level loss function value on a9a dataset when using different values for $\gamma$.

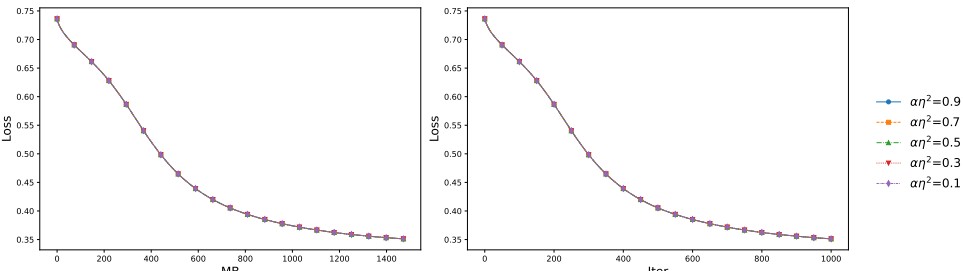

*Figure 8.* The upper-level loss function value on a9a dataset when using different values for $\alpha\eta^2$.

We also tune the hyperparameters $\rho$ and $\beta$ used for compression in Algorithm 1. In Figure 9, we run CE-DSMGD-VR on a9a dataset with $\rho \in \{0.1, 0.3, 0.5, 0.7, 0.9\}$, while keeping all other settings unchanged. In Figure 10, we run CE-DSMGD-VR on a9a dataset with $\beta \in \{0.1, 0.3, 0.5, 0.7, 0.9\}$, while keeping all other settings unchanged. From these two figures, we can conclude that these two hyperparameters do not have a significant influence on the convergence rate.

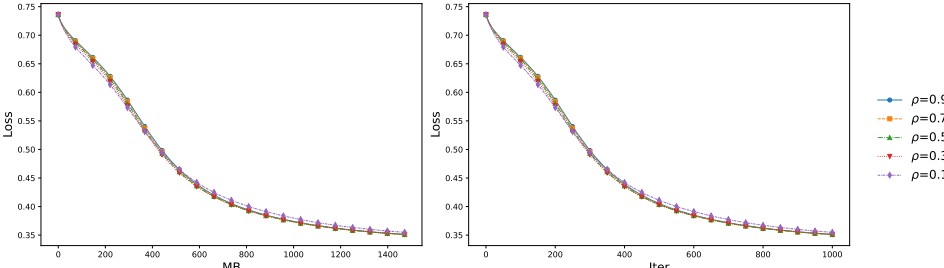

*Figure 9.* The upper-level loss function value on a9a dataset when using different values for $\rho$.

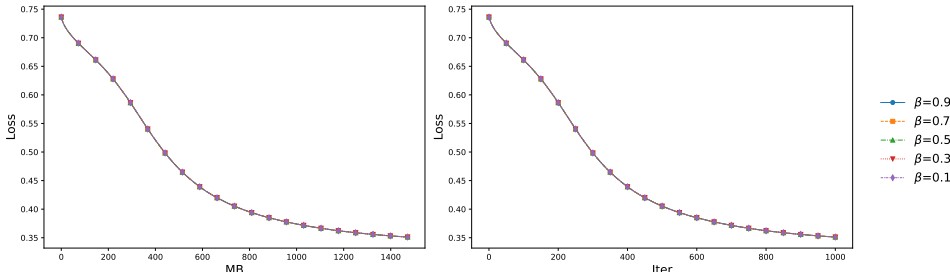

*Figure 10.* The upper-level loss function value on a9a dataset when using different values for $\beta$.

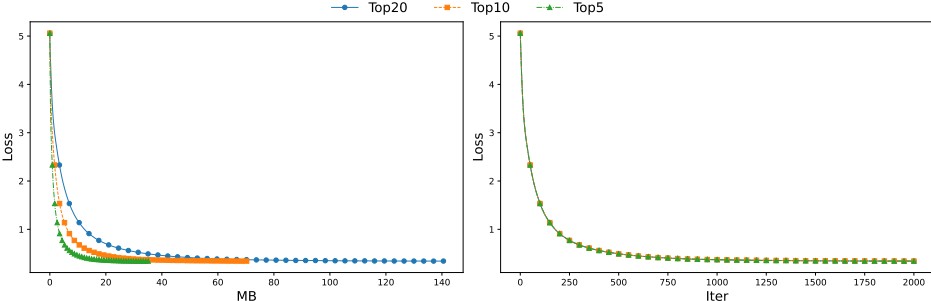

*Figure 11.* The upper-level loss function value with respect to the number of iterations and the communicated megabytes when using different compression ratios for a9a dataset.

To further demonstrate the performance of our algorithm, we use different compression ratios for a9a dataset, including Top-10% and Top-5%. As shown in the left subfigure of Figure 11, when using a large compression ratio, CE-DSMGD-VR consumes less communication costs. Meanwhile, as shown in the right subfigure of Figure 11, our algorithm can maintain its convergence performance even when using a large compression ratio. These observations confirm the robustness of CE-DSMGD-VR .

Moreover, we conducted an additional experiment to demonstrate the performance of our algorithm on different communication topologies. In particular, we use two additional graphs: a torus graph and random graph, where we use the Erdos-Renyi random graph with an edge probability being 0.4. From Figure 12, we can find that the convergence performance of CE-DSMGD-VR is robust to different communication topologies.

In addition, to further demonstrate the effectiveness of our algorithm, we conduct experiments with larger levels $K$, more workers, and heterogeneous data distributions. In particular, we perform an additional experiment with $K = 3$ using 16 workers. To simulate data heterogeneity, we partition the dataset such that each worker maintains a unique class distribution. Specifically, the local training and validation sets are sub-sampled to achieve target positive class ratios linearly spaced from 0.1 to 0.45 across workers. As shown in Figure 13, our algorithm consistently outperforms the baselines while achieving performance comparable to its full-precision counterpart.

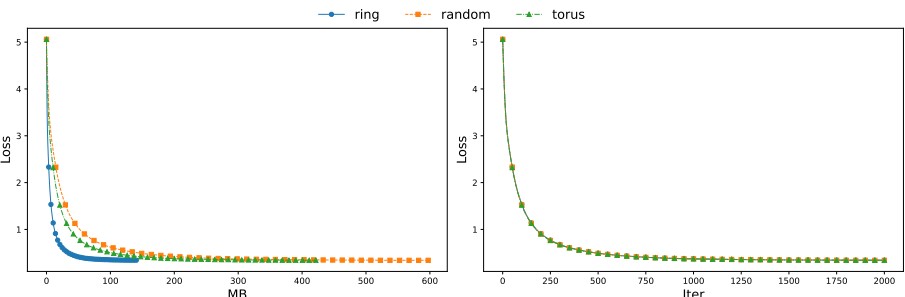

*Figure 12.* The upper-level loss function value with respect to the number of iterations and the communicated megabytes when using different communication topologies for a9a dataset.

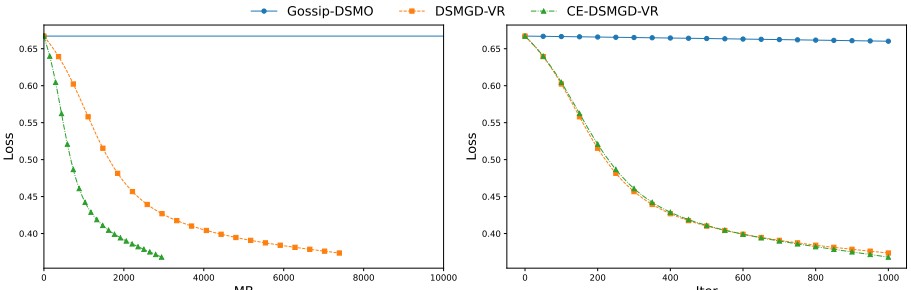

*Figure 13.* The upper-level loss function value with respect to the number of iterations and the communicated megabytes when $K = 3$ for a9a dataset.

## B. Proof Sketch and Unique Challenges of Theorem 5.1

**Step 1: Establish a potential function.** To establish the convergence rate of our Algorithm 1 in Theorem 5.1, we developed a novel potential function as follows:

$$
\begin{aligned}
\mathcal{L}_{t+1} = {} & \mathbb{E}[F_{K+1}(\bar{x}_{t+1})] \\
& + \sum_{k=1}^{K} a_k \mathbb{E}[\|\bar{y}_{k,t+1} - y_k^*(\bar{x}_{t+1})\|^2] + \sum_{k=1}^{K} b_k \mathbb{E}[\|\bar{z}_{k,t+1} - z_k^*(\bar{x}_{t+1})\|^2] \\
& + \sum_{k=1}^{K} c_k \mathbb{E}[\|(M_{k,t+1}^y - U_{k,t+1})\mathbf{1}\tfrac{1}{N}\|^2] + \sum_{k=1}^{K} d_k \mathbb{E}[\|(M_{k,t+1}^z - V_{k,t+1})\mathbf{1}\tfrac{1}{N}\|^2] + e\mathbb{E}[\|(M_{t+1}^x - G_{t+1})\mathbf{1}\tfrac{1}{N}\|^2] \\
& + \sum_{k=1}^{K} \tilde{c}_k \tfrac{1}{N}\mathbb{E}[\|M_{k,t+1}^y - U_{k,t+1}\|_F^2] + \sum_{k=1}^{K} \tilde{d}_k \tfrac{1}{N}\mathbb{E}[\|M_{k,t+1}^z - V_{k,t+1}\|_F^2] + \tilde{e}\tfrac{1}{N}\mathbb{E}[\|M_{t+1}^x - G_{t+1}\|_F^2] \\
& + \omega\tfrac{1}{N}\mathbb{E}[\|X_{t+1}(I - J)\|_F^2] + \sum_{k=1}^{K} \nu_k \tfrac{1}{N}\mathbb{E}[\|Y_{k,t+1}(I - J)\|_F^2] + \sum_{k=1}^{K} \tau_k \tfrac{1}{N}\mathbb{E}[\|Z_{k,t+1}(I - J)\|_F^2] \\
& + \tilde{\omega}\tfrac{1}{N}\mathbb{E}[\|P_{t+1}(I - J)\|_F^2] + \sum_{k=1}^{K} \tilde{\nu}_k \tfrac{1}{N}\mathbb{E}[\|Q_{k,t+1}(I - J)\|_F^2] + \sum_{k=1}^{K} \tilde{\tau}_k \tfrac{1}{N}\mathbb{E}[\|R_{k,t+1}(I - J)\|_F^2] \\
& + \theta\tfrac{1}{N}\mathbb{E}[\|X_{t+1} - \hat{X}_{t+1}\|_F^2] + \sum_{k=1}^{K} \phi_k \tfrac{1}{N}\mathbb{E}[\|Y_{k,t+1} - \hat{Y}_{k,t+1}\|_F^2] + \sum_{k=1}^{K} \psi_k \tfrac{1}{N}\mathbb{E}[\|Z_{k,t+1} - \hat{Z}_{k,t+1}\|_F^2] \\
& + \tilde{\theta}\tfrac{1}{N}\mathbb{E}[\|P_{t+1} - \hat{P}_{t+1}\|_F^2] + \sum_{k=1}^{K} \tilde{\phi}_k \tfrac{1}{N}\mathbb{E}[\|Q_{k,t+1} - \hat{Q}_{k,t+1}\|_F^2] + \sum_{k=1}^{K} \tilde{\psi}_k \tfrac{1}{N}\mathbb{E}[\|R_{k,t+1} - \hat{R}_{k,t+1}\|_F^2] .
\end{aligned}
$$

Other than the loss function $F_{K+1}(\bar{x}_{t+1})$, there are four classes of estimation errors in the potential function $\mathcal{L}_{t+1}$, which are marked with different colors. In particular, there are optimization errors with respect to the lower-level variables and the recursive Hessian-inverse-vector product, gradient estimation errors with respect to the gradient of hypergradient, lower-level gradients, and the gradient of the recursive Hessian-inverse-vector product, consensus errors of all variables and gradient estimators, and compression errors raised from the compression operation.

**Step 2: Establish the upper bound for each term in the potential function (Section D-F).** In this step, we establish the upper bound for each class of estimation errors. Specifically, we established the upper bound for the optimization errors in Lemma D.2 and Lemma D.3, for the gradient estimation errors in Lemmas E.1, E.2, E.3, E.4, E.5, E.6, for the

consensus errors  in Lemmas F.2, F.3, and for the  compression error  in Lemmas F.4, F.5.

Dependence across levels

Dependence across iterations

$$\mathbb{E}[\|\bar{z}_{k,t+1} - z_k^*(\bar{x}_{t+1})\|^2] \leq (1 - \frac{\gamma_z\eta\mu}{8})\mathbb{E}[\|\bar{z}_{k,t} - z_k^*(\bar{x}_t)\|^2] + \frac{15\gamma_z\eta C_J^2}{\mu}\mathbb{E}[\|\bar{z}_{k+1,t} - z_{k+1}^*(\bar{x}_t)\|^2]$$

$$+ \frac{15\gamma_z\eta}{\mu}L_J^2\Delta_{k+1}^2\mathbb{E}[\|\bar{y}_{k+1,t} - y_{k+1}^*(\bar{x}_t)\|^2] + \frac{15\gamma_z\eta}{\mu}(L_H^2\Delta_k^2 + L_J^2\Delta_{k+1}^2)\mathbb{E}[\|\bar{y}_{k,t} - y_k^*(\bar{x}_t)\|^2]$$

$$+ \frac{15\gamma_z\eta}{\mu}L_H^2\Delta_k^2\mathbb{E}[\|\bar{y}_{k-1,t} - y_k^*(\bar{x}_{t-1})\|^2] + 3\eta\gamma_z^2\frac{1}{N}\mathbb{E}[\|R_{k,t}(I-J)\|_F^2] + \frac{9\gamma_x^2\eta}{\gamma_z\mu}C_{z_k}^2\mathbb{E}[\|\bar{m}_t^x\|^2]$$

$$+ \frac{12\gamma_z\eta}{\mu}\mathbb{E}[\|(M_{k,t}^z - V_{k,t})\mathbf{1}\frac{1}{N}\|^2] + \frac{45\gamma_z\eta}{\mu}(\Delta_k^2 L_H^2 + \Delta_{k+1}^2 L_J^2)\frac{1}{N}\mathbb{E}[\|Y_{k,t}(I-J)\|_F^2]$$

$$+ \frac{45\gamma_z\eta}{\mu}\Delta_k^2 L_H^2\frac{1}{N}\mathbb{E}[\|Y_{k-1,t}(I-J)\|_F^2] + \frac{40\gamma_z\eta}{\mu}\Delta_{k+1}^2 L_J^2\frac{1}{N}\mathbb{E}[\|Y_{k+1,t}(I-J)\|_F^2]$$

$$+ (\frac{45\gamma_z\eta}{\mu}L_g^2 + 3)\frac{1}{N}\mathbb{E}[\|Z_{k,t}(I-J)\|_F^2] + \frac{45\gamma_z\eta}{\mu}C_J^2\frac{1}{N}\mathbb{E}[\|Z_{k+1,t}(I-J)\|_F^2] .$$

Dependence across workers

*Figure 14.* The triple dependence across levels, workers, iterations in the upper bound of the the optimization error $\mathbb{E}[\|\bar{z}_{k,t+1} - z_k^*(\bar{x}_{t+1})\|^2]$.

Due to the multi-level structure in the loss function, the resulting upper bounds **exhibit recursive dependencies across levels, workers, and iterations**. For example, the upper bound of the optimization error $\mathbb{E}[\|\bar{z}_{k,t+1} - z_k^*(\bar{x}_{t+1})\|^2]$ in Lemma D.3 involves a triple dependence across levels, workers, and iterations. This triple dependence incurs significant challenges for establishing the upper bound of $\mathcal{L}_{t+1} - \mathcal{L}_t$ in the subsequent analysis.

Moreover, the analysis reveals **a nontrivial interdependence between the  consensus error  and the  compression error** . Specifically, the bounds in Lemma F.2 and Lemma F.4 show that these two error terms depend on each other, which further increases the difficulty of selecting appropriate coefficients in the potential function.

$$\|X_{t+1}(I-J)\|_F^2 \leq (1 - \rho_x\eta(1-\lambda))\|X_t(I-J)\|_F^2 + 8\frac{\rho_x\eta}{(1-\lambda)}\|\hat{X}_t - X_t\|_F^2 + 2\frac{\gamma_x^2\eta}{\rho_x(1-\lambda)}\|P_t(I-J)\|_F^2 .$$

$$\|\hat{X}_{t+1} - X_{t+1}\|_F^2 \leq (1 - \frac{\delta}{4})\|\hat{X}_t - X_t\|_F^2 + \frac{12\gamma_x^2\eta^2}{\delta}\|P_t(I-J)\|_F^2 + \frac{12\gamma_x^2\eta^2}{\delta}N\|\bar{m}_t^x\|^2 + \frac{48\rho_x^2\eta^2}{\delta}\|X_t(I-J)\|_F^2 .$$

**Step 3: Establish the upper bound for $\mathcal{L}_{t+1} - \mathcal{L}_t$ by determining the coefficient of each term in the potential function. (Section G).** To establish the upper bound of $\mathcal{L}_{t+1} - \mathcal{L}_t$, we need to select the coefficients in the potential function, including $\{a_k\}_{k=1}^K$, $\{b_k\}_{k=1}^K$, $\{c_k\}_{k=1}^K$, $\{d_k\}_{k=1}^K$, $\{\tilde{c}_k\}_{k=1}^K$, $\{\tilde{d}_k\}_{k=1}^K$, $\{\nu_k\}_{k=1}^K$, $\{\tau_k\}_{k=1}^K$, $\{\tilde{\nu}_k\}_{k=1}^K$, $\{\tilde{\tau}_k\}_{k=1}^K$, $\{\phi_k\}_{k=1}^K$, $\{\tilde{\phi}_k\}_{k=1}^K$, $\{\psi_k\}_{k=1}^K$, $\{\tilde{\psi}_k\}_{k=1}^K$, $\{e, \tilde{e}, \omega, \tilde{\omega}, \theta, \tilde{\theta}\}$, such that **the coefficients (marked in blue) in Eqs. (35), (38), and analogous expressions, are non-positive**. As a result, we can remove the corresponding terms from the upper bound of $\mathcal{L}_{t+1} - \mathcal{L}_t$.

**The unique challenges in determining those coefficients lie in the recursive dependence in the multi-level optimization problem.** For example, when determining the coefficient $\{b_k\}_{k=1}^K$ for the optimization error $\mathbb{E}[\|\bar{z}_{k,t} - z_k^*(\bar{x}_t)\|^2]$, due to the recursive dependence, $\{b_k\}_{k=1}^K$ should satisfy the following recursive dependence:

$$b_{k-1} \leq b_k \frac{\mu^2}{256C_J^2} .$$

Meanwhile, they should satisfy some conditions, such as

$$b_k \geq \frac{3072\gamma_x\gamma_z}{\mu}\left(\frac{16C_J^2 L_g^2}{\alpha_x N} + \frac{320C_J^2 L_g^2}{\beta_z^2(1-\lambda)^2}\right)\frac{C_J^{2(k-1)}}{L_g^{2(k-1)}} .$$

**These constraints make it more challenging to determine the value of $\{b_k\}_{k=1}^K$ compared to the existing decentralized bilevel optimization algorithms.**

By incorporating various constraints, we finally obtain the following coefficient:

$$b_k = \frac{L_g^{2k}}{\mu^{2k}} \frac{(256C_J)^{2(k-1)}}{L_g^{2(k-1)}} \frac{3072\gamma_x\gamma_z}{\mu} \left( \frac{80C_J^2}{\gamma_z^2} + \frac{80C_J^2 L_g^2}{\alpha_x N} + \frac{320C_J^2 L_g^2}{\beta_z^2(1-\lambda)^2} + \frac{320C_J^4}{\beta_x^2(1-\lambda)^2} \right),$$

which actually also has a recursive structure with level $k$. For other coefficients, we follow a similar strategy to determine their values.

In addition, due to the **interdependence between the consensus error and compression error**, it is more challenging to determine their coefficients. For example, when determining the value of $\{\tau_k\}_{k=1}^K$ and $\{\psi_k\}_{k=1}^K$, they will depend on each other. Specifically, when setting the coefficient of $\mathbb{E}[\|\hat{Z}_{k,t} - Z_{k,t}\|_F^2]$ in Eq. (38) to be non-positive, we have

$$\tau_k \frac{8\rho_z\eta}{(1-\lambda)} + 16\rho_z^2\eta^2 \mathcal{Z}_k - \psi_k \frac{\delta}{4} \le 0 .$$

When setting the coefficient of $\mathbb{E}[\|Z_{k,t}(I-J)\|_F^2]$ in Eq. (38) to be non-positive, we have

$$\begin{aligned}
\mathcal{D}_{z,k} = {} & 3b_k + b_k \frac{48\gamma_z\eta}{\mu}L_g^2 + b_{k-1}\frac{48\gamma_z\eta}{\mu}C_J^2 + \psi_k\frac{48\rho_z^2\eta^2}{\delta} + \psi_k\frac{144\gamma_z^2\eta^2 L_g^2}{\delta} + \psi_{k-1}\frac{144\gamma_z^2\eta^2 C_J^2}{\delta} \\
& + 16\rho_z^2\eta^2\mathcal{Z}_k + 48\gamma_z^2\eta^2 L_g^2\mathcal{Z}_k + 48\gamma_z^2\eta^2 C_J^2\mathcal{Z}_{k-1} - \tau_k\rho_z\eta(1-\lambda) \le 0 .
\end{aligned} \tag{20}$$

From these two inequalities, we observe that $\{\tau_k\}_{k=1}^K$ and $\{\psi_k\}_{k=1}^K$ are mutually dependent, which complicates the determination of the coefficients in the potential function. To address the issue of the interdependence between different coefficients, we propose to use one class of coefficients, e.g., $\{\tau_k\}_{k=1}^K$, to replace the other class, e.g., $\{\psi_k = \tau_k\frac{32\rho_z\eta}{\delta(1-\lambda)} + \frac{64}{\delta}\rho_z^2\eta^2\mathcal{Z}_k\}_{k=1}^K$ and then solve inequalities regarding $\{\tau_k\}_{k=1}^K$ to determine their values. After that, we can obtain the values of $\{\psi_k\}_{k=1}^K$.

All in all, due to the multi-level structure, it is much more challenging to determine the values of the coefficients in the potential function than existing two-level methods. We successfully addressed these unique challenges by leveraging the recursive structure to determine these coefficients. **We believe that this novel design for coefficients can benefit more multi-level optimization algorithms.**

**Step 4: Obtain the convergence rate.** Based on Step 3, we obtain the following upper bound, which leads to the convergence rate of our algorithm.

$$\frac{1}{T}\sum_{t=0}^{T-1}\mathbb{E}[\|\nabla F_{K+1}(\bar{x}_t)\|^2] \le \frac{2(\mathcal{L}_0 - \mathcal{L}_T)}{\eta\gamma_x T} + \frac{2\mathcal{F}_0}{\eta\gamma_x T} . \tag{21}$$

where $\mathcal{F}_0$ is defined in Eq. (34).

## C. Properties of SMO Problems

**Lemma C.1.** *(Restatement of Lemma 8) Given Assumptions 3.1-3.4, for any $x \in \mathbb{R}^{d_0}$, the full hypergradient can be computed as:*

$$\nabla F_{K+1}(x) = \nabla_1 f_{K+1}(x, y_K^*(x)) + (-1)^K (\nabla_{12}^2 f_1(y_1^*(x), x))^T z_1^*(x) . \tag{22}$$

*Proof.* Based on the definition of $\frac{\partial y_k^*(x)}{\partial x}$ and $z_k^*(x)$, the hypergradient $\nabla F(x)$ can be represented as follows:

$$\begin{aligned}
\nabla F_{K+1}(x) = {} & \nabla_1 f_{K+1}(x, y_K^*(x)) + \left(\frac{\partial y_K^*(x)}{\partial x}\right)^T \nabla_2 f_{K+1}(x, y_K^*(x)) \\
= {} & \nabla_1 f_{K+1}(x, y_K^*(x)) - \left(\frac{\partial y_{K-1}^*(x)}{\partial x}\right)^T \left(\nabla_{12}^2 f_K(y_K^*(x), y_{K-1}^*(x))\right)^T \\
& \times \underbrace{[\nabla_{11}^2 f_K(y_K^*(x), y_{K-1}^*(x))]^{-1} \nabla_2 f_{K+1}(x, y_K^*(x))}_{z_K^*(x)}
\end{aligned}$$

$$= \nabla_1 f_{K+1}(x, y_K^*(x)) + \left(\frac{\partial y_{K-2}^*(x)}{\partial x}\right)^T \left(\nabla_{12}^2 f_{K-1}(y_{K-1}^*(x), y_{K-2}^*(x))\right)^T$$

$$\times \underbrace{[\nabla_{11}^2 f_{K-1}(y_{K-1}^*(x), y_{K-2}^*(x))]^{-1} \left(\nabla_{12}^2 f_K(y_K^*(x), y_{K-1}^*(x))\right)^T z_K^*(x)}_{z_{K-1}^*(x)}$$

$$= \cdots$$

$$= \nabla_1 f_{K+1}(x, y_K^*(x)) + (-1)^{K-k} \left(\frac{\partial y_k^*(x)}{\partial x}\right)^T \left(\nabla_{12}^2 f_{k+1}(y_{k+1}^*(x), y_k^*(x))\right)^T z_{k+1}^*(x)$$

$$= \cdots$$

$$= \nabla_1 f_{K+1}(x, y_K^*(x)) + (-1)^K \left(\nabla_{12}^2 f_1(y_1^*(x), x)\right)^T \underbrace{[\nabla_{11}^2 f_1(y_1^*(x), x)]^{-1} \left(\nabla_{12}^2 f_2(y_2^*(x), y_1^*(x))\right)^T z_2^*(x)}_{z_1^*(x)} .$$

$\square$

**Lemma C.2.** (***Continuity of*** $y_k^*(x)$) *Given Assumptions 3.1-3.4, for* $k \in \{1, \cdots, K\}$ *and any* $x \in \mathbb{R}^{d_0}$, $y_k^*(x)$ *is* $C_{y_k}$-*Lipschitz continuous, where* $C_{y_k} = \frac{C_J^k}{\mu^k}$.

*Proof.* According to the definition of $\frac{\partial y_k^*(x)}{\partial x}$, when $k = 1$, we have

$$\left\|\frac{\partial y_1^*(x)}{\partial x}\right\| = \left\|-(\nabla_{12}^2 f_1(y_1^*(x), x))^T [\nabla_{11}^2 f_1(y_1^*(x), x)]^{-1}\right\| \le \frac{C_J}{\mu} ,$$

where the last step follows from Assumption 3.3 and Assumption 3.1.

When $k > 1$, we use the induction approach to prove it. In particular, we assume $\|\frac{\partial y_{k-1}^*(x)}{\partial x}\| \le \frac{C_J^{k-1}}{\mu^{k-1}}$, then it is easy to know that

$$\left\|\frac{\partial y_k^*(x)}{\partial x}\right\| = \left\|-\frac{\partial y_{k-1}^*(x)}{\partial x}(\nabla_{12}^2 f_k(y_k^*(x), y_{k-1}^*(x)))^T [\nabla_{11}^2 f_k(y_k^*(x), y_{k-1}^*(x))]^{-1}\right\|$$

$$\le \left\|-\frac{\partial y_{k-1}^*(x)}{\partial x}\right\| \frac{C_J}{\mu} \le \frac{C_J^k}{\mu^k} ,$$

where the second step follows from Assumption 3.3 and Assumption 3.1. $\square$

**Lemma C.3.** (***Continuity of*** $z_k^*(x)$) *Given Assumptions 3.1-3.4, then for* $k \in \{1, 2, \cdots, K\}$ *and any* $x_1, x_2 \in \mathbb{R}^{d_0}$, *we have*

- $\|z_k^*(x_1)\| \le \Delta_k = \frac{C_J^{K-k+1}}{\mu^{K-k+1}}$ ,

- $\|z_k^*(x_1) - z_k^*(x_2)\| \le C_{z_k}\|x_1 - x_2\|$,

*where* $C_{z_K} = \frac{L_f}{\mu} + \frac{L_f}{\mu}\frac{C_J^K}{\mu^K} + \frac{C_f L_H}{\mu^2}\frac{C_J^K}{\mu^K} + \frac{C_f L_H}{\mu^2}\frac{C_J^{K-1}}{\mu^{K-1}}$ *and* $C_{z_k} = \frac{C_J}{\mu}\left(\frac{L_f}{\mu} + \frac{L_f}{\mu}\frac{C_J^K}{\mu^K} + \frac{C_f L_H}{\mu^2}\frac{C_J^K}{\mu^K} + \frac{C_f L_H}{\mu^2}\frac{C_J^{K-1}}{\mu^{K-1}}\right) +$ $\left(\sum_{j=0}^{K-k-1}\frac{C_J^j}{\mu^j}\right)(\frac{C_J^{K-1}}{\mu^{K-1}}\frac{C_J L_H}{\mu^2} + \frac{C_J^K}{\mu^K}\frac{L_J}{\mu})(\frac{C_J}{\mu} + 1)$ *for* $k \in \{1, 2, \cdots, K-1\}$.

*Proof.* When $k = K$, we have

$$\|z_K^*(x)\| = \|[\nabla_{11}^2 f_K(y_K^*(x), y_{K-1}^*(x))]^{-1}\nabla_2 f_{K+1}(x, y_K^*(x))\| \le \frac{C_f}{\mu} ,$$

where the second step follows from Assumption 3.2 and Assumption 3.1.

When $k \in \{1, \cdots, K-1\}$, we use the inductive approach to prove it. In particular, we assume $\|z_{k+1}^*(x)\| \le \frac{C_J^{K-k}}{\mu^{K-k}}$, then

$$\|z_k^*(x)\| = \|[\nabla_{11}^2 f_k(y_k^*(x), y_{k-1}^*(x))]^{-1}(\nabla_{12}^2 f_{k+1}(y_{k+1}^*(x), y_k^*(x)))^T z_{k+1}^*(x)\| \le \frac{C_J}{\mu}\|z_{k+1}^*(x)\| \le \frac{C_J^{K-k+1}}{\mu^{K-k+1}} ,$$

where the second step follows from Assumption 3.3 and Assumption 3.1.

When $k = K$, we have

$$
\begin{aligned}
\|z_K^*(x_1) - z_K^*(x_2)\| &= \|[\nabla_{11}^2 f_K(y_K^*(x_1), y_{K-1}^*(x_1))]^{-1} \nabla_2 f_{K+1}(x_1, y_K^*(x_1)) \\
&\quad - [\nabla_{11}^2 f_K(y_K^*(x_2), y_{K-1}^*(x_2))]^{-1} \nabla_2 f_{K+1}(x, y_K^*(x_2))\| \\
&\leq \|[\nabla_{11}^2 f_K(y_K^*(x_1), y_{K-1}^*(x_1))]^{-1} \nabla_2 f_{K+1}(x_1, y_K^*(x_1)) - [\nabla_{11}^2 f_K(y_K^*(x_1), y_{K-1}^*(x_1))]^{-1} \nabla_2 f_{K+1}(x_2, y_K^*(x_2))\| \\
&\quad + \|[\nabla_{11}^2 f_K(y_K^*(x_1), y_{K-1}^*(x_1))]^{-1} \nabla_2 f_{K+1}(x_2, y_K^*(x_2)) - [\nabla_{11}^2 f_K(y_K^*(x_2), y_{K-1}^*(x_2))]^{-1} \nabla_2 f_{K+1}(x, y_K^*(x_2))\| \\
&\leq \frac{1}{\mu} \|\nabla_2 f_{K+1}(x_1, y_K^*(x_1)) - \nabla_2 f_{K+1}(x_2, y_K^*(x_2))\| + \frac{C_f}{\mu^2} \|\nabla_{11}^2 f_K(y_K^*(x_2), y_{K-1}^*(x_2)) - \nabla_{11}^2 f_K(y_K^*(x_1), y_{K-1}^*(x_1))\| \\
&\leq \frac{L_f}{\mu} \|x_1 - x_2\| + \frac{L_f}{\mu} \|y_K^*(x_1) - y_K^*(x_2)\| + \frac{C_f L_H}{\mu^2} \|y_K^*(x_1) - y_K^*(x_2)\| + \frac{C_f L_H}{\mu^2} \|y_{K-1}^*(x_1) - y_{K-1}^*(x_2)\| \\
&\leq \left( \frac{L_f}{\mu} + \frac{L_f}{\mu} \frac{C_J^K}{\mu^K} + \frac{C_f L_H}{\mu^2} \frac{C_J^K}{\mu^K} + \frac{C_f L_H}{\mu^2} \frac{C_J^{K-1}}{\mu^{K-1}} \right) \|x_1 - x_2\|,
\end{aligned}
$$

where the third step follows from Assumption 3.2, Assumption 3.3 , Assumption 3.1, and last step follows Lemma C.2.

When $k \in \{1, \cdots, K-1\}$, we have

$$
\begin{aligned}
\|z_k^*(x_1) - z_k^*(x_2)\| &= \|[\nabla_{11}^2 f_k(y_k^*(x_1), y_{k-1}^*(x_1))]^{-1} \nabla_{12}^2 f_{k+1}(y_{k+1}^*(x_1), y_k^*(x_1))^T z_{k+1}^*(x_1) \\
&\quad - [\nabla_{11}^2 f_k(y_k^*(x_2), y_{k-1}^*(x_2))]^{-1} \nabla_{12}^2 f_{k+1}(y_{k+1}^*(x_2), y_k^*(x_2))^T z_{k+1}^*(x_2)\| \\
&\leq \underbrace{\|[\nabla_{11}^2 f_k(y_k^*(x_1), y_{k-1}^*(x_1))]^{-1} \nabla_{12}^2 f_{k+1}(y_{k+1}^*(x_1), y_k^*(x_1))^T z_{k+1}^*(x_1) \\
&\quad - [\nabla_{11}^2 f_k(y_k^*(x_2), y_{k-1}^*(x_2))]^{-1} \nabla_{12}^2 f_{k+1}(y_{k+1}^*(x_1), y_k^*(x_1))^T z_{k+1}^*(x_1)\|}_{T_1} \\
&\quad + \underbrace{\|[\nabla_{11}^2 f_k(y_k^*(x_2), y_{k-1}^*(x_2))]^{-1} \nabla_{12}^2 f_{k+1}(y_{k+1}^*(x_1), y_k^*(x_1))^T z_{k+1}^*(x_1) \\
&\quad - [\nabla_{11}^2 f_k(y_k^*(x_2), y_{k-1}^*(x_2))]^{-1} \nabla_{12}^2 f_{k+1}(y_{k+1}^*(x_2), y_k^*(x_2))^T z_{k+1}^*(x_1)\|}_{T_2} \\
&\quad + \underbrace{\|[\nabla_{11}^2 f_k(y_k^*(x_2), y_{k-1}^*(x_2))]^{-1} \nabla_{12}^2 f_{k+1}(y_{k+1}^*(x_2), y_k^*(x_2))^T z_{k+1}^*(x_1) \\
&\quad - [\nabla_{11}^2 f_k(y_k^*(x_2), y_{k-1}^*(x_2))]^{-1} \nabla_{12}^2 f_{k+1}(y_{k+1}^*(x_2), y_k^*(x_2))^T z_{k+1}^*(x_2)\|}_{T_3}.
\end{aligned}
$$

For $T_1$, we obtain

$$
\begin{aligned}
T_1 &\leq \frac{C_J^{K-k}}{\mu^{K-k}} \frac{C_J}{\mu^2} \|\nabla_{11}^2 f_k(y_k^*(x_1), y_{k-1}^*(x_1)) - \nabla_{11}^2 f_k(y_k^*(x_2), y_{k-1}^*(x_2))\| \\
&\leq \frac{C_J^{K-k}}{\mu^{K-k}} \frac{C_J L_H}{\mu^2} (\|y_k^*(x_1) - y_k^*(x_2)\| + \|y_{k-1}^*(x_1) - y_{k-1}^*(x_2)\|) \\
&\leq \frac{C_J^{K-k}}{\mu^{K-k}} \frac{C_J L_H}{\mu^2} \left( \frac{C_J^k}{\mu^k} \|x_1 - x_2\| + \frac{C_J^{k-1}}{\mu^{k-1}} \|x_1 - x_2\| \right),
\end{aligned}
$$

where the first and second steps follow from Assumption 3.3 and Assumption 3.1, and the last step follows Lemma C.2. Similarly, for $T_2$, we obtain

$$
\begin{aligned}
T_2 &\leq \frac{1}{\mu} \frac{C_J^{K-k}}{\mu^{K-k}} \|\nabla_{12}^2 f_{k+1}(y_{k+1}^*(x_1), y_k^*(x_1)) - \nabla_{12}^2 f_{k+1}(y_{k+1}^*(x_2), y_k^*(x_2))\| \\
&\leq \frac{C_J^{K-k}}{\mu^{K-k}} \frac{L_J}{\mu} (\|y_{k+1}^*(x_1) - y_{k+1}^*(x_2)\| + \|y_k^*(x_1) - y_k^*(x_2)\|) \\
&\leq \frac{C_J^{K-k}}{\mu^{K-k}} \frac{L_J}{\mu} \left( \frac{C_J^{k+1}}{\mu^{k+1}} \|x_1 - x_2\| + \frac{C_J^k}{\mu^k} \|x_1 - x_2\| \right).
\end{aligned}
$$

For $T_3$, we obtain $T_3 \leq \frac{C_J}{\mu} \|z_{k+1}^*(x_1) - z_{k+1}^*(x_2)\|$.

Plugging all these bounds into the inequality yields

$$
\|z_k^*(x_1) - z_k^*(x_2)\| \leq \frac{C_J}{\mu} \|z_{k+1}^*(x_1) - z_{k+1}^*(x_2)\| + \left( \frac{C_J^{K-1}}{\mu^{K-1}} \frac{C_J L_H}{\mu^2} + \frac{C_J^K}{\mu^K} \frac{L_J}{\mu} \right) \left( \frac{C_J}{\mu} + 1 \right) \|x_1 - x_2\|.
$$

Recursively applying above inequality, we obtain

$$\|z_k^*(x_1) - z_k^*(x_2)\| \leq \frac{C_J}{\mu}\|z_K^*(x_1) - z_K^*(x_2)\| + \Big(\sum_{j=0}^{K-k-1}\frac{C_J^j}{\mu^j}\Big)\Big(\frac{C_J^{K-1}}{\mu^{K-1}}\frac{C_J L_H}{\mu^2} + \frac{C_J^K}{\mu^K}\frac{L_J}{\mu}\Big)\Big(\frac{C_J}{\mu} + 1\Big)\|x_1 - x_2\|$$

$$\leq \Bigg(\frac{C_J}{\mu}\Big(\frac{L_f}{\mu} + \frac{L_f}{\mu}\frac{C_J^K}{\mu^K} + \frac{C_f L_H}{\mu^2}\frac{C_J^K}{\mu^K} + \frac{C_f L_H}{\mu^2}\frac{C_J^{K-1}}{\mu^{K-1}}\Big) + \Big(\sum_{j=0}^{K-k-1}\frac{C_J^j}{\mu^j}\Big)\Big(\frac{C_J^{K-1}}{\mu^{K-1}}\frac{C_J L_H}{\mu^2} + \frac{C_J^K}{\mu^K}\frac{L_J}{\mu}\Big)\Big(\frac{C_J}{\mu} + 1\Big)\Bigg)\|x_1 - x_2\| .$$

$\square$

**Lemma C.4.** *(**Smoothness of Hypergradient**) Given Assumptions 3.1-3.4, then $\nabla F_{K+1}(x)$ is $L_F$-Lipschitz continuous, where $L_F = L_f(1 + C_{y_K}) + \Delta_1 L_J(1 + C_{y_1}) + C_J C_{z_1}$.*

*Proof.* For any $x_1, x_2 \in \mathbb{R}^{d_0}$, we have

$$\|\nabla F_{K+1}(x_1) - \nabla F_{K+1}(x_2)\|$$
$$= \|\nabla_1 f_{K+1}(x_1, y_K^*(x_1)) + (-1)^K \nabla_{12}^2 f_1(y_1^*(x_1), x_1)^T z_1^*(x_1) - \nabla_1 f_{K+1}(x_2, y_K^*(x_2)) - (-1)^K \nabla_{12}^2 f_1(y_1^*(x_2), x_2)^T z_1^*(x_2)\|$$
$$\leq \|\nabla_1 f_{K+1}(x_1, y_K^*(x_1)) - \nabla_1 f_{K+1}(x_2, y_K^*(x_2))\| + \|\nabla_{12}^2 f_1(y_1^*(x_1), x_1)^T z_1^*(x_1) - \nabla_{12}^2 f_1(y_1^*(x_2), x_2)^T z_1^*(x_1)\|$$
$$\quad + \|\nabla_{12}^2 f_1(y_1^*(x_2), x_2)^T z_1^*(x_1) - \nabla_{12}^2 f_1(y_1^*(x_2), x_2)^T z_1^*(x_2)\|$$
$$\leq L_f(\|x_1 - x_2\| + \|y_K^*(x_1) - y_K^*(x_2)\|) + \Delta_1 L_J(\|x_1 - x_2\| + \|y_1^*(x_1) - y_1^*(x_2)\|) + C_J\|z_1^*(x_1) - z_1^*(x_2)\|$$
$$\leq L_f(1 + C_{y_K})\|x_1 - x_2\| + \Delta_1 L_J(1 + C_{y_1})\|x_1 - x_2\| + C_J C_{z_1}\|x_1 - x_2\|$$
$$= (L_f(1 + C_{y_K}) + \Delta_1 L_J(1 + C_{y_1}) + C_J C_{z_1})\|x_1 - x_2\| ,$$

where the third step follows from Assumptions 3.1, 3.3, the second to last step follows from Lemmas C.2, C.3. $\square$

**Lemma C.5.** *Assuming that Assumptions 3.1 through 3.5 hold, for any $t \geq 0$, the following inequality is true:*

$$\frac{1}{N}\mathbb{E}[\|\tilde{G}_{t,t} - G_t\|_F^2] \leq \hat{\sigma}_F^2 \triangleq (2 + 2\Delta_1^2)\sigma^2 .$$

*Proof.* From the definition of $G_t$, we derive

$$\frac{1}{N}\mathbb{E}[\|\tilde{G}_{t,t} - G_t\|_F^2] = \frac{1}{N}\sum_{n=1}^N \mathbb{E}[\|\nabla_1 f_{K+1}^{(n)}(x_t^{(n)}, y_{K,t}^{(n)}; \xi_t^{(n)}) + (-1)^K \nabla_{12}^2 f_1^{(n)}(y_{1,t}^{(n)}, x_t^{(n)}; \zeta_{1,t}^{(n)}) z_{1,t}^{(n)}$$
$$- \nabla_1 f_{K+1}^{(n)}(x_t^{(n)}, y_{K,t}^{(n)}) - (-1)^K \nabla_{12}^2 f_1^{(n)}(y_{1,t}^{(n)}, x_t^{(n)}) z_{1,t}^{(n)}\|^2]$$
$$\leq 2\frac{1}{N}\sum_{n=1}^N \mathbb{E}[\|\nabla_1 f_{K+1}^{(n)}(x_t^{(n)}, y_{K,t}^{(n)}; \xi_t^{(n)}) - \nabla_1 f_{K+1}^{(n)}(x_t^{(n)}, y_{K,t}^{(n)})\|^2]$$
$$+ 2\frac{1}{N}\sum_{n=1}^N \mathbb{E}[\|\nabla_{12}^2 f_1^{(n)}(y_{1,t}^{(n)}, x_t^{(n)}; \zeta_{1,t}^{(n)}) z_{1,t}^{(n)} - \nabla_{12}^2 f_1^{(n)}(y_{1,t}^{(n)}, x_t^{(n)}) z_{1,t}^{(n)}\|^2]$$
$$\leq (2 + 2\Delta_1^2)\sigma^2 .$$

$\square$

**Lemma C.6.** *Assuming that Assumptions 3.1 through 3.5 hold, when $k \in \{1, \cdots, K-1\}$, the following inequality is true:*

$$\frac{1}{N}\mathbb{E}[\|V_{k,t} - \tilde{V}_{k,t,t}\|_F^2] \leq 2\Delta_k^2\sigma^2 + 2\Delta_{k+1}^2\sigma^2 .$$

*When $k = K$, the following inequality is true:*

$$\frac{1}{N}\mathbb{E}[\|V_{K,t} - \tilde{V}_{K,t,t}\|_F^2] \leq 2\Delta_K^2\sigma^2 + 2\sigma^2 .$$

*Proof.* When $k \in \{1, \cdots, K-1\}$, we derive

$$\frac{1}{N}\mathbb{E}[\|V_{k,t} - \tilde{V}_{k,t,t}\|_F^2] = \frac{1}{N}\sum_{n=1}^N \mathbb{E}[\|\nabla_{11}^2 f_k^{(n)}(y_{k,t}^{(n)}, y_{k-1,t}^{(n)}) z_{k,t}^{(n)} - \nabla_{12}^2 f_{k+1}^{(n)}(y_{k+1,t}^{(n)}, y_{k,t}^{(n)}) z_{k+1,t}^{(n)}$$

$$- \nabla_{11}^2 f_k^{(n)}(y_{k,t}^{(n)}, y_{k-1,t}^{(n)}; \zeta_{k,t}^{(n)}) z_{k,t}^{(n)} + \nabla_{12}^2 f_{k+1}^{(n)}(y_{k+1,t}^{(n)}, y_{k,t}^{(n)}; \zeta_{k+1,t}^{(n)}) z_{k+1,t}^{(n)} \|^2]$$

$$\leq 2 \frac{1}{N} \sum_{n=1}^N \mathbb{E}[\| \nabla_{11}^2 f_k^{(n)}(y_{k,t}^{(n)}, y_{k-1,t}^{(n)}) z_{k,t}^{(n)} - \nabla_{11}^2 f_k^{(n)}(y_{k,t}^{(n)}, y_{k-1,t}^{(n)}; \zeta_{k,t}^{(n)}) z_{k,t}^{(n)} \|^2]$$

$$+ 2 \frac{1}{N} \sum_{n=1}^N \mathbb{E}[\| - \nabla_{12}^2 f_{k+1}^{(n)}(y_{k+1,t}^{(n)}, y_{k,t}^{(n)}) z_{k+1,t}^{(n)} + \nabla_{12}^2 f_{k+1}^{(n)}(y_{k+1,t}^{(n)}, y_{k,t}^{(n)}; \zeta_{k+1,t}^{(n)}) z_{k+1,t}^{(n)} \|^2]$$

$$\leq 2\Delta_k^2 \sigma^2 + 2\Delta_{k+1}^2 \sigma^2 .$$

When $k = K$, we derive

$$\frac{1}{N}\mathbb{E}[\|V_{K,t} - \tilde{V}_{K,t,t}\|_F^2] = \frac{1}{N} \sum_{n=1}^N \mathbb{E}[\| \nabla_{11}^2 f_K^{(n)}(y_{K,t}^{(n)}, y_{K-1,t}^{(n)}) z_{K,t}^{(n)} - \nabla_2 f_{K+1}^{(n)}(x_t^{(n)}, y_{K,t}^{(n)})$$

$$- \nabla_{11}^2 f_K^{(n)}(y_{K,t}^{(n)}, y_{K-1,t}^{(n)}; \zeta_{K,t+1}^{(n)}) z_{K,t}^{(n)} + \nabla_2 f_{K+1}^{(n)}(x_t^{(n)}, y_{K,t}^{(n)}; \xi_{t+1}^{(n)}) \|^2]$$

$$\leq 2 \frac{1}{N} \sum_{n=1}^N \mathbb{E}[\| \nabla_{11}^2 f_K^{(n)}(y_{K,t}^{(n)}, y_{K-1,t}^{(n)}) z_{K,t}^{(n)} - \nabla_{11}^2 f_K^{(n)}(y_{K,t}^{(n)}, y_{K-1,t}^{(n)}; \zeta_{K,t+1}^{(n)}) z_{K,t}^{(n)} \|^2]$$

$$+ 2 \frac{1}{N} \sum_{n=1}^N \mathbb{E}[\| - \nabla_2 f_{K+1}^{(n)}(x_t^{(n)}, y_{K,t}^{(n)}) + \nabla_2 f_{K+1}^{(n)}(x_t^{(n)}, y_{K,t}^{(n)}; \xi_{t+1}^{(n)}) \|^2]$$

$$\leq 2\Delta_K^2 \sigma^2 + 2\sigma^2 .$$

$\square$

**Lemma C.7.** *Assuming that Assumptions 3.1 through 3.5 hold, for any $t \geq 0$, the following inequality is true:*

$$\frac{1}{N}\mathbb{E}[\|\tilde{G}_{t+1,t+1} - \tilde{G}_{t,t+1}\|_F^2] \leq 3(L_f^2 + \Delta_1^2 L_J^2) \frac{1}{N}\mathbb{E}[\|X_t - X_{t+1}\|_F^2] + 3C_J^2 \frac{1}{N}\mathbb{E}[\|Z_{1,t} - Z_{1,t+1}\|_F^2]$$

$$+ 3\Delta_1^2 L_J^2 \frac{1}{N}\mathbb{E}[\|Y_{1,t} - Y_{1,t+1}\|_F^2] + 3L_f^2 \frac{1}{N}\mathbb{E}[\|Y_{K,t} - Y_{K,t+1}\|_F^2] .$$

*Proof.* From the definition of $\tilde{G}_{t+1,t+1}$, we derive

$$\frac{1}{N}\mathbb{E}[\|\tilde{G}_{t+1,t+1} - \tilde{G}_{t,t+1}\|_F^2] = \frac{1}{N} \sum_{n=1}^N \mathbb{E}[\| \nabla_1 f_{K+1}^{(n)}(x_t^{(n)}, y_{K,t}^{(n)}; \xi_{t+1}^{(n)}) + (-1)^K \nabla_{12}^2 f_1^{(n)}(y_{1,t}^{(n)}, x_t^{(n)}; \zeta_{1,t+1}^{(n)}) z_{1,t}^{(n)}$$

$$- \nabla_1 f_{K+1}^{(n)}(x_{t+1}^{(n)}, y_{K,t+1}^{(n)}; \xi_{t+1}^{(n)}) - (-1)^K \nabla_{12}^2 f_1^{(n)}(y_{1,t+1}^{(n)}, x_{t+1}^{(n)}; \zeta_{1,t+1}^{(n)}) z_{1,t+1}^{(n)} \|^2]$$

$$\leq 3 \frac{1}{N} \sum_{n=1}^N \mathbb{E}[\| \nabla_1 f_{K+1}^{(n)}(x_t^{(n)}, y_{K,t}^{(n)}; \xi_{t+1}^{(n)}) - \nabla_1 f_{K+1}^{(n)}(x_{t+1}^{(n)}, y_{K,t+1}^{(n)}; \xi_{t+1}^{(n)}) \|^2]$$

$$+ 3 \frac{1}{N} \sum_{n=1}^N \mathbb{E}[\| \nabla_{12}^2 f_1^{(n)}(y_{1,t}^{(n)}, x_t^{(n)}; \zeta_{1,t+1}^{(n)}) z_{1,t}^{(n)} - \nabla_{12}^2 f_1^{(n)}(y_{1,t}^{(n)}, x_t^{(n)}; \zeta_{1,t+1}^{(n)}) z_{1,t+1}^{(n)} \|^2]$$

$$+ 3 \frac{1}{N} \sum_{n=1}^N \mathbb{E}[\| \nabla_{12}^2 f_1^{(n)}(y_{1,t}^{(n)}, x_t^{(n)}; \zeta_{1,t+1}^{(n)}) z_{1,t+1}^{(n)} - \nabla_{12}^2 f_1^{(n)}(y_{1,t+1}^{(n)}, x_{t+1}^{(n)}; \zeta_{1,t+1}^{(n)}) z_{1,t+1}^{(n)} \|^2]$$

$$\leq 3L_f^2 \frac{1}{N}\mathbb{E}[\|X_t - X_{t+1}\|_F^2] + 3L_f^2 \frac{1}{N}\mathbb{E}[\|Y_{K,t} - Y_{K,t+1}\|_F^2] + 3C_J^2 \frac{1}{N}\mathbb{E}[\|Z_{1,t} - Z_{1,t+1}\|_F^2]$$

$$+ 3\Delta_1^2 L_J^2 \frac{1}{N}\mathbb{E}[\|Y_{1,t} - Y_{1,t+1}\|_F^2] + 3\Delta_1^2 L_J^2 \frac{1}{N}\mathbb{E}[\|X_t - X_{t+1}\|_F^2] .$$

$\square$

**Lemma C.8.** *Assuming that Assumptions 3.1 through 3.5 hold, when $k \in \{1, \cdots, K\}$, the following inequality is true:*

$$\frac{1}{N}\mathbb{E}[\|\tilde{U}_{k,t+1,t+1} - \tilde{U}_{k,t,t+1}\|_F^2] \leq L_g^2 \frac{1}{N}\mathbb{E}[\|Y_{k,t} - Y_{k,t+1}\|_F^2] + L_g^2 \frac{1}{N}\mathbb{E}[\|Y_{k-1,t} - Y_{k-1,t+1}\|_F^2] .$$

*Note that $Y_{0,t} = X_t$.*

*Proof.* Similarly, we derive

$$\frac{1}{N}\mathbb{E}[\|\tilde{U}_{k,t+1,t+1} - \tilde{U}_{k,t,t+1}\|_F^2] = \frac{1}{N}\sum_{n=1}^{N}\mathbb{E}[\|\nabla_1 f_k^{(n)}(y_{k,t+1}^{(n)}, y_{k-1,t+1}^{(n)}; \zeta_{k,t+1}^{(n)}) - \nabla_1 f_k^{(n)}(y_{k,t}^{(n)}, y_{k-1,t}^{(n)}; \zeta_{k,t+1}^{(n)})\|^2]$$

$$\leq L_g^2 \frac{1}{N}\mathbb{E}[\|Y_{k,t} - Y_{k,t+1}\|_F^2] + L_g^2 \frac{1}{N}\mathbb{E}[\|Y_{k-1,t} - Y_{k-1,t+1}\|_F^2].$$

$\square$

**Lemma C.9.** *Assuming that Assumptions 3.1 through 3.5 hold, when $k \in \{1, \cdots, K-1\}$, the following inequality is true:*

$$\frac{1}{N}\mathbb{E}[\|\tilde{V}_{k,t+1,t+1} - \tilde{V}_{k,t,t+1}\|_F^2] \leq 4L_g^2 \frac{1}{N}\mathbb{E}[\|Z_{k,t+1} - Z_{k,t}\|_F^2]$$

$$+ 4C_J^2 \frac{1}{N}\mathbb{E}[\|Z_{k+1,t+1} - Z_{k+1,t}\|_F^2] + 4\Delta_k^2 L_H^2 \frac{1}{N}\mathbb{E}[\|Y_{k-1,t+1} - Y_{k-1,t}\|_F^2]$$

$$+ 4(\Delta_k^2 L_H^2 + \Delta_{k+1}^2 L_J^2)\frac{1}{N}\mathbb{E}[\|Y_{k,t+1} - Y_{k,t}\|_F^2] + 4\Delta_{k+1}^2 L_J^2 \frac{1}{N}\mathbb{E}[\|Y_{k+1,t+1} - Y_{k+1,t}\|_F^2].$$

*Note that $Y_{0,t} = X_t$. When $k = K$, the following inequality is true:*

$$\frac{1}{N}\mathbb{E}[\|\tilde{V}_{K,t+1,t+1} - \tilde{V}_{K,t,t+1}\|_F^2] \leq 3L_g^2 \frac{1}{N}\mathbb{E}[\|Z_{K,t} - Z_{K,t+1}\|^2] + 3L_f^2 \frac{1}{N}\mathbb{E}[\|X_{t+1} - X_t\|^2]$$

$$+ 3\Delta_K^2 L_H^2 \frac{1}{N}\mathbb{E}[\|Y_{K-1,t} - Y_{K-1,t+1}\|^2] + 3(\Delta_K^2 L_H^2 + L_f^2)\frac{1}{N}\mathbb{E}[\|Y_{K,t} - Y_{K,t+1}\|^2].$$

*Proof.* When $k \in \{1, \cdots, K-1\}$, we derive

$$\frac{1}{N}\mathbb{E}[\|\tilde{V}_{k,t+1,t+1} - \tilde{V}_{k,t,t+1}\|_F^2]$$

$$= \frac{1}{N}\sum_{n=1}^{N}\mathbb{E}[\|\nabla_{11}^2 f_k^{(n)}(y_{k,t+1}^{(n)}, y_{k-1,t+1}^{(n)}; \zeta_{k,t+1}^{(n)})z_{k,t+1}^{(n)} - \nabla_{12}^2 f_{k+1}^{(n)}(y_{k+1,t+1}^{(n)}, y_{k,t+1}^{(n)}; \zeta_{k+1,t+1}^{(n)})z_{k+1,t+1}^{(n)}$$

$$- \nabla_{11}^2 f_k^{(n)}(y_{k,t}^{(n)}, y_{k-1,t}^{(n)}; \zeta_{k,t+1}^{(n)})z_{k,t}^{(n)} + \nabla_{12}^2 f_{k+1}^{(n)}(y_{k+1,t}^{(n)}, y_{k,t}^{(n)}; \zeta_{k+1,t+1}^{(n)})z_{k+1,t}^{(n)}\|^2]$$

$$\leq 4\frac{1}{N}\sum_{n=1}^{N}\mathbb{E}[\|\nabla_{11}^2 f_k^{(n)}(y_{k,t+1}^{(n)}, y_{k-1,t+1}^{(n)}; \zeta_{k,t+1}^{(n)})z_{k,t+1}^{(n)} - \nabla_{11}^2 f_k^{(n)}(y_{k,t+1}^{(n)}, y_{k-1,t+1}^{(n)}; \zeta_{k,t+1}^{(n)})z_{k,t}^{(n)}\|^2]$$

$$+ 4\frac{1}{N}\sum_{n=1}^{N}\mathbb{E}[\|\nabla_{11}^2 f_k^{(n)}(y_{k,t+1}^{(n)}, y_{k-1,t+1}^{(n)}; \zeta_{k,t+1}^{(n)})z_{k,t}^{(n)} - \nabla_{11}^2 f_k^{(n)}(y_{k,t}^{(n)}, y_{k-1,t}^{(n)}; \zeta_{k,t+1}^{(n)})z_{k,t}^{(n)}\|^2]$$

$$+ 4\frac{1}{N}\sum_{n=1}^{N}\mathbb{E}[\| - \nabla_{12}^2 f_{k+1}^{(n)}(y_{k+1,t+1}^{(n)}, y_{k,t+1}^{(n)}; \zeta_{k+1,t+1}^{(n)})z_{k+1,t+1}^{(n)} + \nabla_{12}^2 f_{k+1}^{(n)}(y_{k+1,t+1}^{(n)}, y_{k,t+1}^{(n)}; \zeta_{k+1,t+1}^{(n)})z_{k+1,t}^{(n)}\|^2]$$

$$+ 4\frac{1}{N}\sum_{n=1}^{N}\mathbb{E}[\| - \nabla_{12}^2 f_{k+1}^{(n)}(y_{k+1,t+1}^{(n)}, y_{k,t+1}^{(n)}; \zeta_{k+1,t+1}^{(n)})z_{k+1,t}^{(n)} + \nabla_{12}^2 f_{k+1}^{(n)}(y_{k+1,t}^{(n)}, y_{k,t}^{(n)}; \zeta_{k+1,t+1}^{(n)})z_{k+1,t}^{(n)}\|^2]$$

$$\leq 4L_g^2 \frac{1}{N}\mathbb{E}[\|Z_{k,t+1} - Z_{k,t}\|_F^2] + 4C_J^2 \frac{1}{N}\mathbb{E}[\|Z_{k+1,t+1} - Z_{k+1,t}\|_F^2] + 4\Delta_k^2 L_H^2 \frac{1}{N}\mathbb{E}[\|Y_{k-1,t+1} - Y_{k-1,t}\|_F^2]$$

$$+ 4(\Delta_k^2 L_H^2 + \Delta_{k+1}^2 L_J^2)\frac{1}{N}\mathbb{E}[\|Y_{k,t+1} - Y_{k,t}\|_F^2] + 4\Delta_{k+1}^2 L_J^2 \frac{1}{N}\mathbb{E}[\|Y_{k+1,t+1} - Y_{k+1,t}\|_F^2].$$

When $k = K$, we derive

$$\frac{1}{N}\mathbb{E}[\|\tilde{V}_{K,t+1,t+1} - \tilde{V}_{K,t,t+1}\|_F^2] = \frac{1}{N}\sum_{n=1}^{N}\mathbb{E}[\|\nabla_{11}^2 f_K^{(n)}(y_{K,t}^{(n)}, y_{K-1,t}^{(n)}; \zeta_{K,t+1}^{(n)})z_{K,t}^{(n)} - \nabla_2 f_{K+1}^{(n)}(x_t^{(n)}, y_{K,t}^{(n)}; \xi_{t+1}^{(n)})$$

$$- \nabla_{11}^2 f_K^{(n)}(y_{K,t+1}^{(n)}, y_{K-1,t+1}^{(n)}; \zeta_{K,t+1}^{(n)})z_{K,t+1}^{(n)} + \nabla_2 f_{K+1}^{(n)}(x_{t+1}^{(n)}, y_{K,t+1}^{(n)}; \xi_{t+1}^{(n)})\|^2]$$

$$\leq 3\frac{1}{N}\sum_{n=1}^{N}\mathbb{E}[\|\nabla_{11}^2 f_K^{(n)}(y_{K,t}^{(n)}, y_{K-1,t}^{(n)}; \zeta_{K,t+1}^{(n)})z_{K,t}^{(n)} - \nabla_{11}^2 f_K^{(n)}(y_{K,t}^{(n)}, y_{K-1,t}^{(n)}; \zeta_{K,t+1}^{(n)})z_{K,t+1}^{(n)}\|^2]$$

$$+ 3\frac{1}{N}\sum_{n=1}^{N}\mathbb{E}[\|\nabla_{11}^2 f_K^{(n)}(y_{K,t}^{(n)}, y_{K-1,t}^{(n)}; \zeta_{K,t+1}^{(n)})z_{K,t+1}^{(n)} - \nabla_{11}^2 f_K^{(n)}(y_{K,t+1}^{(n)}, y_{K-1,t+1}^{(n)}; \zeta_{K,t+1}^{(n)})z_{K,t+1}^{(n)}\|^2]$$

$$+ 3\frac{1}{N}\sum_{n=1}^{N}\mathbb{E}[\| -\nabla_2 f_{K+1}^{(n)}(x_t, y_{K,t}; \xi_{t+1}) + \nabla_2 f_{K+1}^{(n)}(x_{t+1}, y_{K,t+1}; \xi_{t+1})\|^2]$$

$$\leq 3L_g^2 \frac{1}{N}\mathbb{E}[\|Z_{K,t} - Z_{K,t+1}\|^2] + 3L_f^2 \frac{1}{N}\mathbb{E}[\|X_{t+1} - X_t\|^2] + 3\Delta_K^2 L_H^2 \frac{1}{N}\mathbb{E}[\|Y_{K-1,t} - Y_{K-1,t+1}\|^2]$$

$$+ 3(\Delta_K^2 L_H^2 + L_f^2)\frac{1}{N}\mathbb{E}[\|Y_{K,t} - Y_{K,t+1}\|^2].$$

□

# D. Key Descent and Optimization Errors

**Lemma D.1.** *Assuming that Assumptions 3.1 through 3.5 hold, and $\eta \leq \frac{1}{2\gamma_x L_F}$, the following inequality is true:*

$$\mathbb{E}[F_{K+1}(\bar{x}_{t+1})] \leq \mathbb{E}[F_{K+1}(\bar{x}_t)] - \frac{\eta\gamma_x}{2}\mathbb{E}[\|\nabla F(\bar{x}_t)\|^2] - \frac{\eta\gamma_x}{4}\mathbb{E}[\|\bar{m}_t^x\|^2] + \eta\gamma_x\mathbb{E}[\|(G_t - M_t^x)\mathbf{1}\frac{1}{N}\|^2]$$

$$+ 6\eta\gamma_x\Delta_1^2 L_J^2 \mathbb{E}[\|y_1^*(\bar{x}_t) - \bar{y}_{1,t}\|^2] + 6\eta\gamma_x L_f^2 \mathbb{E}[\|y_K^*(\bar{x}_t) - \bar{y}_{K,t}\|^2] + 6\eta\gamma_x C_J^2 \mathbb{E}[\|z_1^*(\bar{x}_t) - \bar{z}_{1,t}\|^2]$$

$$+ 6\eta\gamma_x(L_f^2 + \Delta_1^2 L_J^2)\frac{1}{N}\mathbb{E}[\|X_t(I - J)\|_F^2] + 6\eta\gamma_x\Delta_1^2 L_J^2 \frac{1}{N}\mathbb{E}[\|Y_{1,t}(I - J)\|_F^2]$$

$$+ 6\eta\gamma_x L_f^2 \frac{1}{N}\mathbb{E}[\|Y_{K,t}(I - J)\|_F^2] + 6\eta\gamma_x C_J^2 \frac{1}{N}\mathbb{E}[\|Z_{1,t}(I - J)\|_F^2].$$

*Proof.* Since $F(x)$ is $L_F$-smooth, we derive

$$\mathbb{E}[F_{K+1}(\bar{x}_{t+1})] \leq \mathbb{E}[F_{K+1}(\bar{x}_t)] + \mathbb{E}[\langle \nabla F_{K+1}(\bar{x}_t), \bar{x}_{t+1} - \bar{x}_t\rangle] + \frac{L_F}{2}\mathbb{E}[\|\bar{x}_{t+1} - \bar{x}_t\|^2]$$

$$= \mathbb{E}[F_{K+1}(\bar{x}_t)] - \eta\gamma_x\mathbb{E}[\langle \nabla F_{K+1}(\bar{x}_t), \bar{m}_t^x\rangle] + \frac{\eta^2\gamma_x^2 L_F}{2}\mathbb{E}[\|\bar{m}_t^x\|^2]$$

$$= \mathbb{E}[F_{K+1}(\bar{x}_t)] - \frac{\eta\gamma_x}{2}\mathbb{E}[\|\nabla F_{K+1}(\bar{x}_t)\|^2] - \frac{\eta\gamma_x}{2}\mathbb{E}[\|\bar{m}_t^x\|^2] + \frac{\eta\gamma_x}{2}\mathbb{E}[\|\nabla F_{K+1}(\bar{x}_t) - \bar{m}_t^x\|^2] + \frac{\eta^2\gamma_x^2 L_F}{2}\mathbb{E}[\|\bar{m}_t^x\|^2]$$

$$\leq \mathbb{E}[F_{K+1}(\bar{x}_t)] - \frac{\eta\gamma_x}{2}\mathbb{E}[\|\nabla F_{K+1}(\bar{x}_t)\|^2] - \frac{\eta\gamma_x}{4}\mathbb{E}[\|\bar{m}_t^x\|^2] + \frac{\eta\gamma_x}{2}\mathbb{E}[\|\nabla F_{K+1}(\bar{x}_t) - \bar{m}_t^x\|^2]$$

$$\leq \mathbb{E}[F_{K+1}(\bar{x}_t)] - \frac{\eta\gamma_x}{2}\mathbb{E}[\|\nabla F_{K+1}(\bar{x}_t)\|^2] - \frac{\eta\gamma_x}{4}\mathbb{E}[\|\bar{m}_t^x\|^2] + \eta\gamma_x\mathbb{E}[\|(G_t - M_t^x)\mathbf{1}\frac{1}{N}\|^2] + \eta\gamma_x\mathbb{E}[\|\nabla F_{K+1}(\bar{x}_t) - \frac{1}{N}G_t\mathbf{1}\|^2],$$

where the second to last step follows from $\eta \leq \frac{1}{2\gamma_x L_F}$. As for $\mathbb{E}[\|\nabla F_{K+1}(\bar{x}_t) - \frac{1}{N}G_t\mathbf{1}\|^2]$, we decompose it as follows

$$\mathbb{E}[\|\nabla F_{K+1}(\bar{x}_t) - \frac{1}{N}G_t\mathbf{1}\|^2] = \mathbb{E}[\|\frac{1}{N}\sum_{n=1}^{N}\nabla_1 f_{K+1}^{(n)}(\bar{x}_t, y_K^*(\bar{x}_t)) + \frac{1}{N}\sum_{n=1}^{N}(-1)^K\nabla_{12}^2 f_1^{(n)}(y_1^*(\bar{x}_t), \bar{x}_t)z_1^*(\bar{x}_t)$$

$$- \frac{1}{N}\sum_{n=1}^{N}\nabla_1 f_{K+1}^{(n)}(x_t^{(n)}, y_{K,t}^{(n)}) - \frac{1}{N}\sum_{n=1}^{N}(-1)^K\nabla_{12}^2 f_1^{(n)}(y_{1,t}^{(n)}, x_t^{(n)})z_{1,t}^{(n)}\|^2]$$

$$\leq 6\,\mathbb{E}[\|\underbrace{\frac{1}{N}\sum_{n=1}^{N}\nabla_1 f_{K+1}^{(n)}(\bar{x}_t, y_K^*(\bar{x}_t)) - \frac{1}{N}\sum_{n=1}^{N}\nabla_1 f_{K+1}^{(n)}(\bar{x}_t, \bar{y}_{K,t})\|^2]}_{T_1}$$

$$+ 6\,\mathbb{E}[\|\underbrace{\frac{1}{N}\sum_{n=1}^{N}\nabla_1 f_{K+1}^{(n)}(\bar{x}_t, \bar{y}_{K,t}) - \frac{1}{N}\sum_{n=1}^{N}\nabla_1 f_{K+1}^{(n)}(x_t^{(n)}, y_{K,t}^{(n)})\|^2]}_{T_2}$$

$$+ 6\,\mathbb{E}[\|\underbrace{\frac{1}{N}\sum_{n=1}^{N}(-1)^K\nabla_{12}^2 f_1^{(n)}(y_1^*(\bar{x}_t), \bar{x}_t)z_1^*(\bar{x}_t) - \frac{1}{N}\sum_{n=1}^{N}(-1)^K\nabla_{12}^2 f_1^{(n)}(\bar{y}_{1,t}, \bar{x}_t)z_1^*(\bar{x}_t)\|^2]}_{T_3}\qquad(23)$$

$$+ 6\,\mathbb{E}[\|\underbrace{\frac{1}{N}\sum_{n=1}^{N}(-1)^K\nabla_{12}^2 f_1^{(n)}(\bar{y}_{1,t}, \bar{x}_t)z_1^*(\bar{x}_t) - \frac{1}{N}\sum_{n=1}^{N}(-1)^K\nabla_{12}^2 f_1^{(n)}(y_{1,t}^{(n)}, x_t^{(n)})z_1^*(\bar{x}_t)\|^2]}_{T_4}$$

$$+ 6\,\mathbb{E}[\|\frac{1}{N}\sum_{n=1}^{N}(-1)^K \nabla_{12}^2 f_1^{(n)}(y_{1,t}^{(n)}, x_t^{(n)}) z_1^*(\bar{x}_t) - \frac{1}{N}\sum_{n=1}^{N}(-1)^K \nabla_{12}^2 f_1^{(n)}(y_{1,t}^{(n)}, x_t^{(n)}) \bar{z}_{1,t}\|^2]$$
$$\underbrace{\qquad\qquad\qquad\qquad\qquad\qquad\qquad\qquad\qquad\qquad\qquad\qquad\qquad\qquad\qquad\qquad}_{T_5}$$

$$+ 6\,\mathbb{E}[\|\frac{1}{N}\sum_{n=1}^{N}(-1)^K \nabla_{12}^2 f_1^{(n)}(y_{1,t}^{(n)}, x_t^{(n)}) \bar{z}_{1,t} - \frac{1}{N}\sum_{n=1}^{N}(-1)^K \nabla_{12}^2 f_1^{(n)}(y_{1,t}^{(n)}, x_t^{(n)}) z_{1,t}^{(n)}\|^2]\,.$$
$$\underbrace{\qquad\qquad\qquad\qquad\qquad\qquad\qquad\qquad\qquad\qquad\qquad\qquad\qquad\qquad\qquad\qquad}_{T_6}$$

By bounding each term, we derive

$$T_1 \le 6L_f^2 \mathbb{E}[\|y_K^*(\bar{x}_t) - \bar{y}_{K,t}\|^2]\,, \quad T_2 \le 6L_f^2 \frac{1}{N}\sum_{n=1}^{N}\mathbb{E}[\|x_t^{(n)} - \bar{x}_t\|^2] + 6L_f^2 \frac{1}{N}\sum_{n=1}^{N}\mathbb{E}[\|y_{K,t}^{(n)} - \bar{y}_{K,t}\|^2]\,,$$

$$T_3 \le 6\Delta_1^2 L_J^2 \mathbb{E}[\|y_1^*(\bar{x}_t) - \bar{y}_{1,t}\|^2]\,, \quad T_4 \le 6\Delta_1^2 L_J^2 \frac{1}{N}\sum_{n=1}^{N}\mathbb{E}[\|x_t^{(n)} - \bar{x}_t\|^2] + 6\Delta_1^2 L_J^2 \frac{1}{N}\sum_{n=1}^{N}\mathbb{E}[\|y_{1,t}^{(n)} - \bar{y}_{1,t}\|^2]\,,$$

$$T_5 \le 6C_J^2 \mathbb{E}[\|z_1^*(\bar{x}_t) - \bar{z}_{1,t}\|^2]\,, \quad T_6 \le 6C_J^2 \frac{1}{N}\sum_{n=1}^{N}\mathbb{E}[\|\bar{z}_{1,t} - z_{1,t}^{(n)}\|^2]\,.$$

Combining these bounds yields

$$\mathbb{E}[\|\nabla F_{K+1}(\bar{x}_t) - \frac{1}{N}G_t \mathbf{1}\|^2] \le 6\Delta_1^2 L_J^2 \mathbb{E}[\|y_1^*(\bar{x}_t) - \bar{y}_{1,t}\|^2] + 6L_f^2 \mathbb{E}[\|y_K^*(\bar{x}_t) - \bar{y}_{K,t}\|^2] + 6C_J^2 \mathbb{E}[\|z_1^*(\bar{x}_t) - \bar{z}_{1,t}\|^2]$$
$$+ 6(L_f^2 + \Delta_1^2 L_J^2)\frac{1}{N}\mathbb{E}[\|X_t(I - J)\|_F^2] + 6\Delta_1^2 L_J^2 \frac{1}{N}\mathbb{E}[\|Y_{1,t}(I - J)\|_F^2] + 6L_f^2 \frac{1}{N}\mathbb{E}[\|Y_{K,t}(I - J)\|_F^2] + 6C_J^2 \frac{1}{N}\mathbb{E}[\|Z_{1,t}(I - J)\|_F^2]\,.$$

Therefore, we complete the proof. $\qquad\qquad\qquad\qquad\qquad\qquad\qquad\qquad\qquad\qquad\qquad\qquad\qquad\qquad\qquad\square$

**Lemma D.2.** *(**Optimization error regarding** y) Assuming that Assumptions 3.1 through 3.5 hold, and $\gamma_y \le \frac{1}{6L_g}$, when $k \in \{1, \cdots, K\}$, the following inequality is true:*

$$\mathbb{E}[\|\bar{y}_{k,t+1} - y_k^*(\bar{x}_{t+1})\|^2] \le (1 - \frac{\gamma_y \eta \mu}{4})\mathbb{E}[\|y_k^*(\bar{x}_t) - \bar{y}_{k,t}\|^2] - \frac{3\eta\gamma_y^2}{4}\mathbb{E}[\|\bar{m}_{k,t}^y\|^2] + \frac{5\eta\gamma_x^2}{\gamma_y \mu}C_{y_k}^2 \mathbb{E}[\|\bar{m}_t^x\|^2]$$
$$+ \frac{10\gamma_y \eta}{\mu}L_g^2 \mathbb{E}[\|y_{k-1}^*(\bar{x}_t) - \bar{y}_{k-1,t}\|^2] + \frac{20\gamma_y \eta L_g^2}{\mu}\frac{1}{N}\mathbb{E}[\|Y_{k,t}(I - J)\|_F^2] + \frac{20\gamma_y \eta L_g^2}{\mu}\frac{1}{N}\mathbb{E}[\|Y_{k-1,t}(I - J)\|_F^2]$$
$$+ \frac{20\gamma_y \eta}{\mu}\mathbb{E}[\|(M_{k,t}^y - U_{k,t})\mathbf{1}\frac{1}{N}\|^2]\,.$$

*Note that $y_0^*(\bar{x}_t) - \bar{y}_{0,t} = 0$ and $Y_{0,t} = X_t$.*

*Proof.* Here, we introduce a virtual variable $\bar{\bar{y}}_{k,t+1} = \bar{y}_{k,t} - \gamma_y \bar{m}_{k,t}^y$. Then, it is easy to know that $\bar{y}_{k,t+1} = \bar{y}_{k,t} + \eta(\bar{\bar{y}}_{k,t+1} - \bar{y}_{k,t})$. Since $f_k(\bar{y}_k, y_{k-1}^*(\bar{x}))$ is $\mu$-strongly convex regarding $\bar{y}_k$, we derive

$$f_k(y, y_k^*(\bar{x}_t)) \ge f_k(\bar{y}_{k,t}, y_k^*(\bar{x}_t)) + \langle \nabla_1 f_k(\bar{y}_{k,t}, y_k^*(\bar{x}_t)), y - \bar{y}_{k,t}\rangle + \frac{\mu}{2}\|y - \bar{y}_{k,t}\|^2$$
$$= f_k(\bar{y}_{k,t}, y_k^*(\bar{x}_t)) + \langle \nabla_1 f_k(\bar{y}_{k,t}, y_k^*(\bar{x}_t)), \bar{\bar{y}}_{k,t+1} - \bar{y}_{k,t}\rangle + \langle \nabla_1 f_k(\bar{y}_{k,t}, y_k^*(\bar{x}_t)), y - \bar{\bar{y}}_{k,t+1}\rangle + \frac{\mu}{2}\|y - \bar{y}_{k,t}\|^2\,.$$
$$\ge f_k(\bar{\bar{y}}_{k,t+1}, y_k^*(\bar{x}_t)) - \frac{L_g}{2}\|\bar{\bar{y}}_{k,t+1} - \bar{y}_{k,t}\|^2 + \langle \nabla_1 f_k(\bar{y}_{k,t}, y_k^*(\bar{x}_t)), y - \bar{\bar{y}}_{k,t+1}\rangle + \frac{\mu}{2}\|y - \bar{y}_{k,t}\|^2\,.$$

The third term can be decomposed as

$$\langle \nabla_1 f_k(\bar{y}_{k,t}, y_k^*(\bar{x}_t)), y - \bar{\bar{y}}_{k,t+1}\rangle = \langle \nabla_1 f_k(\bar{y}_{k,t}, y_k^*(\bar{x}_t)) - \bar{m}_{k,t}^y, y - \tilde{y}_{k,t+1}\rangle + \langle \bar{m}_{k,t}^y, y - \bar{\bar{y}}_{k,t+1}\rangle$$
$$= \langle \nabla_1 f_k(\bar{y}_{k,t}, y_k^*(\bar{x}_t)) - \bar{m}_{k,t}^y, y - \bar{\bar{y}}_{k,t+1}\rangle + \langle \bar{m}_{k,t}^y, y - \bar{y}_{k,t}\rangle + \langle \bar{m}_{k,t}^y, \bar{y}_{k,t} - \bar{\bar{y}}_{k,t+1}\rangle\,.$$

Combining these results yield

$$f_k(y, y_k^*(\bar{x}_t)) \ge f_k(\bar{\bar{y}}_{k,t+1}, y_k^*(\bar{x}_t)) + \frac{\mu}{2}\|y - \bar{y}_{k,t}\|^2 + (\gamma_y - \frac{\gamma_y^2 L_g}{2})\|\bar{m}_{k,t}^y\|^2$$

$$+ \langle \nabla_1 f_k(\bar{y}_{k,t}, y_k^*(\bar{x}_t)) - \bar{m}_{k,t}^y, y - \bar{\bar{y}}_{k,t+1} \rangle + \langle \bar{m}_{k,t}^y, y - \bar{y}_{k,t} \rangle \ .$$

Then, by setting $y = y_k^*(\bar{x}_t)$, we obtain

$$
\begin{aligned}
f_k(y_k^*(\bar{x}_t), y_{k-1}^*(\bar{x}_t)) &\geq f_k(\bar{\bar{y}}_{k,t+1}, y_{k-1}^*(\bar{x}_t)) + \frac{\mu}{2}\|y_k^*(\bar{x}_t) - \bar{y}_{k,t}\|^2 + (\gamma_y - \frac{\gamma_y^2 L_g}{2})\|\bar{m}_{k,t}^y\|^2 + \langle \bar{m}_{k,t}^y, y_k^*(\bar{x}_t) - \bar{y}_{k,t} \rangle \\
&\quad + \langle \nabla_1 f_k(\bar{y}_{k,t}, y_{k-1}^*(\bar{x}_t)) - \bar{m}_{k,t}^y, y_k^*(\bar{x}_t) - \bar{\bar{y}}_{k,t+1} \rangle \\
&= f_k(\bar{\bar{y}}_{k,t+1}, y_{k-1}^*(\bar{x}_t)) + \frac{\mu}{2}\|y_k^*(\bar{x}_t) - \bar{y}_{k,t}\|^2 + (\gamma_y - \frac{\gamma_y^2 L_g}{2})\|\bar{m}_{k,t}^y\|^2 + \langle \bar{m}_{k,t}^y, y_k^*(\bar{x}_t) - \bar{y}_{k,t} \rangle - \frac{1}{2\gamma_y\eta}\|\bar{y}_{k,t} - y_k^*(\bar{x}_t)\|^2 \\
&\quad - \frac{\gamma_y\eta}{2}\|\bar{m}_{k,t}^y\|^2 + \frac{1}{2\gamma_y\eta}\|\bar{y}_{k,t+1} - y_k^*(\bar{x}_t)\|^2 + \underbrace{\langle \nabla_1 f_k(\bar{y}_{k,t}, y_{k-1}^*(\bar{x}_t)) - \bar{m}_{k,t}^y, y_k^*(\bar{x}_t) - \bar{\bar{y}}_{k,t+1} \rangle}_{T_1} \ ,
\end{aligned}
$$

where the last step follows from $\|\bar{y}_{k,t+1} - y_k^*(\bar{x}_t)\|^2 = \|\bar{y}_{k,t} + \eta(\bar{\bar{y}}_{k,t+1} - \bar{y}_{k,t}) - y_k^*(\bar{x}_t)\|^2 = \|\bar{y}_{k,t} - y_k^*(\bar{x}_t)\|^2 + \gamma_y^2\eta^2\|\bar{m}_{k,t}^y\|^2 - 2\eta\gamma_y\langle \bar{y}_{k,t} - y_k^*(\bar{x}_t), \bar{m}_{k,t}^y \rangle$. In particular, $T_1$ can be decomposed as

$$
\begin{aligned}
T_1 &= \langle \nabla_1 f_k(\bar{y}_{k,t}, y_{k-1}^*(\bar{x}_t)) - \bar{m}_{k,t}^y, y_k^*(\bar{x}_t) - \bar{y}_{k,t} \rangle + \langle \nabla_1 f_k(\bar{y}_{k,t}, y_{k-1}^*(\bar{x}_t)) - \bar{m}_{k,t}^y, \bar{y}_{k,t} - \bar{\bar{y}}_{k,t+1} \rangle \\
&\geq -\frac{1}{\mu}\|\nabla_1 f_k(\bar{y}_{k,t}, y_{k-1}^*(\bar{x}_t)) - \bar{m}_{k,t}^y\|^2 - \frac{\mu}{4}\|y_k^*(\bar{x}_t) - \bar{y}_{k,t}\|^2 - \frac{1}{\mu}\|\nabla_1 f_k(\bar{y}_{k,t}, y_{k-1}^*(\bar{x}_t)) - \bar{m}_{k,t}^y\|^2 - \frac{\mu}{4}\|\bar{y}_{k,t} - \bar{\bar{y}}_{k,t+1}\|^2 \ .
\end{aligned}
$$

Combining these results with $\mu \leq L_g$ yields

$$
\begin{aligned}
f_k(y_k^*(\bar{x}_t), y_{k-1}^*(\bar{x}_t)) &= f_k(\bar{\bar{y}}_{k,t+1}, y_{k-1}^*(\bar{x}_t)) + (\frac{\mu}{4} - \frac{1}{2\gamma_y\eta})\|y_k^*(\bar{x}_t) - \bar{y}_{k,t}\|^2 + (\frac{\gamma_y}{2} - \frac{3\gamma_y^2 L_g}{4})\|\bar{m}_{k,t}^y\|^2 \\
&\quad - \frac{2}{\mu}\|\nabla_1 f_k(\bar{y}_{k,t}, y_{k-1}^*(\bar{x}_t)) - \bar{m}_{k,t}^y\|^2 + \frac{1}{2\gamma_y\eta}\|\bar{y}_{k,t+1} - y_k^*(\bar{x}_t)\|^2 \ .
\end{aligned}
$$

Because $y_k^*(\bar{x}_t)$ is the optimal solution of the strongly convex problem, we have $f_k(\bar{\bar{y}}_{k,t+1}, y_{k-1}^*(\bar{x}_t)) \geq f_k(y_k^*(\bar{x}_t), y_{k-1}^*(\bar{x}_t))$. Then, we obtain

$$(\frac{\mu}{4} - \frac{1}{2\gamma_y\eta})\|y_k^*(\bar{x}_t) - \bar{y}_{k,t}\|^2 + (\frac{\gamma_y}{2} - \frac{3\gamma_y^2 L_g}{4})\|\bar{m}_{k,t}^y\|^2 - \frac{2}{\mu}\|\nabla_1 f_k(\bar{y}_{k,t}, y_{k-1}^*(\bar{x}_t)) - \bar{m}_{k,t}^y\|^2 + \frac{1}{2\gamma_y\eta}\|\bar{y}_{k,t+1} - y_k^*(\bar{x}_t)\|^2 \leq 0 \ .$$

By setting $\gamma_y \leq \frac{1}{6L_g}$, we have

$$\|\bar{y}_{k,t+1} - y_k^*(\bar{x}_t)\|^2 \leq (1 - \frac{\gamma_y\eta\mu}{2})\|y_k^*(\bar{x}_t) - \bar{y}_{k,t}\|^2 - \frac{3\gamma_y^2\eta}{4}\|\bar{m}_{k,t}^y\|^2 + \frac{4\gamma_y\eta}{\mu}\|\nabla_1 f_k(\bar{y}_{k,t}, y_{k-1}^*(\bar{x}_t)) - \bar{m}_{k,t}^y\|^2 \ . \tag{24}$$

Then, we have

$$
\begin{aligned}
\|\bar{y}_{k,t+1} - y_k^*(\bar{x}_{t+1})\|^2 &\leq (1+c)\|\bar{y}_{k,t+1} - y_k^*(\bar{x}_t)\|^2 + (1+1/c)\|y_k^*(\bar{x}_{t+1}) - y_k^*(\bar{x}_t)\|^2 \\
&\leq (1+c)(1 - \frac{\gamma_y\eta\mu}{2})\|y_k^*(\bar{x}_t) - \bar{y}_{k,t}\|^2 - (1+c)\frac{3\gamma_y^2\eta}{4}\|\bar{m}_{k,t}^y\|^2 \\
&\quad + (1+c)\frac{4\gamma_y\eta}{\mu}\|\nabla_1 f_k(\bar{y}_{k,t}, y_{k-1}^*(\bar{x}_t)) - \bar{m}_{k,t}^y\|^2 + (1+1/c)\|y_k^*(\bar{x}_{t+1}) - y_k^*(\bar{x}_t)\|^2 \\
&\leq (1 - \frac{\gamma_y\eta\mu}{4})\|y_k^*(\bar{x}_t) - \bar{y}_{k,t}\|^2 - \frac{3\eta\gamma_y^2}{4}\|\bar{m}_{k,t}^y\|^2 + \frac{5\gamma_y\eta}{\mu}\|\nabla_1 f_k(\bar{y}_{k,t}, y_{k-1}^*(\bar{x}_t)) - \bar{m}_{k,t}^y\|^2 + \frac{5\eta\gamma_x^2}{\gamma_y\mu}C_{y_k}^2\|\bar{m}_t^x\|^2 \ .
\end{aligned}
\tag{25}
$$

where the last step follows from $c = \frac{\gamma_y\eta\mu}{4}$. In addition, we derive:

$$
\begin{aligned}
\|\nabla_1 f_k(\bar{y}_{k,t}, y_{k-1}^*(\bar{x}_t)) - \bar{m}_{k,t}^y\|^2 &\leq 2\|\nabla_1 f_k(\bar{y}_{k,t}, y_{k-1}^*(\bar{x}_t)) - \nabla_1 f_k(\bar{y}_{k,t}, \bar{y}_{k-1,t})\|^2 + 2\|\nabla_1 f_k(\bar{y}_{k,t}, \bar{y}_{k-1,t}) - \bar{m}_{k,t}^y\|^2 \\
&\leq 2L_g^2\|y_{k-1}^*(\bar{x}_t) - \bar{y}_{k-1,t}\|^2 + 4\|\frac{1}{N}\sum_{n=1}^N \nabla_1 f_k^{(n)}(\bar{y}_{k,t}, \bar{y}_{k-1,t}) - \frac{1}{N}\sum_{n=1}^N \nabla_1 f_k^{(n)}(y_{k,t}^{(n)}, y_{k-1,t}^{(n)})\|^2 \\
&\quad + 4\|\frac{1}{N}\sum_{n=1}^N \nabla_1 f_k^{(n)}(y_{k,t}^{(n)}, y_{k-1,t}^{(n)}) - \bar{m}_{k,t}^y\|^2 \ .
\end{aligned}
\tag{26}
$$

$$\leq 2L_g^2\|y_{k-1}^*(\bar{x}_t) - \bar{y}_{k-1,t}\|^2 + \frac{4}{N}\|Y_{k,t}(I-J)\|_F^2 + \frac{4}{N}\|Y_{k-1,t}(I-J)\|_F^2 + 4\|(M_{k,t}^y - U_{k,t})\mathbf{1}\frac{1}{N}\|^2.$$

Finally, combining Eq. (24), (25), and (26), we obtain

$$\mathbb{E}[\|\bar{y}_{k,t+1} - y_k^*(\bar{x}_{t+1})\|^2]$$

$$\leq (1 - \frac{\gamma_y\eta\mu}{4})\mathbb{E}[\|y_k^*(\bar{x}_t) - \bar{y}_{k,t}\|^2] - \frac{3\eta\gamma_y^2}{4}\mathbb{E}[\|\bar{m}_{k,t}^y\|^2] + \frac{5\eta\gamma_x^2}{\gamma_y\mu}C_{y_k}^2\mathbb{E}[\|\bar{m}_t^x\|^2] + \frac{10\gamma_y\eta}{\mu}L_g^2\mathbb{E}[\|y_{k-1}^*(\bar{x}_t) - \bar{y}_{k-1,t}\|^2]$$

$$+ \frac{20\gamma_y\eta L_g^2}{\mu}\frac{1}{N}\mathbb{E}[\|Y_{k,t}(I-J)\|_F^2] + \frac{20\gamma_y\eta L_g^2}{\mu}\frac{1}{N}\mathbb{E}[\|Y_{k-1,t}(I-J)\|_F^2] + \frac{20\gamma_y\eta}{\mu}\mathbb{E}[\|(M_{k,t}^y - U_{k,t})\mathbf{1}\frac{1}{N}\|^2].$$

$\square$

**Lemma D.3.** (*Optimization error regarding* z) *Assuming that Assumptions 3.1 through 3.5 hold, when $\eta < \frac{1}{\gamma_z\mu}$, for $k \in \{1, \cdots, K-1\}$, the following inequality is true:*

$$\mathbb{E}[\|\bar{z}_{k,t+1} - z_k^*(\bar{x}_{t+1})\|^2] \leq (1 - \frac{\gamma_z\eta\mu}{8})\mathbb{E}[\|\bar{z}_{k,t} - z_k^*(\bar{x}_t)\|^2] + \frac{15\gamma_z\eta C_J^2}{\mu}\mathbb{E}[\|\bar{z}_{k+1,t} - z_{k+1}^*(\bar{x}_t)\|^2]$$

$$+ \frac{15\gamma_z\eta}{\mu}L_J^2\Delta_{k+1}^2\mathbb{E}[\|\bar{y}_{k+1,t} - y_{k+1}^*(\bar{x}_t)\|^2] + \frac{15\gamma_z\eta}{\mu}(L_H^2\Delta_k^2 + L_J^2\Delta_{k+1}^2)\mathbb{E}[\|\bar{y}_{k,t} - y_k^*(\bar{x}_t)\|^2]$$

$$+ \frac{15\gamma_z\eta}{\mu}L_H^2\Delta_k^2\mathbb{E}[\|\bar{y}_{k-1,t} - y_k^*(\bar{x}_{t-1})\|^2] + 3\eta\gamma_z^2\frac{1}{N}\mathbb{E}[\|R_{k,t}(I-J)\|_F^2] + \frac{9\gamma_x^2\eta}{\gamma_z\mu}C_{z_k}^2\mathbb{E}[\|\bar{m}_t^x\|^2]$$

$$+ \frac{12\gamma_z\eta}{\mu}\mathbb{E}[\|(M_{k,t}^z - V_{k,t})\mathbf{1}\frac{1}{N}\|^2] + \frac{45\gamma_z\eta}{\mu}(\Delta_k^2 L_H^2 + \Delta_{k+1}^2 L_J^2)\frac{1}{N}\mathbb{E}[\|Y_{k,t}(I-J)\|_F^2]$$

$$+ \frac{45\gamma_z\eta}{\mu}\Delta_k^2 L_H^2\frac{1}{N}\mathbb{E}[\|Y_{k-1,t}(I-J)\|_F^2] + \frac{40\gamma_z\eta}{\mu}\Delta_{k+1}^2 L_J^2\frac{1}{N}\mathbb{E}[\|Y_{k+1,t}(I-J)\|_F^2]$$

$$+ (\frac{45\gamma_z\eta}{\mu}L_g^2 + 3)\frac{1}{N}\mathbb{E}[\|Z_{k,t}(I-J)\|_F^2] + \frac{45\gamma_z\eta}{\mu}C_J^2\frac{1}{N}\mathbb{E}[\|Z_{k+1,t}(I-J)\|_F^2].$$

*Note that $Y_{0,t} = X_t$ and $\bar{y}_{0,t} - y_0^*(\bar{x}_t) = 0$.*

*When $k = K$, the following inequality is true:*

$$\mathbb{E}[\|\bar{z}_{K,t+1} - z_K^*(\bar{x}_{t+1})\|^2] \leq (1 - \frac{\gamma_z\eta\mu}{8})\mathbb{E}[\|\bar{z}_{K,t} - z_K^*(\bar{x}_t)\|^2] + \frac{15\gamma_z\eta}{\mu}(L_H^2\Delta_K^2 + L_J^2\Delta_{K+1}^2)\mathbb{E}[\|\bar{y}_{K,t} - y_K^*(\bar{x}_t)\|^2]$$

$$+ \frac{15\gamma_z\eta}{\mu}L_H^2\Delta_K^2\mathbb{E}[\|\bar{y}_{K-1,t} - y_K^*(\bar{x}_{t-1})\|^2] + 3\eta\gamma_z^2\frac{1}{N}\mathbb{E}[\|R_{K,t}(I-J)\|_F^2] + \frac{9\gamma_x^2\eta}{\gamma_z\mu}C_{z_K}^2\mathbb{E}[\|\bar{m}_t^x\|^2]$$

$$+ \frac{12\gamma_z\eta}{\mu}\mathbb{E}[\|(M_{K,t}^z - V_{K,t})\mathbf{1}\frac{1}{N}\|^2] + \frac{45\gamma_z\eta}{\mu}(\Delta_K^2 L_H^2 + \Delta_{K+1}^2 L_J^2)\frac{1}{N}\mathbb{E}[\|Y_{K,t}(I-J)\|_F^2]$$

$$+ \frac{45\gamma_z\eta}{\mu}\Delta_K^2 L_H^2\frac{1}{N}\mathbb{E}[\|Y_{K-1,t}(I-J)\|_F^2] + (\frac{45\gamma_z\eta}{\mu}L_g^2 + 3)\frac{1}{N}\mathbb{E}[\|Z_{K,t}(I-J)\|_F^2].$$

*Proof.* For any $k \in \{1, \cdots, K-1\}$, we have

$$\mathbb{E}[\|\bar{z}_{k,t+1} - z_k^*(\bar{x}_t)\|^2] = \mathbb{E}[\|\frac{1}{N}\sum_{n=1}^N \mathcal{P}(z_{k,t}^{(n)} - \gamma_z\eta r_{k,t}^{(n)} + \eta\sum_{j\in\mathcal{N}_n} w_{nj}(z_{k,t}^{(j)} - z_{k,t}^{(n)})) - z_k^*(\bar{x}_t)\|^2]$$

$$\leq \frac{1}{N}\sum_{n=1}^N \mathbb{E}[\|\mathcal{P}(z_{k,t}^{(n)} - \gamma_z\eta r_{k,t}^{(n)} + \eta\sum_{j\in\mathcal{N}_n} w_{nj}(z_{k,t}^{(j)} - z_{k,t}^{(n)})) - z_k^*(\bar{x}_t)\|^2]$$

$$\leq \frac{1}{N}\sum_{n=1}^N \mathbb{E}[\|z_{k,t}^{(n)} - \gamma_z\eta r_{k,t}^{(n)} + \eta\sum_{j\in\mathcal{N}_n} w_{nj}(z_{k,t}^{(j)} - z_{k,t}^{(n)}) - z_k^*(\bar{x}_t)\|^2]$$

$$= \frac{1}{N}\sum_{n=1}^N \mathbb{E}[\|\eta\sum_{j\in\mathcal{N}_n} w_{nj}z_{k,t}^{(j)} - \eta\bar{z}_{k,t} - \eta\gamma_z r_{k,t}^{(n)} + \eta\gamma_z\bar{r}_{k,t} + (1-\eta)z_{k,t}^{(n)} - (1-\eta)\bar{z}_{k,t} + \bar{z}_{k,t} - \eta\gamma_z\bar{r}_{k,t} - z_k^*(\bar{x}_t)\|^2]$$

$$\leq \frac{1}{N}\sum_{n=1}^N \mathbb{E}[\|\eta\sum_{j\in\mathcal{N}_n} w_{nj}z_{k,t}^{(j)} - \eta\bar{z}_{k,t} - \eta\gamma_z r_{k,t}^{(n)} + \eta\gamma_z\bar{r}_{k,t} + (1-\eta)z_{k,t}^{(n)} - (1-\eta)\bar{z}_{k,t}\|^2]$$

$$+ 2\frac{1}{N}\sum_{n=1}^{N}\mathbb{E}\Big[\Big\langle \bar{z}_{k,t} - \eta\gamma_z\bar{r}_{k,t} - z_k^*(\bar{x}_t), \eta\sum_{j\in\mathcal{N}_n}w_{nj}z_{k,t}^{(j)} - \eta\bar{z}_{k,t} - \eta\gamma_z r_{k,t}^{(n)} + \eta\gamma_z\bar{r}_{k,t} + (1-\eta)z_{k,t}^{(n)} - (1-\eta)\bar{z}_{k,t}\Big\rangle\Big]$$

$$+ \mathbb{E}[\|\bar{z}_{k,t} - \eta\gamma_z\bar{r}_{k,t} - z_k^*(\bar{x}_t)\|^2]$$

$$= \underbrace{\frac{1}{N}\sum_{n=1}^{N}\mathbb{E}[\|\eta\sum_{j\in\mathcal{N}_n}w_{nj}z_{k,t}^{(j)} - \eta\bar{z}_{k,t} - \eta\gamma_z r_{k,t}^{(n)} + \eta\gamma_z\bar{r}_{k,t} + (1-\eta)z_{k,t}^{(n)} - (1-\eta)\bar{z}_{k,t}\|^2]}_{T_1}$$

$$+ \mathbb{E}[\|\bar{z}_{k,t} - \eta\gamma_z\bar{r}_{k,t} - z_k^*(\bar{x}_t)\|^2]\,,$$

where the sixth step follows from the fact $\frac{1}{N}\sum_{n=1}^{N}(\sum_{j\in\mathcal{N}_n}w_{nj}z_{k,t}^{(j)} - \eta\bar{z}_{k,t} - \eta\gamma_z v_{k,t}^{(n)} + \eta\gamma_z\bar{v}_{z_k,t} + (1-\eta)z_{k,t}^{(n)} - (1-\eta)\bar{z}_{k,t}) = 0$. For $T_1$, we derive

$$T_1 \le \frac{1}{N}\sum_{n=1}^{N}(1+\frac{1-\eta}{\eta})\eta^2\mathbb{E}[\|\sum_{j\in\mathcal{N}_n}w_{nj}z_{k,t}^{(j)} - \bar{z}_{k,t} - \gamma_z r_{k,t}^{(n)} + \gamma_z\bar{r}_{k,t}\|^2] + \frac{1}{N}\sum_{n=1}^{N}(1+\frac{\eta}{1-\eta})(1-\eta)^2\mathbb{E}[\|z_{k,t}^{(n)} - \bar{z}_{k,t}\|^2]$$

$$\le 2\eta\frac{1}{N}\sum_{n=1}^{N}\mathbb{E}[\|\sum_{j\in\mathcal{N}_n}w_{nj}z_{k,t}^{(j)} - \bar{z}_{k,t}\|^2] + 2\eta\gamma_z^2\frac{1}{N}\sum_{n=1}^{N}\mathbb{E}[\|r_{k,t}^{(n)} - \bar{r}_{k,t}\|^2] + (1-\eta)\frac{1}{N}\sum_{n=1}^{N}\mathbb{E}[\|z_{k,t}^{(n)} - \bar{z}_{k,t}\|^2]$$

$$\le 2\frac{1}{N}\mathbb{E}[\|Z_{k,t}(I-J)\|_F^2] + 2\eta\gamma_z^2\frac{1}{N}\mathbb{E}[\|R_{k,t}(I-J)\|_F^2]\,,$$

where the last step follows from $\sum_{n=1}^{N}\|\sum_{j\in\mathcal{N}_n}w_{nj}z_{k,t}^{(j)} - \bar{z}_{k,t}\|^2 \le \lambda^2\sum_{n=1}^{N}\|z_{k,t}^{(n)} - \bar{z}_{k,t}\|^2 = \lambda^2\|Z_{k,t}(I-J)\|_F^2$ and $2\eta\lambda^2 + 1 - \eta \le 2\eta + 1 - \eta \le 1 + \eta \le 2$. Thus, we obtain

$$\mathbb{E}[\|\bar{z}_{k,t+1} - z_k^*(\bar{x}_t)\|^2] \le 2\frac{1}{N}\mathbb{E}[\|Z_{k,t}(I-J)\|_F^2] + 2\eta\gamma_z^2\frac{1}{N}\mathbb{E}[\|R_{k,t}(I-J)\|_F^2] + \mathbb{E}[\|\bar{z}_{k,t} - \eta\gamma_z\bar{r}_{k,t} - z_k^*(\bar{x}_t)\|^2]\,. \quad (27)$$

As for $\|\bar{z}_{k,t} - \eta\gamma_z\bar{r}_{k,t} - z_k^*(\bar{x}_t)\|^2$, we have

$$\mathbb{E}[\|\bar{z}_{k,t} - \eta\gamma_z\bar{r}_{k,t} - z_k^*(\bar{x}_t)\|^2] = \mathbb{E}[\|\bar{z}_{k,t} - \eta\gamma_z\bar{v}_{k,t} - z_k^*(\bar{x}_t)\|^2]$$

$$\le (1+\frac{\gamma_z\eta\mu}{4})\underbrace{\mathbb{E}[\|\bar{z}_{k,t} - z_k^*(\bar{x}_t) - \eta\gamma_z(\nabla_{11}^2 f_k(\bar{y}_{k,t}, \bar{y}_{k-1,t})\bar{z}_{k,t} - \nabla_{12}^2 f_{k+1}(\bar{y}_{k+1,t}, \bar{y}_{k,t})\bar{z}_{k+1,t})\|^2]}_{T_2}$$

$$+ (1+\frac{4}{\gamma_z\eta\mu})\eta^2\gamma_z^2\underbrace{\mathbb{E}[\|\nabla_{11}^2 f_k(\bar{y}_{k,t}, \bar{y}_{k-1,t})\bar{z}_{k,t} - \nabla_{12}^2 f_{k+1}(\bar{y}_{k+1,t}, \bar{y}_{k,t})\bar{z}_{k+1,t} - \bar{v}_{k,t}\|^2]}_{T_3}\,, \quad (28)$$

where the first step follows from $\bar{r}_{k,t} = \bar{v}_{k,t}$.

For $T_2$, we decompose it as follows

$$T_2 = \mathbb{E}[\|\bar{z}_{k,t} - z_k^*(\bar{x}_t) - \eta\gamma_z\Big(\nabla_{11}^2 f_k(\bar{y}_{k,t}, \bar{y}_{k-1,t})\bar{z}_{k,t} - \nabla_{12}^2 f_{k+1}(\bar{y}_{k+1,t}, \bar{y}_{k,t})\bar{z}_{k+1,t}\Big)$$

$$+ \eta\gamma_z\Big(\nabla_{11}^2 f_k(y_k^*(\bar{x}_t), y_{k-1}^*(\bar{x}_t))z_k^*(\bar{x}_t) - \nabla_{12}^2 f_{k+1}(y_{k+1}^*(\bar{x}_t), y_k^*(\bar{x}_t))z_{k+1}^*(\bar{x}_t)\Big)\|^2]$$

$$= \mathbb{E}[\|(I - \eta\gamma_z\nabla_{11}^2 f_k(\bar{y}_{k,t}, \bar{y}_{k-1,t}))\bar{z}_{k,t} - (I - \eta\gamma_z\nabla_{11}^2 f_k(y_k^*(\bar{x}_t), y_{k-1}^*(\bar{x}_t)))z_k^*(\bar{x}_t)$$

$$+ \eta\gamma_z\Big(\nabla_{12}^2 f_{k+1}(\bar{y}_{k+1,t}, \bar{y}_{k,t})\bar{z}_{k+1,t} - \nabla_{12}^2 f_{k+1}(y_{k+1}^*(\bar{x}_t), y_k^*(\bar{x}_t))z_{k+1}^*(\bar{x}_t)\Big)\|^2]$$

$$\le (1+\frac{\gamma_z\eta\mu}{2})\underbrace{\mathbb{E}[\|(I - \eta\gamma_z\nabla_{11}^2 f_k(\bar{y}_{k,t}, \bar{y}_{k-1,t}))\bar{z}_{k,t} - (I - \eta\gamma_z\nabla_{11}^2 f_k(\bar{y}_{k,t}, \bar{y}_{k-1,t}))z_k^*(\bar{x}_t)\|^2]}_{T_{2,1}}$$

$$+ 3(1+\frac{2}{\gamma_z\eta\mu})\underbrace{\mathbb{E}[\|(I - \eta\gamma_z\nabla_{11}^2 f_k(\bar{y}_{k,t}, \bar{y}_{k-1,t}))z_k^*(\bar{x}_t) - (I - \eta\gamma_z\nabla_{11}^2 f_k(y_k^*(\bar{x}_t), y_{k-1}^*(\bar{x}_t)))z_k^*(\bar{x}_t)\|^2]}_{T_{2,2}}$$

$$+ 3\eta^2\gamma_z^2(1+\frac{2}{\gamma_z\eta\mu})\underbrace{\mathbb{E}[\|\nabla_{12}^2 f_{k+1}(\bar{y}_{k+1,t}, \bar{y}_{k,t})\bar{z}_{k+1,t} - \nabla_{12}^2 f_{k+1}(\bar{y}_{k+1,t}, \bar{y}_{k,t})z_{k+1}^*(\bar{x}_t)\|^2]}_{T_{2,3}}$$

$$+ 3\eta^2\gamma_z^2(1+\frac{2}{\gamma_z\eta\mu})\underbrace{\mathbb{E}[\|\nabla_{12}^2 f_{k+1}(\bar{y}_{k+1,t}, \bar{y}_{k,t})z_{k+1}^*(\bar{x}_t) - \nabla_{12}^2 f_{k+1}(y_{k+1}^*(\bar{x}_t), y_k^*(\bar{x}_t))z_{k+1}^*(\bar{x}_t)\|^2]}_{T_{2,4}}\,.$$

By bounding each term, we derive

$$T_{2,1} \leq (1 - \gamma_z \eta \mu)^2 \mathbb{E}[\|\bar{z}_{k,t} - z_k^*(\bar{x}_t)\|^2] , \quad T_{2,2} \leq C_J^2 \mathbb{E}[\|\bar{z}_{k+1,t} - z_{k+1}^*(\bar{x}_t)\|^2] ,$$
$$T_{2,3} \leq L_H^2 \Delta_k^2 \mathbb{E}[\|\bar{y}_{k,t} - y_k^*(\bar{x}_t)\|^2] + L_H^2 \Delta_k^2 \mathbb{E}[\|\bar{y}_{k-1,t} - y_k^*(\bar{x}_{t-1})\|^2] ,$$
$$T_{2,4} \leq L_J^2 \Delta_{k+1}^2 \mathbb{E}[\|\bar{y}_{k+1,t} - y_{k+1}^*(\bar{x}_t)\|^2] + L_J^2 \Delta_{k+1}^2 \mathbb{E}[\|\bar{y}_{k,t} - y_k^*(\bar{x}_t)\|^2] .$$

Combining these bounds with $\eta < \frac{1}{\gamma_z \mu}$ yields

$$T_2 \leq (1 - \frac{\gamma_z \eta \mu}{2}) \mathbb{E}[\|\bar{z}_{k,t} - z_k^*(\bar{x}_t)\|^2] + \frac{9\gamma_z \eta C_J^2}{\mu} \mathbb{E}[\|\bar{z}_{k+1,t} - z_{k+1}^*(\bar{x}_t)\|^2] + \frac{9\gamma_z \eta}{\mu} L_J^2 \Delta_{k+1}^2 \mathbb{E}[\|\bar{y}_{k+1,t} - y_{k+1}^*(\bar{x}_t)\|^2]$$
$$+ \frac{9\gamma_z \eta}{\mu}(L_H^2 \Delta_k^2 + L_J^2 \Delta_{k+1}^2) \mathbb{E}[\|\bar{y}_{k,t} - y_k^*(\bar{x}_t)\|^2] + \frac{9\gamma_z \eta}{\mu} L_H^2 \Delta_k^2 \mathbb{E}[\|\bar{y}_{k-1,t} - y_k^*(\bar{x}_{t-1})\|^2] .$$

For $T_3$, following Eq. (23), we decompose it and bounding each term, and obtain

$$T_3 \leq 2\mathbb{E}[\|(M_{k,t}^z - V_{k,t})\mathbf{1}\frac{1}{N}\|^2] + 8(\Delta_k^2 L_H^2 + \Delta_{k+1}^2 L_J^2)\frac{1}{N}\mathbb{E}[\|Y_{k,t}(I - J)\|_F^2] + 8\Delta_k^2 L_H^2 \frac{1}{N}\mathbb{E}[\|Y_{k-1,t}(I - J)\|_F^2]$$
$$+ 8\Delta_{k+1}^2 L_J^2 \frac{1}{N}\mathbb{E}[\|Y_{k+1,t}(I - J)\|_F^2] + 8L_g^2 \frac{1}{N}\mathbb{E}[\|Z_{k,t}(I - J)\|_F^2] + 8C_J^2 \frac{1}{N}\mathbb{E}[\|Z_{k+1,t}(I - J)\|_F^2] .$$

Finally, plugging $T_2$ and $T_3$ into Eq. (28), combining Eq. (27) with $\eta < \frac{1}{\gamma_z \mu}$, we obtain

$$\mathbb{E}[\|\bar{z}_{k,t+1} - z_k^*(\bar{x}_{t+1})\|^2] \leq (1 + \frac{\gamma_z \eta \mu}{8}) \mathbb{E}[\|\bar{z}_{k,t+1} - z_k^*(\bar{x}_t)\|^2] + (1 + \frac{8}{\gamma_z \eta \mu}) \mathbb{E}[\|z_k^*(\bar{x}_{t+1}) - z_k^*(\bar{x}_t)\|^2]$$

$$\leq (1 + \frac{\gamma_z \eta \mu}{8}) \mathbb{E}[\|\bar{z}_{k,t} - \eta\gamma_z \bar{r}_{k,t} - z_k^*(\bar{x}_t)\|^2] + 2(1 + \frac{\gamma_z \eta \mu}{8})\frac{1}{N}\mathbb{E}[\|Z_{k,t}(I - J)\|_F^2]$$
$$+ 2\eta\gamma_z^2(1 + \frac{\gamma_z \eta \mu}{8})\frac{1}{N}\mathbb{E}[\|R_{k,t}(I - J)\|_F^2] + (1 + \frac{8}{\gamma_z \eta \mu})\mathbb{E}[\|z_k^*(\bar{x}_{t+1}) - z_k^*(\bar{x}_t)\|^2]$$

$$\leq (1 - \frac{\gamma_z \eta \mu}{8})\mathbb{E}[\|\bar{z}_{k,t} - z_k^*(\bar{x}_t)\|^2] + \frac{15\gamma_z \eta C_J^2}{\mu}\mathbb{E}[\|\bar{z}_{k+1,t} - z_{k+1}^*(\bar{x}_t)\|^2] + \frac{15\gamma_z \eta}{\mu} L_J^2 \Delta_{k+1}^2 \mathbb{E}[\|\bar{y}_{k+1,t} - y_{k+1}^*(\bar{x}_t)\|^2]$$
$$+ \frac{15\gamma_z \eta}{\mu}(L_H^2 \Delta_k^2 + L_J^2 \Delta_{k+1}^2)\mathbb{E}[\|\bar{y}_{k,t} - y_k^*(\bar{x}_t)\|^2] + \frac{15\gamma_z \eta}{\mu} L_H^2 \Delta_k^2 \mathbb{E}[\|\bar{y}_{k-1,t} - y_k^*(\bar{x}_{t-1})\|^2]$$
$$+ \frac{12\gamma_z \eta}{\mu}\mathbb{E}[\|(M_{k,t}^z - V_{k,t})\mathbf{1}\frac{1}{N}\|^2] + \frac{45\gamma_z \eta}{\mu}(\Delta_k^2 L_H^2 + \Delta_{k+1}^2 L_J^2)\frac{1}{N}\mathbb{E}[\|Y_{k,t}(I - J)\|_F^2]$$
$$+ \frac{45\gamma_z \eta}{\mu}\Delta_k^2 L_H^2 \frac{1}{N}\mathbb{E}[\|Y_{k-1,t}(I - J)\|_F^2] + \frac{40\gamma_z \eta}{\mu}\Delta_{k+1}^2 L_J^2 \frac{1}{N}\mathbb{E}[\|Y_{k+1,t}(I - J)\|_F^2]$$
$$+ (\frac{45\gamma_z \eta}{\mu}L_g^2 + 3)\frac{1}{N}\mathbb{E}[\|Z_{k,t}(I - J)\|_F^2] + \frac{45\gamma_z \eta}{\mu}C_J^2 \frac{1}{N}\mathbb{E}[\|Z_{k+1,t}(I - J)\|_F^2]$$
$$+ 3\eta\gamma_z^2 \frac{1}{N}\mathbb{E}[\|R_{k,t}(I - J)\|_F^2] + \frac{9\gamma_x^2 \eta}{\gamma_z \mu}C_{z_k}^2 \mathbb{E}[\|\bar{m}_t^x\|^2] .$$

With the same approach, we prove the case for $k = K$ so that we omit its proof. $\square$

**Lemma D.4.** *(**Upper bound regarding momentum of z**) Assuming that Assumptions 3.1 through 3.5 hold, for any $k \in \{1, \cdots, K-1\}$, the following inequality is true:*

$$\mathbb{E}[\|\bar{m}_{k,t}^z\|^2] \leq 3\mathbb{E}[\|(M_{k,t}^z - V_{k,t})\mathbf{1}\frac{1}{N}\|^2] + 12L_g^2 \frac{1}{N}\mathbb{E}[\|Z_{k,t}(I - J)\|_F^2] + 12C_J^2 \frac{1}{N}\mathbb{E}[\|Z_{k+1,t}(I - J)\|_F^2]$$
$$+ 12(\Delta_k^2 L_H^2 + \Delta_{k+1}^2 L_J^2)\frac{1}{N}\mathbb{E}[\|Y_{k,t}(I - J)\|_F^2] + 12\Delta_k^2 L_H^2 \frac{1}{N}\mathbb{E}[\|Y_{k-1,t}(I - J)\|_F^2]$$
$$+ 12\Delta_{k+1}^2 L_J^2 \frac{1}{N}\mathbb{E}[\|Y_{k+1,t}(I - J)\|_F^2] + 12(\Delta_k^2 L_H^2 + \Delta_{k+1}^2 L_J^2)\mathbb{E}[\|\bar{y}_{k,t} - y_k^*(\bar{x}_t)\|^2]$$
$$+ 12\Delta_k^2 L_H^2 \mathbb{E}[\|\bar{y}_{k-1,t} - y_{k-1}^*(\bar{x}_t)\|^2] + 12\Delta_{k+1}^2 L_J^2 \mathbb{E}[\|\bar{y}_{k+1,t} - y_{k+1}^*(\bar{x}_t)\|^2]$$
$$+ 12L_g^2 \mathbb{E}[\|\bar{z}_{k,t} - z_k^*(\bar{x}_t)\|^2] + 12C_J^2 \mathbb{E}[\|\bar{z}_{k+1,t} - z_{k+1}^*(\bar{x}_t)\|^2] .$$

*Note that $\bar{y}_{0,t} - y_0^*(\bar{x}_t) = 0$. When $k = K$, the following inequality is true:*

$$\mathbb{E}[\|\bar{m}_{K,t}^z\|^2] \leq 3\mathbb{E}[\|(M_{K,t}^z - V_{K,t})\mathbf{1}\frac{1}{N}\|^2] + 12L_g^2 \frac{1}{N}\mathbb{E}[\|Z_{K,t}(I - J)\|_F^2] + 12L_f^2 \frac{1}{N}\mathbb{E}[\|X_t(I - J)\|_F^2]$$

$$+ 12(\Delta_K^2 L_H^2 + L_f^2)\frac{1}{N}\mathbb{E}[\|Y_{K,t}(I-J)\|_F^2] + 12\Delta_K^2 L_H^2 \frac{1}{N}\mathbb{E}[\|Y_{K-1,t}(I-J)\|_F^2]$$

$$+ 12(\Delta_K^2 L_H^2 + L_f^2)\mathbb{E}[\|\bar{y}_{K,t} - y_K^*(\bar{x}_t)\|^2] + 12\Delta_K^2 L_H^2 \mathbb{E}[\|\bar{y}_{K-1,t} - y_{K-1}^*(\bar{x}_t)\|^2] + 12L_g^2 \mathbb{E}[\|\bar{z}_{K,t} - z_K^*(\bar{x}_t)\|^2].$$

*Proof.* When $k \in \{1, \cdots, K-1\}$, we obtain

$$\frac{1}{N}\mathbb{E}[\|\bar{V}_k(X_t, \{Y_{k',t}\}_{k'=1}^K, \{Z_{k',t}\}_{k'=1}^K) - \bar{V}_k(\bar{X}_t, \{\bar{Y}_{k',t}\}_{k'=1}^K, \{\bar{Z}_{k',t}\}_{k'=1}^K)\|_F^2]$$

$$= \mathbb{E}[\|\frac{1}{N}\sum_{n=1}^N \nabla_{11}^2 f_k^{(n)}(y_{k,t}^{(n)}, y_{k-1,t}^{(n)})z_{k,t}^{(n)} - \frac{1}{N}\sum_{n=1}^N \nabla_{12}^2 f_{k+1}^{(n)}(y_{k+1,t}^{(n)}, y_{k,t}^{(n)})z_{k+1,t}^{(n)}$$

$$- \nabla_{11}^2 g_k(\bar{y}_{k,t}, \bar{y}_{k-1,t})\bar{z}_{k,t} + \nabla_{12}^2 g_{k+1}(\bar{y}_{k+1,t}, \bar{y}_{k,t})\bar{z}_{k+1,t}\|^2]$$

$$\leq 4\mathbb{E}[\|\frac{1}{N}\sum_{n=1}^N \nabla_{11}^2 f_k^{(n)}(y_{k,t}^{(n)}, y_{k-1,t}^{(n)})z_{k,t}^{(n)} - \frac{1}{N}\sum_{n=1}^N \nabla_{11}^2 f_k^{(n)}(y_{k,t}^{(n)}, y_{k-1,t}^{(n)})\bar{z}_{k,t}\|^2]$$

$$+ 4\mathbb{E}[\|\frac{1}{N}\sum_{n=1}^N \nabla_{11}^2 f_k^{(n)}(y_{k,t}^{(n)}, y_{k-1,t}^{(n)})\bar{z}_{k,t} - \nabla_{11}^2 g_k(\bar{y}_{k,t}, \bar{y}_{k-1,t})\bar{z}_{k,t}\|^2]$$

$$+ 4\mathbb{E}[\| - \frac{1}{N}\sum_{n=1}^N \nabla_{12}^2 f_{k+1}^{(n)}(y_{k+1,t}^{(n)}, y_{k,t}^{(n)})z_{k+1,t}^{(n)} + \frac{1}{N}\sum_{n=1}^N \nabla_{12}^2 f_{k+1}^{(n)}(y_{k+1,t}^{(n)}, y_{k,t}^{(n)})\bar{z}_{k+1,t}\|^2]$$

$$+ 4\mathbb{E}[\| - \frac{1}{N}\sum_{n=1}^N \nabla_{12}^2 f_{k+1}^{(n)}(y_{k+1,t}^{(n)}, y_{k,t}^{(n)})\bar{z}_{k+1,t} + \nabla_{12}^2 g_{k+1}(\bar{y}_{k+1,t}, \bar{y}_{k,t})\bar{z}_{k+1,t}\|^2]$$

$$\leq 4L_g^2 \frac{1}{N}\mathbb{E}[\|Z_{k,t}(I-J)\|_F^2] + 4C_J^2 \frac{1}{N}\mathbb{E}[\|Z_{k+1,t}(I-J)\|_F^2] + 4(\Delta_k^2 L_H^2 + \Delta_{k+1}^2 L_J^2)\frac{1}{N}\mathbb{E}[\|Y_{k,t}(I-J)\|_F^2]$$

$$+ 4\Delta_k^2 L_H^2 \frac{1}{N}\mathbb{E}[\|Y_{k-1,t}(I-J)\|_F^2] + 4\Delta_{k+1}^2 L_J^2 \frac{1}{N}\mathbb{E}[\|Y_{k+1,t}(I-J)\|_F^2]. \tag{29}$$

Moreover, when $k \in \{1, \cdots, K-1\}$, we derive

$$\frac{1}{N}\mathbb{E}[\|\bar{V}_k(\bar{X}_t, \{\bar{Y}_{k',t}\}_{k'=1}^K, \{\bar{Z}_{k',t}\}_{k'=1}^K)\|_F^2] = \mathbb{E}[\|\nabla_{11}^2 f_k(\bar{y}_{k,t}, \bar{y}_{k-1,t})\bar{z}_{k,t} - \nabla_{12}^2 f_{k+1}(\bar{y}_{k+1,t}, \bar{y}_{k,t})\bar{z}_{k+1,t}\|^2]$$

$$= \mathbb{E}[\|\nabla_{11}^2 f_k(\bar{y}_{k,t}, \bar{y}_{k-1,t})\bar{z}_{k,t} - \nabla_{12}^2 f_{k+1}(\bar{y}_{k+1,t}, \bar{y}_{k,t})\bar{z}_{k+1,t}$$

$$- \nabla_{11}^2 f_k(y_k^*(\bar{x}_t), y_{k-1}^*(\bar{x}_t))z_k^*(\bar{x}_t) + \nabla_{12}^2 f_{k+1}(y_{k+1}^*(\bar{x}_t), y_k^*(\bar{x}_t))z_{k+1}^*(\bar{x}_t)\|^2]$$

$$\leq 4\mathbb{E}[\|\nabla_{11}^2 f_k(\bar{y}_{k,t}, \bar{y}_{k-1,t})\bar{z}_{k,t} - \nabla_{11}^2 f_k(y_k^*(\bar{x}_t), y_{k-1}^*(\bar{x}_t))\bar{z}_{k,t}\|^2]$$

$$+ 4\mathbb{E}[\|\nabla_{11}^2 f_k(y_k^*(\bar{x}_t), y_{k-1}^*(\bar{x}_t))\bar{z}_{k,t} - \nabla_{11}^2 f_k(y_k^*(\bar{x}_t), y_{k-1}^*(\bar{x}_t))z_k^*(\bar{x}_t)\|^2]$$

$$+ 4\mathbb{E}[\| - \nabla_{12}^2 f_{k+1}(\bar{y}_{k+1,t}, \bar{y}_{k,t})\bar{z}_{k+1,t} + \nabla_{12}^2 f_{k+1}(y_{k+1}^*(\bar{x}_t), y_k^*(\bar{x}_t))\bar{z}_{k+1,t}\|^2]$$

$$+ 4\mathbb{E}[\| - \nabla_{12}^2 f_{k+1}(y_{k+1}^*(\bar{x}_t), y_k^*(\bar{x}_t))\bar{z}_{k+1,t} + \nabla_{12}^2 f_{k+1}(y_{k+1}^*(\bar{x}_t), y_k^*(\bar{x}_t))z_{k+1}^*(\bar{x}_t)\|^2]$$

$$\leq 4(\Delta_k^2 L_H^2 + \Delta_{k+1}^2 L_J^2)\mathbb{E}[\|\bar{y}_{k,t} - y_k^*(\bar{x}_t)\|^2] + 4\Delta_k^2 L_H^2 \mathbb{E}[\|\bar{y}_{k-1,t} - y_{k-1}^*(\bar{x}_t)\|^2]$$

$$+ 4\Delta_{k+1}^2 L_J^2 \mathbb{E}[\|\bar{y}_{k+1,t} - y_{k+1}^*(\bar{x}_t)\|^2] + 4L_g^2 \mathbb{E}[\|\bar{z}_{k,t} - z_k^*(\bar{x}_t)\|^2] + 4C_J^2 \mathbb{E}[\|\bar{z}_{k+1,t} - z_{k+1}^*(\bar{x}_t)\|^2], \tag{30}$$

where the second step follows from $\nabla_{11}^2 f_k(y_k^*(\bar{x}_t), y_{k-1}^*(\bar{x}_t))z_k^*(\bar{x}_t) - \nabla_{12}^2 f_{k+1}(y_{k+1}^*(\bar{x}_t), y_k^*(\bar{x}_t))z_{k+1}^*(\bar{x}_t) = 0$, the last step follows from Assumption 3.3 and the projection regarding $z$.

By combining Eq. (29) and Eq. (30), we derive

$$\mathbb{E}[\|\bar{m}_{k,t}^z\|^2] = \frac{1}{N}\mathbb{E}[\|\bar{M}_{k,t}^z\|_F^2]$$

$$\leq 3\frac{1}{N}\mathbb{E}[\|\bar{M}_{k,t}^z - \bar{V}_k(X_t, \{Y_{k',t}\}_{k'=1}^K, \{Z_{k',t}\}_{k'=1}^K)\|_F^2] + 3\frac{1}{N}\mathbb{E}[\|\bar{V}_k(\bar{X}_t, \{\bar{Y}_{k',t}\}_{k'=1}^K, \{\bar{Z}_{k',t}\}_{k'=1}^K)\|_F^2]$$

$$+ 3\frac{1}{N}\mathbb{E}[\|\bar{V}_k(X_t, \{Y_{k',t}\}_{k'=1}^K, \{Z_{k',t}\}_{k'=1}^K) - \bar{V}_k(\bar{X}_t, \{\bar{Y}_{k',t}\}_{k'=1}^K, \{\bar{Z}_{k',t}\}_{k'=1}^K)\|_F^2]$$

$$\leq 3\mathbb{E}[\|(M_{k,t}^z - V_{k,t})\mathbf{1}\frac{1}{N}\|^2] + 12L_g^2 \frac{1}{N}\mathbb{E}[\|Z_{k,t}(I-J)\|_F^2] + 12C_J^2 \frac{1}{N}\mathbb{E}[\|Z_{k+1,t}(I-J)\|_F^2]$$

$$+ 12(\Delta_k^2 L_H^2 + \Delta_{k+1}^2 L_J^2)\frac{1}{N}\mathbb{E}[\|Y_{k,t}(I-J)\|_F^2] + 12\Delta_k^2 L_H^2 \frac{1}{N}\mathbb{E}[\|Y_{k-1,t}(I-J)\|_F^2]$$

$$+ 12\Delta_{k+1}^2 L_J^2 \frac{1}{N}\mathbb{E}[\|Y_{k+1,t}(I-J)\|_F^2] + 12(\Delta_k^2 L_H^2 + \Delta_{k+1}^2 L_J^2)\mathbb{E}[\|\bar{y}_{k,t} - y_k^*(\bar{x}_t)\|^2]$$

$$+ 12\Delta_k^2 L_H^2 \mathbb{E}[\|\bar{y}_{k-1,t} - y_{k-1}^*(\bar{x}_t)\|^2] + 12\Delta_{k+1}^2 L_J^2 \mathbb{E}[\|\bar{y}_{k+1,t} - y_{k+1}^*(\bar{x}_t)\|^2]$$
$$+ 12L_g^2 \mathbb{E}[\|\bar{z}_{k,t} - z_k^*(\bar{x}_t)\|^2] + 12C_J^2 \mathbb{E}[\|\bar{z}_{k+1,t} - z_{k+1}^*(\bar{x}_t)\|^2].$$

Similarly, we prove the cases for $k = K$. Therefore, we omit it. □

## E. Gradient Estimation Errors

**Lemma E.1.** *(Global gradient estimation error regarding* x*) Assuming that Assumptions 3.1 through 3.5 hold, and $\alpha_x \eta^2 \in (0,1)$, the following inequality is true:*

$$\mathbb{E}[\|(G_{t+1} - M_{t+1}^x)\mathbf{1}\frac{1}{N}\|^2] \leq (1 - \alpha_x\eta^2)\mathbb{E}[\|(M_t^x - G_t)\mathbf{1}\frac{1}{N}\|^2] + 3(\Delta_1^2 L_J^2 + L_f^2)\frac{1}{N^2}\mathbb{E}[\|X_t - X_{t+1}\|_F^2]$$

$$+ 3C_J^2 \frac{1}{N^2}\mathbb{E}[\|Z_{1,t} - Z_{1,t+1}\|_F^2] + 3L_f^2 \frac{1}{N^2}\mathbb{E}[\|Y_{K,t} - Y_{K,t+1}\|_F^2] + 3\Delta_1^2 L_J^2 \frac{1}{N^2}\mathbb{E}[\|Y_{1,t} - Y_{1,t+1}\|_F^2] + 2\alpha_x^2\eta^4\frac{\hat{\sigma}_F^2}{N}.$$

*When $t = 0$, $\mathbb{E}[\|(M_0^x - G_0)\mathbf{1}\frac{1}{N}\|^2] \leq \frac{\hat{\sigma}_F^2}{NS_0}$, $S_0$ is the batch size for initialization.*

*Proof.* According to the definition of $m_{x,t+1}^{(n)}$, we derive

$$\mathbb{E}[\|(M_{t+1}^x - G_{t+1})\mathbf{1}\frac{1}{N}\|^2] = \mathbb{E}[\|((1 - \alpha_x\eta^2)(M_t^x - \tilde{G}_{t,t+1}) + \tilde{G}_{t+1,t+1} - G_{t+1})\mathbf{1}\frac{1}{N}\|^2]$$

$$= \mathbb{E}[\|((1 - \alpha_x\eta^2)(M_t^x - G_t) + (1 - \alpha_x\eta^2)(G_t - \tilde{G}_{t,t+1} + \tilde{G}_{t+1,t+1} - G_{t+1}) + \alpha_x\eta^2(\tilde{G}_{t+1,t+1} - G_{t+1}))\mathbf{1}\frac{1}{N}\|^2]$$

$$= \mathbb{E}[\|((1 - \alpha_x\eta^2)(M_t^x - G_t))\mathbf{1}\frac{1}{N}\|^2] + \frac{1}{N^2}\mathbb{E}[\|(1 - \alpha_x\eta^2)(G_t - \tilde{G}_{t,t+1} + \tilde{G}_{t+1,t+1} - G_{t+1}) + \alpha_x\eta^2(\tilde{G}_{t+1,t+1} - G_{t+1})\|_F^2]$$

$$\leq (1 - \alpha_x\eta^2)\mathbb{E}[\|(M_t^x - G_t)\mathbf{1}\frac{1}{N}\|^2] + 2\frac{1}{N^2}\mathbb{E}[\|\tilde{G}_{t+1,t+1} - \tilde{G}_{t,t+1}\|_F^2] + 2\alpha_x^2\eta^4\frac{1}{N^2}\mathbb{E}[\|\tilde{G}_{t+1,t+1} - G_{t+1}\|_F^2]$$

$$\leq (1 - \alpha_x\eta^2)\mathbb{E}[\|(M_t^x - G_t)\mathbf{1}\frac{1}{N}\|^2] + 3(\Delta_1^2 L_J^2 + L_f^2)\frac{1}{N^2}\mathbb{E}[\|X_t - X_{t+1}\|_F^2] + 3C_J^2 \frac{1}{N^2}\mathbb{E}[\|Z_{1,t} - Z_{1,t+1}\|_F^2]$$

$$+ 3L_f^2 \frac{1}{N^2}\mathbb{E}[\|Y_{K,t} - Y_{K,t+1}\|_F^2] + 3\Delta_1^2 L_J^2 \frac{1}{N^2}\mathbb{E}[\|Y_{1,t} - Y_{1,t+1}\|_F^2] + 2\alpha_x^2\eta^4\frac{\hat{\sigma}_F^2}{N},$$

where the third step follows from the fact $\mathbb{E}[(1 - \alpha_x\eta^2)(G_t - \tilde{G}_{t,t+1} + \tilde{G}_{t+1,t+1} - G_{t+1}) + \alpha_x\eta^2(\tilde{G}_{t+1,t+1} - G_{t+1})] = 0$, and the fact that the sampling operation on different workers is independent, the fourth step follows from $\alpha_x\eta^2 \in (0,1)$ and the fact $\mathbb{E}[\|a - \mathbb{E}[a]\|^2] \leq \mathbb{E}[\|a\|^2]$ for any random variable $a$, the last step follows from Lemma C.5 and Lemma C.7. When $t = 0$, it directly follows from Lemma C.5, with batch size as $S_0$. □

**Lemma E.2.** *(Local gradient estimation error regarding* x*) Given Assumptions 3.1-3.4 and $\alpha_x\eta^2 \in (0,1)$, the following inequality is true:*

$$\frac{1}{N}\mathbb{E}[\|M_{t+1}^x - G_{t+1}\|_F^2] \leq (1 - \alpha_x\eta^2)\frac{1}{N}\mathbb{E}[\|M_t^x - G_t\|_F^2] + 3(\Delta_1^2 L_J^2 + L_f^2)\frac{1}{N}\mathbb{E}[\|X_t - X_{t+1}\|_F^2]$$

$$+ 3C_J^2 \frac{1}{N}\mathbb{E}[\|Z_{1,t} - Z_{1,t+1}\|_F^2] + 3L_f^2 \frac{1}{N}\mathbb{E}[\|Y_{K,t} - Y_{K,t+1}\|_F^2] + 3\Delta_1^2 L_J^2 \frac{1}{N}\mathbb{E}[\|Y_{1,t} - Y_{1,t+1}\|_F^2] + 2\alpha_x^2\eta^4\hat{\sigma}_F^2.$$

*When $t = 0$, $\frac{1}{N}\mathbb{E}[\|M_0^x - G_0\|_F^2] \leq \frac{\hat{\sigma}_F^2}{S_0}$, $S_0$ is the batch size for initialization.*

This lemma can be proved as Lemma E.1. Therefore, we ignore the proof.

**Lemma E.3.** *(Global gradient estimation error regarding* y*) Assuming that Assumptions 3.1 through 3.5 hold, and $\alpha_y\eta^2 \in (0,1)$, for $k \in \{1, \cdots, K\}$, the following inequality is true:*

$$\mathbb{E}[\|(M_{k,t+1}^y - U_{k,t+1})\mathbf{1}\frac{1}{N}\|^2] \leq (1 - \alpha_y\eta^2)\mathbb{E}[\|(M_{k,t}^y - U_{k,t})\mathbf{1}\frac{1}{N}\|^2]$$

$$+ 2L_g^2 \frac{1}{N^2}\mathbb{E}[\|Y_{k,t} - Y_{k,t+1}\|_F^2] + 2L_g^2 \frac{1}{N^2}\mathbb{E}[\|Y_{k-1,t} - Y_{k-1,t+1}\|_F^2] + 2\alpha_y^2\eta^4\frac{\sigma^2}{N}.$$

*Note that $Y_{0,t} = X_t$. When $t = 0$, $\mathbb{E}[\|(M_{k,0}^y - U_{k,0})\mathbf{1}\frac{1}{N}\|^2] \leq \frac{\sigma^2}{NS_0}$.*

*Proof.* Following the proof of Lemma E.1, we derive

$$\mathbb{E}[\|(M_{k,t+1}^y - U_{k,t+1})\mathbf{1}\frac{1}{N}\|^2] \leq (1 - \alpha_y\eta^2)\mathbb{E}[\|(M_{k,t}^y - U_{k,t})\mathbf{1}\frac{1}{N}\|^2] + 2\frac{1}{N^2}\mathbb{E}[\|\tilde{U}_{k,t+1,t+1} - \tilde{U}_{k,t,t+1}\|_F^2]$$

$$+ 2\alpha_y^2\eta^4\frac{1}{N^2}\mathbb{E}[\|\tilde{U}_{k,t+1,t+1} - U_{k,t+1}\|_F^2]$$

$$\leq (1 - \alpha_y\eta^2)\mathbb{E}[\|(M_{k,t}^y - U_{k,t})\mathbf{1}\frac{1}{N}\|^2] + 2L_g^2\frac{1}{N^2}\mathbb{E}[\|Y_{k,t} - Y_{k,t+1}\|_F^2] + 2L_g^2\frac{1}{N^2}\mathbb{E}[\|Y_{k-1,t} - Y_{k-1,t+1}\|_F^2] + 2\alpha_y^2\eta^4\frac{\sigma^2}{N} .$$

□

**Lemma E.4.** *(**Local gradient estimation error regarding** y) Assuming that Assumptions 3.1 through 3.5 hold, and $\alpha_y\eta^2 \in (0, 1)$, for $k \in \{1, \cdots, K\}$, the following inequality is true:*

$$\frac{1}{N}\mathbb{E}[\|M_{k,t+1}^y - U_{k,t+1}\|_F^2] \leq (1 - \alpha_y\eta^2)\frac{1}{N}\mathbb{E}[\|M_{k,t}^y - U_{k,t}\|_F^2]$$

$$+ 2L_g^2\frac{1}{N}\mathbb{E}[\|Y_{k,t} - Y_{k,t+1}\|_F^2] + 2L_g^2\frac{1}{N}\mathbb{E}[\|Y_{k-1,t} - Y_{k-1,t+1}\|_F^2] + 2\alpha_y^2\eta^4\sigma^2 .$$

*Note that $Y_{0,t} = X_t$. When $t = 0$, $\frac{1}{N}\mathbb{E}[\|M_{k,0}^y - U_{k,0}\|_F^2] \leq \frac{\sigma^2}{S_0}$.*

This lemma can be proved as Lemma E.3. Therefore, we omit its proof.

**Lemma E.5.** *(**Global gradient estimation error regarding** z) Assuming that Assumptions 3.1 through 3.5 hold, and $\alpha_z\eta^2 \in (0, 1)$, when $k \in \{1, \cdots, K - 1\}$, the following inequality is true:*

$$\mathbb{E}[\|(M_{k,t+1}^z - V_{k,t+1})\mathbf{1}\frac{1}{N}\|^2] \leq (1 - \alpha_z\eta^2)\mathbb{E}[\|(M_{k,t}^z - V_{k,t})\mathbf{1}\frac{1}{N}\|^2] + 4\alpha_z^2\eta^4(\Delta_k^2 + \Delta_{k+1}^2)\frac{\sigma^2}{N}$$

$$+ 8L_g^2\frac{1}{N^2}\mathbb{E}[\|Z_{k,t+1} - Z_{k,t}\|_F^2] + 8C_J^2\frac{1}{N^2}\mathbb{E}[\|Z_{k+1,t+1} - Z_{k+1,t}\|_F^2] + 8\Delta_k^2 L_H^2\frac{1}{N^2}\mathbb{E}[\|Y_{k-1,t+1} - Y_{k-1,t}\|_F^2]$$

$$+ 8(\Delta_k^2 L_H^2 + \Delta_{k+1}^2 L_J^2)\frac{1}{N^2}\mathbb{E}[\|Y_{k,t+1} - Y_{k,t}\|_F^2] + 8\Delta_{k+1}^2 L_J^2\frac{1}{N^2}\mathbb{E}[\|Y_{k+1,t+1} - Y_{k+1,t}\|_F^2] ,$$

*When $t = 0$, $\mathbb{E}[\|(M_{k,0}^z - V_{k,0})\mathbf{1}\frac{1}{N}\|^2] \leq 2(\Delta_k^2 + \Delta_{k+1}^2)\frac{\sigma^2}{NS_0}$. When $k = K$, the following inequality is true:*

$$\mathbb{E}[\|(M_{K,t+1}^z - V_{K,t+1})\mathbf{1}\frac{1}{N}\|^2] \leq (1 - \alpha_z\eta^2)\mathbb{E}[\|(M_{K,t}^z - V_{K,t})\mathbf{1}\frac{1}{N}\|^2] + 4\alpha_z^2\eta^4(\Delta_K^2 + 1)\frac{\sigma^2}{N}$$

$$+ 8L_g^2\frac{1}{N^2}\mathbb{E}[\|Z_{K,t} - Z_{K,t+1}\|_F^2] + 8L_f^2\frac{1}{N^2}\mathbb{E}[\|X_{t+1} - X_t\|_F^2] + 8\Delta_K^2 L_H^2\frac{1}{N^2}\sum_{n=1}^{N}\mathbb{E}[\|Y_{K-1,t} - Y_{K-1,t+1}\|_F^2]$$

$$+ 8(\Delta_K^2 L_H^2 + L_f^2)\frac{1}{N^2}\mathbb{E}[\|Y_{K,t} - Y_{K,t+1}\|_F^2] .$$

*When $t = 0$, $\mathbb{E}[\|(M_{K,0}^z - V_{K,0})\mathbf{1}\frac{1}{N}\|^2] \leq 2(\Delta_K^2 + 1)\frac{\sigma^2}{NS_0}$.*

*Proof.* Following the proof of Lemma E.1, when $k \in \{1, \cdots, K - 1\}$, we derive

$$\mathbb{E}[\|(M_{k,t+1}^z - V_{k,t+1})\mathbf{1}\frac{1}{N}\|^2]$$

$$\leq (1 - \alpha_z\eta^2)\mathbb{E}[\|(M_{k,t}^z - V_{k,t})\mathbf{1}\frac{1}{N}\|^2] + 2\frac{1}{N^2}\mathbb{E}[\|\tilde{V}_{k,t+1,t+1} - \tilde{V}_{k,t,t+1}\|_F^2] + 2\alpha_z^2\eta^4\frac{1}{N^2}\mathbb{E}[\|\tilde{V}_{k,t+1,t+1} - V_{k,t+1}\|_F^2]$$

$$\leq (1 - \alpha_z\eta^2)\mathbb{E}[\|(M_{k,t}^z - V_{k,t})\mathbf{1}\frac{1}{N}\|^2] + 8L_g^2\frac{1}{N^2}\mathbb{E}[\|Z_{k,t+1} - Z_{k,t}\|_F^2] + 8C_J^2\frac{1}{N^2}\mathbb{E}[\|Z_{k+1,t+1} - Z_{k+1,t}\|_F^2]$$

$$+ 8\Delta_k^2 L_H^2\frac{1}{N^2}\mathbb{E}[\|Y_{k-1,t+1} - Y_{k-1,t}\|_F^2] + 8(\Delta_k^2 L_H^2 + \Delta_{k+1}^2 L_J^2)\frac{1}{N^2}\mathbb{E}[\|Y_{k,t+1} - Y_{k,t}\|_F^2]$$

$$+ 8\Delta_{k+1}^2 L_J^2\frac{1}{N^2}\mathbb{E}[\|Y_{k+1,t+1} - Y_{k+1,t}\|_F^2] + 4\alpha_z^2\eta^4(\Delta_k^2 + \Delta_{k+1}^2)\frac{\sigma^2}{N} ,$$

where the last step follows from Lemma C.9 and Lemma C.6.

Then, we use the same approach to prove the cases where $k = K$. Here, we omit the proof. □

**Lemma E.6.** *(Local gradient estimation error regarding* z*)* *Assuming that Assumptions 3.1 through 3.5 hold, and* $\alpha_z \eta^2 \in (0,1)$, *when* $k \in \{1, \cdots, K-1\}$, *the following inequality is true:*

$$\frac{1}{N}\mathbb{E}[\|M_{k,t+1}^z - V_{k,t+1}\|_F^2] \leq (1 - \alpha_z \eta^2)\frac{1}{N}\mathbb{E}[\|M_{k,t}^z - V_{k,t}\|_F^2] + 4\alpha_z^2 \eta^4 (\Delta_k^2 + \Delta_{k+1}^2)\sigma^2$$

$$+ 8L_g^2 \frac{1}{N}\mathbb{E}[\|Z_{k,t+1} - Z_{k,t}\|_F^2] + 8C_J^2 \frac{1}{N}\mathbb{E}[\|Z_{k+1,t+1} - Z_{k+1,t}\|_F^2] + 8\Delta_k^2 L_H^2 \frac{1}{N}\mathbb{E}[\|Y_{k-1,t+1} - Y_{k-1,t}\|_F^2]$$

$$+ 8(\Delta_k^2 L_H^2 + \Delta_{k+1}^2 L_J^2)\frac{1}{N}\mathbb{E}[\|Y_{k,t+1} - Y_{k,t}\|_F^2] + 8\Delta_{k+1}^2 L_J^2 \frac{1}{N}\mathbb{E}[\|Y_{k+1,t+1} - Y_{k+1,t}\|_F^2] \, .$$

*When* $t = 0$, $\frac{1}{N}\mathbb{E}[\|M_{k,0}^z - V_{k,0}\|_F^2] \leq 2(\Delta_k^2 + \Delta_{k+1}^2)\frac{\sigma^2}{S_0}$. *When* $k = K$, *the following inequality is true:*

$$\frac{1}{N}\mathbb{E}[\|M_{K,t+1}^z - V_{K,t+1}\|_F^2] \leq (1 - \alpha_z \eta^2)\frac{1}{N}\mathbb{E}[\|M_{K,t}^z - V_{K,t}\|_F^2] + 4\alpha_z^2 \eta^4 (\Delta_K^2 + 1)\sigma^2$$

$$+ 8L_g^2 \frac{1}{N}\mathbb{E}[\|Z_{K,t} - Z_{K,t+1}\|_F^2] + 8L_f^2 \frac{1}{N}\mathbb{E}[\|X_{t+1} - X_t\|_F^2] + 8\Delta_K^2 L_H^2 \frac{1}{N}\sum_{n=1}^{N}\mathbb{E}[\|Y_{K-1,t} - Y_{K-1,t+1}\|_F^2]$$

$$+ 8(\Delta_K^2 L_H^2 + L_f^2)\frac{1}{N}\mathbb{E}[\|Y_{K,t} - Y_{K,t+1}\|_F^2] \, .$$

*When* $t = 0$, $\frac{1}{N}\mathbb{E}[\|M_{K,0}^z - V_{K,0}\|_F^2] \leq 2(\Delta_K^2 + 1)\frac{\sigma^2}{S_0}$.

This lemma can be proved by following Lemma E.5. Thus, we omit the proof.

## F. Consensus and Feedback Errors

**Lemma F.1.** *(Zhao et al., 2022) Assuming that Assumption 3.5 holds, the spectral gap of* $I + \alpha(W - I)$ *is* $\alpha(1 - \lambda)$ *when* $\alpha \in (0,1)$.

Additionally, since $W$ is a double stochastic matrix, it is easy to know $(W - I)J = 0$ where $I$ is an identity matrix. Then, based on $\bar{X}_t = X_t J$ where $J = \frac{1}{N}\mathbf{1}\mathbf{1}^T$, we obtain

$$\bar{X}_{t+1} = \bar{X}_t - \gamma_x \eta \bar{P}_t \, , \ \bar{P}_{t+1} = \bar{P}_t + \bar{M}_{t+1}^x - \bar{M}_{x,t}^x \, ,$$
$$\bar{Y}_{k,t+1} = \bar{Y}_{k,t} - \gamma_y \eta \bar{Q}_{k,t} \, , \ \bar{Q}_{k,t+1} = \bar{Q}_{k,t} + \bar{M}_{k,t+1}^y - \bar{M}_{k,t}^y \, ,$$
$$\bar{Z}_{k,t+1} = \bar{Z}_{k,t} - \gamma_z \eta \bar{R}_{k,t} \, , \ \bar{R}_{k,t+1} = \bar{R}_{k,t} + \bar{M}_{k,t+1}^z - \bar{M}_{k,t}^z \, ,$$
$$\bar{P}_t = \bar{M}_t^x \, , \bar{Q}_{k,t} = \bar{M}_{k,t}^y \, , \bar{R}_{k,t} = \bar{M}_{k,t}^z \, . \tag{31}$$

**Lemma F.2.** *(Consensus error regarding variables) Assuming that Assumptions 3.1 through 3.6 hold, when* $\eta \leq \min\{\frac{1}{\rho_x}, \frac{1}{\rho_y}, \frac{1}{\rho_z}\}$, *when* $k \in \{1, \cdots, K\}$, *the following inequalities are true:*

$$\|X_{t+1}(I - J)\|_F^2 \leq (1 - \rho_x \eta(1 - \lambda))\|X_t(I - J)\|_F^2 + \frac{8\rho_x \eta}{(1 - \lambda)}\|\hat{X}_t - X_t\|_F^2 + \frac{2\gamma_x^2 \eta}{\rho_x(1 - \lambda)}\|P_t(I - J)\|_F^2 \, .$$

$$\|Y_{k,t+1}(I - J)\|_F^2 \leq (1 - \rho_y \eta(1 - \lambda))\|Y_{k,t}(I - J)\|_F^2 + \frac{8\rho_y \eta}{(1 - \lambda)}\|\hat{Y}_{k,t} - Y_{k,t}\|_F^2 + \frac{2\gamma_y^2 \eta}{\rho_y(1 - \lambda)}\|Q_{k,t}(I - J)\|_F^2 \, .$$

$$\|Z_{k,t+1}(I - J)\|_F^2 \leq (1 - \rho_z \eta(1 - \lambda))\|Z_{k,t}(I - J)\|_F^2 + \frac{8\rho_z \eta}{(1 - \lambda)}\|\hat{Z}_{k,t} - Z_{k,t}\|_F^2 + \frac{2\gamma_z^2 \eta}{\rho_z(1 - \lambda)}\|R_{k,t}(I - J)\|_F^2 \, .$$

*Proof.* From the update rule of $X_{t+1}$, we derive

$$\|X_{t+1}(I - J)\|_F^2 = \|(X_t - \gamma_x \eta P_t + \rho_x \eta \hat{X}_t(W - I))(I - J)\|_F^2$$
$$= \|X_t + \rho_x \eta X_t(W - I) - \rho_x \eta X_t(W - I) + \rho_x \eta \hat{X}_t(W - I) - \bar{X}_t + \gamma_x \eta \bar{P}_t - \gamma_x \eta P_t\|_F^2$$
$$= \|X_t(I + \rho_x \eta(W - I)) - \bar{X}_t + \rho_x \eta(\hat{X}_t - X_t)(W - I) + \gamma_x \eta(\bar{P}_t - P_t)\|_F^2$$
$$\leq (1 + a)\|X_t(I + \rho_x \eta(W - I)) - \bar{X}_t\|_F^2 + (1 + 1/a)\|\rho_x \eta(\hat{X}_t - X_t)(W - I) + \gamma_x \eta(\bar{P}_t - P_t)\|_F^2$$
$$\leq (1 + a)(1 - \rho_x \eta(1 - \lambda))^2\|X_t(I - J)\|_F^2 + 2(1 + 1/a)\rho_x^2 \eta^2\|(\hat{X}_t - X_t)(W - I)\|_F^2 + 2(1 + 1/a)\gamma_x^2 \eta^2\|P_t(I - J)\|_F^2$$
$$\leq (1 + a)(1 - \rho_x \eta(1 - \lambda))^2\|X_t(I - J)\|_F^2 + 8(1 + 1/a)\rho_x^2 \eta^2\|\hat{X}_t - X_t\|_F^2 + 2(1 + 1/a)\gamma_x^2 \eta^2\|P_t(I - J)\|_F^2$$

$$\leq (1 - \rho_x \eta(1-\lambda))\|X_t(I-J)\|_F^2 + 8\frac{\rho_x \eta}{(1-\lambda)}\|\hat{X}_t - X_t\|_F^2 + 2\frac{\gamma_x^2 \eta}{\rho_x(1-\lambda)}\|P_t(I-J)\|_F^2 \,,$$

where first inequality follows from Young's inequality, the second inequality holds due to Lemma F.1, the last step holds due to $a = \frac{\rho_x \eta(1-\lambda)}{1-\rho_x \eta(1-\lambda)}$ and $\rho_x \eta(1-\lambda) < 1$.

In addition, by denoting $Z_{k,t} = \mathcal{P}(\tilde{Z}_{k,t})$, we derive

$$\frac{1}{N}\mathbb{E}[\|Z_{k,t+1}(I-J)\|_F^2] = \frac{1}{N}\sum_{n=1}^{N}\mathbb{E}[\|\mathcal{P}(\tilde{z}_{k,t+1}^{(n)}) - \frac{1}{N}\sum_{m=1}^{N}\mathcal{P}(\tilde{z}_{k,t+1}^{(m)})\|^2]$$

$$= \frac{1}{N}\sum_{n=1}^{N}\mathbb{E}[\|\mathcal{P}(\tilde{z}_{k,t+1}^{(n)}) - \mathcal{P}(\frac{1}{N}\sum_{m=1}^{N}\tilde{z}_{k,t+1}^{(m)}) + \mathcal{P}(\frac{1}{N}\sum_{m=1}^{N}\tilde{z}_{k,t+1}^{(m)}) - \frac{1}{N}\sum_{m=1}^{N}\mathcal{P}(\tilde{z}_{k,t+1}^{(m)})\|^2]$$

$$= \underbrace{\frac{1}{N}\sum_{n=1}^{N}\mathbb{E}[\|\mathcal{P}(\tilde{z}_{k,t+1}^{(n)}) - \mathcal{P}(\frac{1}{N}\sum_{m=1}^{N}\tilde{z}_{k,t+1}^{(m)})\|^2]}_{T_1} + \underbrace{\mathbb{E}[\|\mathcal{P}(\frac{1}{N}\sum_{m=1}^{N}\tilde{z}_{k,t+1}^{(m)}) - \frac{1}{N}\sum_{m=1}^{N}\mathcal{P}(\tilde{z}_{k,t+1}^{(m)})\|^2]}_{T_2}$$

$$+ \underbrace{2\frac{1}{N}\sum_{n=1}^{N}\left\langle \mathcal{P}(\tilde{z}_{k,t+1}^{(n)}) - \mathcal{P}(\frac{1}{N}\sum_{m=1}^{N}\tilde{z}_{k,t+1}^{(m)}), \mathcal{P}(\frac{1}{N}\sum_{m=1}^{N}\tilde{z}_{k,t+1}^{(m)}) - \frac{1}{N}\sum_{m=1}^{N}\mathcal{P}(\tilde{z}_{k,t+1}^{(m)})\right\rangle}_{T_3} \,.$$

For $T_3$, we derive

$$T_3 = 2\left\langle \frac{1}{N}\sum_{n=1}^{N}\left(\mathcal{P}(\tilde{z}_{k,t+1}^{(n)}) - \mathcal{P}(\frac{1}{N}\sum_{m=1}^{N}\tilde{z}_{k,t+1}^{(m)})\right), \mathcal{P}(\frac{1}{N}\sum_{m=1}^{N}\tilde{z}_{k,t+1}^{(m)}) - \frac{1}{N}\sum_{m=1}^{N}\mathcal{P}(\tilde{z}_{k,t+1}^{(m)})\right\rangle$$

$$= -2\mathbb{E}[\|\mathcal{P}(\frac{1}{N}\sum_{m=1}^{N}\tilde{z}_{k,t+1}^{(m)}) - \frac{1}{N}\sum_{m=1}^{N}\mathcal{P}(\tilde{z}_{k,t+1}^{(m)})\|^2]\,.$$

Combining $T_1$, $T_2$, and $T_3$ yields

$$\frac{1}{N}\mathbb{E}[\|Z_{k,t+1}(I-J)\|_F^2] \leq \frac{1}{N}\sum_{n=1}^{N}\mathbb{E}[\|\mathcal{P}(\tilde{z}_{k,t+1}^{(n)}) - \mathcal{P}(\frac{1}{N}\sum_{m=1}^{N}\tilde{z}_{k,t+1}^{(m)})\|^2]$$

$$\leq \frac{1}{N}\sum_{n=1}^{N}\mathbb{E}[\|\tilde{z}_{k,t+1}^{(n)} - \frac{1}{N}\sum_{m=1}^{N}\tilde{z}_{k,t+1}^{(m)}\|^2] = \frac{1}{N}\mathbb{E}[\|\tilde{Z}_{k,t+1}(I-J)\|_F^2]\,,$$

where the second step follows from the non-expansiveness of the projection operation. $\qquad\square$

**Lemma F.3.** *(Consensus error regarding momentum)* *Assuming that Assumptions 3.1 through 3.6 hold, when $\beta_x < 1$, $\beta_y < 1$, $\beta_z < 1$, when $k \in \{1, \cdots, K\}$, the following inequalities are true:*

$$\|P_{t+1}(I-J)\|_F^2 \leq (1 - \beta_x(1-\lambda))\|P_t(I-J)\|_F^2 + \frac{8\beta_x}{(1-\lambda)}\|\hat{P}_t - P_t\|_F^2 + \frac{2}{\beta_x(1-\lambda)}\|M_{t+1}^x - M_t^x\|_F^2\,.$$

$$\|Q_{k,t+1}(I-J)\|_F^2 \leq (1 - \beta_y(1-\lambda))\|Q_{k,t}(I-J)\|_F^2 + \frac{8\beta_y}{(1-\lambda)}\|\hat{Q}_{k,t} - Q_{k,t}\|_F^2 + \frac{2}{\beta_y(1-\lambda)}\|M_{k,t+1}^y - M_{k,t}^y\|_F^2\,.$$

$$\|R_{k,t+1}(I-J)\|_F^2 \leq (1 - \beta_z(1-\lambda))\|R_{k,t}(I-J)\|_F^2 + \frac{8\beta_z}{(1-\lambda)}\|\hat{R}_{k,t} - R_{k,t}\|_F^2 + \frac{2}{\beta_z(1-\lambda)}\|M_{k,t+1}^z - M_{k,t}^z\|_F^2\,.$$

*When $t = 0$, the following inequalities are true:*

$$\frac{1}{N}\mathbb{E}[\|P_0(I-J)\|_F^2] \leq 12(1 + \Delta_1^2)\frac{\sigma^2}{S} + 24\Delta_1^2 C_J^2 + 24\frac{1}{N}\sum_{n=1}^{N}\mathbb{E}[\|\nabla_1 f_{K+1}^{(n)}(x_0, y_{K,0})\|^2]\,.$$

$$\frac{1}{N}\mathbb{E}[\|Q_{k,0}(I-J)\|_F^2] \leq \frac{6\sigma^2}{S} + 12\frac{1}{N}\sum_{n=1}^{N}\mathbb{E}[\|\nabla_1 f_k^{(n)}(y_{k,0}, y_{k-1,0})\|^2]\,.$$

$$\frac{1}{N}\mathbb{E}[\|R_{k,0}(I-J)\|_F^2] \leq 12(\Delta_k^2 + \Delta_{k+1}^2)\frac{\sigma^2}{S} + 24C_J^2(\Delta_k^2 + \Delta_{k+1}^2)\,.$$

*Proof.* From the update rule of $P_{t+1}$, we derive

$$\|P_{t+1}(I - J)\|_F^2 = \|(P_t + M_{t+1}^x - M_t^x + \beta_x \hat{P}_t(W - I))(I - J)\|_F^2$$

$$= \|P_t + \beta_x P_t(W - I) - \beta_x P_t(W - I) + \beta_x \hat{P}_t(W - I) - \bar{P}_t + M_{t+1}^x - M_t^x - \bar{M}_{t+1}^x + \bar{M}_t^x\|_F^2$$

$$= \|P_t(I + \beta_x(W - I)) - \bar{P}_t + \beta_x(\hat{P}_t - P_t)(W - I) + M_{t+1}^x - M_t^x - \bar{M}_{t+1}^x + \bar{M}_t^x\|_F^2$$

$$\leq (1 + a)\|P_t(I + \beta_x(W - I)) - \bar{P}_t\|_F^2 + 2(1 + 1/a)\|\beta_x(\hat{P}_t - P_t)(W - I)\|_F^2 + 2(1 + 1/a)\|M_{t+1}^x - M_t^x - \bar{M}_{t+1}^x + \bar{M}_t^x\|_F^2$$

$$\leq (1 + a)(1 - \beta_x(1 - \lambda))^2\|P_t(I - J)\|_F^2 + 8(1 + 1/a)\beta_x^2\|\hat{P}_t - P_t\|_F^2 + 2(1 + 1/a)\|M_{t+1}^x - M_t^x\|_F^2$$

$$\leq (1 - \beta_x(1 - \lambda))\|P_t(I - J)\|_F^2 + \frac{8\beta_x}{(1 - \lambda)}\|\hat{P}_t - P_t\|_F^2 + \frac{2}{\beta_x(1 - \lambda)}\|M_{t+1}^x - M_t^x\|_F^2 \,,$$

where the first inequality follows from Young's inequality, the second inequality holds due to Lemma F.1, the last step holds due to $a = \frac{\beta_x(1-\lambda)}{1-\beta_x(1-\lambda)}$ and $\beta_x < 1$. When $t = 0$, we derive

$$\frac{1}{N}\mathbb{E}[\|P_0(I - J)\|_F^2] \leq 2\frac{1}{N}\sum_{n=1}^N \mathbb{E}[\|\nabla_1 f_{K+1}^{(n)}(x_0, y_{K,0}; \xi_0^{(n)}) - \frac{1}{N}\sum_{n'=1}^N \nabla_1 f_{K+1}^{(n')}(x_0, y_{K,0}; \xi_0^{(n')})\|^2]$$

$$+ 2\frac{1}{N}\sum_{n=1}^N \mathbb{E}[\|(-1)^K \nabla_{12}^2 f_1^{(n)}(y_{1,0}, x_0; \zeta_{1,0}^{(n)})z_{1,0} - \frac{1}{N}\sum_{n'=1}^N (-1)^K \nabla_{12}^2 f_1^{(n')}(y_{1,0}, x_0; \zeta_{1,0}^{(n')})z_{1,0}\|^2]$$

$$\leq 6\frac{1}{N}\sum_{n=1}^N \mathbb{E}[\|\nabla_1 f_{K+1}^{(n)}(x_0, y_{K,0}; \xi_0^{(n)}) - \nabla_1 f_{K+1}^{(n)}(x_0, y_{K,0})\|^2]$$

$$+ 6\frac{1}{N}\sum_{n=1}^N \mathbb{E}[\|\nabla_1 f_{K+1}^{(n)}(x_0, y_{K,0}) - \frac{1}{N}\sum_{n'=1}^N \nabla_1 f_{K+1}^{(n')}(x_0, y_{K,0})\|^2]$$

$$+ 6\frac{1}{N}\sum_{n=1}^N \mathbb{E}[\|\frac{1}{N}\sum_{n'=1}^N \nabla_1 f_{K+1}^{(n')}(x_0, y_{K,0}) - \frac{1}{N}\sum_{n'=1}^N \nabla_1 f_{K+1}^{(n')}(x_0, y_{K,0}; \xi_0^{(n')})\|^2]$$

$$+ 6\frac{1}{N}\sum_{n=1}^N \mathbb{E}[\|(-1)^K \nabla_{12}^2 f_1^{(n)}(y_{1,0}, x_0; \zeta_{1,0}^{(n)})z_{1,0} - (-1)^K \nabla_{12}^2 f_1^{(n)}(y_{1,0}, x_0)z_{1,0}\|^2]$$

$$+ 6\frac{1}{N}\sum_{n=1}^N \mathbb{E}[\|(-1)^K \nabla_{12}^2 f_1^{(n)}(y_{1,0}, x_0)z_{1,0} - \frac{1}{N}\sum_{n'=1}^N (-1)^K \nabla_{12}^2 f_1^{(n')}(y_{1,0}, x_0)z_{1,0}\|^2]$$

$$+ 6\frac{1}{N}\sum_{n=1}^N \mathbb{E}[\|\frac{1}{N}\sum_{n'=1}^N (-1)^K \nabla_{12}^2 f_1^{(n')}(y_{1,0}, x_0)z_{1,0} - \frac{1}{N}\sum_{n'=1}^N (-1)^K \nabla_{12}^2 f_1^{(n')}(y_{1,0}, x_0; \zeta_{1,0}^{(n')})z_{1,0}\|^2]$$

$$\leq 12(1 + \Delta_1^2)\frac{\sigma^2}{S} + 24\Delta_1^2 C_J^2 + 24\frac{1}{N}\sum_{n=1}^N \mathbb{E}[\|\nabla_1 f_{K+1}^{(n)}(x_0, y_{K,0})\|^2] \,. \tag{32}$$

$\square$

**Lemma F.4.** *(Feedback error regarding variables) Assuming that Assumptions 3.1 through 3.6 and $\{\rho_x, \rho_y, \rho_z\} \leq \frac{\delta}{8\sqrt{3}}$ hold, the following inequalities are true:*

$$\|\hat{X}_{t+1} - X_{t+1}\|_F^2 \leq (1 - \frac{\delta}{4})\|\hat{X}_t - X_t\|_F^2 + \frac{12\gamma_x^2\eta^2}{\delta}\|P_t(I - J)\|_F^2 + \frac{12\gamma_x^2\eta^2}{\delta}N\|\bar{m}_t^x\|^2 + \frac{48\rho_x^2\eta^2}{\delta}\|X_t(I - J)\|_F^2 \,.$$

$$\|\hat{Y}_{k,t+1} - Y_{k,t+1}\|_F^2 \leq (1 - \frac{\delta}{4})\|\hat{Y}_{k,t} - Y_{k,t}\|_F^2 + \frac{12\gamma_y^2\eta^2}{\delta}\|Q_{k,t}(I - J)\|_F^2 + \frac{12\gamma_y^2\eta^2}{\delta}N\|\bar{m}_{k,t}^y\|^2 + \frac{48\rho_y^2\eta^2}{\delta}\|Y_{k,t}(I - J)\|_F^2 \,.$$

$$\|\hat{Z}_{k,t+1} - Z_{k,t+1}\|_F^2 \leq (1 - \frac{\delta}{4})\|\hat{Z}_{k,t} - Z_{k,t}\|_F^2 + \frac{12\gamma_z^2\eta^2}{\delta}\|R_{k,t}(I - J)\|_F^2 + \frac{12\gamma_z^2\eta^2}{\delta}N\|\bar{m}_{k,t}^z\|^2 + \frac{48\rho_z^2\eta^2}{\delta}\|Z_{k,t}(I - J)\|_F^2 \,.$$

*Proof.* From the update rule of $\hat{X}_{t+1}$, we derive

$$\|\hat{X}_{t+1} - X_{t+1}\|_F^2 = \|\hat{X}_t + \mathcal{C}(X_{t+1} - \hat{X}_t) - X_{t+1}\|_F^2$$

$$\leq (1 - \delta)\|X_{t+1} - \hat{X}_t\|_F^2$$

$$\leq (1 - \delta)(1 + a)\|X_t - \hat{X}_t\|_F^2 + (1 - \delta)(1 + 1/a)\|X_{t+1} - X_t\|_F^2$$

$$\leq (1 - \frac{\delta}{2})\|\hat{X}_t - X_t\|_F^2 + \frac{3}{\delta}\|X_{t+1} - X_t\|_F^2 \, ,$$

$$\leq (1 - \frac{\delta}{2})\|\hat{X}_t - X_t\|_F^2 + \frac{3}{\delta}\Big(4\gamma_x^2\eta^2\|P_t(I-J)\|_F^2 + 4N\gamma_x^2\eta^2\|\bar{m}_t^x\|^2 + 16\rho_x^2\eta^2\|\hat{X}_t - X_t\|_F^2 + 16\rho_x^2\eta^2\|X_t(I-J)\|_F^2\Big) \, ,$$

$$\leq (1 - \frac{\delta}{4})\|\hat{X}_t - X_t\|_F^2 + \frac{12\gamma_x^2\eta^2}{\delta}\|P_t(I-J)\|_F^2 + \frac{12\gamma_x^2\eta^2}{\delta}N\|\bar{m}_t^x\|^2 + \frac{48\rho_x^2\eta^2}{\delta}\|X_t(I-J)\|_F^2 \, ,$$

where the second step holds due to Assumption 3.6, $a = \frac{\delta}{2}$, the second to last step follows from Lemma F.6 and the last step follows from $\rho_x \leq \frac{\delta}{8\sqrt{3}}$. □

**Lemma F.5.** *(Feedback error regarding momentum)* *Assuming that Assumptions 3.1 through 3.6 hold and* $\{\beta_x, \beta_y, \beta_z \leq \frac{\delta}{12}$, *the following inequalities are true:*

$$\|\hat{P}_{t+1} - P_{t+1}\|_F^2 \leq (1 - \frac{\delta}{4})\|P_t - \hat{P}_t\|_F^2 + \frac{9}{\delta}\|M_{t+1}^x - M_t^x\|_F^2 + \frac{36\beta_x^2}{\delta}\|P_t(I-J)\|_F^2 \, .$$

$$\|\hat{Q}_{k,t+1} - Q_{k,t+1}\|_F^2 \leq (1 - \frac{\delta}{4})\|Q_{k,t} - \hat{Q}_{k,t}\|_F^2 + \frac{9}{\delta}\|M_{k,t+1}^y - M_{k,t}^y\|_F^2 + \frac{36\beta_y^2}{\delta}\|Q_{k,t}(I-J)\|_F^2 \, .$$

$$\|\hat{R}_{k,t+1} - R_{k,t+1}\|_F^2 \leq (1 - \frac{\delta}{4})\|R_{k,t} - \hat{R}_{k,t}\|_F^2 + \frac{9}{\delta}\|M_{k,t+1}^z - M_{k,t}^z\|_F^2 + \frac{36\beta_z^2}{\delta}\|R_{k,t}(I-J)\|_F^2 \, .$$

*When* $t = 0$, *the following inequalities are true:*

$$\frac{1}{N}\mathbb{E}[\|P_0 - \hat{P}_0\|_F^2] \leq 2(1 + \Delta_1^2)\frac{\sigma^2}{S} + 4\Delta_1^2 C_J^2 + 4\frac{1}{N}\sum_{n=1}^{N}\mathbb{E}[\|\nabla_1 f_{K+1}^{(n)}(x_0, y_{K,0})\|^2] \, .$$

$$\frac{1}{N}\mathbb{E}[\|Q_{k,0} - \hat{Q}_{k,0}\|_F^2] \leq \frac{\sigma^2}{S} + 2\frac{1}{N}\sum_{n=1}^{N}\mathbb{E}[\|\nabla_1 f_k^{(n)}(y_{k,0}, y_{k-1,0})\|^2] \, .$$

$$\frac{1}{N}\mathbb{E}[\|R_{k,0} - \hat{R}_{k,0}\|_F^2] \leq 2(\Delta_k^2 + \Delta_{k+1}^2)\frac{\sigma^2}{S} + 4C_J^2(\Delta_k^2 + \Delta_{k+1}^2) \, .$$

*Proof.* From the update rule of $\hat{P}_{t+1}$, we derive

$$\|\hat{P}_{t+1} - P_{t+1}\|_F^2 = \|\hat{P}_t + \mathcal{C}(P_{t+1} - \hat{P}_t) - P_{t+1}\|_F^2$$
$$\leq (1 - \delta)\|P_{t+1} - \hat{P}_t\|_F^2$$
$$\leq (1 - \delta)(1 + a)\|P_t - \hat{P}_t\|_F^2 + (1 - \delta)(1 + 1/a)\|P_{t+1} - P_t\|_F^2$$
$$\leq (1 - \frac{\delta}{2})\|P_t - \hat{P}_t\|_F^2 + \frac{3}{\delta}\|P_{t+1} - P_t\|_F^2$$
$$\leq (1 - \frac{\delta}{2})\|P_t - \hat{P}_t\|_F^2 + \frac{3}{\delta}\Big(3\|M_{t+1}^x - M_t^x\|_F^2 + 12\beta_x^2\|\hat{P}_t - P_t\|_F^2 + 12\beta_x^2\|P_t(I-J)\|_F^2\Big)$$
$$\leq (1 - \frac{\delta}{4})\|P_t - \hat{P}_t\|_F^2 + \frac{9}{\delta}\|M_{t+1}^x - M_t^x\|_F^2 + \frac{36\beta_x^2}{\delta}\|P_t(I-J)\|_F^2 \, ,$$

where the second step holds due to Assumption 3.6, $a = \frac{\delta}{2}$, the second to last step follows from Lemma F.7 and the last step follows from $\beta_x \leq \frac{\delta}{12}$. When $t = 0$, it can be handled in the same manner as in Eq. (32). □

**Lemma F.6.** *(Descent rule regarding variables)* *Assuming that Assumptions 3.1 through 3.6 hold, when* $k \in \{1, \cdots, K\}$, *the following inequalities are true:*

$$\|X_{t+1} - X_t\|_F^2 \leq 4\gamma_x^2\eta^2\|P_t(I-J)\|_F^2 + 4N\gamma_x^2\eta^2\|\bar{m}_t^x\|^2 + 16\rho_x^2\eta^2\|\hat{X}_t - X_t\|_F^2 + 16\rho_x^2\eta^2\|X_t(I-J)\|_F^2 \, .$$
$$\|Y_{k,t+1} - Y_{k,t}\|_F^2 \leq 4\gamma_y^2\eta^2\|Q_{k,t}(I-J)\|_F^2 + 4N\gamma_y^2\eta^2\|\bar{m}_{k,t}^y\|^2 + 16\rho_y^2\eta^2\|\hat{Y}_{k,t} - Y_{k,t}\|_F^2 + 16\rho_y^2\eta^2\|Y_{k,t}(I-J)\|_F^2 \, .$$
$$\|Z_{k,t+1} - Z_{k,t}\|_F^2 \leq 4\gamma_z^2\eta^2\|R_{k,t}(I-J)\|_F^2 + 4N\gamma_z^2\eta^2\|\bar{m}_{k,t}^z\|^2 + 16\rho_z^2\eta^2\|\hat{Z}_{k,t} - Z_{k,t}\|_F^2 + 16\rho_z^2\eta^2\|Z_{k,t}(I-J)\|_F^2 \, .$$

*Proof.* From the update rule of $X_{t+1}$, we derive

$$\|X_{t+1} - X_t\|_F^2 = \|X_t - \gamma_x\eta P_t + \rho_x\eta\hat{X}_t(W - I) - X_t\|_F^2$$
$$= \| - \gamma_x\eta(P_t - \bar{P}_t + \bar{P}_t) + \rho_x\eta(\hat{X}_t - X_t + X_t - \bar{X}_t)(W - I)\|_F^2$$
$$\leq 4\gamma_x^2\eta^2\|P_t(I-J)\|_F^2 + 4N\gamma_x^2\eta^2\|\bar{m}_t^x\|^2 + 16\rho_x^2\eta^2\|\hat{X}_t - X_t\|_F^2 + 16\rho_x^2\eta^2\|X_t(I-J)\|_F^2 \, ,$$

where the last step follows from $\|(\hat{X}_t - X_t)(W - I)\|_F^2 \leq \|W - I\|_2^2\|\hat{X}_t - X_t\|_F^2 \leq 4\|\hat{X}_t - X_t\|_F^2$. □

**Lemma F.7.** *(Descent rule regarding momentum)* *Assuming that Assumptions 3.1 through 3.6 hold, when $k \in \{1, \cdots, K\}$, the following inequalities are true:*

$$\|P_{t+1} - P_t\|_F^2 \le 3\|M_{t+1}^x - M_t^x\|_F^2 + 12\beta_x^2\|\hat{P}_t - P_t\|_F^2 + 12\beta_x^2\|P_t(I - J)\|_F^2 .$$

$$\|Q_{k,t+1} - Q_{k,t}\|_F^2 \le 3\|M_{k,t+1}^y - M_{k,t}^y\|_F^2 + 12\beta_y^2\|\hat{Q}_{k,t} - Q_{k,t}\|_F^2 + 12\beta_y^2\|Q_{k,t}(I - J)\|_F^2 .$$

$$\|R_{k,t+1} - R_{k,t}\|_F^2 \le 3\|M_{k,t+1}^z - M_{k,t}^z\|_F^2 + 12\beta_z^2\|\hat{R}_{k,t} - R_{k,t}\|_F^2 + 12\beta_z^2\|R_{k,t}(I - J)\|_F^2 .$$

*Proof.* From the update rule of $P_{t+1}$, we derive

$$
\begin{aligned}
\|P_{t+1} - P_t\|_F^2 &= \|P_t + M_{t+1}^x - M_t^x + \beta_x \hat{P}_t(W - I) - P_t\|_F^2 \\
&= \|M_{t+1}^x - M_t^x + \beta_x \hat{P}_t(W - I)\|_F^2 \\
&= \|M_{t+1}^x - M_t^x + \beta_x(\hat{P}_t - P_t)(W - I) + \beta_x P_t(W - I)\|_F^2 \\
&= \|M_{t+1}^x - M_t^x + \beta_x(\hat{P}_t - P_t)(W - I) + \beta_x(P_t - \bar{P}_t)(W - I)\|_F^2 \\
&\le 3\|M_{t+1}^x - M_t^x\|_F^2 + 12\beta_x^2\|\hat{P}_t - P_t\|_F^2 + 12\beta_x^2\|P_t(I - J)\|_F^2 .
\end{aligned}
$$

$\square$

# G. Proof of Theorem 5.1

*Proof.* To establish the convergence rate, we introduce the following potential function:

$$
\begin{aligned}
\mathcal{L}_{t+1} = {}& \underbrace{\mathbb{E}[F_{K+1}(\bar{x}_{t+1})]}_{\text{Lemma D.1}} + \sum_{k=1}^K a_k \underbrace{\mathbb{E}[\|\bar{y}_{k,t+1} - y_k^*(\bar{x}_{t+1})\|^2]}_{\text{Lemma D.2}} + \sum_{k=1}^K b_k \underbrace{\mathbb{E}[\|\bar{z}_{k,t+1} - z_k^*(\bar{x}_{t+1})\|^2]}_{\text{Lemma D.3}} \\
&+ \sum_{k=1}^K c_k \underbrace{\mathbb{E}[\|(M_{k,t+1}^y - U_{k,t+1})\mathbf{1}\tfrac{1}{N}\|^2]}_{\text{Lemma E.3}} + \sum_{k=1}^K d_k \underbrace{\mathbb{E}[\|(M_{k,t+1}^z - V_{k,t+1})\mathbf{1}\tfrac{1}{N}\|^2]}_{\text{Lemma E.5}} + e \underbrace{\mathbb{E}[\|(M_{t+1}^x - G_{t+1})\mathbf{1}\tfrac{1}{N}\|^2]}_{\text{Lemma E.1}} \\
&+ \sum_{k=1}^K \tilde{c}_k \underbrace{\tfrac{1}{N}\mathbb{E}[\|M_{k,t+1}^y - U_{k,t+1}\|_F^2]}_{\text{Lemma E.4}} + \sum_{k=1}^K \tilde{d}_k \underbrace{\tfrac{1}{N}\mathbb{E}[\|M_{k,t+1}^z - V_{k,t+1}\|_F^2]}_{\text{Lemma E.6}} + \tilde{e} \underbrace{\tfrac{1}{N}\mathbb{E}[\|M_{t+1}^x - G_{t+1}\|_F^2]}_{\text{Lemma E.2}} \\
&+ \underbrace{\omega \tfrac{1}{N}\mathbb{E}[\|X_{t+1}(I - J)\|_F^2] + \sum_{k=1}^K \nu_k \tfrac{1}{N}\mathbb{E}[\|Y_{k,t+1}(I - J)\|_F^2] + \sum_{k=1}^K \tau_k \tfrac{1}{N}\mathbb{E}[\|Z_{k,t+1}(I - J)\|_F^2]}_{\text{Lemma F.2}} \\
&+ \underbrace{\tilde{\omega} \tfrac{1}{N}\mathbb{E}[\|P_{t+1}(I - J)\|_F^2] + \sum_{k=1}^K \tilde{\nu}_k \tfrac{1}{N}\mathbb{E}[\|Q_{k,t+1}(I - J)\|_F^2] + \sum_{k=1}^K \tilde{\tau}_k \tfrac{1}{N}\mathbb{E}[\|R_{k,t+1}(I - J)\|_F^2]}_{\text{Lemma F.3}} \\
&+ \underbrace{\theta \tfrac{1}{N}\mathbb{E}[\|X_{t+1} - \hat{X}_{t+1}\|_F^2] + \sum_{k=1}^K \phi_k \tfrac{1}{N}\mathbb{E}[\|Y_{k,t+1} - \hat{Y}_{k,t+1}\|_F^2] + \sum_{k=1}^K \psi_k \tfrac{1}{N}\mathbb{E}[\|Z_{k,t+1} - \hat{Z}_{k,t+1}\|_F^2]}_{\text{Lemma F.4}} \\
&+ \underbrace{\tilde{\theta} \tfrac{1}{N}\mathbb{E}[\|P_{t+1} - \hat{P}_{t+1}\|_F^2] + \sum_{k=1}^K \tilde{\phi}_k \tfrac{1}{N}\mathbb{E}[\|Q_{k,t+1} - \hat{Q}_{k,t+1}\|_F^2] + \sum_{k=1}^K \tilde{\psi}_k \tfrac{1}{N}\mathbb{E}[\|R_{k,t+1} - \hat{R}_{k,t+1}\|_F^2]}_{\text{Lemma F.5}} .
\end{aligned}
\tag{33}
$$

Applying above Lemmas, with descent rule regarding variables in Lemma F.6, descent rule regarding momentum in Lemma F.7, and upper bound regarding momentum of $z$ in Lemma D.4, we obtain the upper-bound of $\mathcal{L}_{t+1} - \mathcal{L}_t$.

The right-hand side can be expressed as the sum of six terms, which we bound separately: estimation errors ($\mathcal{F}_1$), other errors related to $\mathbf{z}$ ($\mathcal{F}_2$), optimization errors ($\mathcal{F}_3$), other errors related to $\mathbf{x}$ ($\mathcal{F}_4$) and $\mathbf{y}$ ($\mathcal{F}_5$), together with the momentum term ($\mathcal{F}_6$).

Thus, we derive

$$\mathcal{L}_{t+1} - \mathcal{L}_t \le -\frac{\eta\gamma_x}{2}\mathbb{E}[\|\nabla F_{K+1}(\bar{x}_t)\|^2] + \mathcal{F}_0 + \mathcal{F}_1 + \mathcal{F}_2 + \mathcal{F}_3 + \mathcal{F}_4 + \mathcal{F}_5 + \mathcal{F}_6 ,$$

where $\mathcal{F}_0$ is the term denoted by

$$
\begin{aligned}
\mathcal{F}_0 &= 2\alpha_y^2\eta^4\frac{\sigma^2}{N}\sum_{k=1}^{K}c_k + \sum_{k=1}^{K-1}d_k4\alpha_z^2\eta^4(\Delta_k^2+\Delta_{k+1}^2)\frac{\sigma^2}{N} + 4d_K\alpha_z^2\eta^4(\Delta_K^2+1)\frac{\sigma^2}{N} + 2e\alpha_x^2\eta^4\frac{\hat{\sigma}_F^2}{N} + 2\alpha_y^2\eta^4\sigma^2\sum_{k=1}^{K}\tilde{c}_k \\
&\quad + \sum_{k=1}^{K-1}\tilde{d}_k4\alpha_z^2\eta^4(\Delta_k^2+\Delta_{k+1}^2)\sigma^2 + 4\tilde{d}_K\alpha_z^2\eta^4(\Delta_K^2+1)\sigma^2 + 2\tilde{e}\alpha_x^2\eta^4\hat{\sigma}_F^2 + \tilde{\omega}\frac{8\alpha_x^2\eta^4}{\beta_x(1-\lambda)}\hat{\sigma}_F^2 + \sum_{k=1}^{K}\tilde{\nu}_k\frac{8\alpha_y^2\eta^4}{\beta_y(1-\lambda)}\sigma^2 \\
&\quad + \sum_{k=1}^{K-1}\tilde{\tau}_k\frac{16\alpha_z^2\eta^4}{\beta_z(1-\lambda)}(\Delta_k^2+\Delta_{k+1}^2)\sigma^2 + \tilde{\tau}_K\frac{16\alpha_z^2\eta^4}{\beta_z(1-\lambda)}(\Delta_K^2+1)\sigma^2 + \tilde{\theta}\frac{36\alpha_x^2\eta^4}{\delta}\hat{\sigma}_F^2 + \sum_{k=1}^{K}\tilde{\phi}_k\frac{36\alpha_y^2\eta^4}{\delta}\sigma^2 \\
&\quad + \sum_{k=1}^{K-1}\tilde{\psi}_k\frac{72\alpha_z^2\eta^4}{\delta}(\Delta_k^2+\Delta_{k+1}^2)\sigma^2 + \tilde{\psi}_K\frac{72\alpha_z^2\eta^4}{\delta}(\Delta_K^2+1)\sigma^2 \,.
\end{aligned}
\tag{34}
$$

We next bound these terms one by one.

## G.1. Estimation errors ($\mathcal{F}_1$)

For $\mathcal{F}_1$, it consists of three types of errors: the momentum feedback error, the local estimation error, and the global estimation error. Specifically, we have

$$
\begin{aligned}
\mathcal{F}_1 &= \left(\tilde{\omega}\frac{8\beta_x}{(1-\lambda)}-\tilde{\theta}\frac{\delta}{4}\right)\frac{1}{N}\mathbb{E}[\|\hat{P}_t-P_t\|_F^2] + \sum_{k=1}^{K}\left(\tilde{\nu}_k\frac{8\beta_y}{(1-\lambda)}-\tilde{\phi}_k\frac{\delta}{4}\right)\frac{1}{N}\mathbb{E}[\|\hat{Q}_{k,t}-Q_{k,t}\|_F^2] \\
&\quad + \sum_{k=1}^{K}\left(\tilde{\tau}_k\frac{8\beta_z}{(1-\lambda)}-\tilde{\psi}_k\frac{\delta}{4}\right)\frac{1}{N}\mathbb{E}[\|\hat{R}_{k,t}-R_{k,t}\|_F^2] + \left(\tilde{\omega}\frac{8\alpha_x^2\eta^4}{\beta_x(1-\lambda)}+\tilde{\theta}\frac{36\alpha_x^2\eta^4}{\delta}-\alpha_x\eta^2\tilde{e}\right)\frac{1}{N}\mathbb{E}[\|M_t^x-G_t\|_F^2] \\
&\quad + \left(\eta\gamma_x-\alpha_x\eta^2 e\right)\mathbb{E}[\|(M_t^x-G_t)\mathbf{1}\frac{1}{N}\|^2] + \sum_{k=1}^{K}\left(\tilde{\nu}_k\frac{8\alpha_y^2\eta^4}{\beta_y(1-\lambda)}+\tilde{\phi}_k\frac{36\alpha_y^2\eta^4}{\delta}-\tilde{c}_k\alpha_y\eta^2\right)\frac{1}{N}\mathbb{E}[\|M_{k,t}^y-U_{k,t}\|_F^2] \\
&\quad + \sum_{k=1}^{K}\left(a_k\frac{20\gamma_y\eta}{\mu}-c_k\alpha_y\eta^2\right)\mathbb{E}[\|(M_{k,t}^y-U_{k,t})\mathbf{1}\frac{1}{N}\|^2] + \sum_{k=1}^{K}\left(\tilde{\tau}_k\frac{8\alpha_z^2\eta^4}{\beta_z(1-\lambda)}+\tilde{\psi}_k\frac{36\alpha_z^2\eta^4}{\delta}-\tilde{d}_k\alpha_z\eta^2\right)\frac{1}{N}\mathbb{E}[\|M_{k,t}^z-V_{k,t}\|_F^2] \\
&\quad + \sum_{k=1}^{K}\left(b_k\frac{12\gamma_z\eta}{\mu}+\psi_k\frac{36\gamma_z^2\eta^2}{\delta}+12\gamma_z^2\eta^2\mathcal{Z}_k-d_k\alpha_z\eta^2\right)\mathbb{E}[\|(M_{k,t}^z-V_{k,t})\mathbf{1}\frac{1}{N}\|^2]\,,
\end{aligned}
\tag{35}
$$

where

$$
\begin{aligned}
\mathcal{Z}_1 &= \frac{8L_g^2 d_1}{N} + \frac{3eC_J^2}{N} + 8L_g^2\tilde{d}_1 + 3\tilde{e}C_J^2 + \frac{12\tilde{\omega}C_J^2}{\beta_x(1-\lambda)} + \frac{16L_g^2\tilde{\tau}_1}{\beta_z(1-\lambda)} + \tilde{\theta}\frac{54C_J^2}{\delta} + \frac{72L_g^2\tilde{\psi}_1}{\delta}\,, \\
\mathcal{Z}_k &= \frac{8L_g^2 d_k}{N} + \frac{8C_J^2 d_{k-1}}{N} + 8L_g^2\tilde{d}_k + 8C_J^2\tilde{d}_{k-1} + \frac{16L_g^2\tilde{\tau}_k}{\beta_z(1-\lambda)} + \frac{16C_J^2\tilde{\tau}_{k-1}}{\beta_z(1-\lambda)} + \frac{72L_g^2\tilde{\psi}_k}{\delta} + \frac{72C_J^2\tilde{\psi}_{k-1}}{\delta}\,, k\in\{2,\cdots,K\}.
\end{aligned}
$$

### (1.1) For the coefficient of feedback errors:

We set $\tilde{\theta} = \tilde{\omega}\frac{32\beta_x}{\delta(1-\lambda)}$ such that the coefficient of $\frac{1}{N}\mathbb{E}[\|\hat{P}_t-P_t\|_F^2]$ is zero, i.e., $\tilde{\omega}\frac{8\beta_x}{(1-\lambda)}-\tilde{\theta}\frac{\delta}{4}=0$. Similarly, we set $\tilde{\phi}_k = \tilde{\nu}_k\frac{32\beta_y}{\delta(1-\lambda)}$ and $\tilde{\psi}_k = \tilde{\tau}_k\frac{32\beta_z}{\delta(1-\lambda)}$, such that the coefficients of $\frac{1}{N}\mathbb{E}[\|\hat{Q}_{k,t}-Q_{k,t}\|_F^2]$ and $\frac{1}{N}\mathbb{E}[\|\hat{R}_{k,t}-R_{k,t}\|_F^2]$ are zero for any $k\in\{1,\cdots,K\}$.

### (1.2) For the local estimation errors:

We set $\tilde{e} = \frac{16\gamma_x}{\beta_x^2(1-\lambda)^2}\frac{3C_J^2}{8L_g^2}$ and $\tilde{\omega} = \frac{\gamma_x}{\beta_x(1-\lambda)}\frac{3C_J^2}{8L_g^2}$ such that the coefficient of $\frac{1}{N}\mathbb{E}[\|M_t^x-G_t\|_F^2]$ is non-positive, i.e., $\tilde{\omega}\frac{8\alpha_x^2\eta^4}{\beta_x(1-\lambda)}+\tilde{\theta}\frac{36\alpha_x^2\eta^4}{\delta}-\tilde{e}\alpha_x\eta^2 \leq \tilde{\omega}\frac{16\alpha_x\eta^2}{\beta_x(1-\lambda)}-\tilde{e}\alpha_x\eta^2 = \frac{\gamma_x}{\beta_x(1-\lambda)}\frac{3C_J^2}{8L_g^2}\frac{16\alpha_x\eta^2}{\beta_x(1-\lambda)}-\frac{16\gamma_x}{\beta_x^2(1-\lambda)^2}\frac{3C_J^2}{8L_g^2}\alpha_x\eta^2 = 0$, where we use $\alpha_x\eta^2 \leq 1$ and $\beta_x \leq \frac{\delta}{12}$ in the first step.

Similarly, we set $\tilde{d}_k = \frac{16\gamma_x}{\beta_z^2(1-\lambda)^2}\frac{3C_J^{2k}}{8L_g^{2k}}$ and $\tilde{\tau}_k = \frac{\gamma_x}{\beta_z(1-\lambda)}\frac{3C_J^{2k}}{8L_g^{2k}}$ such that the coefficient of $\frac{1}{N}\mathbb{E}[\|M_{k,t}^z-V_{k,t}\|_F^2]$ is non-positive for any $k\in\{1,\cdots,K\}$.

Moreover, we set $\tilde{c}_k = \frac{16\gamma_x}{\beta_y^2(1-\lambda)^2} \frac{3C_J^{2k}}{8L_g^{2k}}$ and $\tilde{\nu}_k = \frac{\gamma_x}{\beta_y(1-\lambda)} \frac{3C_J^{2k}}{8L_g^{2k}}$ such that the coefficient of $\frac{1}{N}\mathbb{E}[\|M_{k,t}^y - U_{k,t}\|_F^2]$ is non-positive for any $k \in \{1, \cdots, K\}$.

**(1.3) For global estimation errors:**

**(1.3.a) Global estimation error regarding x:** Here, we set $e = \frac{\gamma_x}{\alpha_x \eta}$ such that the coefficient of $\mathbb{E}[\|(M_t^x - G_t)\mathbf{1}\frac{1}{N}\|^2]$ is zero, i.e., $\eta\gamma_x - \alpha_x\eta^2 e = 0$.

**(1.3.b) Global estimation error regarding y**: To make the coefficient of $\mathbb{E}[\|(M_{k,t}^y - U_{k,t})\mathbf{1}\frac{1}{N}\|^2]$ non-positive, i.e., $a_k\frac{20\gamma_y\eta}{\mu} - c_k\alpha_y\eta^2 \leq 0$, we derive $c_k = a_k\frac{20\gamma_y}{\alpha_y\eta\mu}$.

**(1.3.c) Global estimation error regarding z**: In the following, we aim to set the coefficient of $\mathbb{E}[\|(M_{k,t}^z - V_{k,t})\mathbf{1}\frac{1}{N}\|^2]$ to be non-positive,i.e.,

$$b_k\frac{12\gamma_z\eta}{\mu} + \psi_k\frac{36\gamma_z^2\eta^2}{\delta} + 12\gamma_z^2\eta^2\mathcal{Z}_k - d_k\alpha_z\eta^2 \leq 0 \,.$$

Specifically, we enforce

$$b_k\frac{12\gamma_z\eta}{\mu} \leq \frac{1}{2}d_k\alpha_z\eta^2 \,, \quad \psi_k\frac{36\gamma_z^2\eta^2}{\delta} + 12\gamma_z^2\eta^2\mathcal{Z}_k \leq \frac{1}{2}d_k\alpha_z\eta^2 \,, \tag{36}$$

Then, we obtain

$$d_k = \frac{\gamma_x}{\alpha_x\eta}\Delta_{d_k} \,, \quad \text{where} \quad \Delta_{d_k} = \frac{3C_J^{2k}}{8L_g^{2k}} \,, \quad k \in \{1, \cdots, K\} \,, \quad \text{and} \quad b_k \leq \frac{\alpha_z\eta\mu}{24\gamma_z}d_k \,. \tag{37}$$

We will solve the second inequality after determining the value of $\psi_k$.

**G.2. Other errors related to z ($\mathcal{F}_2$)**

The term $\mathcal{F}_2$ comprises three types of errors related to $z$: the variable feedback error, the variable consensus error, and the momentum consensus error. Specifically, we have

$$\mathcal{F}_2 \leq \sum_{k=1}^{K}\left(\tau_k\frac{8\rho_z\eta}{(1-\lambda)} + 16\rho_z^2\eta^2\mathcal{Z}_k - \psi_k\frac{\delta}{4}\right)\frac{1}{N}\mathbb{E}[\|\hat{Z}_{k,t} - Z_{k,t}\|_F^2] + \sum_{k=1}^{K}\mathcal{D}_{z,k}\frac{1}{N}\mathbb{E}[\|Z_{k,t}(I - J)\|_F^2]$$

$$+ \sum_{k=1}^{K}\left(3b_k\eta\gamma_z^2 + \tau_k\frac{2\gamma_z^2\eta}{\rho_z(1-\lambda)} + \psi_k\frac{12\gamma_z^2\eta^2}{\delta} + \tilde{\psi}_k\frac{36\beta_z^2}{\delta} + 4\gamma_z^2\eta^2\mathcal{Z}_k - \tilde{\tau}_k\beta_z(1-\lambda)\right)\frac{1}{N}\mathbb{E}[\|R_{k,t}(I - J)\|_F^2] \,, \tag{38}$$

where

$$\mathcal{D}_{z,1} = 3b_1 + b_1\frac{48\gamma_z\eta}{\mu}L_g^2 + 6\eta\gamma_xC_J^2 + \psi_1\frac{48\rho_z^2\eta^2}{\delta} + \psi_1\frac{144\gamma_z^2\eta^2L_g^2}{\delta} + 16\rho_z^2\eta^2\mathcal{Z}_1 + 48\gamma_z^2\eta^2L_g^2\mathcal{Z}_1 - \tau_1\rho_z\eta(1-\lambda) \,,$$

$$\mathcal{D}_{z,k} = 3b_k + b_k\frac{48\gamma_z\eta}{\mu}L_g^2 + b_{k-1}\frac{48\gamma_z\eta}{\mu}C_J^2 + \psi_k\frac{48\rho_z^2\eta^2}{\delta} + \psi_k\frac{144\gamma_z^2\eta^2L_g^2}{\delta} + \psi_{k-1}\frac{144\gamma_z^2\eta^2C_J^2}{\delta} + 16\rho_z^2\eta^2\mathcal{Z}_k$$

$$+ 48\gamma_z^2\eta^2L_g^2\mathcal{Z}_k + 48\gamma_z^2\eta^2C_J^2\mathcal{Z}_{k-1} - \tau_k\rho_z\eta(1-\lambda) \,, k \in \{2, \cdots, K\} \,.$$

**(2.1) Feedback error regarding z**: To eliminate $\frac{1}{N}\mathbb{E}[\|\hat{Z}_{k,t} - Z_{k,t}\|_F^2]$, when $k \in \{1, \cdots, K\}$, we enforce

$$\tau_k\frac{8\rho_z\eta}{(1-\lambda)} + 16\rho_z^2\eta^2\mathcal{Z}_k - \psi_k\frac{\delta}{4} \leq 0 \,, \quad \text{and obtain} \quad \psi_k = \tau_k\frac{32\rho_z\eta}{\delta(1-\lambda)} + \frac{64}{\delta}\rho_z^2\eta^2\mathcal{Z}_k \,.$$

**(2.2) Consensus error regarding z**: When $k \in \{2, \cdots, K\}$, to eliminate $\frac{1}{N}\mathbb{E}[\|Z_{k,t}(I - J)\|_F^2]$, from the definition of $\psi_k$, we enforce

$$\mathcal{D}_{z,k} = \eta\gamma_x\frac{3b_k}{\eta\gamma_x} + \eta\gamma_xb_k\frac{48\gamma_z}{\gamma_x\mu}L_g^2 + \eta\gamma_xb_{k-1}\frac{48\gamma_z}{\gamma_x\mu}C_J^2 + \tau_k\frac{32\rho_z\eta}{\delta(1-\lambda)}\frac{144\gamma_z^2\eta^2L_g^2}{\delta} + \frac{64}{\delta}\rho_z^2\eta^2\mathcal{Z}_k\frac{144\gamma_z^2\eta^2L_g^2}{\delta}$$

$$+ \tau_{k-1} \frac{32\rho_z\eta}{\delta(1-\lambda)} \frac{144\gamma_z^2\eta^2 C_J^2}{\delta} + \frac{64}{\delta}\rho_z^2\eta^2 \mathcal{Z}_{k-1} \frac{144\gamma_z^2\eta^2 C_J^2}{\delta} + 48\gamma_z^2\eta^2 L_g^2 \mathcal{Z}_k + 48\gamma_z^2\eta^2 C_J^2 \mathcal{Z}_{k-1}$$

$$+ \tau_k \frac{32\rho_z\eta}{\delta(1-\lambda)} \frac{48\rho_z^2\eta^2}{\delta} + \frac{64}{\delta}\rho_z^2\eta^2 \mathcal{Z}_k \frac{48\rho_z^2\eta^2}{\delta} + 16\rho_z^2\eta^2 \mathcal{Z}_k - \tau_k\rho_z\eta(1-\lambda) \leq 0 \,.$$

Firstly, we enforce

$$\tau_k \frac{32\rho_z\eta}{\delta(1-\lambda)} \frac{144\gamma_z^2\eta^2 L_g^2}{\delta} \leq \frac{1}{12}\tau_k\rho_z\eta(1-\lambda) \,, \qquad \tau_k \frac{32\rho_z\eta}{\delta(1-\lambda)} \frac{48\rho_z^2\eta^2}{\delta} \leq \frac{1}{12}\tau_k\rho_z\eta(1-\lambda) \,.$$

Then we obtain

$$\gamma_z \leq \frac{\delta(1-\lambda)}{96\sqrt{6}L_g} \,, \qquad \rho_z \leq \frac{\delta(1-\lambda)}{96\sqrt{2}} \,. \tag{39}$$

Then, from $\rho_z \leq \frac{\delta}{8\sqrt{3}}$, and by setting $\tau_{k-1}\frac{C_J^2}{L_g^2} \leq \tau_k$, $\mathcal{Z}_{k-1}\frac{C_J^2}{L_g^2} = \mathcal{Z}_k$, we have

$$\eta\gamma_x \frac{3b_k}{\eta\gamma_x} + \eta\gamma_x b_k \frac{48\gamma_z}{\gamma_x\mu}L_g^2 + \eta\gamma_x b_{k-1}\frac{48\gamma_z}{\gamma_x\mu}C_J^2 + 96\gamma_z^2\eta^2 L_g^2 \mathcal{Z}_k + 32\rho_z^2\eta^2 \mathcal{Z}_k \leq \frac{1}{2}\tau_k\rho_z\eta(1-\lambda) \,.$$

We will prove this setting hold after determining the value of $\tau_k$.

Therefore, we set

$$\mathcal{E}_{1,zk} = \frac{3b_k}{\eta\gamma_x} + b_k\frac{48\gamma_z}{\gamma_x\mu}L_g^2 + b_{k-1}\frac{48\gamma_z}{\gamma_x\mu}C_J^2 \,, \qquad \mathcal{E}_{2,zk} = 96\gamma_z^2\eta^2 L_g^2 \mathcal{Z}_k + 32\rho_z^2\eta^2 \mathcal{Z}_k \,,$$

$$\tau_k = \frac{2\gamma_x\mathcal{E}_{1,zk}}{\rho_z(1-\lambda)} + \frac{2\mathcal{E}_{2,zk}}{\rho_z\eta(1-\lambda)} \,.$$

When $k = 1$, to eliminate $\frac{1}{N}\mathbb{E}[\|Z_{1,t}(I-J)\|_F^2]$, we enforce

$$\mathcal{D}_{z,1} = 3b_1 + b_1\frac{48\gamma_z\eta}{\mu}L_g^2 + 6\eta\gamma_x C_J^2 + \psi_1\frac{48\rho_z^2\eta^2}{\delta} + \psi_1\frac{144\gamma_z^2\eta^2 L_g^2}{\delta} + 16\rho_z^2\eta^2 \mathcal{Z}_1 + 48\gamma_z^2\eta^2 L_g^2 \mathcal{Z}_1 - \tau_1\rho_z\eta(1-\lambda) \leq 0 \,.$$

From the definition of $\psi_k$, we have

$$\mathcal{D}_{z,1} = \eta\gamma_x \frac{3b_1}{\eta\gamma_x} + \eta\gamma_x b_1 \frac{48\gamma_z}{\gamma_x\mu}L_g^2 + 6\eta\gamma_x C_J^2 + \tau_1\frac{32\rho_z\eta}{(1-\lambda)}\frac{144\gamma_z^2\eta^2 L_g^2}{\delta^2} + \frac{64}{\delta}\rho_z^2\eta^2 \mathcal{Z}_1 \frac{144\gamma_z^2\eta^2 L_g^2}{\delta}$$

$$+ 48\gamma_z^2\eta^2 L_g^2 \mathcal{Z}_1 + \tau_1 \frac{32\rho_z\eta}{\delta(1-\lambda)}\frac{48\rho_z^2\eta^2}{\delta} + \frac{64}{\delta}\rho_z^2\eta^2 \mathcal{Z}_1 \frac{48\rho_z^2\eta^2}{\delta} + 16\rho_z^2\eta^2 \mathcal{Z}_1 - \tau_1\rho_z\eta(1-\lambda) \leq 0 \,.$$

Firstly, from Eq. (39), we obtain

$$\tau_1 \frac{32\rho_z\eta}{(1-\lambda)}\frac{144\gamma_z^2\eta^2 L_g^2}{\delta^2} \leq \frac{1}{9}\tau_1\rho_z\eta(1-\lambda) \,, \qquad \tau_1 \frac{32\rho_z\eta}{\delta(1-\lambda)}\frac{48\rho_z^2\eta^2}{\delta} \leq \frac{1}{9}\tau_1\rho_z\eta(1-\lambda) \,.$$

Then, since $\rho_z \leq \frac{\delta}{8\sqrt{3}}$, we have

$$\eta\gamma_x \frac{3b_1}{\eta\gamma_x} + \eta\gamma_x b_1 \frac{48\gamma_z}{\gamma_x\mu}L_g^2 + 6\eta\gamma_x C_J^2 + 96\gamma_z^2\eta^2 L_g^2 \mathcal{Z}_1 + 32\rho_z^2\eta^2 \mathcal{Z}_1 \leq \frac{1}{2}\tau_1\rho_z\eta(1-\lambda) \,.$$

Therefore, we set

$$\mathcal{E}_{1,z1} = \frac{3b_1}{\eta\gamma_x} + b_1\frac{48\gamma_z}{\gamma_x\mu}L_g^2 + 6C_J^2 \,, \qquad \mathcal{E}_{2,z1} = 96\gamma_z^2\eta^2 L_g^2 \mathcal{Z}_1 + 32\rho_z^2\eta^2 \mathcal{Z}_1 \,,$$

$$\tau_1 = \frac{2\gamma_x\mathcal{E}_{1,z1}}{\rho_z(1-\lambda)} + \frac{2\mathcal{E}_{2,z1}}{\rho_z\eta(1-\lambda)} \,.$$

**(2.3) Consensus error regarding momentum of z**: In the following, we aim to set the coefficient of $\frac{1}{N}\mathbb{E}[\|R_{k,t}(I-J)\|_F^2]$ to be non-positive, when $k \in \{1, \cdots, K\}$, i.e.,

$$3b_k\eta\gamma_z^2 + \tau_k\frac{2\gamma_z^2\eta}{\rho_z(1-\lambda)} + \psi_k\frac{12\gamma_z^2\eta^2}{\delta} + \tilde{\psi}_k\frac{36\beta_z^2}{\delta} + 4\gamma_z^2\eta^2 \mathcal{Z}_k - \tilde{\tau}_k\beta_z(1-\lambda) \leq 0 \,.$$

Firstly, we enforce

$$\tilde{\tau}_k \frac{32\beta_z}{\delta(1-\lambda)} \frac{36\beta_z^2}{\delta} \leq \frac{1}{3}\tilde{\tau}_k \beta_z (1-\lambda) , \quad \text{and we obtain } \beta_z \leq \frac{\delta(1-\lambda)}{24\sqrt{6}} .$$

Then, from the definition of $\tau_k$, we enforce

$$3b_k \eta \gamma_z^2 + \frac{\gamma_x \mathcal{E}_{1,zk}}{\rho_z(1-\lambda)} \frac{2\gamma_z^2 \eta}{\rho_z(1-\lambda)} + \mathcal{E}_{2,zk} \frac{2\gamma_z^2}{\rho_z^2(1-\lambda)^2} + \psi_k \frac{12\gamma_z^2 \eta^2}{\delta} + 4\gamma_z^2 \eta^2 \mathcal{Z}_k \leq \frac{2}{3}\tilde{\tau}_k \beta_z (1-\lambda) .$$

Specifically, from the definition of $\tilde{\tau}_k = \frac{\gamma_x}{\beta_z(1-\lambda)} \frac{3C_J^{2k}}{8L_g^{2k}}$, we enforce

$$3b_k \eta \gamma_z^2 \leq \frac{\gamma_x}{9} \frac{3C_J^{2k}}{8L_g^{2k}} , \qquad \frac{\gamma_x \mathcal{E}_{1,zk}}{\rho_z(1-\lambda)} \frac{2\gamma_z^2 \eta}{\rho_z(1-\lambda)} \leq \frac{\gamma_x}{9} \frac{3C_J^{2k}}{8L_g^{2k}} ,$$

$$\mathcal{E}_{2,zk} \frac{2\gamma_z^2}{\rho_z^2(1-\lambda)^2} \leq \frac{\gamma_x}{9} \frac{3C_J^{2k}}{8L_g^{2k}} , \qquad \psi_k \frac{12\gamma_z^2 \eta^2}{\delta} \leq \frac{\gamma_x}{9} \frac{3C_J^{2k}}{8L_g^{2k}} , \qquad 4\gamma_z^2 \eta^2 \mathcal{Z}_k \leq \frac{\gamma_x}{18} \frac{3C_J^{2k}}{8L_g^{2k}} . \tag{40}$$

To solve the first inequality in Eq. (40), from $b_k \leq \frac{\alpha_z \eta \mu}{24\gamma_z} d_k$, we obtain $\gamma_z \leq \frac{\alpha_x}{72\alpha_z \mu}$.

To solve the second inequality in Eq. (40), when $k \in \{2, \cdots, K\}$, from the definition of $\mathcal{E}_{1,zk}$, we enforce

$$\frac{3b_k}{\eta\gamma_x} \frac{2\gamma_z^2 \eta \gamma_x}{\rho_z^2(1-\lambda)^2} \leq \frac{\gamma_x}{27} \frac{3C_J^{2k}}{8L_g^{2k}} , \qquad b_k \frac{48\gamma_z}{\gamma_x \mu} L_g^2 \frac{2\gamma_z^2 \eta \gamma_x}{\rho_z^2(1-\lambda)^2} \leq \frac{\gamma_x}{27} \frac{3C_J^{2k}}{8L_g^{2k}} ,$$

$$b_{k-1} \frac{48\gamma_z}{\gamma_x \mu} C_J^2 \frac{2\gamma_z^2 \eta \gamma_x}{\rho_z^2(1-\lambda)^2} \leq \frac{\gamma_x}{27} \frac{3C_J^{2k}}{8L_g^{2k}} .$$

Therefore, from $b_k \leq \frac{\alpha_x \eta \mu}{24\gamma_z} d_k$, we obtain $\gamma_z \leq \left\{ \frac{4\alpha_x \rho_z^2(1-\lambda)^2}{27\alpha_z \mu} , \frac{\sqrt{\alpha_x}\rho_z(1-\lambda)}{6\sqrt{3}\alpha_z L_g} \right\}$.

When $k = 1$, from the definition of $\mathcal{E}_{1,z1}$, we only enforce the unique inequality here, i.e.,

$$6C_J^2 \frac{2\gamma_z^2 \eta \gamma_x}{\rho_z^2(1-\lambda)^2} \leq \frac{\gamma_x}{27} \frac{3C_J^2}{8L_g^2} , \quad \text{and we obtain } \gamma_z \leq \frac{\rho_z(1-\lambda)}{12\sqrt{6}L_g} .$$

To solve the last inequality in Eq. (40), from the definition of $\mathcal{Z}_k$, when $k \in \{2, \cdots, K\}$, we have

$$\mathcal{Z}_k = \frac{8L_g^2 d_k}{N} + \frac{8C_J^2 d_{k-1}}{N} + 8L_g^2 \tilde{d}_k + 8C_J^2 \tilde{d}_{k-1} + \frac{16L_g^2 \tilde{\tau}_k}{\beta_z(1-\lambda)} + \frac{16C_J^2 \tilde{\tau}_{k-1}}{\beta_z(1-\lambda)} + \frac{72L_g^2 \tilde{\psi}_k}{\delta} + \frac{72C_J^2 \tilde{\psi}_{k-1}}{\delta}$$

$$\leq \frac{8L_g^2 d_k}{N} + \frac{8C_J^2 d_{k-1}}{N} + 10L_g^2 \tilde{d}_k + 10C_J^2 \tilde{d}_{k-1}$$

$$= \frac{8L_g^2}{N} \frac{\gamma_x}{\alpha_x \eta} \frac{3C_J^{2k}}{8L_g^{2k}} + \frac{8C_J^2}{N} \frac{\gamma_x}{\alpha_x \eta} \frac{3C_J^{2(k-1)}}{8L_g^{2(k-1)}} + 10L_g^2 \frac{16\gamma_x}{\beta_z^2(1-\lambda)^2} \frac{3C_J^{2k}}{8L_g^{2k}} + 10C_J^2 \frac{16\gamma_x}{\beta_z^2(1-\lambda)^2} \frac{3C_J^{2(k-1)}}{8L_g^{2(k-1)}}$$

$$= \left( \frac{16C_J^2}{N} \frac{\gamma_x}{\alpha_x \eta} + \frac{320\gamma_x C_J^2}{\beta_z^2(1-\lambda)^2} \right) \frac{3C_J^{2(k-1)}}{8L_g^{2(k-1)}} . \tag{41}$$

Here, it is easy to prove that $\mathcal{Z}_{k-1} \frac{C_J^2}{L_g^2} = \mathcal{Z}_k$.

Therefore, we enforce

$$4\gamma_z^2 \eta^2 \left( \frac{16C_J^2}{N} \frac{\gamma_x}{\alpha_x \eta} + \frac{320\gamma_x C_J^2}{\beta_z^2(1-\lambda)^2} \right) \frac{3C_J^{2(k-1)}}{8L_g^{2(k-1)}} \leq \frac{\gamma_x}{18} \frac{3C_J^{2k}}{8L_g^{2k}} .$$

Specifically, we enforce

$$4\gamma_z^2 \eta^2 \frac{16C_J^2}{N} \frac{\gamma_x}{\alpha_x \eta} \frac{3C_J^{2(k-1)}}{8L_g^{2(k-1)}} \leq \frac{\gamma_x}{36} \frac{3C_J^{2k}}{8L_g^{2k}} , \qquad 4\gamma_z^2 \eta^2 \frac{320\gamma_x}{\beta_z^2(1-\lambda)^2} C_J^2 \frac{3C_J^{2(k-1)}}{8L_g^{2(k-1)}} \leq \frac{\gamma_x}{36} \frac{3C_J^{2k}}{8L_g^{2k}} .$$

Then, we obtain $\gamma_z \leq \left\{ \frac{\sqrt{\alpha_x N}}{48 L_g}, \frac{\beta_z(1-\lambda)}{96\sqrt{5}L_g} \right\}$.

When $k = 1$, we have

$$
\begin{aligned}
\mathcal{Z}_1 &= \frac{8L_g^2 d_1}{N} + \frac{3e C_J^2}{N} + 8L_g^2 \tilde{d}_1 + 3\tilde{e}C_J^2 + \frac{12\tilde{\omega}C_J^2}{\beta_x(1-\lambda)} + \frac{16L_g^2\tilde{\tau}_1}{\beta_z(1-\lambda)} + \tilde{\theta}\frac{54C_J^2}{\delta} + \frac{72L_g^2\tilde{\psi}_1}{\delta} \\
&\leq \frac{8L_g^2 d_1}{N} + \frac{3eC_J^2}{N} + 10L_g^2\tilde{d}_1 + 6\tilde{e}C_J^2 \\
&= \frac{8L_g^2}{N}\frac{\gamma_x}{\alpha_x\eta}\frac{3C_J^2}{8L_g^2} + \frac{3C_J^2}{N}\frac{\gamma_x}{\alpha_x\eta} + 10L_g^2\frac{16\gamma_x}{\beta_z^2(1-\lambda)^2}\frac{3C_J^2}{8L_g^2} + 6C_J^2\frac{16\gamma_x}{\beta_x^2(1-\lambda)^2}\frac{3C_J^2}{8L_g^2} \\
&= \left( \frac{16L_g^2}{N}\frac{\gamma_x}{\alpha_x\eta} + \frac{160\gamma_x L_g^2}{\beta_z^2(1-\lambda)^2} + \frac{96\gamma_x C_J^2}{\beta_x^2(1-\lambda)^2} \right)\frac{3C_J^2}{8L_g^2}.
\end{aligned}
\tag{42}
$$

Therefore, we enforce

$$
4\gamma_z^2\eta^2\left( \frac{16L_g^2}{N}\frac{\gamma_x}{\alpha_x\eta} + \frac{160\gamma_x L_g^2}{\beta_z^2(1-\lambda)^2} + \frac{96\gamma_x C_J^2}{\beta_x^2(1-\lambda)^2} \right)\frac{3C_J^2}{8L_g^2} \leq \frac{\gamma_x}{18}\frac{3C_J^2}{8L_g^2}.
$$

Specifically, we enforce

$$
4\gamma_z^2\eta^2\frac{16L_g^2}{N}\frac{\gamma_x}{\alpha_x\eta}\frac{3C_J^2}{8L_g^2} \leq \frac{\gamma_x}{36}\frac{3C_J^2}{8L_g^2}, \qquad 4\gamma_z^2\eta^2\frac{160\gamma_x L_g^2}{\beta_z^2(1-\lambda)^2}\frac{3C_J^2}{8L_g^2} \leq \frac{\gamma_x}{72}\frac{3C_J^2}{8L_g^2},
$$

$$
4\gamma_z^2\eta^2\frac{96\gamma_x C_J^2}{\beta_x^2(1-\lambda)^2}\frac{3C_J^2}{8L_g^2} \leq \frac{\gamma_x}{64}\frac{3C_J^2}{8L_g^2}.
$$

The first two inequalities are already satisfied. And from the last inequality in Eq. (42), we obtain $\gamma_z \leq \frac{\beta_x(1-\lambda)}{144C_J}$.

To solve the fourth inequality in Eq. (40), we enforce

$$
\left( \tau_k\frac{32\rho_z\eta}{\delta(1-\lambda)} + \frac{64}{\delta}\rho_z^2\eta^2\mathcal{Z}_k \right)\frac{12\gamma_z^2\eta^2}{\delta} \leq \frac{\gamma_x}{9}\frac{3C_J^{2k}}{8L_g^{2k}}.
$$

Specifically, we enforce

$$
\tau_k\frac{32\rho_z\eta}{\delta(1-\lambda)}\frac{12\gamma_z^2\eta^2}{\delta} \leq \frac{\gamma_x}{18}\frac{3C_J^{2k}}{8L_g^{2k}}, \qquad \frac{64}{\delta}\rho_z^2\eta^2\mathcal{Z}_k\frac{12\gamma_z^2\eta^2}{\delta} \leq \frac{\gamma_x}{18}\frac{3C_J^{2k}}{8L_g^{2k}}.
\tag{43}
$$

Since $\rho_z \leq \frac{\delta}{8\sqrt{3}}$, the second inequality in Eq. (43) holds.

To solve the first inequality in Eq. (43), from the definition of $\tau_k$, we enforce

$$
\frac{2\gamma_x\mathcal{E}_{1,zk}}{\rho_z(1-\lambda)}\frac{32\rho_z\eta}{\delta(1-\lambda)}\frac{12\gamma_z^2\eta^2}{\delta} \leq \frac{\gamma_x}{36}\frac{3C_J^{2k}}{8L_g^{2k}}, \qquad \frac{2\mathcal{E}_{2,zk}}{\rho_z\eta(1-\lambda)}\frac{32\rho_z\eta}{\delta(1-\lambda)}\frac{12\gamma_z^2\eta^2}{\delta} \leq \frac{\gamma_x}{36}\frac{3C_J^{2k}}{8L_g^{2k}}.
$$

From the definition of $\mathcal{E}_{2,zk}$ and Eq. (39), the second inequality is already satisfied.

To solve the first inequality, when $k \in \{2, \cdots, K\}$, from the definition of $\mathcal{E}_{1,zk}$, we enforce

$$
\frac{3b_k}{\eta\gamma_x}\frac{64\gamma_x\eta}{(1-\lambda)^2}\frac{4\gamma_z^2\eta^2}{\delta^2} \leq \frac{\gamma_x}{36}\frac{3C_J^{2k}}{8L_g^{2k}}, \qquad b_k\frac{48\gamma_z}{\gamma_x\mu}L_g^2\frac{64\gamma_x\eta}{(1-\lambda)^2}\frac{4\gamma_z^2\eta^2}{\delta^2} \leq \frac{\gamma_x}{36}\frac{3C_J^{2k}}{8L_g^{2k}},
$$

$$
b_{k-1}\frac{48\gamma_z}{\gamma_x\mu}C_J^2\frac{64\gamma_x\eta}{(1-\lambda)^2}\frac{4\gamma_z^2\eta^2}{\delta^2} \leq \frac{\gamma_x}{36}\frac{3C_J^{2k}}{8L_g^{2k}}.
$$

Therefore, from $b_k \leq \frac{\alpha_z\eta\mu}{24\gamma_z}d_k$, we obtain $\gamma_z \leq \left\{ \frac{\alpha_x\delta^2(1-\lambda)^2}{1152\alpha_z\mu}, \frac{\sqrt{\alpha_x}\delta(1-\lambda)}{96\sqrt{2\alpha_z}L_g} \right\}$.

When $k = 1$, from the definition of $\mathcal{E}_{1,z1}$, we only enforce the unique inequality here, i.e.,

$$
6C_J^2\frac{64\gamma_x\eta}{(1-\lambda)^2}\frac{4\gamma_z^2\eta^2}{\delta^2} \leq \frac{\gamma_x}{36}\frac{3C_J^2}{8L_g^2}, \quad \text{and we obtain } \gamma_z \leq \frac{\delta(1-\lambda)}{384L_g}.
$$

To solve the last inequality in Eq. (40), from the definition of $\mathcal{E}_{2,zk}$, we enforce

$$96\gamma_z^2\eta^2 L_g^2 \mathcal{Z}_k \frac{2\gamma_z^2}{\rho_z^2(1-\lambda)^2} \leq \frac{\gamma_x}{18}\frac{3C_J^{2k}}{8L_g^{2k}} , \qquad 32\mathcal{Z}_k \frac{2\gamma_z^2\eta^2}{(1-\lambda)^2} \leq \frac{\gamma_x}{18}\frac{3C_J^{2k}}{8L_g^{2k}} .$$

Since $\gamma_z \leq \frac{\rho_z(1-\lambda)}{12\sqrt{6}L_g}$, the first inequality is already satisfied.

To solve the second inequality, from the definition of $\mathcal{Z}_k$, when $k \in \{2,\cdots,K\}$, we enforce

$$32\frac{16C_J^2}{N}\frac{\gamma_x}{\alpha_x\eta}\frac{3C_J^{2(k-1)}}{8L_g^{2(k-1)}}\frac{2\gamma_z^2\eta^2}{(1-\lambda)^2} \leq \frac{\gamma_x}{36}\frac{3C_J^{2k}}{8L_g^{2k}} ,$$

$$32\frac{320\gamma_x}{\beta_z^2(1-\lambda)^2}C_J^2\frac{3C_J^{2(k-1)}}{8L_g^{2(k-1)}}\frac{2\gamma_z^2\eta^2}{(1-\lambda)^2} \leq \frac{\gamma_x}{36}\frac{3C_J^{2k}}{8L_g^{2k}} .$$

We obtain $\gamma_z \leq \left\{ \frac{\sqrt{\alpha_x N}(1-\lambda)}{192L_g} , \frac{\beta_z(1-\lambda)^2}{384\sqrt{5}L_g} \right\}$.

When $k=1$, we only enforce the unique inequality

$$32\frac{96\gamma_x C_J^2}{\beta_x^2(1-\lambda)^2}\frac{3C_J^2}{8L_g^2}\frac{2\gamma_z^2\eta^2}{(1-\lambda)^2} \leq \frac{\gamma_x}{54}\frac{3C_J^2}{8L_g^2} , \quad \text{and we obtain } \gamma_z \leq \frac{\beta_x(1-\lambda)^2}{576C_J} .$$

Here, after determining the value of $\psi_k$, we now turn to the second inequality in Eq. (36). Specifically, we enforce

$$\psi_k\frac{36\gamma_z^2\eta^2}{\delta} \leq \frac{1}{3}\frac{\gamma_x}{\alpha_x\eta}\frac{3C_J^{2k}}{8L_g^{2k}}\alpha_z\eta^2 , \qquad 12\gamma_z^2\eta^2\mathcal{Z}_k \leq \frac{1}{6}\frac{\gamma_x}{\alpha_x\eta}\frac{3C_J^{2k}}{8L_g^{2k}}\alpha_z\eta^2 . \tag{44}$$

To solve the second inequality in Eq. (44), from the definition of $\mathcal{Z}_k$, we enforce

$$12\gamma_z^2\eta^2\frac{16C_J^2}{N}\frac{\gamma_x}{\alpha_x\eta}\frac{3C_J^{2(k-1)}}{8L_g^{2(k-1)}} \leq \frac{1}{12}\frac{\gamma_x}{\alpha_x\eta}\frac{3C_J^{2k}}{8L_g^{2k}}\alpha_z\eta^2 ,$$

$$12\gamma_z^2\eta^2\frac{320\gamma_x}{\beta_z^2(1-\lambda)^2}C_J^2\frac{3C_J^{2(k-1)}}{8L_g^{2(k-1)}} \leq \frac{1}{12}\frac{\gamma_x}{\alpha_x\eta}\frac{3C_J^{2k}}{8L_g^{2k}}\alpha_z\eta^2 ,$$

$$12\gamma_z^2\eta^2\frac{96\gamma_x C_J^2}{\beta_x^2(1-\lambda)^2}\frac{3C_J^2}{8L_g^2} \leq \frac{1}{54}\frac{\gamma_x}{\alpha_x\eta}\frac{3C_J^2}{8L_g^2}\alpha_z\eta^2 .$$

Then, we obtain $\gamma_z \leq \left\{ \frac{\sqrt{\alpha_z N}}{48L_g} , \frac{\sqrt{\alpha_z}\beta_z(1-\lambda)}{96\sqrt{5\alpha_x}L_g} , \frac{\beta_x(1-\lambda)\sqrt{\alpha_z}}{144\sqrt{3\alpha_x}C_J} \right\}$.

To solve the first inequality in Eq. (44), we enforce

$$(\tau_k\frac{32\rho_z\eta}{\delta(1-\lambda)} + \frac{64}{\delta}\rho_z^2\eta^2\mathcal{Z}_k)\frac{36\gamma_z^2\eta^2}{\delta} \leq \frac{1}{3}\frac{\gamma_x}{\alpha_x\eta}\frac{3C_J^{2k}}{8L_g^{2k}}\alpha_z\eta^2 .$$

Specifically, we enforce

$$\tau_k\frac{32\rho_z\eta}{\delta(1-\lambda)}\frac{36\gamma_z^2\eta^2}{\delta} \leq \frac{1}{6}\frac{\gamma_x}{\alpha_x\eta}\frac{3C_J^{2k}}{8L_g^{2k}}\alpha_z\eta^2 , \qquad \frac{64}{\delta}\rho_z^2\eta^2\mathcal{Z}_k\frac{36\gamma_z^2\eta^2}{\delta} \leq \frac{1}{6}\frac{\gamma_x}{\alpha_x\eta}\frac{3C_J^{2k}}{8L_g^{2k}}\alpha_z\eta^2 .$$

Since $\rho_z \leq \frac{\delta}{8\sqrt{3}}$, the second inequality holds.

To solve the first inequality, from the definition of $\tau_k$, we enforce

$$\frac{2\gamma_x\mathcal{E}_{1,zk}}{\rho_z(1-\lambda)}\frac{32\rho_z\eta}{\delta(1-\lambda)}\frac{36\gamma_z^2\eta^2}{\delta} \leq \frac{1}{12}\frac{\gamma_x}{\alpha_x\eta}\frac{3C_J^{2k}}{8L_g^{2k}}\alpha_z\eta^2 ,$$

$$\frac{2\mathcal{E}_{2,zk}}{\rho_z\eta(1-\lambda)}\frac{32\rho_z\eta}{\delta(1-\lambda)}\frac{36\gamma_z^2\eta^2}{\delta} \leq \frac{1}{12}\frac{\gamma_x}{\alpha_x\eta}\frac{3C_J^{2k}}{8L_g^{2k}}\alpha_z\eta^2 .$$

From the definition of $\mathcal{E}_{2,zk}$ and Eq. (39), the second inequality is already satisfied.

To solve the first inequality, from the definition of $\mathcal{E}_{1,zk}$, we enforce

$$\frac{3b_k}{\eta\gamma_x}\frac{64\gamma_x\eta}{(1-\lambda)^2}\frac{12\gamma_z^2\eta^2}{\delta^2} \leq \frac{1}{12}\frac{\gamma_x}{\alpha_x\eta}\frac{3C_J^{2k}}{8L_g^{2k}}\alpha_z\eta^2 \,, \qquad b_k\frac{48\gamma_z}{\gamma_x\mu}L_g^2\frac{64\gamma_x\eta}{(1-\lambda)^2}\frac{12\gamma_z^2\eta^2}{\delta^2} \leq \frac{1}{12}\frac{\gamma_x}{\alpha_x\eta}\frac{3C_J^{2k}}{8L_g^{2k}}\alpha_z\eta^2 \,,$$

$$b_{k-1}\frac{48\gamma_z}{\gamma_x\mu}C_J^2\frac{64\gamma_x\eta}{(1-\lambda)^2}\frac{12\gamma_z^2\eta^2}{\delta^2} \leq \frac{1}{12}\frac{\gamma_x}{\alpha_x\eta}\frac{3C_J^{2k}}{8L_g^{2k}}\alpha_z\eta^2 \,, \qquad 6C_J^2\frac{64\gamma_x\eta}{(1-\lambda)^2}\frac{12\gamma_z^2\eta^2}{\delta^2} \leq \frac{1}{12}\frac{\gamma_x}{\alpha_x\eta}\frac{3C_J^2}{8L_g^2}\alpha_z\eta^2 \,.$$

Therefore, from $b_k \leq \frac{\alpha_z\eta\mu}{24\gamma_z}d_k$ ,we obtain $\gamma_z \leq \left\{ \frac{\delta^2(1-\lambda)^2}{1152\mu} \,, \frac{\delta(1-\lambda)}{96\sqrt{2}L_g} \,, \frac{\sqrt{\alpha_x}\delta(1-\lambda)}{384\sqrt{\alpha_x}L_g} \right\}$

## G.3. Optimization errors ($\mathcal{F}_3$)

The term $\mathcal{F}_3$ comprises three optimization errors related to $x$, $y$, and $z$, respectively. Specifically, we have

$$\mathcal{F}_3 = \sum_{k=1}^{K}\mathcal{D}_{y^*,k}\mathbb{E}[\|y_k^*(\bar{x}_t) - \bar{y}_{k,t}\|^2] + \sum_{k=1}^{K}\mathcal{D}_{z^*,k}\mathbb{E}[\|\bar{z}_{k,t} - z_k^*(\bar{x}_t)\|^2] \,,$$

where

$$\mathcal{D}_{y^*,1} = 6\eta\gamma_x\Delta_1^2L_J^2 + a_2\frac{10\gamma_y\eta}{\mu}L_g^2 + b_1\frac{15\gamma_z\eta}{\mu}(L_H^2\Delta_1^2 + L_J^2\Delta_2^2) + b_2\frac{15\gamma_z\eta}{\mu}L_H^2\Delta_2^2 + \psi_1\frac{144\gamma_z^2\eta^2}{\delta}(\Delta_1^2L_H^2 + \Delta_2^2L_J^2)$$

$$+ \psi_2\frac{144\gamma_z^2\eta^2}{\delta}\Delta_2^2L_H^2 + 48\gamma_z^2\eta^2\mathcal{Z}_1(\Delta_1^2L_H^2 + \Delta_2^2L_J^2) + 48\gamma_z^2\eta^2\mathcal{Z}_2\Delta_2^2L_H^2 - a_1\frac{\gamma_y\eta\mu}{4} \,,$$

$$\mathcal{D}_{y^*,k} = a_{k+1}\frac{10\gamma_y\eta}{\mu}L_g^2 + b_{k+1}\frac{15\gamma_z\eta}{\mu}L_H^2\Delta_{k+1}^2 + b_k\frac{15\gamma_z\eta}{\mu}(L_H^2\Delta_k^2 + L_J^2\Delta_{k+1}^2) + b_{k-1}\frac{15\gamma_z\eta}{\mu}L_J^2\Delta_k^2$$

$$+ \psi_k\frac{144\gamma_z^2\eta^2}{\delta}(\Delta_k^2L_H^2 + \Delta_{k+1}^2L_J^2) + \psi_{k+1}\frac{144\gamma_z^2\eta^2}{\delta}\Delta_{k+1}^2L_H^2 + \psi_{k-1}\frac{144\gamma_z^2\eta^2}{\delta}\Delta_k^2L_J^2$$

$$+ 48\gamma_z^2\eta^2\mathcal{Z}_k(\Delta_k^2L_H^2 + \Delta_{k+1}^2L_J^2) + 48\gamma_z^2\eta^2\mathcal{Z}_{k+1}\Delta_{k+1}^2L_H^2 + 48\gamma_z^2\eta^2\mathcal{Z}_{k-1}\Delta_k^2L_J^2 - a_k\frac{\gamma_y\eta\mu}{4} \,, k \in \{2,\cdots,K-1\} \,,$$

$$\mathcal{D}_{y^*,K} = 6\eta\gamma_xL_f^2 + b_{K-1}\frac{15\gamma_z\eta}{\mu}L_J^2\Delta_K + b_K\frac{15\gamma_z\eta}{\mu}(L_H^2\Delta_K + L_J^2\Delta_{K+1}^2) + \psi_{K-1}\frac{144\gamma_z^2\eta^2}{\delta}\Delta_K^2L_J^2$$

$$+ \psi_K\frac{144\gamma_z^2\eta^2}{\delta}(\Delta_K^2L_H^2 + L_f^2) + 48\gamma_z^2\eta^2\mathcal{Z}_{K-1}\Delta_K^2L_J^2 + 48\gamma_z^2\eta^2\mathcal{Z}_K(\Delta_K^2L_H^2 + L_f^2) - a_K\frac{\gamma_y\eta\mu}{4} \,,$$

$$\mathcal{D}_{z^*,1} = 6\eta\gamma_xC_J^2 + \psi_1\frac{144\gamma_z^2\eta^2L_g^2}{\delta} + 48\gamma_z^2\eta^2L_g^2\mathcal{Z}_1 - b_1\frac{\gamma_z\eta\mu}{8} \,,$$

$$\mathcal{D}_{z^*,k} = b_{k-1}\frac{15\gamma_z\eta C_J^2}{\mu} + \psi_k\frac{144\gamma_z^2\eta^2L_g^2}{\delta} + \psi_{k-1}\frac{144C_J^2\gamma_z^2\eta^2}{\delta} + 48\gamma_z^2\eta^2L_g^2\mathcal{Z}_k + 48\gamma_z^2\eta^2C_J^2\mathcal{Z}_{k-1} - b_k\frac{\gamma_z\eta\mu}{8} \,, k \in \{2,\cdots,K\} \,.$$

**(3.1) Optimization error regarding z**: In the following, we aim to enforce the coefficient of $\mathbb{E}[\|\bar{z}_{k,t} - z_k^*(\bar{x}_t)\|^2]$ to be non-positive when $k \in \{2,\cdots,K\}$. From the definition of $\psi_k$ and $\rho_z \leq \frac{\delta}{8\sqrt{3}}$, we have

$$\mathcal{D}_{z^*,k} \leq b_{k-1}\frac{15\gamma_z\eta C_J^2}{\mu} + \tau_k\frac{32\rho_z\eta}{(1-\lambda)}\frac{144\gamma_z^2\eta^2L_g^2}{\delta^2} + \tau_{k-1}\frac{32\rho_z\eta}{(1-\lambda)}\frac{144C_J^2\gamma_z^2\eta^2}{\delta^2}$$

$$+ 96\gamma_z^2\eta^2L_g^2\mathcal{Z}_k + 96\gamma_z^2\eta^2C_J^2\mathcal{Z}_{k-1} - b_k\frac{\gamma_z\eta\mu}{8} \leq 0 \,.$$

Then, from $\tau_{k-1}\frac{C_J^2}{L_g^2} \leq \tau_k$, $\mathcal{Z}_{k-1}\frac{C_J^2}{L_g^2} = \mathcal{Z}_k$, we obtain

$$\mathcal{D}_{z^*,k} \leq b_{k-1}\frac{15\gamma_z\eta C_J^2}{\mu} + \tau_k\frac{64\rho_z\eta}{(1-\lambda)}\frac{144\gamma_z^2\eta^2L_g^2}{\delta^2} + 192\gamma_z^2\eta^2L_g^2\mathcal{Z}_k - b_k\frac{\gamma_z\eta\mu}{8} \,.$$

Specifically, we enforce

$$b_{k-1}\frac{15\gamma_z\eta C_J^2}{\mu} \leq b_k\frac{\gamma_z\eta\mu}{16} \,, \qquad \tau_k\frac{64\rho_z\eta}{(1-\lambda)}\frac{144\gamma_z^2\eta^2L_g^2}{\delta^2} + 192\gamma_z^2\eta^2L_g^2\mathcal{Z}_k \leq b_k\frac{\gamma_z\eta\mu}{16} \,. \tag{45}$$

To solve the first inequality in Eq. (45), we obtain $b_{k-1} \leq b_k\frac{\mu^2}{256C_J^2}$. In addition, we derive $\tau_{k-1}\frac{C_J^2}{L_g^2} \leq \tau_k$, since $\tau_k$ depends on $b_k$ and $\mathcal{Z}_k$, and $b_{k-1}\frac{C_J^2}{L_g^2} \leq b_k$ and $\frac{L_g^2}{\mu^2} \geq 1$ are satisfied.

To solve the second inequality in Eq. (45), firstly, from the definition of $\tau_k$, we obtain

$$\frac{2\gamma_x \mathcal{E}_{1,zk}}{\rho_z(1-\lambda)}\frac{64\rho_z\eta}{(1-\lambda)}\frac{144\gamma_z^2\eta^2 L_g^2}{\delta^2} \leq b_k\frac{\gamma_z\eta\mu}{32} ,$$

$$\frac{2\mathcal{E}_{2,zk}}{\rho_z\eta(1-\lambda)}\frac{64\rho_z\eta}{(1-\lambda)}\frac{144\gamma_z^2\eta^2 L_g^2}{\delta^2} + 192\gamma_z^2\eta^2 L_g^2\mathcal{Z}_k \leq b_k\frac{\gamma_z\eta\mu}{32} . \quad (46)$$

Specifically, to solve the first inequality in Eq. (46), we enforce

$$64\frac{3b_k}{\eta\gamma_x}\frac{2\gamma_x}{(1-\lambda)^2}\frac{144\gamma_z^2\eta^2 L_g^2}{\delta^2} \leq b_k\frac{\gamma_z\eta\mu}{96} , \qquad 64b_k\frac{48\gamma_z}{\gamma_x\mu}L_g^2\frac{2\gamma_x}{(1-\lambda)^2}\frac{144\gamma_z^2\eta^2 L_g^2}{\delta^2} \leq b_k\frac{\gamma_z\eta\mu}{96} ,$$

$$64b_{k-1}\frac{48\gamma_z}{\gamma_x\mu}C_J^2\frac{2\gamma_x}{(1-\lambda)^2}\frac{144\gamma_z^2\eta^2 L_g^2}{\delta^2} \leq b_k\frac{\gamma_z\eta\mu}{96} . \quad (47)$$

Therefore, from $b_{k-1} \leq b_k\frac{\mu^2}{256C_J^2}$ , we obtain $\gamma_z \leq \left\{ \frac{\mu\delta^2(1-\lambda)^2}{384\times144\times96L_g^2} , \frac{\mu\delta(1-\lambda)}{9216L_g^2} , \frac{\delta(1-\lambda)}{576L_g} \right\}$.

To solve the second inequality in Eq. (46), from the definition of $\mathcal{E}_{2,zk}$, and Eq. (39), we obtain

$$96\gamma_z^2\eta^2 L_g^2\mathcal{Z}_k\frac{128}{(1-\lambda)^2}\frac{144\gamma_z^2\eta^2 L_g^2}{\delta^2} + 32\rho_z^2\eta^2\mathcal{Z}_k\frac{128}{(1-\lambda)^2}\frac{144\gamma_z^2\eta^2 L_g^2}{\delta^2} + 192\gamma_z^2\eta^2 L_g^2\mathcal{Z}_k$$

$$\leq 32\gamma_z^2\eta^2 L_g^2\mathcal{Z}_k + 32\gamma_z^2\eta^2 L_g^2\mathcal{Z}_k + 192\gamma_z^2\eta^2 L_g^2\mathcal{Z}_k$$

$$= 256\gamma_z^2\eta^2 L_g^2\mathcal{Z}_k \leq b_k\frac{\gamma_z\eta\mu}{32} .$$

Therefore, we obtain $b_k \geq \frac{3072\gamma_x\gamma_z}{\mu}\left(\frac{16C_J^2 L_g^2}{\alpha_x N} + \frac{320C_J^2 L_g^2}{\beta_z^2(1-\lambda)^2}\right)\frac{C_J^{2(k-1)}}{L_g^{2(k-1)}}$. Since $L_g/\mu > 1$, for any $k \in \{2,\cdots,K\}$, we define

$$b_k = \frac{L_g^{2k}}{\mu^{2k}}\frac{(256C_J)^{2(k-1)}}{L_g^{2(k-1)}}\frac{3072\gamma_x\gamma_z}{\mu}\left(\frac{80C_J^2}{\gamma_z^2} + \frac{80C_J^2 L_g^2}{\alpha_x N} + \frac{320C_J^2 L_g^2}{\beta_z^2(1-\lambda)^2} + \frac{320C_J^4}{\beta_x^2(1-\lambda)^2}\right)$$

$$\triangleq \Delta_{b_k}\frac{\gamma_x\gamma_z}{\mu}\left(\frac{80C_J^2}{\gamma_z^2} + \frac{80C_J^2 L_g^2}{\alpha_x N} + \frac{320C_J^2 L_g^2}{\beta_z^2(1-\lambda)^2} + \frac{320C_J^4}{\beta_x^2(1-\lambda)^2}\right) .$$

We verify that $b_{k-1} \leq b_k\frac{\mu^2}{256C_J^2}$ for any $k \in \{3,\cdots,K\}$.

In the following, we aim to enforce the coefficient of $\mathbb{E}[\|\bar{z}_{k,t} - z_k^*(\bar{x}_t)\|^2]$ to be non-positive when $k = 1$, from the definition of $\psi_1$, $\rho_z \leq \frac{\delta}{8\sqrt{3}}$, we have

$$\mathcal{D}_{z^*,1} \leq 6\eta\gamma_x C_J^2 + \tau_1\frac{32\rho_z\eta}{\delta(1-\lambda)}\frac{144\gamma_z^2\eta^2 L_g^2}{\delta} + 96\gamma_z^2\eta^2 L_g^2\mathcal{Z}_1 - b_1\frac{\gamma_z\eta\mu}{8} \leq 0 .$$

This inequality can be handled analogously, and we omit the detailed derivation for brevity. We therefore obtain $b_1 \leq b_2\frac{\mu^2}{256C_J^2}$, with the same definition of $b_1$. Finally, to ensure $b_k \leq \frac{\alpha_z\eta\mu}{24\gamma_z}d_k$ in Eq. (37), we enforce

$$\frac{L_g^{2k}}{\mu^{2k}}\frac{(256C_J)^{2(k-1)}}{L_g^{2(k-1)}}\frac{3072\gamma_x\gamma_z}{\mu}\left(\frac{80C_J^2}{\gamma_z^2} + \frac{80C_J^2 L_g^2}{\alpha_x N} + \frac{320C_J^2 L_g^2}{\beta_z^2(1-\lambda)^2} + \frac{320C_J^4}{\beta_x^2(1-\lambda)^2}\right) \leq \frac{\alpha_z\eta\mu}{24\gamma_z}\frac{\gamma_x}{\alpha_x\eta}\frac{3C_J^{2k}}{8L_g^{2k}} .$$

To make this inequality hold, we enforce

$$\frac{L_g^{2k}}{\mu^{2k}}\frac{(256C_J)^{2(k-1)}}{L_g^{2(k-1)}}\frac{3072\gamma_x\gamma_z}{\mu}\frac{80C_J^2}{\gamma_z^2} \leq \frac{\alpha_z\eta\mu}{48\gamma_z}\frac{\gamma_x}{\alpha_x\eta}\frac{3C_J^{2k}}{8L_g^{2k}} ,$$

$$\frac{L_g^{2k}}{\mu^{2k}}\frac{(256C_J)^{2(k-1)}}{L_g^{2(k-1)}}\frac{3072\gamma_x\gamma_z}{\mu}\left(\frac{80C_J^2 L_g^2}{\alpha_x N} + \frac{320C_J^2 L_g^2}{\beta_z^2(1-\lambda)^2} + \frac{320C_J^4}{\beta_x^2(1-\lambda)^2}\right) \leq \frac{\alpha_z\eta\mu}{48\gamma_z}\frac{\gamma_x}{\alpha_x\eta}\frac{3C_J^{2k}}{8L_g^{2k}} ,$$

which leads to

$$\alpha_x \leq \frac{\alpha_z}{1024\times640\times48}\frac{\mu^{2(k+1)}}{L_g^{2(k+1)}256^{2(k-1)}} ,$$

$$\gamma_z \le \frac{\mu^{k+1}}{L_g^{k+1} 256^{(k-1)}} \sqrt{\frac{\alpha_z}{3072 \times 128\alpha_x}} \Bigg/ \sqrt{\frac{80L_g^2}{\alpha_x N} + \frac{320L_g^2}{\beta_z^2 (1-\lambda)^2} + \frac{320C_J^2}{\beta_x^2 (1-\lambda)^2}} .$$

**(2.2) Optimization error regarding y**: In the following, we aim to enforce the coefficient of $\mathbb{E}[\|y_k^*(\bar{x}_t) - \bar{y}_{k,t}\|^2]$ to be non-positive. Firstly, from the definition of $\psi_k$, $\tau_k$, Eq. (39) and Eq. (47), we obtain

$$
\begin{aligned}
\psi_k \frac{144\gamma_z^2 \eta^2}{\delta} &= \tau_k \frac{32\rho_z \eta}{\delta(1-\lambda)} \frac{144\gamma_z^2 \eta^2}{\delta} + \frac{64}{\delta} \rho_z^2 \eta^2 \mathcal{Z}_k \frac{144\gamma_z^2 \eta^2}{\delta} \\
&\le \frac{2\eta\gamma_x \mathcal{E}_{1,zk}}{(1-\lambda)^2} 32 \frac{144\gamma_z^2 \eta^2}{\delta^2} + \frac{2\mathcal{E}_{2,zk}}{(1-\lambda)^2} 32 \frac{144\gamma_z^2 \eta^2}{\delta^2} + 48\gamma_z^2 \eta^2 \mathcal{Z}_k \\
&\le \frac{3b_k}{\eta\gamma_x} \frac{2\eta\gamma_x}{(1-\lambda)^2} 32 \frac{144\gamma_z^2 \eta^2}{\delta^2} + b_k \frac{48\gamma_z}{\gamma_x \mu} L_g^2 \frac{2\eta\gamma_x}{(1-\lambda)^2} 32 \frac{144\gamma_z^2 \eta^2}{\delta^2} + b_{k-1} \frac{48\gamma_z}{\gamma_x \mu} C_J^2 \frac{2\eta\gamma_x}{(1-\lambda)^2} 32 \frac{144\gamma_z^2 \eta^2}{\delta^2} \\
&\quad + 96\gamma_z^2 \eta^2 L_g^2 \mathcal{Z}_k \frac{2}{(1-\lambda)^2} 32 \frac{144\gamma_z^2 \eta^2}{\delta^2} + 32\rho_z^2 \eta^2 \mathcal{Z}_k \frac{2}{(1-\lambda)^2} 32 \frac{144\gamma_z^2 \eta^2}{\delta^2} + 48\gamma_z^2 \eta^2 \mathcal{Z}_k \\
&\le b_k \frac{\gamma_z \eta}{192\mu} + b_k \frac{8\gamma_z \eta}{\mu} + b_{k-1} \frac{8\gamma_z \eta}{\mu} \frac{C_J^2}{L_g^2} + 16\gamma_z^2 \eta^2 \mathcal{Z}_k + 16\gamma_z^2 \eta^2 \mathcal{Z}_k + 48\gamma_z^2 \eta^2 \mathcal{Z}_k \\
&\le b_k \frac{17\gamma_z \eta}{\mu} + 80\gamma_z^2 \eta^2 \mathcal{Z}_k .
\end{aligned}
\tag{48}
$$

Therefore, when $k \in \{2, \cdots, K-1\}$, we enforce

$$
\begin{aligned}
\mathcal{D}_{y^*,k} &= a_{k+1} \frac{10\gamma_y \eta}{\mu} L_g^2 + b_{k+1} \frac{32\gamma_z \eta}{\mu} L_H^2 \Delta_{k+1}^2 + b_k \frac{32\gamma_z \eta}{\mu} (L_H^2 \Delta_k^2 + L_J^2 \Delta_{k+1}^2) + b_{k-1} \frac{32\gamma_z \eta}{\mu} L_J^2 \Delta_k^2 \\
&\quad + 128\gamma_z^2 \eta^2 \mathcal{Z}_k (\Delta_k^2 L_H^2 + \Delta_{k+1}^2 L_J^2) + 128\gamma_z^2 \eta^2 \mathcal{Z}_{k+1} \Delta_{k+1}^2 L_H^2 + 128\gamma_z^2 \eta^2 \mathcal{Z}_{k-1} \Delta_k^2 L_J^2 - a_k \frac{\gamma_y \eta\mu}{4} \le 0 .
\end{aligned}
$$

Specifically, we enforce

$$
\begin{aligned}
a_{k+1} \frac{10\gamma_y \eta}{\mu} L_g^2 &\le \frac{\gamma_y \eta\mu}{16} a_k , \\
b_{k+1} \frac{32\gamma_z \eta}{\mu} \Delta_{k+1}^2 L_H^2 &+ b_k \frac{32\gamma_z \eta}{\mu} (\Delta_k^2 L_H^2 + \Delta_{k+1}^2 L_J^2) + b_{k-1} \frac{32\gamma_z \eta}{\mu} \Delta_k^2 L_J^2 \\
&+ 128\gamma_z^2 \eta^2 (\Delta_k^2 L_H^2 + \Delta_{k+1}^2 L_J^2) \mathcal{Z}_k + 128\gamma_z^2 \eta^2 \Delta_{k+1}^2 L_H^2 \mathcal{Z}_{k+1} + 128\gamma_z^2 \eta^2 \Delta_k^2 L_J^2 \mathcal{Z}_{k-1} \\
&\le \frac{\alpha_z \eta\mu}{24\gamma_z} d_{k+1} \frac{32\gamma_z \eta}{\mu} \Delta_{k+1}^2 L_H^2 + \frac{\alpha_z \eta\mu}{24\gamma_z} d_k \frac{32\gamma_z \eta}{\mu} (\Delta_k^2 L_H^2 + \Delta_{k+1}^2 L_J^2) + \frac{\alpha_z \eta\mu}{24\gamma_z} d_{k-1} \frac{32\gamma_z \eta}{\mu} \Delta_k^2 L_J^2 \\
&+ 128\gamma_z^2 \eta^2 (\Delta_k^2 L_H^2 + \Delta_{k+1}^2 L_J^2) \mathcal{Z}_k + 128\gamma_z^2 \eta^2 \Delta_{k+1}^2 L_H^2 \mathcal{Z}_{k+1} + 128\gamma_z^2 \eta^2 \Delta_k^2 L_J^2 \mathcal{Z}_{k-1} \\
&\le \frac{3\gamma_y \eta\mu}{16} a_k ,
\end{aligned}
$$

where we use $b_k \le \frac{\alpha_z \eta\mu}{24\gamma_z} d_k$ in the second step of the last inequality.

For the second inequality, from $L_g/\mu > 1$, for any $k \in \{2, \cdots, K-1\}$, we define

$$
\begin{aligned}
\mathcal{E}_a &= \frac{80C_J^2}{\alpha_x N} + \frac{480C_J^2 L_g^2}{\beta_z^2 (1-\lambda)^2 \mu^2} + \frac{80L_g^4}{\mu^2 \alpha_x N} + \frac{640L_g^4}{\beta_z^2 (1-\lambda)^2 \mu^2} + \frac{96C_J^2 L_g^2}{\beta_x^2 (1-\lambda)^2 \mu^2} \\
a_k &= \frac{160^{K-k} C_J^{2K}}{\mu^{2K}} \left( L_f^2 + \frac{C_J^2}{\mu^2} L_H^2 + \frac{L_g^2}{\mu^2} L_J^2 + L_H^2 \right) \frac{\gamma_x}{\gamma_y \mu} \left( 32 + \frac{16\alpha_z}{3\alpha_x} + 256\gamma_z^2 \mathcal{E}_a \right) \\
&\triangleq \Delta_{a_k} \frac{\gamma_x}{\gamma_y \mu} \left( 32 + \frac{16\alpha_z}{3\alpha_x} + 256\gamma_z^2 \mathcal{E}_a \right) .
\end{aligned}
$$

The cases $k = 1$ and $k = K$ follow by analogous arguments.

## G.4. Other errors related to x ($\mathcal{F}_4$)

The term $\mathcal{F}_4$ comprises three types of errors related to $x$: the variable feedback error, the variable consensus error, and the momentum consensus error. Specifically, we have

$$\mathcal{F}_4 \le \left( \omega \frac{8\rho_x \eta}{(1-\lambda)} + 16\rho_x^2 \eta^2 \mathcal{X} - \theta \frac{\delta}{4} \right) \frac{1}{N} \mathbb{E}[\|\hat{X}_t - X_t\|_F^2] + \mathcal{D}_x \frac{1}{N} \mathbb{E}[\|X_t(I - J)\|_F^2]$$

$$+ \left( \omega \frac{2\gamma_x^2 \eta}{\rho_x(1-\lambda)} + \theta \frac{12\gamma_x^2 \eta^2}{\delta} + \tilde{\theta} \frac{36\beta_x^2}{\delta} + 4\gamma_x^2 \eta^2 \mathcal{X} - \tilde{\omega}\beta_x(1-\lambda) \right) \frac{1}{N} \mathbb{E}[\|P_t(I-J)\|_F^2] \,.$$

where

$$\mathcal{X} = \frac{2L_g^2 c_1}{N} + \frac{8L_H^2 d_1}{N}\Delta_1^2 + \frac{8L_f^2 d_K}{N} + \frac{3e}{N}(\Delta_1^2 L_J^2 + L_f^2) + 2L_g^2 \tilde{c}_1 + 8L_H^2 \tilde{d}_1 \Delta_1^2 + 8L_f^2 \tilde{d}_K + 3\tilde{e}(\Delta_1^2 L_J^2 + L_f^2)$$

$$+ \frac{4L_g^2 \tilde{\nu}_1}{\beta_y(1-\lambda)} + \frac{16L_H^2 \tilde{\tau}_1 \Delta_1^2}{\beta_z(1-\lambda)} + \frac{16L_f^2 \tilde{\tau}_K}{\beta_z(1-\lambda)} + \frac{12\tilde{\omega}(\Delta_1^2 L_J^2 + L_f^2)}{\beta_x(1-\lambda)} + \frac{18L_g^2 \tilde{\phi}_1}{\delta} + \frac{72L_H^2 \tilde{\psi}_1}{\delta}\Delta_1^2 + \frac{72L_f^2 \tilde{\psi}_K}{\delta} + \frac{54\tilde{\theta}}{\delta}(\Delta_1^2 L_J^2 + L_f^2) \,,$$

$$\mathcal{D}_x = 6\eta\gamma_x(L_f^2 + \Delta_1^2 L_J^2) + a_1 \frac{20\gamma_y \eta L_g^2}{\mu} + b_1 \frac{48\gamma_z \eta}{\mu}\Delta_1^2 L_H^2 + b_K \frac{48\gamma_z \eta}{\mu}L_f^2 + \theta\frac{48\rho_x^2 \eta^2}{\delta} + 16\rho_x^2 \eta^2 \mathcal{X}$$

$$+ \psi_1 \frac{144\gamma_z^2 \eta^2}{\delta}\Delta_1^2 L_H^2 + \psi_K \frac{144 L_f^2 \gamma_z^2 \eta^2}{\delta} + 48\gamma_z^2 \eta^2 \mathcal{Z}_1 \Delta_1^2 L_H^2 + 48\gamma_z^2 \eta^2 L_f^2 \mathcal{Z}_K - \omega\rho_x\eta(1-\lambda) \,.$$

**(4.1) Feedback error regarding x**: To eliminate $\frac{1}{N}\mathbb{E}[\|\hat{X}_t - X_t\|_F^2]$, we enforce

$$\omega\frac{8\rho_x \eta}{(1-\lambda)} + 16\rho_x^2 \eta^2 \mathcal{X} - \theta\frac{\delta}{4} \leq 0 \,, \quad \text{and obtain} \quad \theta = \omega\frac{32\rho_x \eta}{\delta(1-\lambda)} + \frac{64}{\delta}\rho_x^2 \eta^2 \mathcal{X} \,. \tag{49}$$

**(4.2) Consensus error regarding x**: To eliminate $\frac{1}{N}\mathbb{E}[\|X_t(I-J)\|_F^2]$, following Eq. (48), we enforce

$$\mathcal{D}_x \leq 6\eta\gamma_x(L_f^2 + \Delta_1^2 L_J^2) + a_1 \frac{20\gamma_y \eta L_g^2}{\mu} + b_1 \frac{48\gamma_z \eta}{\mu}\Delta_1^2 L_H^2 + b_K \frac{48\gamma_z \eta}{\mu}L_f^2 + \theta\frac{48\rho_x^2 \eta^2}{\delta} + 16\rho_x^2 \eta^2 \mathcal{X} + b_1 \frac{17\gamma_z \eta}{\mu}\Delta_1^2 L_H^2$$

$$+ 80\gamma_z^2 \eta^2 \mathcal{Z}_1 \Delta_1^2 L_H^2 + b_K \frac{17\gamma_z \eta}{\mu}L_f^2 + 80\gamma_z^2 \eta^2 \mathcal{Z}_K L_f^2 + 48\gamma_z^2 \eta^2 \mathcal{Z}_1 \Delta_1^2 L_H^2 + 48\gamma_z^2 \eta^2 L_f^2 \mathcal{Z}_K - \omega\rho_x\eta(1-\lambda)$$

$$= 6\eta\gamma_x(L_f^2 + \Delta_1^2 L_J^2) + a_1 \frac{20\gamma_y \eta L_g^2}{\mu} + b_1 \frac{65\gamma_z \eta}{\mu}\Delta_1^2 L_H^2 + b_K \frac{65\gamma_z \eta}{\mu}L_f^2 + \theta\frac{48\rho_x^2 \eta^2}{\delta} + 16\rho_x^2 \eta^2 \mathcal{X}$$

$$+ 128\gamma_z^2 \eta^2 \mathcal{Z}_1 \Delta_1^2 L_H^2 + 128\gamma_z^2 \eta^2 L_f^2 \mathcal{Z}_K - \omega\rho_x\eta(1-\lambda) \leq 0 \,.$$

From the definition of $\theta$, we have

$$6\eta\gamma_x(L_f^2 + \Delta_1^2 L_J^2) + a_1 \frac{20\gamma_y \eta L_g^2}{\mu} + b_1 \frac{65\gamma_z \eta}{\mu}\Delta_1^2 L_H^2 + b_K \frac{65\gamma_z \eta}{\mu}L_f^2 + \omega\frac{32\rho_x \eta}{\delta(1-\lambda)}\frac{48\rho_x^2 \eta^2}{\delta} + \frac{64}{\delta}\rho_x^2 \eta^2 \mathcal{X}\frac{48\rho_x^2 \eta^2}{\delta}$$

$$+ 16\rho_x^2 \eta^2 \mathcal{X} + 128\gamma_z^2 \eta^2 \mathcal{Z}_1 \Delta_1^2 L_H^2 + 128\gamma_z^2 \eta^2 L_f^2 \mathcal{Z}_K - \omega\rho_x\eta(1-\lambda) \leq 0 \,.$$

Firstly, we enforce

$$\omega\frac{32\rho_x \eta}{\delta(1-\lambda)}\frac{48\rho_x^2 \eta^2}{\delta} \leq \frac{1}{12}\omega\rho_x\eta(1-\lambda) \,, \quad \text{and obtain} \quad \rho_x \leq \frac{\delta(1-\lambda)}{96\sqrt{2}} \,. \tag{50}$$

Then from $\rho_x \leq \frac{\delta}{8\sqrt{3}}$, we have

$$6\eta\gamma_x(L_f^2 + \Delta_1^2 L_J^2) + a_1 \frac{20\gamma_y \eta L_g^2}{\mu} + b_1 \frac{65\gamma_z \eta}{\mu}\Delta_1^2 L_H^2 + b_K \frac{65\gamma_z \eta}{\mu}L_f^2$$

$$+ 32\rho_x^2 \eta^2 \mathcal{X} + 128\gamma_z^2 \eta^2 \mathcal{Z}_1 \Delta_1^2 L_H^2 + 128\gamma_z^2 \eta^2 L_f^2 \mathcal{Z}_K \leq \frac{11}{12}\omega\rho_x\eta(1-\lambda) \,.$$

Then, due to $b_k \leq \frac{\alpha_z \eta\mu}{24\gamma_z}d_k$ and the upper bound of $\gamma_z$, we have

$$6\eta\gamma_x L_f^2 + \frac{20\gamma_x \eta L_g^2}{\mu^2}\Delta_{a_1}\left( 32 + \frac{5\alpha_z}{2\alpha_x} + \frac{12C_J^2}{\mu^2} + \frac{15L_g^2}{\mu^2} \right)$$

$$+ \frac{3C_J^{2K}}{8\mu^{2K}}\left( 16\eta\gamma_x L_J^2 + \eta\gamma_x \frac{3\alpha_z}{\alpha_x}(L_H^2 \frac{C_J^2}{\mu^2} + L_f^2) + 4\eta\gamma_x(\frac{C_J^2}{\mu^2}L_H^2 + L_f^2) \right) + 32\rho_x^2 \eta^2 \mathcal{X} - \frac{1}{2}\omega\rho_x\eta(1-\lambda) \leq 0 \,.$$

Then, we set

$$\Delta_{aa} = \left( 32 + \frac{5\alpha_z}{2\alpha_x} + \frac{12C_J^2}{\mu^2} + \frac{15L_g^2}{\mu^2} \right) \,, \qquad \Delta_{xx} = \left( 16L_J^2 + \frac{3\alpha_z}{\alpha_x}(L_H^2 \frac{C_J^2}{\mu^2} + L_f^2) + 4(\frac{C_J^2}{\mu^2}L_H^2 + L_f^2) \right) \,,$$

$$\mathcal{E}_{1,x} = 6L_f^2 + \frac{20L_g^2}{\mu^2}\Delta_{a_1}\Delta_{aa} + \frac{3C_J^{2K}}{8\mu^{2K}}\Delta_{xx}\,, \qquad \mathcal{E}_{2,x} = 32\rho_x^2\eta^2\mathcal{X}\,, \qquad \omega = \frac{2\gamma_x\mathcal{E}_{1,x}}{\rho_x(1-\lambda)} + \frac{2\mathcal{E}_{2,x}}{\rho_x\eta(1-\lambda)}\,.$$

**(4.3) Consensus error regarding momentum of x**: In the following, we aim to set the coefficient of $\frac{1}{N}\mathbb{E}[\|P_t(I-J)\|_F^2]$ to be non-positive, i.e.,

$$\omega\frac{2\gamma_x^2\eta}{\rho_x(1-\lambda)} + \theta\frac{12\gamma_x^2\eta^2}{\delta} + \tilde{\theta}\frac{36\beta_x^2}{\delta} + 4\gamma_x^2\eta^2\mathcal{X} - \tilde{\omega}\beta_x(1-\lambda) \le 0\,.$$

Firstly, we enforce

$$\tilde{\omega}\frac{32\beta_x}{\delta(1-\lambda)}\frac{36\beta_x^2}{\delta} \le \frac{1}{3}\tilde{\omega}\beta_x(1-\lambda)\,, \quad \text{and obtain} \quad \beta_x \le \frac{\delta(1-\lambda)}{24\sqrt{6}}\,.$$

Then, from the definition of $\omega$, we enforce

$$\frac{2\gamma_x\mathcal{E}_{1,x}}{\rho_x(1-\lambda)}\frac{2\gamma_x^2\eta}{\rho_x(1-\lambda)} + \frac{4\gamma_x^2\mathcal{E}_{2,x}}{\rho_x^2(1-\lambda)^2} + \theta\frac{12\gamma_x^2\eta^2}{\delta} + 4\gamma_x^2\eta^2\mathcal{X} \le \frac{2}{3}\tilde{\omega}\beta_x(1-\lambda)\,.$$

Specifically, from the definition of $\tilde{\omega} = \frac{\gamma_x}{\beta_x(1-\lambda)}\frac{3C_J^2}{8L_g^2}$, we enforce

$$\frac{2\gamma_x\mathcal{E}_{1,x}}{\rho_x(1-\lambda)}\frac{2\gamma_x^2\eta}{\rho_x(1-\lambda)} \le \frac{\gamma_x}{6}\frac{3C_J^2}{8L_g^2}\,, \qquad \frac{2\mathcal{E}_{2,x}}{\rho_x\eta(1-\lambda)}\frac{2\gamma_x^2\eta}{\rho_x(1-\lambda)} \le \frac{\gamma_x}{6}\frac{3C_J^2}{8L_g^2}\,,$$

$$\theta\frac{12\gamma_x^2\eta^2}{\delta} \le \frac{\gamma_x}{6}\frac{3C_J^2}{8L_g^2}\,, \qquad 4\gamma_x^2\eta^2\mathcal{X} \le \frac{\gamma_x}{12}\frac{3C_J^2}{8L_g^2}\,. \tag{51}$$

To solve the last inequality in Eq. (51), from the definition of $\mathcal{X}$, we have

$$\mathcal{X} \le \frac{3C_J^{2K}}{8\mu^{2K}}\left(\frac{8\gamma_x}{\alpha_x\eta N}(L_f^2 + \frac{C_J^2}{\mu^2}L_H^2 + L_J^2) + \left(\frac{160L_H^2\gamma_x}{\beta_z^2(1-\lambda)^2} + \frac{72L_J^2\gamma_x}{\beta_x^2(1-\lambda)^2}\right)\frac{C_J^2}{L_g^2}\right) + \frac{3C_J^{2K}}{8L_g^{2K}}\frac{160L_f^2\gamma_x}{\beta_z^2(1-\lambda)^2}$$

$$+ \left(\frac{72L_f^2\gamma_x}{\beta_x^2(1-\lambda)^2} + \frac{40L_g^2\gamma_x}{\beta_y^2(1-\lambda)^2}\right)\frac{3C_J^2}{8L_g^2} + \frac{3\gamma_x}{\alpha_x\eta N}L_f^2 + \frac{2}{N}\frac{20\gamma_xL_g^2}{\alpha_y\eta\mu^2}\Delta_{a_1}\Delta_{aa}\,.$$

Therefore, by setting $\Delta_L^2 = (L_f^2 + \frac{C_J^2}{\mu^2}L_H^2 + L_J^2)$, we enforce

$$4\gamma_x^2\eta^2\frac{3C_J^{2K}}{8\mu^{2K}}\frac{8\gamma_x}{\alpha_x\eta N}\Delta_L^2 \le \frac{\gamma_x}{96}\frac{3C_J^2}{8L_g^2}\,, \qquad 4\gamma_x^2\eta^2\frac{3C_J^{2K}}{8\mu^{2K}}\frac{160L_H^2\gamma_x}{\beta_z^2(1-\lambda)^2}\frac{C_J^2}{L_g^2} \le \frac{\gamma_x}{96}\frac{3C_J^2}{8L_g^2}\,,$$

$$4\gamma_x^2\eta^2\frac{3C_J^{2K}}{8\mu^{2K}}\frac{72L_J^2\gamma_x}{\beta_x^2(1-\lambda)^2}\frac{C_J^2}{L_g^2} \le \frac{\gamma_x}{96}\frac{3C_J^2}{8L_g^2}\,, \qquad 4\gamma_x^2\eta^2\frac{3C_J^{2K}}{8L_g^{2K}}\frac{160L_f^2\gamma_x}{\beta_z^2(1-\lambda)^2} \le \frac{\gamma_x}{96}\frac{3C_J^2}{8L_g^2}\,,$$

$$4\gamma_x^2\eta^2\frac{72L_f^2\gamma_x}{\beta_x^2(1-\lambda)^2}\frac{3C_J^2}{8L_g^2} \le \frac{\gamma_x}{96}\frac{3C_J^2}{8L_g^2}\,, \qquad 4\gamma_x^2\eta^2\frac{40L_g^2\gamma_x}{\beta_y^2(1-\lambda)^2}\frac{3C_J^2}{8L_g^2} \le \frac{\gamma_x}{96}\frac{3C_J^2}{8L_g^2}\,,$$

$$4\gamma_x^2\eta^2\frac{3\gamma_x}{\alpha_x\eta N}L_f^2 \le \frac{\gamma_x}{96}\frac{3C_J^2}{8L_g^2}\,, \qquad 4\gamma_x^2\eta^2\frac{2}{N}\frac{20\gamma_xL_g^2}{\alpha_y\eta\mu^2}\Delta_{a_1}\Delta_{aa} \le \frac{\gamma_x}{96}\frac{3C_J^2}{8L_g^2}\,. \tag{52}$$

Then, we obtain

$$\gamma_x \le \Big\{\frac{\mu^K}{L_gC_J^{K-1}}\frac{\sqrt{\alpha_xN}}{32\sqrt{3}\Delta_L}\,, \frac{\mu^K}{C_J^K}\frac{\beta_z(1-\lambda)}{64\sqrt{15}L_H}\,, \frac{\mu^K}{C_J^K}\frac{\beta_x(1-\lambda)}{96\sqrt{3}L_J}\,, \frac{L_g^{K-1}}{C_J^{K-1}}\frac{\beta_z(1-\lambda)}{64\sqrt{15}L_f}\,,$$

$$\frac{\beta_x(1-\lambda)}{96\sqrt{3}L_f}\,, \frac{\beta_y(1-\lambda)}{32\sqrt{15}L_g}\,, \frac{\sqrt{\alpha_xN}C_J}{32\sqrt{3}L_fL_g}\,, \frac{\sqrt{\alpha_yN}\mu}{64\sqrt{10}\Delta_{a_1}\Delta_{aa}}\frac{C_J}{L_g^2}\Big\}\,.$$

To solve the third inequality in Eq. (51), from the definition of $\theta$, we enforce

$$\omega\frac{32\rho_x\eta}{\delta(1-\lambda)}\frac{12\gamma_x^2\eta^2}{\delta} \le \frac{\gamma_x}{12}\frac{3C_J^2}{8L_g^2}\,, \qquad \frac{64}{\delta}\rho_x^2\eta^2\mathcal{X}\frac{12\gamma_x^2\eta^2}{\delta} \le \frac{\gamma_x}{12}\frac{3C_J^2}{8L_g^2}\,.$$

Since $\rho_x \leq \frac{\delta}{8\sqrt{3}}$, the second inequality holds. To solve the first inequality, from the definition of $\omega$, we enforce

$$\frac{2\gamma_x \mathcal{E}_{1,x}}{\rho_x(1-\lambda)} \frac{32\rho_x \eta}{\delta(1-\lambda)} \frac{12\gamma_x^2 \eta^2}{\delta} \leq \frac{\gamma_x}{16} \frac{3C_J^2}{8L_g^2} \,, \qquad \frac{2\mathcal{E}_{2,x}}{\rho_x \eta(1-\lambda)} \frac{32\rho_x \eta}{\delta(1-\lambda)} \frac{12\gamma_x^2 \eta^2}{\delta} \leq \frac{\gamma_x}{64} \frac{3C_J^2}{8L_g^2} \,.$$

From Eq. (50), the second inequality is already satisfied. To solve the first one, from the definition of $\mathcal{E}_{1,x}$, we enforce

$$6L_f^2 \frac{64\eta\gamma_x}{(1-\lambda)^2} \frac{12\gamma_x^2 \eta^2}{\delta^2} \leq \frac{\gamma_x}{48} \frac{3C_J^2}{8L_g^2} \,, \qquad \frac{20L_g^2}{\mu^2} \Delta_{a_1} \Delta_{aa} \frac{64\eta\gamma_x}{(1-\lambda)^2} \frac{12\gamma_x^2 \eta^2}{\delta^2} \leq \frac{\gamma_x}{48} \frac{3C_J^2}{8L_g^2} \,,$$

$$\frac{3C_J^{2K}}{8\mu^{2K}} \Delta_{xx} \frac{64\eta\gamma_x}{(1-\lambda)^2} \frac{12\gamma_x^2 \eta^2}{\delta^2} \leq \frac{\gamma_x}{48} \frac{3C_J^2}{8L_g^2} \,.$$

Then, we obtain $\gamma_x \leq \left\{ \frac{\delta(1-\lambda)C_J}{768L_f L_g} \,, \frac{\mu\delta(1-\lambda)C_J}{256\sqrt{30\Delta_{a_1}\Delta_{aa}}L_g^2} \,, \frac{\mu^K}{L_g C_J^{K-1}} \frac{\delta(1-\lambda)}{192\sqrt{\Delta_{xx}}} \right\}$.

To solve the first inequality in Eq. (51), from the definition of $\mathcal{E}_{1,x}$, we enforce

$$6L_f^2 \frac{2\gamma_x}{\rho_x(1-\lambda)} \frac{2\gamma_x^2 \eta}{\rho_x(1-\lambda)} \leq \frac{\gamma_x}{18} \frac{3C_J^2}{8L_g^2} \,, \qquad \frac{20L_g^2}{\mu^2} \Delta_{a_1} \Delta_{aa} \frac{2\gamma_x}{\rho_x(1-\lambda)} \frac{2\gamma_x^2 \eta}{\rho_x(1-\lambda)} \leq \frac{\gamma_x}{18} \frac{3C_J^2}{8L_g^2} \,,$$

$$\frac{3C_J^{2K}}{8\mu^{2K}} \Delta_{xx} \frac{2\gamma_x}{\rho_x(1-\lambda)} \frac{2\gamma_x^2 \eta}{\rho_x(1-\lambda)} \leq \frac{\gamma_x}{18} \frac{3C_J^2}{8L_g^2} \,.$$

Then, we obtain $\gamma_x \leq \left\{ \frac{\rho_x(1-\lambda)C_J}{24\sqrt{2}L_f L_g} \,, \frac{\mu\rho_x(1-\lambda)C_J}{16\sqrt{15\Delta_{a_1}\Delta_{aa}}L_g^2} \,, \frac{\mu^K}{L_g C_J^{K-1}} \frac{\rho_x(1-\lambda)}{6\sqrt{2\Delta_{xx}}} \right\}$.

To solve the second inequality in Eq. (51), from the definition of $\mathcal{E}_{2,x}$, we have

$$32\mathcal{X} \frac{4\gamma_x^2 \eta^2}{(1-\lambda)^2} \leq \frac{\gamma_x}{6} \frac{3C_J^2}{8L_g^2} \,.$$

Then, we solve it as in Eq. (52). We omit the details here and obtain

$$\gamma_x \leq \left\{ \frac{\mu^K}{L_g C_J^{K-1}} \frac{(1-\lambda)\sqrt{\alpha_x N}}{128\sqrt{3}\Delta_L} \,, \frac{\mu^K}{C_J^K} \frac{\beta_z(1-\lambda)^2}{256\sqrt{15}L_H} \,, \frac{\mu^K}{C_J^K} \frac{\beta_x(1-\lambda)^2}{384\sqrt{3}L_J} \,, \frac{L_g^{K-1}}{C_J^{K-1}} \frac{\beta_z(1-\lambda)^2}{256\sqrt{15}L_f} \,, \right.$$

$$\left. \frac{\beta_x(1-\lambda)^2}{384\sqrt{3}L_f} \,, \frac{\beta_y(1-\lambda)^2}{128\sqrt{15}L_g} \,, \frac{(1-\lambda)\sqrt{\alpha_x N}C_J}{128\sqrt{3}L_f L_g} \,, \frac{(1-\lambda)\sqrt{\alpha_y N}\mu}{256\sqrt{10\Delta_{a_1}\Delta_{aa}}} \frac{C_J}{L_g^2} \right\} \,.$$

## G.5. Other errors related to y ($\mathcal{F}_5$)

The term $\mathcal{F}_5$ comprises three types of errors related to $y$: the variable feedback error, the variable consensus error, and the momentum consensus error. Specifically, we have

$$\mathcal{F}_5 \leq \sum_{k=1}^{K} \left( \nu_k \frac{8\rho_y \eta}{(1-\lambda)} + 16\rho_y^2 \eta^2 \mathcal{Y}_k - \phi_k \frac{\delta}{4} \right) \frac{1}{N} \mathbb{E}[\|\hat{Y}_{k,t} - Y_{k,t}\|_F^2] + \sum_{k=1}^{K} \mathcal{D}_{y,k} \frac{1}{N} \mathbb{E}[\|Y_{k,t}(I-J)\|_F^2]$$

$$+ \sum_{k=1}^{K} \left( \nu_k \frac{2\gamma_y^2 \eta}{\rho_y(1-\lambda)} + \phi_k \frac{12\gamma_y^2 \eta^2}{\delta} + \tilde{\phi}_k \frac{36\beta_y^2}{\delta} + 4\gamma_y^2 \eta^2 \mathcal{Y}_k - \tilde{\nu}_k \beta_y(1-\lambda) \right) \frac{1}{N} \mathbb{E}[\|Q_{k,t}(I-J)\|_F^2] \,,$$

where the definitions of $\mathcal{D}_{y,k}$ and $\mathcal{Y}_k$ will be specified in the following analysis.

**(5.1) Feedback error regarding y**: To eliminate $\frac{1}{N}\mathbb{E}[\|\hat{Y}_{k,t} - Y_{k,t}\|_F^2]$, when $k \in \{1, \cdots, K\}$,

$$\nu_k \frac{8\rho_y \eta}{(1-\lambda)} + 16\rho_y^2 \eta^2 \mathcal{Y}_k - \phi_k \frac{\delta}{4} \leq 0 \,, \quad \text{and obtain} \quad \phi_k = \nu_k \frac{32\rho_y \eta}{\delta(1-\lambda)} + \frac{64}{\delta} \rho_y^2 \eta^2 \mathcal{Y}_k \,.$$

**(5.2) Consensus error regarding y**: When $k = 1$, to eliminate $\frac{1}{N}\mathbb{E}[\|Y_{1,t}(I-J)\|_F^2]$, we enforce

$$\mathcal{D}_{y,1} = 6\eta\gamma_x \Delta_1^2 L_J^2 + a_1 \frac{20\gamma_y \eta L_g^2}{\mu} + a_2 \frac{20\gamma_y \eta L_g^2}{\mu} + b_1 \frac{48\gamma_z \eta}{\mu}(\Delta_1^2 L_H^2 + \Delta_2^2 L_J^2) + b_2 \frac{48\gamma_z \eta}{\mu} \Delta_2^2 L_H^2 + \phi_1 \frac{48\rho_y^2 \eta^2}{\delta}$$

$$+ \psi_1 \frac{144\gamma_z^2\eta^2}{\delta}(\Delta_1^2 L_H^2 + \Delta_2^2 L_J^2) + \psi_2 \frac{144\gamma_z^2\eta^2}{\delta}\Delta_2^2 L_H^2 + 48\gamma_z^2\eta^2(\Delta_1^2 L_H^2 + \Delta_2^2 L_J^2)\mathcal{Z}_1 + 48\gamma_z^2\eta^2 \mathcal{Z}_2\Delta_2^2 L_H^2$$
$$+ 16\rho_y^2\eta^2 \mathcal{Y}_1 - \nu_1\rho_y\eta(1-\lambda) \leq 0 \,.$$

From Eq. (48) and the definition of $\phi_k$, we have

$$\mathcal{D}_{y,1} \leq 6\eta\gamma_x\Delta_1^2 L_J^2 + (a_1 + a_2)\frac{20\gamma_y\eta L_g^2}{\mu} + b_1\frac{65\gamma_z\eta}{\mu}(\Delta_1^2 L_H^2 + \Delta_2^2 L_J^2) + b_2\frac{65\gamma_z\eta}{\mu}\Delta_2^2 L_H^2 + 128\gamma_z^2\eta^2(\Delta_1^2 L_H^2 + \Delta_2^2 L_J^2)\mathcal{Z}_1$$
$$+ 128\gamma_z^2\eta^2 \mathcal{Z}_2\Delta_2^2 L_H^2 + \nu_1\frac{32\rho_y\eta}{\delta(1-\lambda)}\frac{48\rho_y^2\eta^2}{\delta} + \frac{64}{\delta}\rho_y^2\eta^2 \mathcal{Y}_1\frac{48\rho_y^2\eta^2}{\delta} + 16\rho_y^2\eta^2 \mathcal{Y}_1 - \nu_1\rho_y\eta(1-\lambda) \leq 0 \,.$$

Firstly, we enforce

$$\nu_1\frac{32\rho_y\eta}{\delta(1-\lambda)}\frac{48\rho_y^2\eta^2}{\delta} \leq \frac{1}{12}\nu_1\rho_y\eta(1-\lambda) \,, \quad \text{and obtain} \quad \rho_y \leq \frac{\delta(1-\lambda)}{96\sqrt{2}} \,. \tag{53}$$

Then from $\rho_y \leq \frac{\delta}{8\sqrt{3}}$, we have

$$6\eta\gamma_x\Delta_1^2 L_J^2 + (a_1 + a_2)\frac{20\gamma_y\eta L_g^2}{\mu} + b_1\frac{65\gamma_z\eta}{\mu}(\Delta_1^2 L_H^2 + \Delta_2^2 L_J^2) + b_2\frac{65\gamma_z\eta}{\mu}\Delta_2^2 L_H^2$$
$$+ 128\gamma_z^2\eta^2(\Delta_1^2 L_H^2 + \Delta_2^2 L_J^2)\mathcal{Z}_1 + 128\gamma_z^2\eta^2 \mathcal{Z}_2\Delta_2^2 L_H^2 + 32\rho_y^2\eta^2 \mathcal{Y}_1 \leq \frac{11}{12}\nu_1\rho_y\eta(1-\lambda) \,.$$

Then, due to $b_k \leq \frac{\alpha_z\eta\mu}{24\gamma_z}d_k$ and the upper bound of $\gamma_z$, we have

$$\frac{20\gamma_x\eta L_g^2}{\mu^2}(\Delta_{a_1} + \Delta_{a_2})\left(32 + \frac{5\alpha_z}{2\alpha_x} + \frac{12C_J^2}{\mu^2} + \frac{15L_g^2}{\mu^2}\right)$$
$$+ \frac{3C_J^{2K}}{8\mu^{2K}}\left(16\eta\gamma_x L_J^2 + \eta\gamma_x\frac{6\alpha_z}{\alpha_x}(\frac{C_J^2}{\mu^2}L_H^2 + L_J^2) + 8\eta\gamma_x(\frac{C_J^2}{\mu^2}L_H^2 + L_J^2)\right) + 32\rho_y^2\eta^2 \mathcal{Y}_1 - \frac{1}{2}\nu_1\rho_y\eta(1-\lambda) \leq 0 \,.$$

Therefore, we set

$$\Delta_{yy} = \left(16L_J^2 + \frac{6\alpha_z}{\alpha_x}(L_f^2 + \frac{C_J^2}{\mu^2}L_H^2 + L_J^2) + 8(L_f^2 + \frac{C_J^2}{\mu^2}L_H^2 + L_J^2)\right)$$
$$\mathcal{E}_{1,y1} = \frac{20L_g^2}{\mu^2}(\Delta_{a_1} + \Delta_{a_2})\Delta_{aa} + \frac{3C_J^{2K}}{8\mu^{2K}}\Delta_{yy} \,, \qquad \mathcal{E}_{2,y1} = 32\rho_y^2\eta^2 \mathcal{Y}_1 \,, \qquad \nu_1 = \frac{2\gamma_x\mathcal{E}_{1,y1}}{\rho_y(1-\lambda)} + \frac{2\mathcal{E}_{2,y1}}{\rho_y\eta(1-\lambda)} \,.$$

When $k = K$, to eliminate $\frac{1}{N}\mathbb{E}[\|Y_{K,t}(I-J)\|_F^2]$, we enforce

$$\mathcal{D}_{y,K} = 6\eta\gamma_x L_f^2 + a_K\frac{20\gamma_y\eta L_g^2}{\mu} + b_K\frac{48\gamma_z\eta}{\mu}(\Delta_K^2 L_H^2 + L_f^2) + b_{K-1}\frac{48\gamma_z\eta}{\mu}\Delta_K^2 L_J^2 + \phi_K\frac{48\rho_y^2\eta^2}{\delta}$$
$$+ \psi_{K-1}\frac{144\gamma_z^2\eta^2}{\delta}\Delta_K^2 L_J^2 + \psi_K\frac{144\gamma_z^2\eta^2}{\delta}(\Delta_K^2 L_H^2 + L_f^2) + 48\gamma_z^2\eta^2 \mathcal{Z}_{K-1}\Delta_K^2 L_J^2 + 48\gamma_z^2\eta^2 \mathcal{Z}_K(\Delta_K^2 L_H^2 + L_f^2)$$
$$+ 16\rho_y^2\eta^2 \mathcal{Y}_K - \nu_K\rho_y\eta(1-\lambda) \leq 0 \,.$$

This inequality can be handled analogously. Therefore, we set

$$\mathcal{E}_{1,yK} = 6L_f^2 + \frac{20L_g^2}{\mu^2}\Delta_{a_K}\Delta_{aa} + \frac{3C_J^{2K}}{8\mu^{2K}}\Delta_{yy} \,, \qquad \mathcal{E}_{2,yK} = 32\rho_y^2\eta^2 \mathcal{Y}_K \,, \qquad \nu_K = \frac{2\gamma_x\mathcal{E}_{1,yK}}{\rho_y(1-\lambda)} + \frac{2\mathcal{E}_{2,yK}}{\rho_y\eta(1-\lambda)} \,.$$

When $k \in \{2, \cdots, K-1\}$, to eliminate $\frac{1}{N}\mathbb{E}[\|Y_{k,t}(I-J)\|_F^2]$, we enforce

$$\mathcal{D}_{y,k} = a_k\frac{20\gamma_y\eta L_g^2}{\mu} + a_{k+1}\frac{20\gamma_y\eta L_g^2}{\mu} + b_k\frac{48\gamma_z\eta}{\mu}(\Delta_k^2 L_H^2 + \Delta_{k+1}^2 L_J^2) + b_{k+1}\frac{48\gamma_z\eta}{\mu}\Delta_{k+1}^2 L_H^2 + b_{k-1}\frac{40\gamma_z\eta}{\mu}\Delta_k^2 L_J^2$$
$$+ \psi_k\frac{144\gamma_z^2\eta^2}{\delta}(\Delta_k^2 L_H^2 + \Delta_{k+1}^2 L_J^2) + \psi_{k+1}\frac{144\gamma_z^2\eta^2}{\delta}\Delta_{k+1}^2 L_H^2 + \psi_{k-1}\frac{144\gamma_z^2\eta^2}{\delta}\Delta_k^2 L_J^2$$
$$+ 48\gamma_z^2\eta^2(\Delta_k^2 L_H^2 + \Delta_{k+1}^2 L_J^2)\mathcal{Z}_k + 48\gamma_z^2\eta^2 \mathcal{Z}_{k+1}\Delta_{k+1}^2 L_H^2 + 48\gamma_z^2\eta^2 \mathcal{Z}_{k-1}\Delta_k^2 L_J^2$$

$$+ \phi_k \frac{48\rho_y^2\eta^2}{\delta} + 16\rho_y^2\eta^2 \mathcal{Y}_k - \nu_k\rho_y\eta(1-\lambda) \leq 0 \,,$$

and we obtain

$$\mathcal{E}_{1,yk} = \frac{20L_g^2}{\mu^2}(\Delta_{a_k} + \Delta_{a_{k+1}})\Delta_{aa} + \frac{3C_J^{2K}}{8\mu^{2K}}\Delta_{yy} \,, \qquad \mathcal{E}_{2,yk} = 32\rho_y^2\eta^2\mathcal{Y}_k \,, \qquad \nu_k = \frac{2\gamma_x\mathcal{E}_{1,yk}}{\rho_y(1-\lambda)} + \frac{2\mathcal{E}_{2,yk}}{\rho_y\eta(1-\lambda)} \,.$$

**(5.3) Consensus error regarding momentum of y**: In the following, we aim to set the coefficient of $\frac{1}{N}\mathbb{E}[\|Q_{t,k}(I-J)\|_F^2]$ to be non-positive when $k \in \{1, \cdots, K\}$, i.e.,

$$\nu_k \frac{2\gamma_y^2\eta}{\rho_y(1-\lambda)} + \phi_k \frac{12\gamma_y^2\eta^2}{\delta} + \tilde{\phi}_k \frac{36\beta_y^2}{\delta} + 4\gamma_y^2\eta^2\mathcal{Y}_k - \tilde{\nu}_k\beta_y(1-\lambda) \leq 0 \,.$$

Firstly, from the definition of $\tilde{\phi}_k$, we enforce

$$\tilde{\nu}_k \frac{32\beta_y}{\delta(1-\lambda)} \frac{36\beta_y^2}{\delta} \leq \frac{1}{3}\tilde{\nu}_k\beta_y(1-\lambda) \,, \quad \text{and obtain} \quad \beta_y \leq \frac{\delta}{24\sqrt{6}}(1-\lambda) \,.$$

Then, from the definition of $\nu_k$, we enforce

$$\frac{2\gamma_x\mathcal{E}_{1,yk}}{\rho_y(1-\lambda)} \frac{2\gamma_y^2\eta}{\rho_y(1-\lambda)} + \frac{2\mathcal{E}_{2,yk}}{\rho_y\eta(1-\lambda)} \frac{2\gamma_y^2\eta}{\rho_y(1-\lambda)} + \phi_k \frac{12\gamma_y^2\eta^2}{\delta} + 4\gamma_y^2\eta^2\mathcal{Y}_k \leq \frac{2}{3}\tilde{\nu}_k\beta_y(1-\lambda) \,.$$

Specifically, from the definition of $\tilde{\nu}_k = \frac{\gamma_x}{\beta_y(1-\lambda)}\frac{3C_J^{2k}}{8L_g^{2k}}$, we enforce

$$\frac{2\gamma_x\mathcal{E}_{1,yk}}{\rho_y(1-\lambda)} \frac{2\gamma_y^2\eta}{\rho_y(1-\lambda)} \leq \frac{\gamma_x}{6}\frac{3C_J^{2k}}{8L_g^{2k}} \,, \qquad \frac{2\mathcal{E}_{2,yk}}{\rho_y\eta(1-\lambda)} \frac{2\gamma_y^2\eta}{\rho_y(1-\lambda)} \leq \frac{\gamma_x}{6}\frac{3C_J^{2k}}{8L_g^{2k}} \,,$$

$$\phi_k \frac{12\gamma_y^2\eta^2}{\delta} \leq \frac{\gamma_x}{6}\frac{3C_J^{2k}}{8L_g^{2k}} \,, \qquad 4\gamma_y^2\eta^2\mathcal{Y}_k \leq \frac{\gamma_x}{12}\frac{3C_J^{2k}}{8L_g^{2k}} \,. \tag{54}$$

When $k \in \{2, \cdots, K-1\}$, we have

$$
\begin{aligned}
\mathcal{Y}_k ={}& \frac{2L_g^2 c_k}{N} + \frac{2L_g^2 c_{k+1}}{N} + \frac{8d_k}{N}(\Delta_k^2 L_H^2 + \Delta_{k+1}^2 L_J^2) + \frac{8d_{k-1}}{N}\Delta_k^2 L_J^2 + \frac{8d_{k+1}}{N}\Delta_{k+1}^2 L_H^2 \\
&+ 2L_g^2\tilde{c}_k + 2L_g^2\tilde{c}_{k+1} + 8\tilde{d}_k(\Delta_k^2 L_H^2 + \Delta_{k+1}^2 L_J^2) + 8\tilde{d}_{k-1}\Delta_k^2 L_J^2 + 8\tilde{d}_{k+1}\Delta_{k+1}^2 L_H^2 \\
&+ \frac{4L_g^2\tilde{\nu}_k}{\beta_y(1-\lambda)} + \frac{4L_g^2\tilde{\nu}_{k+1}}{\beta_y(1-\lambda)} + \frac{16\tilde{\tau}_k}{\beta_z(1-\lambda)}(\Delta_k^2 L_H^2 + \Delta_{k+1}^2 L_J^2) + \frac{16\tilde{\tau}_{k-1}\Delta_k^2 L_J^2}{\beta_z(1-\lambda)} + \frac{16\tilde{\tau}_{k+1}\Delta_{k+1}^2 L_H^2}{\beta_z(1-\lambda)} \\
&+ \frac{18L_g^2\tilde{\phi}_k}{\delta} + \frac{18L_g^2\tilde{\phi}_{k+1}}{\delta} + \frac{72\tilde{\psi}_k}{\delta}(\Delta_k^2 L_H^2 + \Delta_{k+1}^2 L_J^2) + \frac{72\tilde{\psi}_{k-1}}{\delta}\Delta_k^2 L_J^2 + \frac{72\tilde{\psi}_{k+1}}{\delta}\Delta_{k+1}^2 L_H^2 \,, \\
\leq{}& \frac{2}{N}\frac{20\gamma_x L_g^2}{\alpha_y\eta\mu^2}(\Delta_{a_k} + \Delta_{a_{k+1}})\Delta_{aa} + \frac{40\gamma_x(L_g^2 + C_J^2)}{\beta_y^2(1-\lambda)^2}\frac{3C_J^{2k}}{8L_g^{2k}} + \frac{3C_J^{2K}}{8\mu^{2K}}\frac{16\gamma_x}{\alpha_x\eta N}\Delta_L^2 + \frac{3C_J^{2K}}{8\mu^{2(K-k+1)}L_g^{2(k-1)}}\frac{320\gamma_x}{\beta_z^2(1-\lambda)^2}\Delta_L^2 \,.
\end{aligned}
$$

Therefore, to solve the last inequality in Eq. (54), we enforce

$$4\gamma_y^2\eta^2 \frac{2}{N}\frac{20\gamma_x L_g^2}{\alpha_y\eta\mu^2}(\Delta_{a_k} + \Delta_{a_{k+1}})\Delta_{aa} \leq \frac{\gamma_x}{60}\frac{3C_J^{2k}}{8L_g^{2k}} \,, \qquad 4\gamma_y^2\eta^2 \frac{40\gamma_x(L_g^2 + C_J^2)}{\beta_y^2(1-\lambda)^2}\frac{3C_J^{2k}}{8L_g^{2k}} \leq \frac{\gamma_x}{60}\frac{3C_J^{2k}}{8L_g^{2k}} \,,$$

$$4\gamma_y^2\eta^2 \frac{3C_J^{2K}}{8\mu^{2K}}\frac{16\gamma_x}{\alpha_x\eta N}\Delta_L^2 \leq \frac{\gamma_x}{60}\frac{3C_J^{2k}}{8L_g^{2k}} \,, \qquad 4\gamma_y^2\eta^2 \frac{3C_J^{2K}}{8\mu^{2(K-k+1)}L_g^{2(k-1)}}\frac{320\gamma_x}{\beta_z^2(1-\lambda)^2}\Delta_L^2 \leq \frac{\gamma_x}{60}\frac{3C_J^{2k}}{8L_g^{2k}} \,. \tag{55}$$

Then, we obtain

$$\gamma_y \leq \left\{ \frac{C_J^k}{L_g^{(k+1)}}\frac{\sqrt{\alpha_y N}\mu}{160\sqrt{(\Delta_{a_k} + \Delta_{a_{k+1}})}} \,, \frac{\beta_y(1-\lambda)}{40\sqrt{6(L_g^2 + C_J^2)}} \,, \frac{\mu^K}{C_J^{K-k}L_g^k}\frac{\sqrt{\alpha_x N}}{16\sqrt{15}\Delta_L} \,, \frac{\mu^{K-k+1}}{C_J^{K-k}L_g}\frac{\beta_z(1-\lambda)}{160\sqrt{3}\Delta_L} \right\} \,.$$

Similarly, when $k = 1$, we have

$$\mathcal{Y}_1 = \frac{2L_g^2 c_1}{N} + \frac{2L_g^2 c_2}{N} + \frac{8d_1}{N}(\Delta_1^2 L_H^2 + \Delta_2^2 L_J^2) + \frac{8d_2}{N}\Delta_2^2 L_H^2 + \frac{3e}{N}\Delta_1^2 L_J^2$$

$$+ 2L_g^2\tilde{c}_1 + 2L_g^2\tilde{c}_2 + 8\tilde{d}_1(\Delta_1^2 L_H^2 + \Delta_2^2 L_J^2) + 8\tilde{d}_2\Delta_2^2 L_H^2 + 3\tilde{e}\Delta_1^2 L_J^2$$

$$+ \frac{4L_g^2\tilde{\nu}_1}{\beta_y(1-\lambda)} + \frac{4L_g^2\tilde{\nu}_2}{\beta_y(1-\lambda)} + \frac{16\tilde{\tau}_1}{\beta_z(1-\lambda)}(\Delta_1^2 L_H^2 + \Delta_2^2 L_J^2) + \frac{16\tilde{\tau}_2\Delta_2^2 L_H^2}{\beta_z(1-\lambda)} + \frac{12\tilde{\omega}\Delta_1^2 L_J^2}{\beta_x(1-\lambda)}$$

$$+ \frac{18L_g^2\tilde{\phi}_1}{\delta} + \frac{18L_g^2\tilde{\phi}_2}{\delta} + \frac{72\tilde{\psi}_1}{\delta}(\Delta_1^2 L_H^2 + \Delta_2^2 L_J^2) + \frac{72\tilde{\psi}_2}{\delta}\Delta_2^2 L_H^2 + \frac{54\tilde{\theta}}{\delta}\Delta_1^2 L_J^2\,,$$

$$\leq \frac{2}{N}\frac{20\gamma_x L_g^2}{\alpha_y\eta\mu^2}(\Delta_{a_1}+\Delta_{a_2})\Delta_{aa} + \frac{40\gamma_x(L_g^2+C_J^2)}{\beta_y^2(1-\lambda)^2}\frac{3C_J^2}{8L_g^2} + \frac{3C_J^{2K}}{8\mu^{2K}}\frac{16\gamma_x}{\alpha_x\eta N}(L_f^2 + \frac{C_J^2}{\mu^2}L_H^2 + L_J^2)$$

$$+ \frac{3C_J^{2K}}{8\mu^{2K}}\Big(\frac{320\gamma_x}{\beta_z^2(1-\lambda)^2}(L_f^2 + \frac{C_J^2}{\mu^2}L_H^2 + L_J^2) + \frac{72\gamma_x}{\beta_x^2(1-\lambda)^2}\frac{C_J^2}{L_g^2}L_J^2\Big)\,.$$

We only enforce the unique inequality, *i.e.*

$$4\gamma_y^2\eta^2\frac{3C_J^{2K}}{8\mu^{2K}}\frac{72\gamma_x}{\beta_x^2(1-\lambda)^2}\frac{C_J^2}{L_g^2}L_J^2 \leq \frac{\gamma_x}{60}\frac{3C_J^2}{8L_g^2}\,, \quad \text{and obtain} \quad \gamma_y \leq \frac{\mu^K}{C_J^K}\frac{\beta_x(1-\lambda)}{24\sqrt{30}L_J}\,.$$

When $k = K$, we have

$$\mathcal{Y}_K = \frac{2L_g^2 c_K}{N} + \frac{8d_K}{N}(\Delta_K^2 L_H^2 + L_f^2) + \frac{8d_{K-1}}{N}\Delta_K^2 L_J^2 + \frac{3eL_f^2}{N} + 2L_g^2\tilde{c}_K + 8\tilde{d}_K(\Delta_K^2 L_H^2 + L_f^2) + 8\tilde{d}_{K-1}\Delta_K^2 L_J^2 + 3\tilde{e}L_f^2$$

$$+ \frac{4L_g^2\tilde{\nu}_K}{\beta_y(1-\lambda)} + \frac{16\tilde{\tau}_K}{\beta_z(1-\lambda)}(\Delta_K^2 L_H^2 + L_f^2) + \frac{16\tilde{\tau}_{K-1}\Delta_K^2 L_J^2}{\beta_z(1-\lambda)} + \frac{12L_f^2\tilde{\omega}}{\beta_x(1-\lambda)}$$

$$+ \frac{18L_g^2\tilde{\phi}_K}{\delta} + \frac{72\tilde{\psi}_K}{\delta}(\Delta_K^2 L_H^2 + L_f^2) + \frac{72\tilde{\psi}_{K-1}}{\delta}\Delta_K^2 L_J^2 + \frac{54L_f^2\tilde{\theta}}{\delta}$$

$$\leq \frac{2}{N}\frac{20\gamma_x L_g^2}{\alpha_y\eta\mu^2}\Delta_{a_K}\Delta_{aa} + \frac{3L_f^2}{N}\frac{\gamma_x}{\alpha_x\eta} + \frac{40L_g^2\gamma_x}{\beta_y^2(1-\lambda)^2}\frac{3C_J^{2K}}{8L_g^{2K}} + \frac{3C_J^{2K}}{8\mu^{2K}}\frac{16\gamma_x}{\alpha_x\eta N}(L_f^2 + \frac{C_J^2}{\mu^2}L_H^2 + L_J^2)$$

$$+ \frac{3C_J^{2K}}{8\mu^2 L_g^{2(K-1)}}\Big(\frac{320\gamma_x}{\beta_z^2(1-\lambda)^2}(L_f^2 + \frac{C_J^2}{\mu^2}L_H^2 + L_J^2)\Big) + \frac{72L_f^2\gamma_x}{\beta_x^2(1-\lambda)^2}\frac{3C_J^2}{8L_g^2}\,.$$

We only enforce the unique inequality, *i.e.*

$$4\gamma_y^2\eta^2\frac{3L_f^2}{N}\frac{\gamma_x}{\alpha_x\eta} \leq \frac{\gamma_x}{60}\frac{3C_J^{2K}}{8L_g^{2K}}\,, \qquad 4\gamma_y^2\eta^2\frac{72L_f^2\gamma_x}{\beta_x^2(1-\lambda)^2}\frac{3C_J^2}{8L_g^2} \leq \frac{\gamma_x}{60}\frac{3C_J^{2K}}{8L_g^{2K}}\,.$$

Then, we obtain $\gamma_y \leq \Big\{ \frac{C_J^K}{L_g^K}\frac{\sqrt{\alpha_x N}}{8\sqrt{30}L_f}\,, \frac{C_J^{K-1}}{L_g^{K-1}}\frac{\beta_x(1-\lambda)}{24\sqrt{30}L_f}\Big\}$.

To solve the third inequality in Eq. (54), from the definition of $\phi_k$, we enforce

$$\nu_k\frac{32\rho_y\eta}{\delta(1-\lambda)}\frac{12\gamma_y^2\eta^2}{\delta} \leq \frac{\gamma_x}{12}\frac{3C_J^{2k}}{8L_g^{2k}}\,, \qquad \frac{64}{\delta}\rho_y^2\eta^2\mathcal{Y}_k\frac{12\gamma_y^2\eta^2}{\delta} \leq \frac{\gamma_x}{12}\frac{3C_J^{2k}}{8L_g^{2k}}\,. \tag{56}$$

Since $\rho_y \leq \frac{\delta}{8\sqrt{3}}$, the second inequality holds. To solve the first inequality, from the definition of $\nu_k$, we enforce

$$\frac{2\gamma_x\mathcal{E}_{1,yk}}{\rho_y(1-\lambda)}\frac{32\rho_y\eta}{\delta(1-\lambda)}\frac{12\gamma_y^2\eta^2}{\delta} \leq \frac{\gamma_x}{16}\frac{3C_J^{2k}}{8L_g^{2k}}\,, \qquad \frac{2\mathcal{E}_{2,yk}}{\rho_y\eta(1-\lambda)}\frac{32\rho_y\eta}{\delta(1-\lambda)}\frac{12\gamma_y^2\eta^2}{\delta} \leq \frac{\gamma_x}{64}\frac{3C_J^{2k}}{8L_g^{2k}}\,.$$

From Eq. (53), the second inequality is already satisfied. To solve the first inequality, when $k \in \{1, \cdots, K-1\}$, from the definition of $\mathcal{E}_{1,yk}$, we enforce

$$\frac{20L_g^2}{\mu^2}(\Delta_{a_k}+\Delta_{a_{k+1}})\Delta_{aa}\frac{64\gamma_x\eta}{(1-\lambda)^2}\frac{12\gamma_y^2\eta^2}{\delta^2} \leq \frac{\gamma_x}{32}\frac{3C_J^{2k}}{8L_g^{2k}}\,,$$

$$\frac{3C_J^{2K}}{8\mu^{2K}}\Delta_{yy}\frac{64\gamma_x\eta}{(1-\lambda)^2}\frac{12\gamma_y^2\eta^2}{\delta^2} \leq \frac{\gamma_x}{32}\frac{3C_J^{2k}}{8L_g^{2k}}\,.$$

Then, we obtain

$$\gamma_y \leq \Big\{ \frac{\mu C_J^k}{L_g^{k+1}}\frac{\delta(1-\lambda)}{512\sqrt{5(\Delta_{a_k}+\Delta_{a_{k+1}})\Delta_{aa}}}\,, \frac{\mu^K}{C_J^{K-k}L_g^k}\frac{\delta(1-\lambda)}{64\sqrt{6}\Delta_{yy}}\Big\}\,.$$

When $k = K$, from the definition of $\mathcal{E}_{1,yK}$, we only enforce the unique inequalities here,

$$6L_f^2 \frac{64\gamma_x\eta}{(1-\lambda)^2}\frac{12\gamma_y^2\eta^2}{\delta^2} \le \frac{\gamma_x}{64}\frac{3C_J^{2K}}{8L_g^{2K}}\,, \qquad \frac{20L_g^2}{\mu^2}\Delta_{a_K}\Delta_{aa}\frac{64\gamma_x\eta}{(1-\lambda)^2}\frac{12\gamma_y^2\eta^2}{\delta^2} \le \frac{\gamma_x}{64}\frac{3C_J^{2K}}{8L_g^{2K}}\,.$$

We obtain $\gamma_y \le \left\{ \frac{C_J^K}{L_g^K}\frac{\delta(1-\lambda)}{512\sqrt{3}L_f}\,,\, \frac{C_J^K\mu}{L_g^{K+1}}\frac{\delta(1-\lambda)}{512\sqrt{10\Delta_{a_K}\Delta_{aa}}} \right\}$.

To solve the first inequality in Eq. (54), from the definition of $\mathcal{E}_{1,yk}$, we have

$$\frac{4\gamma_x\eta\gamma_y^2\mathcal{E}_{1,yk}}{\rho_y^2(1-\lambda)^2} \le \frac{\gamma_x}{6}\frac{3C_J^{2k}}{8L_g^{2k}}\,, \qquad \frac{20L_g^2}{\mu^2}(\Delta_{a_k}+\Delta_{a_{k+1}})\Delta_{aa}\frac{4\gamma_x\eta\gamma_y^2}{\rho_y^2(1-\lambda)^2} \le \frac{\gamma_x}{12}\frac{3C_J^{2k}}{8L_g^{2k}}\,,$$

$$\frac{3C_J^{2K}}{8\mu^{2K}}\Delta_{yy}\frac{4\gamma_x\eta\gamma_y^2}{\rho_y^2(1-\lambda)^2} \le \frac{\gamma_x}{12}\frac{3C_J^{2k}}{8L_g^{2k}}\,, \qquad 6L_f^2\frac{4\gamma_x\eta\gamma_y^2}{\rho_y^2(1-\lambda)^2} \le \frac{\gamma_x}{24}\frac{3C_J^{2K}}{8L_g^{2K}}\,,$$

$$\frac{20L_g^2}{\mu^2}\Delta_{a_K}\Delta_{aa}\frac{4\gamma_x\eta\gamma_y^2}{\rho_y^2(1-\lambda)^2} \le \frac{\gamma_x}{24}\frac{3C_J^{2K}}{8L_g^{2K}}\,,$$

we obtain

$$\gamma_y \le \left\{ \frac{\mu C_J^k}{L_g^{k+1}}\frac{\rho_y(1-\lambda)}{16\sqrt{10(\Delta_{a_k}+\Delta_{a_{k+1}})\Delta_{aa}}}\,,\, \frac{\mu^K}{C_J^{K-k}L_g^k}\frac{\rho_y(1-\lambda)}{4\sqrt{3\Delta_{yy}}}\,,\, \frac{C_J^K}{L_g^K}\frac{\rho_y(1-\lambda)}{16\sqrt{6}L_f}\,,\, \frac{C_J^K\mu}{L_g^{K+1}}\frac{\rho_y(1-\lambda)}{32\sqrt{5\Delta_{a_K}\Delta_{aa}}} \right\}\,.$$

To solve the second inequality in Eq. (54), from the definition of $\mathcal{E}_{2,yk}$, we have

$$32\mathcal{Y}_k\frac{4\gamma_y^2\eta^2}{(1-\lambda)^2} \le \frac{\gamma_x}{6}\frac{3C_J^{2k}}{8L_g^{2k}}\,.$$

Then, we solve it as in Eq. (55). We omit the details here and obtain

$$\gamma_y \le \left\{ \frac{C_J^k}{L_g^{(k+1)}}\frac{(1-\lambda)\sqrt{\alpha_y N}\mu}{640\sqrt{(\Delta_{a_k}+\Delta_{a_{k+1}})}}\,,\, \frac{\beta_y(1-\lambda)^2}{160\sqrt{6(L_g^2+C_J^2)}}\,,\, \frac{\mu^K}{C_J^{K-k}L_g^k}\frac{(1-\lambda)\sqrt{\alpha_x N}}{64\sqrt{15}\Delta_L}\,,\, \frac{\mu^{K-k+1}}{C_J^{K-k}L_g}\frac{\beta_z(1-\lambda)^2}{640\sqrt{3}\Delta_L}\,, \right.$$

$$\left. \frac{C_J^K}{L_g^K}\frac{(1-\lambda)\sqrt{\alpha_x N}}{32\sqrt{30}L_f}\,,\, \frac{\mu^K}{C_J^K}\frac{\beta_x(1-\lambda)^2}{96\sqrt{30}L_J}\,,\, \frac{C_J^{K-1}}{L_g^{K-1}}\frac{\beta_x(1-\lambda)^2}{96\sqrt{30}L_f} \right\}\,.$$

### G.6. Momentum ($\mathcal{F}_6$)

For $\mathcal{F}_6$, it consists of two terms: the momentum for variables $x$ and $y$. Specifically, we have

$$\mathcal{F}_6 \le \left( \sum_{k=1}^K a_k\frac{5\eta\gamma_x^2}{\gamma_y\mu}C_{y_k}^2 + \sum_{k=1}^K b_k\frac{9\gamma_x^2\eta}{\gamma_z\mu}C_{z_k}^2 + \theta\frac{12\gamma_x^2\eta^2}{\delta} + 4\gamma_x^2\eta^2\mathcal{X} - \frac{\eta\gamma_x}{4} \right)\mathbb{E}[\|\bar{m}_t^x\|^2]$$

$$+ \sum_{k=1}^K \left( \phi_k\frac{12\gamma_y^2\eta^2}{\delta} + 4\gamma_y^2\eta^2\mathcal{Y}_k - a_k\frac{3\eta\gamma_y^2}{4} \right)\mathbb{E}[\|\bar{m}_{k,t}^y\|^2]\,.$$

**(6.1) Momentum for x**: Firstly, we set the coefficient of $\mathbb{E}[\|\bar{m}_t^x\|^2]$ to be non-positive:

$$\sum_{k=1}^K a_k\frac{5\eta\gamma_x^2}{\gamma_y\mu}C_{y_k}^2 + \sum_{k=1}^K \frac{\alpha_z\eta\mu}{24\gamma_z}d_k\frac{9\eta\gamma_x^2}{\gamma_z\mu}C_{z_k}^2 + \theta\frac{12\gamma_x^2\eta^2}{\delta} + 4\gamma_x^2\eta^2\mathcal{X} - \frac{\eta\gamma_x}{4} \le 0\,.$$

Specifically, we enforce

$$\sum_{k=1}^K a_k\frac{5\eta\gamma_x^2}{\gamma_y\mu}C_{y_k}^2 \le \frac{\eta\gamma_x}{16}\,, \qquad \sum_{k=1}^K \frac{\alpha_z\eta\mu}{24\gamma_z}d_k\frac{9\eta\gamma_x^2}{\gamma_z\mu}C_{z_k}^2 \le \frac{\eta\gamma_x}{16}\,,$$

$$\theta\frac{12\gamma_x^2\eta^2}{\delta} \le \frac{\eta\gamma_x}{16}\,, \qquad 4\gamma_x^2\eta^2\mathcal{X} \le \frac{\eta\gamma_x}{32}\,.$$

From the first and second inequalities, we derive $\gamma_x \leq \left\{ \frac{\gamma_y \mu}{4 C_{y_k} \sqrt{5 \sum_{k=1}^{K} \Delta_{a_k} \tilde{A}}}, \frac{\sqrt{\alpha_x} \gamma_z}{C_{z_k} \sqrt{6 \alpha_z \Delta_{d_k}}} \right\}$. Then, we use a similar approach for the third and fourth inequalities as Eq. (51). We omit the details here and obtain

$$\gamma_x \leq \Big\{ \frac{\mu^K}{C_J^K} \frac{\sqrt{\alpha_x N}}{32\sqrt{3}\Delta_L}, \frac{\mu^K L_g}{C_J^{K+1}} \frac{\beta_z(1-\lambda)}{64\sqrt{15}L_H}, \frac{\mu^K L_g}{C_J^{K+1}} \frac{\beta_x(1-\lambda)}{96\sqrt{3}L_J}, \frac{L_g^K}{C_J^K} \frac{\beta_z(1-\lambda)}{64\sqrt{15}L_f}, \frac{\beta_x(1-\lambda)L_g}{96\sqrt{3}L_f C_J},$$

$$\frac{\beta_y(1-\lambda)}{32\sqrt{15}C_J}, \frac{\sqrt{\alpha_x N}}{32\sqrt{3}L_f}, \frac{\sqrt{\alpha_y N}\mu}{64 L_g \sqrt{10\Delta_{a_1}\Delta_{aa}}}, \frac{\delta(1-\lambda)}{768 L_f}, \frac{\mu\delta(1-\lambda)}{256\sqrt{30\Delta_{a_1}\Delta_{aa}}L_g}, \frac{\mu^K}{C_J^K} \frac{\delta(1-\lambda)}{192\sqrt{\Delta_{xx}}} \Big\}.$$

**(6.2) Momentum of y**: Then, we set the coefficient of $\mathbb{E}[\|\bar{m}_{k,t}^y\|^2]$ to be non-positive:

$$\phi_k \frac{12\gamma_y^2\eta^2}{\delta} + 4\gamma_y^2\eta^2 \mathcal{Y}_k - a_k \frac{3\eta\gamma_y^2}{4} \leq 0.$$

Specifically, we enforce

$$\phi_k \frac{12\gamma_y^2\eta^2}{\delta} \leq a_k \frac{3\eta\gamma_y^2}{8}, \quad 4\gamma_y^2\eta^2 \mathcal{Y}_k \leq a_k \frac{3\eta\gamma_y^2}{16}.$$

To solve the second inequality, when $k \in \{2, \cdots, K-1\}$, we have

$$4\gamma_y^2\eta^2 \frac{2L_g^2}{N} \frac{20\gamma_y}{\alpha_y\eta\mu}(a_k + a_{k+1}) + 4\gamma_y^2\eta^2 \frac{40\gamma_x(L_g^2 + C_J^2)}{\beta_y^2(1-\lambda)^2} \frac{3C_J^{2k}}{8L_g^{2k}} + 4\gamma_y^2\eta^2 \frac{3C_J^{2K}}{8\mu^{2K}} \frac{16\gamma_x}{\alpha_x\eta N}\Delta_L^2$$

$$+ 4\gamma_y^2\eta^2 \frac{3C_J^{2K}}{8\mu^{2(K-k+1)}L_g^{2(k-1)}} \frac{320\gamma_x}{\beta_z^2(1-\lambda)^2}\Delta_L^2 \leq a_k \frac{3\eta\gamma_y^2}{16}.$$

Specifically, we enforce

$$4\gamma_y^2\eta^2 \frac{2L_g^2}{N} \frac{20\gamma_y}{\alpha_y\eta\mu}(a_k + a_{k+1}) \leq a_k \frac{3\eta\gamma_y^2}{80}, \quad 4\gamma_y^2\eta^2 \frac{40\gamma_x(L_g^2 + C_J^2)}{\beta_y^2(1-\lambda)^2} \frac{3C_J^{2k}}{8L_g^{2k}} \leq a_k \frac{3\eta\gamma_y^2}{80},$$

$$4\gamma_y^2\eta^2 \frac{3C_J^{2K}}{8\mu^{2K}} \frac{16\gamma_x}{\alpha_x\eta N}\Delta_L^2 \leq a_k \frac{3\eta\gamma_y^2}{80}, \quad 4\gamma_y^2\eta^2 \frac{3C_J^{2K}}{8\mu^{2(K-k+1)}L_g^{2(k-1)}} \frac{320\gamma_x}{\beta_z^2(1-\lambda)^2}\Delta_L^2 \leq a_k \frac{3\eta\gamma_y^2}{80}.$$

Then, we obtain

$$\gamma_y \leq \Big\{ \frac{3\Delta_{a_k}\alpha_y\mu N}{12800 L_g^2(\Delta_{a_k} + \Delta_{a_{k+1}})}, \frac{\beta_y^2(1-\lambda)^2 L_g^{2k}\Delta_{a_k}\tilde{A}}{1600\mu(L_g^2 + C_J^2)C_J^{2k}}, \frac{\alpha_x N\mu^{2K-1}\Delta_{a_k}\tilde{A}}{640 C_J^{2K}\Delta_L^2}, \frac{\mu^{2K-2k+1}L_g^{2(k-1)}\Delta_{a_k}\tilde{A}}{12800 C_J^{2K}\Delta_L^2}\beta_z^2(1-\lambda)^2 \Big\}.$$

Similarly, when $k = 1$ and $k = K$, we only enforce the unique inequalities, i,e,

$$4\gamma_y^2\eta^2 \frac{3C_J^{2K}}{8\mu^{2K}} \frac{72\gamma_x}{\beta_x^2(1-\lambda)^2} \frac{C_J^2}{L_g^2}L_J^2 \leq a_1 \frac{3\eta\gamma_y^2}{80}, \quad 4\gamma_y^2\eta^2 \frac{3L_f^2}{N} \frac{\gamma_x}{\alpha_x\eta} \leq a_K \frac{3\eta\gamma_y^2}{80}, \quad 4\gamma_y^2\eta^2 \frac{72L_f^2\gamma_x}{\beta_x^2(1-\lambda)^2} \frac{3C_J^2}{8L_g^2} \leq a_K \frac{3\eta\gamma_y^2}{80}.$$

Then, we obtain

$$\gamma_y \leq \Big\{ \frac{\mu^{2K-1}L_g^2\Delta_{a_1}\tilde{A}}{2880 L_J^2 C_J^{2(K+1)}}\beta_x^2(1-\lambda)^2, \frac{\alpha_x N\Delta_{a_K}\tilde{A}}{320 L_f^2}, \frac{L_g^2\Delta_{a_K}\tilde{A}}{2880\mu C_J^2 L_f^2}\beta_x^2(1-\lambda)^2 \Big\}.$$

To solve the first inequality, we follow the approach in Eq. (56). Here, we omit the details and obtain

$$\gamma_y \leq \Big\{ \frac{\mu\delta^2(1-\lambda)^2\Delta_{a_k}\tilde{A}}{1280 \times 96 L_g^2(\Delta_{a_k} + \Delta_{a_{k+1}})\Delta_{aa}}, \frac{\mu^{2K-1}\delta^2(1-\lambda)^2\Delta_{a_k}\tilde{A}}{2304 C_J^{2K}\Delta_{yy}}, \frac{\delta^2(1-\lambda)^2\Delta_{a_K}\tilde{A}}{384 \times 192\mu L_f^2}, \frac{1280 \times 192\mu\delta^2(1-\lambda)^2\tilde{A}}{L_g^2\Delta_{aa}} \Big\}.$$

### G.7. Final Step

In summary, by choosing the parameters appropriately so that all coefficients are non-negative, all terms $\mathcal{F}_i$ vanish:

$$\mathcal{E}_a = \frac{80 C_J^2}{\alpha_x N} + \frac{480 C_J^2 L_g^2}{\beta_z^2(1-\lambda)^2\mu^2} + \frac{80 L_g^4}{\mu^2\alpha_x N} + \frac{640 L_g^4}{\beta_z^2(1-\lambda)^2\mu^2} + \frac{96 C_J^2 L_g^2}{\beta_x^2(1-\lambda)^2\mu^2},$$

$$a_k = \Delta_{a_k} \frac{\gamma_x}{\gamma_y \mu} \tilde{A}, \quad \tilde{A} = \left(32 + \frac{16\alpha_z}{3\alpha_x} + 256\gamma_z^2 \mathcal{E}_a\right), \quad \Delta_{a_k} = \frac{160^{K-k} C_J^{2K}}{\mu^{2K}} \left(L_f^2 + \frac{C_J^2}{\mu^2} L_H^2 + \frac{L_g^2}{\mu^2} L_J^2 + L_H^2\right)$$

$$b_k = \Delta_{b_k} \frac{\gamma_x \gamma_z}{\mu} \tilde{B}, \quad \Delta_{b_k} = 3072 \frac{L_g^{2k}}{\mu^{2k}} \frac{(256 C_J)^{2(k-1)}}{L_g^{2(k-1)}},$$

$$\tilde{B} = \left(\frac{80 C_J^2}{\gamma_z^2} + \frac{80 C_J^2 L_g^2}{\alpha_x N} + \frac{320 C_J^2 L_g^2}{\beta_z^2 (1-\lambda)^2} + \frac{320 C_J^4}{\beta_x^2 (1-\lambda)^2}\right),$$

$$c_k = a_k \frac{20\gamma_y}{\alpha_y \eta \mu}, \quad d_k = \frac{3 C_J^{2k}}{8 L_g^{2k}} \frac{\gamma_x}{\alpha_x \eta} \triangleq \frac{\gamma_x}{\alpha_x \eta} \Delta_{d_k},$$

$$e = \frac{\gamma_x}{\alpha_x \eta}, \quad \tilde{e} = \frac{16\gamma_x \Delta_{d1}}{\beta_x^2 (1-\lambda)^2}, \quad \tilde{\omega} = \frac{\gamma_x \Delta_{d_1}}{\beta_x (1-\lambda)}, \quad \tilde{\theta} = \tilde{\omega} \frac{32\beta_x}{\delta(1-\lambda)}$$

$$\tilde{c}_k = \frac{16\gamma_x \Delta_{d_k}}{\beta_y^2 (1-\lambda)^2}, \quad \tilde{\nu}_k = \frac{\gamma_x \Delta_{d_k}}{\beta_y (1-\lambda)}, \quad \tilde{\phi}_k = \tilde{\nu}_k \frac{32\beta_y}{\delta(1-\lambda)},$$

$$\tilde{d}_k = \frac{16\gamma_x \Delta_{d_k}}{\beta_z^2 (1-\lambda)^2}, \quad \tilde{\tau}_k = \frac{\gamma_x \Delta_{d_k}}{\beta_z (1-\lambda)}, \quad \tilde{\psi}_k = \tilde{\tau}_k \frac{32\beta_z}{\delta(1-\lambda)},$$

$$\Delta_{aa} = \left(32 + \frac{5\alpha_z}{2\alpha_x} + \frac{12 C_J^2}{\mu^2} + \frac{15 L_g^2}{\mu^2}\right), \quad \Delta_{xx} = \left(16 L_J^2 + \frac{3\alpha_z}{\alpha_x}(L_H^2 \frac{C_J^2}{\mu^2} + L_f^2) + 4(\frac{C_J^2}{\mu^2} L_H^2 + L_f^2)\right)$$

$$\mathcal{E}_{1,x} = 6 L_f^2 + \frac{20 L_g^2}{\mu^2} \Delta_{a_1} \Delta_{aa} + \frac{3 C_J^{2K}}{8\mu^{2K}} \Delta_{xx}, \quad \mathcal{E}_{2,x} = 32\rho_x^2 \eta^2 \mathcal{X},$$

$$\omega = \frac{2\gamma_x \mathcal{E}_{1,x}}{\rho_x (1-\lambda)} + \frac{2\mathcal{E}_{2,x}}{\rho_x \eta (1-\lambda)}, \quad \theta = \omega \frac{32\rho_x \eta}{\delta(1-\lambda)} + \frac{64}{\delta} \rho_x^2 \eta^2 \mathcal{X},$$

$$\Delta_{yy} = \left(16 L_J^2 + \frac{6\alpha_z}{\alpha_x}(L_f^2 + \frac{C_J^2}{\mu^2} L_H^2 + L_J^2) + 8(L_f^2 + \frac{C_J^2}{\mu^2} L_H^2 + L_J^2)\right), \quad \mathcal{E}_{1,yK} = 6 L_f^2 + \frac{20 L_g^2}{\mu^2} \Delta_{a_K} \Delta_{aa} + \frac{3 C_J^{2K}}{8\mu^{2K}} \Delta_{yy}$$

$$\mathcal{E}_{1,yk} = \frac{20 L_g^2}{\mu^2}(\Delta_{a_k} + \Delta_{a_{k+1}})\Delta_{aa} + \frac{3 C_J^{2K}}{8\mu^{2K}} \Delta_{yy}, \quad \mathcal{E}_{2,yk} = 32\rho_y^2 \eta^2 \mathcal{Y}_k,$$

$$\nu_k = \frac{2\gamma_x \mathcal{E}_{1,yk}}{\rho_y (1-\lambda)} + \frac{2\mathcal{E}_{2,yk}}{\rho_y \eta (1-\lambda)}, \quad \phi_k = \nu_k \frac{32\rho_y \eta}{\delta(1-\lambda)} + \frac{64}{\delta} \rho_y^2 \eta^2 \mathcal{Y}_k,$$

$$\mathcal{E}_{1,z1} = \frac{3 b_1}{\eta \gamma_x} + b_1 \frac{48\gamma_z}{\gamma_x \mu} L_g^2 + 6 C_J^2, \quad \mathcal{E}_{1,zk} = \frac{3 b_k}{\eta \gamma_x} + b_k \frac{48\gamma_z}{\gamma_x \mu} L_g^2 + b_{k-1} \frac{48\gamma_z}{\gamma_x \mu} C_J^2, \quad \mathcal{E}_{2,zk} = 96\gamma_z^2 \eta^2 L_g^2 \mathcal{Z}_k + 32\rho_z^2 \eta^2 \mathcal{Z}_k$$

$$\tau_k = \frac{2\gamma_x \mathcal{E}_{1,zk}}{\rho_z (1-\lambda)} + \frac{2\mathcal{E}_{2,zk}}{\rho_z \eta (1-\lambda)}, \quad \psi_k = \tau_k \frac{32\rho_z \eta}{\delta(1-\lambda)} + \frac{64}{\delta} \rho_z^2 \eta^2 \mathcal{Z}_k, \quad \Delta_L^2 = (L_f^2 + \frac{C_J^2}{\mu^2} L_H^2 + L_J^2), \quad (57)$$

and we obtain

$$\mathcal{L}_{t+1} - \mathcal{L}_t \leq -\frac{\eta\gamma_x}{2} \mathbb{E}[\|\nabla F_{K+1}(\bar{x}_t)\|^2] + \mathcal{F}_0.$$

By summing over $t$ from 0 to $T-1$, we have

$$\frac{1}{T} \sum_{t=0}^{T-1} \mathbb{E}[\|\nabla F_{K+1}(\bar{x}_t)\|^2] \leq \frac{2(\mathcal{L}_0 - \mathcal{L}_T)}{\eta\gamma_x T} + \frac{2\mathcal{F}_0}{\eta\gamma_x T}.$$

For the hyperparameter setting, by denoting

$$\ell = \max\{L_f, L_g, C_J, C_{z_k} \Delta_{d_k}, \mu, \frac{C_J^{K-1}}{\mu^{K-1}}\left(\Delta_L + \frac{C_J^2}{\mu^2}\Delta_L + \frac{C_J + L_g}{\mu}(\Delta_L + \Delta_L \frac{\mu}{\Delta_{a_k}} + \sqrt{\Delta_{xx}})\right),$$

$$\frac{L_f C_J}{\mu}, \frac{L_g^k}{\mu^k} \frac{C_J^{K-k}}{\mu^{K-k}}(\Delta_L + \sqrt{\Delta_{yy}}), \sum_{k=1}^{K} \frac{L_g^k}{C_J^k}\left(L_f(1 + \sqrt{\Delta_{yy}}) + \frac{\sqrt{(\Delta_{a_k} + \Delta_{a_{k+1}})} L_g}{\mu(1 + \sqrt{\Delta_{aa}})^{-1}}\right), \frac{(\Delta_{a_k} + \Delta_{a_{k+1}}) L_g^2}{\Delta_{a_k} \mu}\}.$$

we have

$$\rho_x \leq \min\left\{\frac{\delta}{8\sqrt{3}}, \frac{\delta(1-\lambda)}{96\sqrt{2}}\right\}, \quad \rho_y \leq \min\left\{\frac{\delta}{8\sqrt{3}}, \frac{\delta(1-\lambda)}{96\sqrt{2}}\right\}, \quad \rho_z \leq \min\left\{\frac{\delta}{8\sqrt{3}}, \frac{\delta(1-\lambda)}{96\sqrt{2}}\right\},$$

$$\beta_x \leq \min\left\{\frac{\delta}{12}, \frac{\delta(1-\lambda)}{24\sqrt{6}}\right\}, \quad \beta_y \leq \min\left\{\frac{\delta}{12}, \frac{\delta(1-\lambda)}{24\sqrt{6}}\right\}, \quad \beta_z \leq \min\left\{\frac{\delta}{12}, \frac{\delta(1-\lambda)}{24\sqrt{6}}\right\},$$

$$\alpha_x \leq \frac{\alpha_z}{1024 \times 640 \times 48} \frac{\mu^{2(k+1)}}{L_g^{2(k+1)} 256^{2(k-1)}} \ ,$$

$$\gamma_x \leq \min \left\{ \frac{(1-\lambda)\sqrt{\alpha_x N}}{128\sqrt{3}\ell} \ , \ \frac{(1-\lambda)\sqrt{\alpha_y N}}{64\sqrt{10}\ell} \ , \ \frac{\beta_z(1-\lambda)^2}{256\sqrt{15}\ell} \ , \ \frac{\beta_x(1-\lambda)^2}{384\sqrt{3}\ell} \ , \ \frac{\beta_y(1-\lambda)^2}{128\sqrt{15}\ell} \ , \ \frac{\delta(1-\lambda)}{256\sqrt{30}\ell} \ , \ \frac{\rho_x(1-\lambda)}{24\sqrt{2}\ell} \ , \right.$$

$$\left. \frac{\sqrt{\alpha_x}\gamma_z}{\sqrt{6\alpha_z}\ell} \ , \ \frac{\gamma_y}{4\sqrt{5}\tilde{A}\ell} \right\} \ ,$$

$$\gamma_y \leq \min \left\{ \frac{(1-\lambda)\sqrt{\alpha_x N}}{64\sqrt{15}\ell} \ , \ \frac{(1-\lambda)\sqrt{\alpha_y N}}{640} \ , \ \frac{\beta_x^2(1-\lambda)^2\tilde{A}}{2880\ell} \ , \ \frac{\beta_y^2(1-\lambda)^2\tilde{A}}{320\sqrt{6}\ell} \ , \ \frac{\beta_z^2(1-\lambda)^2\tilde{A}}{12800\ell} \ , \ \frac{\delta^2(1-\lambda)^2\tilde{A}}{1280 \times 96\ell} \ , \ \frac{\rho_y(1-\lambda)}{32\sqrt{5}\ell} \ , \right.$$

$$\left. \frac{\alpha_y N \tilde{A}}{12800\ell} \ , \ \frac{\alpha_x N \tilde{A}}{640\ell} \right\} \ ,$$

$$\gamma_z \leq \min \left\{ \frac{\sqrt{\alpha_x}\delta(1-\lambda)}{9216\sqrt{\alpha_x}\ell} \ , \ \frac{4\alpha_x \rho_z^2(1-\lambda)^2}{27\alpha_z\ell} \ , \ \frac{(1-\lambda)\sqrt{\alpha_x N}}{192\ell} \ , \ \frac{(1-\lambda)\sqrt{\alpha_z N}}{48\ell} \ , \ \frac{\beta_z(1-\lambda)^2}{384\sqrt{5}\ell} \ , \ \frac{\beta_x(1-\lambda)^2}{144\ell} \ , \right.$$

$$\left. \frac{\alpha_x \delta^2(1-\lambda)^2}{384 \times 144 \times 96\alpha_z\ell} \ , \ \frac{\mu^{k+1}}{L_g^{k+1}256^{(k-1)}} \sqrt{\frac{\alpha_z}{3072 \times 128\alpha_x}} \middle/ \sqrt{\frac{80L_g^2}{\alpha_x N} + \frac{320L_g^2}{\beta_z^2(1-\lambda)^2} + \frac{320C_J^2}{\beta_x^2(1-\lambda)^2}} \right\} \ . \tag{58}$$

Taking Big-O notation, we obtain $\gamma_x = \gamma_y = \gamma_z = O(\delta^2(1-\lambda)^4)$.

For the term $\frac{2\mathcal{F}_0}{\eta\gamma_x T}$, from the definition of $\mathcal{F}_0$ in Eq. (34), taking the Big-O notation, we obtain

$$O\left(\frac{2\mathcal{F}_0}{\eta\gamma_x T}\right) = O\left(\frac{\alpha_z \alpha_y \eta^2}{\alpha_x N}\right) + O\left(\frac{\alpha_z^2 \eta^2}{\alpha_x N}\right) + O\left(\frac{\alpha_x \eta^2}{N}\right)$$

$$+ O\left(\frac{\alpha_x^2 \eta^3}{\beta^2(1-\lambda)^2}\right) + O\left(\frac{\alpha_y^2 \eta^3}{\beta^2(1-\lambda)^2}\right) + O\left(\frac{\alpha_z^2 \eta^3}{\beta^2(1-\lambda)^2}\right) \ .$$

When $t = 0$, we have

$$\mathcal{L}_0 = \mathbb{E}[F_{K+1}(\bar{x}_0)] + \sum_{k=1}^{K} \Delta_{a_k} \frac{\gamma_x}{\gamma_y \mu} \tilde{A} \mathbb{E}[\|\bar{y}_{k,0} - y_k^*(\bar{x}_0)\|^2] + \sum_{k=1}^{K} \Delta_{b_k} \frac{\gamma_x \gamma_z}{\mu} \tilde{B} \mathbb{E}[\|\bar{z}_{k,0} - z_k^*(\bar{x}_0)\|^2]$$

$$+ \sum_{k=1}^{K} \Delta_{a_k} \frac{\gamma_x}{\gamma_y \mu} \tilde{A} \frac{20\gamma_y}{\alpha_y \eta \mu} \mathbb{E}[\|(M_{k,0}^y - U_{k,0})\mathbf{1}\frac{1}{N}\|^2] + \sum_{k=1}^{K} \frac{\gamma_x}{\alpha_x \eta} \Delta_{d_k} \mathbb{E}[\|(M_{k,0}^z - V_{k,0})\mathbf{1}\frac{1}{N}\|^2] + \frac{\gamma_x}{\alpha_x \eta} \mathbb{E}[\|(M_0^x - G_0)\mathbf{1}\frac{1}{N}\|^2]$$

$$+ \sum_{k=1}^{K} \frac{16\gamma_x \Delta_{d_k}}{\beta_y^2(1-\lambda)^2} \frac{1}{N} \mathbb{E}[\|M_{k,0}^y - U_{k,0}\|_F^2] + \sum_{k=1}^{K} \frac{16\gamma_x \Delta_{d_k}}{\beta_z^2(1-\lambda)^2} \frac{1}{N} \mathbb{E}[\|M_{k,0}^z - V_{k,0}\|_F^2] + \frac{16\gamma_x \Delta_{d1}}{\beta_x^2(1-\lambda)^2} \frac{1}{N} \mathbb{E}[\|M_0^x - G_0\|_F^2]$$

$$+ \frac{\gamma_x \Delta_{d1}}{\beta_x(1-\lambda)} \frac{1}{N} \mathbb{E}[\|P_0(I-J)\|_F^2] + \sum_{k=1}^{K} \frac{\gamma_x \Delta_{d_k}}{\beta_y(1-\lambda)} \frac{1}{N} \mathbb{E}[\|Q_{k,0}(I-J)\|_F^2] + \sum_{k=1}^{K} \frac{\gamma_x \Delta_{d_k}}{\beta_z(1-\lambda)} \frac{1}{N} \mathbb{E}[\|R_{k,0}(I-J)\|_F^2]$$

$$+ \frac{32\gamma_x \Delta_{d1}}{\delta(1-\lambda)^2} \frac{1}{N} \mathbb{E}[\|P_0 - \hat{P}_0\|_F^2] + \sum_{k=1}^{K} \frac{32\gamma_x \Delta_{d_k}}{\delta(1-\lambda)^2} \frac{1}{N} \mathbb{E}[\|Q_{k,0} - \hat{Q}_{k,0}\|_F^2] + \sum_{k=1}^{K} \frac{32\gamma_x \Delta_{d_k}}{\delta(1-\lambda)^2} \frac{1}{N} \mathbb{E}[\|R_{k,0} - \hat{R}_{k,0}\|_F^2] \ . \tag{59}$$

Applying Lemma E.1- E.6, Lemma F.3, Lemma F.5 with $t = 0$, and taking the Big-O bound, we derive

$$O\left(\frac{2(\mathcal{L}_0 - \mathcal{L}_T)}{\eta\gamma_x T}\right) = O\left(\frac{F_{K+1}(x_0) - F_{K+1}(x_*)}{\eta\gamma_x T}\right) + \frac{\alpha_z}{\alpha_x \eta\gamma_y T} \sum_{k=1}^{K} \mathbb{E}[\|\bar{y}_{k,0} - y_k^*(\bar{x}_0)\|^2] + O\left(\frac{1}{\gamma_z \eta T}\right) \sum_{k=1}^{K} \mathbb{E}[\|\bar{z}_{k,0} - z_k^*(\bar{x}_0)\|^2]$$

$$+ O\left(\frac{\gamma_z}{\alpha_x N\eta T}\right) \sum_{k=1}^{K} \mathbb{E}[\|\bar{z}_{k,0} - z_k^*(\bar{x}_0)\|^2] + O\left(\frac{\gamma_z}{\beta^2(1-\lambda)^2\eta T}\right) \sum_{k=1}^{K} \mathbb{E}[\|\bar{z}_{k,0} - z_k^*(\bar{x}_0)\|^2]$$

$$+ O\left(\frac{1}{\beta(1-\lambda)\eta T}\right) \frac{1}{N} \sum_{n=1}^{N} \mathbb{E}[\|\nabla_1 f_{K+1}^{(n)}(x_0, y_{K,0})\|^2] + O\left(\frac{1}{\beta(1-\lambda)\eta T}\right) \frac{1}{N} \sum_{n=1}^{N} \mathbb{E}[\|\nabla_1 f_k^{(n)}(y_{k,0}, y_{k-1,0})\|^2]$$

$$+ O\left(\frac{\alpha_z}{\alpha_x \alpha_y \eta^2 T} \frac{1}{NS_0}\right) + O\left(\frac{1}{\alpha_x \eta^2 T} \frac{1}{NS_0}\right) + O\left(\frac{1}{\beta^2(1-\lambda)^2 \eta T S_0}\right) + O\left(\frac{1}{\beta^2(1-\lambda)^2 \eta T}\right) \ .$$

Finally, combining these results, we complete the proof of the Theorem

$$
\frac{1}{T} \sum_{t=0}^{T-1} \mathbb{E}[\|\nabla F_{K+1}(\bar{x}_t)\|^2]
$$

$$
\leq \frac{2(F_{K+1}(x_0) - F_{K+1}(x_*))}{\eta \gamma_x T} + \frac{\alpha_z}{\alpha_x \eta \gamma_y T} \sum_{k=1}^{K} \mathbb{E}[\|\bar{y}_{k,0} - y_k^*(\bar{x}_0)\|^2] + O\left(\frac{1}{\gamma_z \eta T}\right) \sum_{k=1}^{K} \mathbb{E}[\|\bar{z}_{k,0} - z_k^*(\bar{x}_0)\|^2]
$$

$$
+ O\left(\frac{\gamma_z}{\alpha_x N \eta T}\right) \sum_{k=1}^{K} \mathbb{E}[\|\bar{z}_{k,0} - z_k^*(\bar{x}_0)\|^2] + O\left(\frac{\gamma_z}{\beta^2(1-\lambda)^2 \eta T}\right) \sum_{k=1}^{K} \mathbb{E}[\|\bar{z}_{k,0} - z_k^*(\bar{x}_0)\|^2]
$$

$$
+ O\left(\frac{1}{\beta(1-\lambda)\eta T}\right) \frac{1}{N} \sum_{n=1}^{N} \mathbb{E}[\|\nabla_1 f_{K+1}^{(n)}(x_0, y_{K,0})\|^2] + O\left(\frac{1}{\beta(1-\lambda)\eta T}\right) \frac{1}{N} \sum_{n=1}^{N} \mathbb{E}[\|\nabla_1 f_k^{(n)}(y_{k,0}, y_{k-1,0})\|^2]
$$

$$
+ O\left(\frac{\alpha_z}{\alpha_x \alpha_y \eta^2 T} \frac{1}{N S_0}\right) + O\left(\frac{1}{\alpha_x \eta^2 T} \frac{1}{N S_0}\right) + O\left(\frac{1}{\beta^2(1-\lambda)^2 \eta T S_0}\right) + O\left(\frac{1}{\beta^2(1-\lambda)^2 \eta T}\right)
$$

$$
+ O\left(\frac{\alpha_z \alpha_y \eta^2}{\alpha_x N}\right) + O\left(\frac{\alpha_z^2 \eta^2}{\alpha_x N}\right) + O\left(\frac{\alpha_x \eta^2}{N}\right) + O\left(\frac{\alpha_x^2 \eta^3}{\beta^2(1-\lambda)^2}\right) + O\left(\frac{\alpha_y^2 \eta^3}{\beta^2(1-\lambda)^2}\right) + O\left(\frac{\alpha_z^2 \eta^3}{\beta^2(1-\lambda)^2}\right) .
$$

$\square$

