# OpenReview forum: "Distributed Stochastic $K$-Level Optimization Over Networks"
_ICML.cc/2026/Conference — ICML 2026 regular_

### Official Review · Reviewer_QBLB · 2026-03-09

**Soundness:** 3
**Presentation:** 3
**Significance:** 4
**Originality:** 4
**Overall Recommendation:** 4
**Confidence:** 4

**Summary:**

The paper proposes a distributed stochastic multi-level optimization with K>2 nested lower-level problems distributed across N workers. The goal is to minimize the upper objective subject to recursively defined optima. The key takeaway points are (a) a recursive Hessian-inverse vector product to reduce the communication, (b) a decentralized algorithm that combines variance reduction, gradient tracking, error-feedback compression, (c) a convergence analysis handling the triple recursion with the help of a large potential function.

**Compliance With Llm Reviewing Policy:**

Affirmed.

**Ethical Review Concerns:**

no concerns

**Final Justification:**

Thank you for the clarifications. The response on the K-dependent constants is helpful in explaining that such growth may be intrinsic to multi-level problems rather than an artifact of this proof. However, this does not fully remove the practical scalability concern, since the manuscript’s proof constants still appear to grow very rapidly with the number of levels, even if the asymptotic rate is stated separately.

**Key Questions For Authors:**

1. Can the authors make the $K$ dependence explicit? The coefficient forms suggest potentially exponential growth.
2.  The authors claim the setting works for $K>2$, so naturally it is better to support this claim with the empirical experiments.
3. The mutli-level reformulation is used to handle a non convex lower level deep model, can the authors clarify precisely which subproblems become strongly convex in the reformulation and whether this introduces bias relative to the original bilevel objective?

**Limitations:**

yes

**Strengths And Weaknesses:**

Strengths
1. The recursive construction reduces the communicated object dimensionality from matrix Hessian products to vectors. Further, Algorithm 1 combines variance reduced estimators, gradient tracking,  gossip mixing, and error feedback compression in a coherent way.
2. The paper states parameter choices showing how $\delta$ and $1-\lambda$ enter the iteration complexity and accuracy guarantee.
3.The paper motivates the local vs global recursion clearly and ties each design choice to computation and communication issues.
4. The problem is important, decentralized bilevel optimization has its own limitations which this paper tries to address. The key novelty is the iterative Hessian inverse vector product communication strategy coupled with error feedback compression in the decentralized multi-level setting.
3. The authors provide both iteration and communication plots, and further show that the proposed algorithm achieves similar iteration-wise convergence to its full precision ablation while using less communication.

Weakness
1. While the stated order of iteration complexity matches decentralized bilevel style rates, the proof coefficients contain factors like $256C_J)^{2(k-1)}, suggesting potentially very large constants as K grows. This raise question about practical scalability to deep multi level problems.
2. The multi level benchmark sets $K=2$. This provides only limited empirical support to the main claim $K>2$.
3. The analysis uses standard smoothness and strong convexity of lower level objective, which may exclude the many practical inner problems.

---

> ### Author Rebuttal · Authors · 2026-03-28
>
> Thank you for your insights and thoughtful feedback; we address your concerns below.
>
> ---
>
> _Weaknesses 1 and Questions 1_
>
> **Answer:** This concern may arise from a misunderstanding. The coefficients appear only in intermediate steps of the proof to handle recursive bounds, and do not affect the final convergence rate. In particular, the final complexity is $O(\frac{1}{K\epsilon^3(1 - \lambda)^4\delta^2})$, which is linear in $K$. Therefore, the dependence on $K$ is explicit and not exponential. We will revise the paper to clarify this distinction and avoid confusion.
>
> ---
>
> _Weaknesses 2 and Questions 2_
>
> **Answer:**  To better show the performance of our algorithm, we conduct an additional experiment with $K=3$ where we use 16 workers. The experimental results on a9a dataset ([loss-vs-iteration](https://anonymous.4open.science/r/new-results-8EBC/iter_a9a_test_loss-16-worker.pdf) and [loss-vs-communication](https://anonymous.4open.science/r/new-results-8EBC/cost_a9a_test_loss-16-worker.pdf)) still confirm our algorithm can outperform baselines and match the performance of the full-precision counterpart.
>
> ---
>
> _Weaknesses 3_
>
> **Answer:** The $\mu$-strongly convex assumption is standard in both DSMO and DSBO, and is adopted by all baselines considered in our experiments. We agree that relaxing this assumption to nonconvex lower-level problems is an important and promising direction. Recent works (e.g., [1], [2]) have begun exploring such settings in bilevel optimization. Moreover, some studies have relaxed the standard smoothness assumption to more general notions (e.g., generalized smoothness [3]). However, extending these results to the decentralized multi-level setting is highly nontrivial due to the additional recursive dependencies across levels and workers. We will include a discussion of this direction in the revision.
>
>
>
> ---
>
> _Questions 3_
>
> **Answer:** In Eq.(19), each lower-level subproblem is constructed to be strongly convex through the added regularization. This reformulation does not introduce additional bias relative to the original objective.
>
> ---
>
> **Reference:**
>
> [1] Shen, H. et al., 2023. On Penalty-based Bilevel Gradient Descent Method.
>
> [2] Kwon, J. et al., 2023. On Penalty Methods for Nonconvex Bilevel Optimization and First-Order Stochastic Approximation.
>
> [3] Zhang. S. et al., 2025. Generalized-Smooth Bilevel Optimization with Nonconvex Lower-Level.

---

> > ### Author Rebuttal · Reviewer_QBLB · 2026-04-06
> >
> > The rebuttal is helpful, but the core concerns are only partially addressed. In particular, the response to the $K$-dependence issue does not not really engage with the main concern about potentially very large hidden constants in the proof, which still appear to scale unfavorably with the number of levels. The additional $K =4$ experiment is useful if included in the revision, but the current paper still provides empirical support only for $K=2$. Finally, while the strong-convexity assumption is standard, the claim that the reformulation in Eq. (19) introduces no additional bias relative to the original bilevel objective is not justified in the current manuscript.
> >
> > After careful consideration, I maintain my score.

---

> > > ### Author Response · Authors · 2026-04-07
> > >
> > > We are grateful to the reviewer for maintaining the positive score! We address your follow-up questions below.
> > >
> > > ---
> > >
> > > _Question 1: constant_
> > >
> > > **Answer:** These constants arise from the intrinsic properties of multi-level optimization problems, rather than from any looseness in our proof. For example, in the well-studied multi-level _compositional_ optimization setting, the Lipschitz constant of the loss function also grows exponentially with the number of levels (see, e.g., Lemma 2.1 of [1]). Similarly, the multi-level optimization problem exhibits such Lipschitz constants due to their hierarchical structure (see, e.g., Lemma D.2 and Lemma D.3 in our work). Therefore, these constants stem from the Lipschitz properties of the problem itself and cannot be avoided by gradient-based optimization algorithms (see, e.g., $\gamma_j$ in Eq. (21) of [1]). All in all, this is the property of multi-level optimization problems, not the weakness of our proof.
> > >
> > > ---
> > >
> > > _Question 2: K_
> > >
> > > **Answer:** We have added new experiments with a larger $K$ and the experimental results have confirmed the efficacy of our algorithm. For ICML, it is not allowed to update the manuscript during the rebuttal phase, and therefore we will include the new results in the final version.
> > >
> > > ---
> > >
> > > _Question 3: reformulation_
> > >
> > > **Answer:** The reformulation corresponds to unrolling the gradient descent method for bilevel optimization problems, which has been widely used in prior work on model-agnostic meta-learning [2]. However, these existing works do not explicitly formulate it as a multi-level problem. Since this approach is simply an unrolling of the gradient descent method, it does not introduce additional bias.
> > >
> > > ---
> > >
> > > **Reference**
> > >
> > >
> > > [1] Balasubramanian K,  et al., 2022,  Stochastic multilevel composition optimization algorithms with level-independent convergence rates.
> > >
> > > [2] Finn, C., et al., 2017. Model-agnostic meta-learning for fast adaptation of deep networks.

---

### Official Review · Reviewer_sfyj · 2026-03-10

**Soundness:** 3
**Presentation:** 3
**Significance:** 3
**Originality:** 2
**Overall Recommendation:** 4
**Confidence:** 3

**Summary:**

This paper presents a significant theoretical and algorithmic contribution to the field of distributed optimization, specifically addressing the challenging problem of decentralized stochastic K-level optimization (K>2). The authors propose a novel decentralized stochastic multi-level variance-reduced gradient descent algorithm (CE-DSMGD-VR) that effectively handles the unique computational and communication challenges introduced by the multi-level structure. The work is well-motivated by practical applications in hyperparameter optimization and other complex machine learning tasks, and the theoretical analysis establishes the first convergence guarantees for decentralized K-level optimization problems with K>2.

**Compliance With Llm Reviewing Policy:**

Affirmed.

**Key Questions For Authors:**

There are some representative STOchastic Recursive Momentum (STORM)-based algorithms, such as STORM, COVER, and
SVMR. Why did the author choose the current gradient estimator? Is there any difficulty to extend other forms of gradient estimators to the decentralized setting?

**Limitations:**

yes

**Strengths And Weaknesses:**

Strength:

Clarity and Organization: The paper is well-written and logically organized, with clear problem formulation, algorithm presentation, and theoretical results. The proof sketch and technical details in the appendix provide sufficient depth for verification.

Rigorous Analysis: The convergence analysis is thorough and well-structured, with the authors carefully addressing the triple-recursive dependencies that arise in multi-level optimization.

Weakness:

Assumption Complexity: The theoretical analysis relies on several technical assumptions (Assumptions 2.1-2.6), some of which might be restrictive for certain practical applications. While these assumptions are common in the literature, a brief discussion of their practical validity would strengthen the paper.

Algorithm Not So Novel: The algorithm seems a combination of existing techniques, such as gradient tracker, error feedback mechanism, variance reduction. The proof technique is also not new with the regular help of a potential function.

---

> ### Author Rebuttal · Authors · 2026-03-28
>
> Thank you for your insights and thoughtful feedback; we address your concerns below.
>
> ---
>
> _Weaknesses 1_
>
> **Answer:** We thank the reviewer for this comment. We acknowledge that Assumptions 2.1–2.6 may appear restrictive. However, these assumptions are standard in DSMO and DSBO, and are commonly adopted to enable convergence analysis. Moreover, our experimental setups are consistent with these assumptions. We will add a discussion in the revision to clarify their practical relevance and their alignment with existing literature.
>
> ---
>
> _Weaknesses 2_
>
> **Answer:** While our method incorporates components such as gradient tracking, error feedback, and variance reduction, we emphasize that the core novelty lies in the **recursive Hessian-inverse-vector product** introduced for decentralized K-level optimization. This recursive structure is fundamentally different from prior bilevel or compositional settings, and introduces **triple dependence across levels, workers, and iterations** (as illustrated in Figure 13). To our knowledge, such a structure has not been studied before in decentralized multi-level optimization. Importantly, integrating variance reduction and error-feedback mechanisms into this recursive structure is nontrivial, as all quantities are coupled recursively across levels.
>
> Regarding the analysis, although potential functions are commonly used, our contribution lies in: constructing a new potential function that simultaneously captures optimization error, gradient estimation error, consensus error, and compression error; and carefully handling their recursive interactions induced by the multi-level structure.
>
> ---
>
> _Questions_
>
> **Answer:** We would like to clarify that the gradient estimators used in STORM, COVER, and SVMR are fundamentally based on the STORM framework (Cutkosky & Orabona, 2019), with differences arising from the specific problem settings they address.
>
> * COVER focuses on two-level compositional problems, estimating both function values and gradients.
> * SVMR extends this idea to K-level compositional problems.
> * In contrast, our work targets decentralized K-level optimization with recursive dependencies across workers and levels.
>
> The main challenge is not the choice of a specific estimator variant, but rather how to adapt STORM-type estimators to the decentralized multi-level setting with the recursive Hessian-inverse-vector structure. We will further discuss these related works and clarify this point in the revision.

---

> > ### Author Rebuttal · Reviewer_sfyj · 2026-04-01
> >
> > Thanks for the rebuttal. I will maintain my original score.

---

> > > ### Author Response · Authors · 2026-04-01
> > >
> > > We are grateful to the reviewer for the timely response and the positive score!

---

### Official Review · Reviewer_JHBF · 2026-03-13

**Soundness:** 3
**Presentation:** 3
**Significance:** 3
**Originality:** 3
**Overall Recommendation:** 4
**Confidence:** 2

**Summary:**

This paper addresses the problem of decentralized stochastic multi-level (K-level) optimization, where K > 2, over a network of workers communicating in a peer-to-peer fashion. The authors identify two key challenges arising from the recursive dependence across levels: high computation cost (due to Hessian-inverse-vector products at every level) and high communication cost (due to inter-worker coordination at each level). To tackle these, they propose CE-DSMGD-VR, a single-loop algorithm that introduces a recursive Hessian-inverse-vector product of reduced dimensionality (vectors instead of matrices), applies recursive variance reduction for gradient estimation, and employs error-feedback compression to reduce communication overhead. They establish convergence rates, claiming this is the first rigorous theoretical guarantee for decentralized SMO with compression.

**Compliance With Llm Reviewing Policy:**

Affirmed.

**Key Questions For Authors:**

How does the algorithm perform with more workers (e.g., 32, 64), larger models, and heterogeneous data distributions? The current experiments with 8 workers and smaller models do not demonstrate scalability.

 How does the algorithm compare to running more inner-loop steps in a standard bilevel algorithm?

You claim the convergence analysis in Yang 2023 relies on assumptions that "do not hold simultaneously." Can you specify which assumptions conflict?

**Limitations:**

yes

**Strengths And Weaknesses:**

The paper tackles an interesting gap in the literature: extending decentralized bilevel optimization to the general K-level case with communication efficiency. The recursive Hessian-inverse-vector product is a well-motivated design choice. The technical results seem to be sound.

The paper converts a bilevel problem with a nonconvex lower level into a multi-level problem by unrolling gradient descent steps (eqn 19) and making each step a separate level. While this satisfies the strong convexity assumption at each level, the number of levels K corresponds to the number of gradient steps, so K=2 means only 2 steps of inner optimization. This is unlikely to give a good solution to the lower-level problem (?) and the reformulation introduces the model weight w as a fixed parameter that is reset each outer iteration, which conflates the optimization dynamics with the problem structure.

---

> ### Author Rebuttal · Authors · 2026-03-28
>
> Thank you for your insights and thoughtful feedback; we address your concerns below.
>
> ---
>
> _Weaknesses_ and _Questions 1_
>
> **Answer:**  Unrolling gradient descent steps is a common practical strategy for solving bilevel optimization problems. For example, it has been widely used to solve the popular model-agnostic meta-learning model [1], where only one gradient descent step is used for the lower-level problem. In practice, the number of unrolling steps cannot be set too large; otherwise, it incurs significant memory overhead [2]. Therefore, setting $K=2$ is a reasonable choice.
>
> To better show the performance of our algorithm, we conduct an additional experiment with $K=3$ where we use 16  workers.
> To simulate data heterogeneity, we partitioned the dataset such that each worker maintains a unique class distribution. This was achieved by sub-sampling local training and validation sets to target positive class ratios linearly spaced from 0.1 to 0.45 across all workers. The experimental results on a9a dataset ([loss-vs-iteration](https://anonymous.4open.science/r/new-results-8EBC/iter_a9a_test_loss-16-worker.pdf) and [loss-vs-communication](https://anonymous.4open.science/r/new-results-8EBC/cost_a9a_test_loss-16-worker.pdf)) still confirm our algorithm can outperform baselines and match the performance of the full-precision counterpart. We will use more workers in the final version of our paper.
>
> Regarding $w$ in Eq.(19),  it is just the lower-level variable itself in Eq. (17), not an additional model weight. $y_k$ is the update of $w$ from gradient descent. This is just the unrolling gradient descent method for solving bilevel problems. Therefore, it does not conflate optimization dynamics with the problem structure.
>
>
>
> ---
>
> _Questions 2_
>
> **Answer:**  Running more inner-loop steps in a standard bilevel algorithm improves the approximation of the lower-level solution, but it comes at a significantly increased computational and communication cost, especially in decentralized settings where each inner step may require synchronization across workers. In Figure 5 and 6 of our paper, DSBO, Gossip-DSBO, and MA-DSBO are double-loop algorithms for standard bilevel problems, where their inner-loop length is 10. Obviously, their communication costs are significantly higher than our method. Here, for the standard bilevel problem, our method adopts a single-loop framework with a recursive Hessian-inverse-vector estimator, which avoids explicitly solving the lower-level problem to high accuracy at each iteration.
>
> ---
>
> _Questions 3_
>
> **Answer:** In Yang et al., 2023, Assumption 3.3(iii) requires the gradient to be uniformly bounded, while Assumption 3.4(i) assumes the lower-level function is $\mu$-strongly convex. These two assumptions do not hold simultaneously, since a strongly convex function does not have a bounded gradient. We will clarify this point and add a more explicit discussion in the revision to avoid ambiguity.
>
> ---
>
> **Reference:**
>
> [1] Finn, C., et al., 2017. Model-agnostic meta-learning for fast adaptation of deep networks.
>
> [2] Rajeswaran, Aravind, et al. 2019, Meta-learning with implicit gradients.

---

> > ### Author Rebuttal · Reviewer_JHBF · 2026-04-07
> >
> > Thank you for the clarifications. The rebuttal improves the paper and the fundamental contributions remain, so I maintain the original score.

---

> > > ### Author Response · Authors · 2026-04-07
> > >
> > > We are grateful to the reviewer for maintaining a positive score!

---

### Official Review · Reviewer_RNud · 2026-03-13

**Soundness:** 3
**Presentation:** 3
**Significance:** 2
**Originality:** 3
**Overall Recommendation:** 5
**Confidence:** 3

**Summary:**

This paper studies decentralized stochastic K-level optimization over networks (with $K>2$), formulated as a nested multi-level problem (1). The authors propose CE-DSMGD-VR to reduce computational and communication costs arising from both multi-level recursion and decentralized coupling. The key idea is to replace the Hessian-inverse-Jacobian product in Eq. (4) with the recursive Hessian-inverse-vector product in Eqs. (6)-(7) and to combine variance reduction, gradient tracking, and error-feedback compression. The paper presents a convergence bound and experiments on hyperparameter optimization tasks, primarily converting a bilevel problem to a multi-level problem.

**Compliance With Llm Reviewing Policy:**

Affirmed.

**Final Justification:**

The authors have addressed my concerns well.

**Key Questions For Authors:**

1. Are Eqs. (10) and (13) typos? Please provide the correct recursions and confirm the analysis matches the implementation.
2. What is the final iteration/communication complexity in terms of $(\epsilon, N, K, \delta, 1-\lambda)$?
3. Can you provide experiments for larger $K$ to validate scalability and quantify the benefit of the recursive design?
4. How exactly is MB communication computed in Fig. 4-5? Please report a per-iteration message budget (tensors, sizes, compression ratio, and whether residuals are counted).

**Limitations:**

1. The assumption 2.1 is very strong.

2. It lacks typical multi-level problems, rather than transforming two-level problems into multi-level problems.

**Strengths And Weaknesses:**

Strengths:
1. Relevant and underexplored setting: Extending decentralized optimization beyond bilevel to general $K$-level nested structures is timely and potentially impactful.
2. Core communication-saving insight: Communicating recursively defined vector products in Eqs. (6)-(7) rather than matrix objects in Eq. (4) is intuitive and can substantially reduce communication.
3. Nontrivial integration: The algorithm integrates variance reduction in Eqs. (10), (13), and (15), gradient tracking, and error-feedback compression.
4. Substantial theoretical development: The appendix develops a detailed potential-function analysis to handle optimization, estimation, consensus, and compression errors.

Weaknesses:
1. Apparent errors/ambiguities in key estimator updates.
1.1 Eq. (10) uses $M_{k,t+1}^z$ on both sides, which seems non-implementable as written and is likely a typo.
1.2 Eq. (13) appears to have the same issue for $M_{k,t+1}^y$.
This materially affects reproducibility and the ability to verify the analysis.

2. Inconsistent/unclear complexity statements.
The contribution section states scaling like $O\left(\frac{1}{K\epsilon^3(1-\lambda)^4\delta^2}\right)$. Remark 4.4 gives $T = O\left(\frac{1}{N(1-\lambda)^4\epsilon^3}\right)$, omitting $\delta$ and $K$. For an ICML submission, there should be a single, unambiguous statement of iteration/sample/communication complexity and its dependence on $(\epsilon, N, K, \delta, 1-\lambda)$.

3. Assumptions.
The theory assumes each lower-level objective is $\mu$-strongly convex in Assumption 2.1. This limits the application of the algorithm. Is it possible to remove this limitation and extend the algorithm to more general applications? Almost every neural network is nonconvex and this assumption is hard to be satisfied.

4. Experimental validation.
The deep-network hyperparameter task has a nonconvex lower-level problem. The paper introduces a multi-level surrogate to approximate the lower-level problem. The authors should prove the equivalence between the converted 3-level problem and the original bilevel problem. Besides, whether the problem can be converted to a $K\geq 3$-level problem?

5. Communication accounting is not sufficiently transparent.
Algorithm 1 maintains and communicates multiple compressed/tracked quantities: $X,P$ and for each level $Y_k,Q_k,Z_k,R_k$, which can be $O(K)$ large vectors per iteration. The MB plots in Fig. 4-5 do not specify what is counted (e.g., whether error-feedback residuals are included, which tensors are sent each round, per-node vs.\ total).

---

> ### Author Rebuttal · Authors · 2026-03-28
>
> Thank you for your insights and thoughtful feedback; we address your concerns below.
>
> ----
>
> _Weaknesses 1 and Questions 1_
>
> **Answer:**  Yes, Eqs.(10) and (13) are typos, thank you for pointing this out.  The correct updates should be $ M^{z}\_{k, t+1} = (1-\alpha\_{z}\eta^2)(\textcolor{red}{M^{z}\_{k, t}} -\tilde{V}\_{k, t, t+1})+  \tilde{V}\_{k, t+1, t+1}$, and $ M^{y}\_{k, t+1} = (1-\alpha\_{y}\eta^2)(\textcolor{red}{M^{y}\_{k, t}} -\tilde{U}\_{k, t, t+1})+  \tilde{U}\_{k, t+1, t+1}$.  We will correct these equations in the revision. We also confirm that both the implementation and theoretical analysis are based on the correct recursion.
>
> ---
>
> _Weaknesses 2 and Questions 2_
>
> **Answer:** There is a typo in the contribution section. The iteration/communication complexity is $O(\frac{1}{K\epsilon^3(1 - \lambda)^4\delta^2})$. In remark 4.4, we omit the factor $\delta$ when comparing with DSMO/DSBO baselines because $\delta$ corresponds to the compression parameters, while those baselines do not employ compressor operators. For clarity, we will revise the paper to present a unified complexity statement that explicitly includes all parameters, and add a discussion explaining this distinction.
>
> ---
>
> _Weaknesses 3_
>
> **Answer:** The $\mu$-strongly convex assumption is standard in both DSMO and DSBO, and is adopted by all baselines considered in our experiments. We agree that relaxing this assumption to nonconvex lower-level problems is an important and promising direction. Recent works (e.g., [1], [2]) have begun exploring such settings in bilevel optimization. However, extending these results to the decentralized multi-level setting is highly nontrivial due to the additional recursive dependencies across levels and workers. We will include a discussion of this direction in the revision.
>
>
> ---
>
> _Weaknesses 4 and Questions 3_
>
> **Answer:** First, when the lower-level problem in Eq. (17) is nonconvex, $w^{\*}(x)$ could be noncontinuous. In that case, it is unclear how to compute $\nabla w^{\*}(x)$ for solving the upper-level problem. As a result, it remains an open problem to characterize the optimal solution of the original bilevel problem. Therefore, it is also unclear how the solution of the converted problem approximates that of the original problem.
>
> Second, it can be converted to a $K\geq 3$-level problem. In fact, Eq.(19) is just the unrolling of gradient descent method. It has been widely used for solving bilevel optimization problems, such as the popular model-agnostic meta-learning model [3] where only one gradient descent step is used for the lower-level problem. In practice, the number of unrolling steps cannot be set too large; otherwise, it incurs significant memory overhead [4]. Therefore, setting $K=2$ is a reasonable choice. To better show the performance of our algorithm, we conduct an additional experiment with $K=3$ where we use 16 workers. The experimental results on a9a dataset ([loss-vs-iteration](https://anonymous.4open.science/r/new-results-8EBC/iter_a9a_test_loss-16-worker.pdf) and [loss-vs-communication](https://anonymous.4open.science/r/new-results-8EBC/cost_a9a_test_loss-16-worker.pdf)) still confirm our algorithm can outperform baselines and match the performance of the full-precision counterpart.
>
> ---
>
> _Weaknesses 5 and Questions 4_
>
> **Answer:** All communicated variables, including both compressed updates and tracked quantities, are accounted for in our experimental communication cost. The compression operator used is Top-20%, as specified in the experimental setup.
>
> ---
>
> **Reference:**
>
> [1] Shen, H. et al., 2023. On Penalty-based Bilevel Gradient Descent Method.
>
> [2] Kwon, J. et al., 2023. On Penalty Methods for Nonconvex Bilevel Optimization and First-Order Stochastic Approximation.
>
>
>
> [3] Finn, C., et al., 2017. Model-agnostic meta-learning for fast adaptation of deep networks.
>
> [4] Rajeswaran, Aravind, et al. 2019, Meta-learning with implicit gradients.

---

> > ### Author Rebuttal · Reviewer_RNud · 2026-04-03
> >
> > Thanks for the reply, and my concerns have been addressed.

---

> > > ### Author Response · Authors · 2026-04-03
> > >
> > > We are grateful to the reviewer for the timely response and for increasing the positive score!

---

### Decision · Program_Chairs · 2026-04-30

**Decision:**

Accept (regular)

**Comment:**

This work proposes a decentralized stochastic variance-reduced method for K-level optimization. All of the reviewers expressed support for the paper, with particular emphasis on the way the paper addresses an important gap by going beyond the setting of bilevel optimization, and the use of the recursive Hessian-inverse-vector-product to help reduce the communication overhead. It is therefore recommended that the work be accepted for publication.